# OMNI-EPIC: Open-endedness via Models of human Notions of Interestingness with Environments Programmed in Code

**Maxence Faldor**[*]
Department of Computing
Imperial College London
London, United Kingdom
m.faldor22@imperial.ac.uk

**Jenny Zhang**[*]
Department of Computer Science
University of British Columbia
Vector Institute
jennyzzt@cs.ubc.ca

**Antoine Cully**[†]
Department of Computing
Imperial College London
London, United Kingdom
a.cully@imperial.ac.uk

**Jeff Clune**[†]
Department of Computer Science
University of British Columbia
Vector Institute
Canada CIFAR AI Chair
jeff.clune@ubc.ca

## Abstract

Open-ended and AI-generating algorithms aim to continuously *generate* and *solve* increasingly complex tasks indefinitely, offering a promising path toward more general intelligence. To accomplish this grand vision, learning must occur within a vast array of potential tasks. Existing approaches to automatically generating environments are constrained within manually predefined, often narrow distributions of environments, limiting their ability to create *any* learning environment. To address this limitation, we introduce a novel framework, OMNI-EPIC, that augments previous work in Open-endedness via Models of human Notions of Interestingness (OMNI) with Environments Programmed in Code (EPIC). OMNI-EPIC leverages foundation models to autonomously generate code specifying the next learnable (i.e., not too easy or difficult for the agent's current skill set) and interesting (e.g., worthwhile and novel) tasks. OMNI-EPIC generates both environments (e.g., an obstacle course) and reward functions (e.g., progress through the obstacle course quickly without touching red objects), enabling it, in principle, to create any simulatable learning task. We showcase the explosive creativity of OMNI-EPIC, which continuously innovates to suggest new, interesting learning challenges. We also highlight how OMNI-EPIC can adapt to reinforcement learning agents' learning progress, generating tasks that are of suitable difficulty. Overall, OMNI-EPIC has the potential to endlessly create learnable and interesting environments, further propelling the development of self-improving AI systems and AI-Generating Algorithms. Project website with videos: https://dub.sh/omniepic.

## 1 Introduction

In recent years, the field of artificial intelligence (AI) has witnessed groundbreaking achievements, particularly with advancements in reinforcement learning (RL) (Akkaya et al., 2019; Silver et al., 2016; Colas et al., 2019) and foundation models (FMs) (Brown, 2020; Radford et al., 2019). Yet, the most ambitious goal of AI — building generalist agents capable of autonomously perceiving, reasoning, deciding, and acting within complex environments — remains a formidable challenge. The creation of general-purpose agents promises immense societal benefits, provided we can address the critical safety and existential risks involved (Bengio et al., 2023), including those unique to

---

[*]co-authors
[†]co-senior authors

open-ended and AI-generating algorithms (AI-GAs) (Clune, 2019; Ecoffet et al., 2020). While researchers have made substantial progress in developing powerful learning architectures (Hafner et al., 2023; Jaegle et al., 2021; Vaswani, 2017), these algorithms have been applied to a relatively narrow range of tasks due to the limited availability of diverse datasets. Compounding this issue, current models require extensive amounts of data and fine-tuning to be trained. Consequently, over the past decade, the bottleneck in advancing AI has shifted from improving learning algorithms to acquiring the necessary data to train them (Jiang et al., 2023). In other words, the main challenge has become: how to effectively acquire large amounts of data for diverse tasks.

A fundamentally different approach involves developing open-ended algorithms that continuously generate and solve new challenges endlessly (Stanley et al., 2017). The objective of open-ended algorithms is to ignite an explosion of creativity and complexity in a computer, akin to the processes observed in biological evolution and human culture, including science and technology. Replicating open-ended evolution *in silico* extends beyond understanding biological life or developing engaging simulations. It delves into fundamental questions about the nature of creativity, the potential for machines to exhibit life-like characteristics, and the emergence of general intelligence. Given that biological evolution is the only known process that has successfully created general intelligence, its principles hold invaluable insights for advancing the field of AI, and creating AI-GAs (Clune, 2019).

For open-ended algorithms and AI-GAs to succeed, they must operate within a *vast* task space capable of generating an infinite array of potential challenges. Previous open-ended algorithms (Sudhakaran et al., 2024; Wang et al., 2023a; 2019; 2020; Zhang et al., 2023) have been applied to limited domains with specific types of worlds (e.g., obstacle courses) and confined to predefined parameterizations (e.g., obstacle size and type), restricting their potential to exhibit true open-endedness. Achieving *Darwin Completeness* — the potential to generate *any possible learning environment* — is essential for realizing the full potential of AI-GAs (Clune, 2019). However, Darwin Completeness presents a significant challenge: the ability to generate infinitely many environments makes the search potentially intractable, or even impossible. Exploring this immense environment search space may involve endlessly generating trivial, redundant, or overly complex tasks that do not effectively contribute to the agent's learning progress. The main challenges are to ensure that generated environments are novel, interesting, solvable, and appropriately matched to the current capabilities of the learning agents. Biological evolution required an unfathomable amount of computation over billions of years to produce intelligence. Therefore, one key scientific challenge is to determine how we can optimize the process of generating and solving interesting tasks within a Darwin Complete environment search space, so as to create an AI-GA given the computational capabilities we expect to have in the future.

Previous work in Open-endedness via Models of human Notions of Interestingness (OMNI) (Zhang et al., 2023) leverages FMs to improve open-ended learning by focusing on tasks that are both learnable and interesting. However, OMNI, like all prior open-ended works (Sudhakaran et al., 2024; Wang et al., 2019; 2020; Zhang et al., 2023; Wang et al., 2023a), was confined to generating tasks within a narrow environment search space, inhibiting the generation of *any* possible learning environment. This paper introduces a novel framework, OMNI-EPIC, that augments OMNI with Environments Programmed in Code (EPIC). OMNI-EPIC utilizes FMs to choose the next interesting and learnable task and subsequently generate environment code to enable the agent to learn how to solve that task. Our approach generates not only the simulated world but also the reward and termination functions, allowing it, in principle, to create any simulatable task. We take advantage of pre-existing simulators and OMNI-EPIC writes code to create tasks within it. For example, if the task is to kick a ball to hit a moving target, OMNI-EPIC would generate the environment code to simulate the physics, the agent, the ball, and the moving target, rewarding the agent when the ball hits the target. Conversely, for a task involving maneuvering the ball around a moving target, the simulated world remains the same, but the reward function could differ, penalizing the agent for any contact with the target. A model of interestingness (MoI) is employed both when generating the next task and checking if any newly proposed task is interestingly new compared to similar ones in the archive. Finally, we introduce a success detector that can automatically determine whether the agent has successfully completed any proposed task.

Our vision is for this algorithm to generate any code, including installing and modifying any existing simulator, or even writing the code for a new simulator. Given that the programming language used here (Python) is Turing complete, OMNI-EPIC could potentially create any computable environment (e.g., logic and math problems to quests in virtual worlds, such as building a computer in Minecraft). As a first step toward this ambitious goal, in this work, we constrain our method to write code for

one simulator, namely PyBullet (Coumans & Bai, 2016). By continuously generating learnable and interesting environments, OMNI-EPIC advances the development of self-improving AI systems, bringing us closer to achieving Darwin Completeness and realizing AI-GAs.

## 2 RELATED WORK

**Unsupervised Environment Design.** Unsupervised environment design has garnered increasing interest in RL. Several works have explored the development of auto-curricula to perpetually generate new training environments for agents (Dennis et al., 2020; Jiang et al., 2021; Parker-Holder et al., 2022; Samvelyan et al., 2023; Wang et al., 2019; 2020). However, a significant limitation of these methods is their reliance on predefined or manually curated distributions of tasks or environment parameters (Dennis et al., 2020; Heess et al., 2017; Jiang et al., 2021; Parker-Holder et al., 2022; Samvelyan et al., 2023; Wang et al., 2019; 2020), inhibiting the generation of any possible learning environment. Regret-based approaches (Jiang et al., 2021; Parker-Holder et al., 2022; Samvelyan et al., 2023) prioritize tasks with high regret, measured by the difference between the highest known return and the mean return across simulations. Alternative methods calculate learning progress by measuring the difference in the agent's task success rates across training steps (Kanitscheider et al., 2021). By focusing on tasks with high learning progress, these approaches aim to guide the agent's learning towards the most promising areas of the task space (Oudeyer et al., 2007; Oudeyer & Kaplan, 2007; Baranes & Oudeyer, 2013). However, a critical challenge remains in distinguishing which environments are genuinely interesting (Jiang et al., 2023; Zhang et al., 2023; Colas et al., 2022). In the vast space of any task describable in natural language, there may be countless learnable but not meaningful environments (e.g., kicking a ball into a goal at slightly different positions). Inspired by Zhang et al. (2023), OMNI-EPIC uses human notions of interestingness distilled into FMs to generate environments that are not only learnable, but also interesting.

**Foundation Models for Environment Design.** Recent advancements in FMs have showcased their remarkable ability to capture extensive knowledge across diverse subjects by training on vast text corpora (Bommasani et al., 2021). Consequently, this has sparked interest in applying FMs to environment design. Ma et al. (2023) utilize FMs to generate code for reward functions while Wang et al. (2023b) employ FMs to generate simulation environments and expert demonstrations. However, these methods do not build upon the agent's previous performance on different tasks, and hence lack an auto-curriculum that can provide an endless stream of environments and tasks. In procedural content generation (Shaker et al., 2016; Juliani et al., 2019; Justesen et al., 2018), Sudhakaran et al. (2024) and Todd et al. (2023) fine-tune FMs to create domain-specific levels. Bruce et al. (2024) propose training a world model to generate game environments. These approaches focus on level creation but do not train agents or adapt the difficulty based on the agent's performance or learning progress. Zala et al. (2024) generate environments as a curriculum to learn a fixed set of tasks but is limited in its ability to generate truly open-ended environments and adapt to the agent's evolving capabilities. Wang et al. (2023c) rely on a predefined set of objects and is limited in reflecting real-world dynamics in its simulation. Despite significant progress in applying FMs to environment design, opportunities remain for further exploration, such as integrating the generative capabilities of FMs with adaptive auto-curricula. OMNI-EPIC addresses this challenge, aiming to unlock the potential for truly open-ended and effective learning environments.

**Foundation Models in Open-Endedness.** The field of open-endedness seeks to create algorithmic systems that produce never-ending innovation (Stanley et al., 2017). There has been increasing interest in leveraging FMs to generate intelligent variations for code or text in evolutionary algorithms. However, these approaches are often confined to a fixed archive with hand-crafted characteristics (Bradley et al., 2023; Ding et al., 2023; Lehman et al., 2023; Lim et al., 2024). The core objective of open-endedness algorithms is to generate and solve an endless stream of tasks. One way to do so is to keep an ever-expanding archive of tasks or solutions. Zhang et al. (2023) and Wang et al. (2023a) have adopted FMs as a mechanism for auto-curricula, proposing both learnable and interesting tasks for agent training. By restricting the agent's interactions to a predefined range of environmental conditions, these methods may hinder the development of truly adaptable and versatile agents capable of handling the complexities of real-world scenarios. OMNI-EPIC addresses this limitation by leveraging FMs to generate not only tasks but also the simulated worlds and reward functions, potentially exposing agents to a wider range of challenges and learning opportunities.

# 3    METHOD

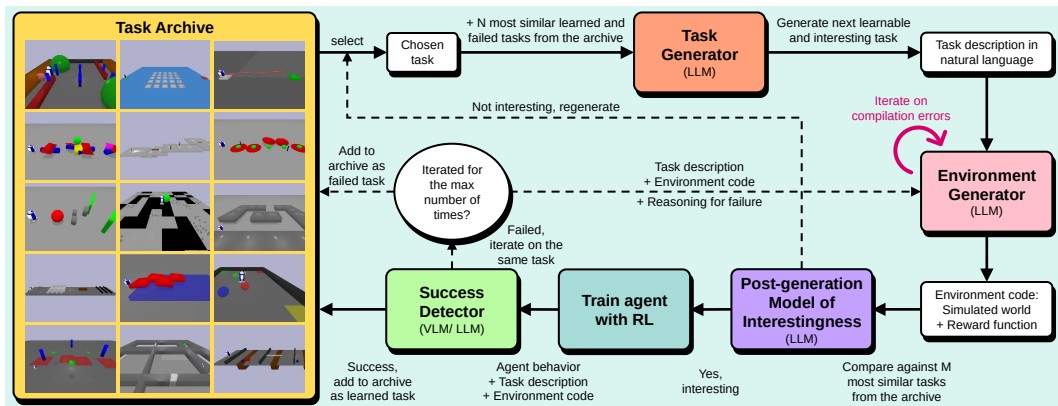

Figure 1: **OMNI-EPIC overview.** OMNI-EPIC continuously generates and solves new, interesting tasks in simulation. Our approach maintains a task archive of learned and failed tasks.

OMNI-EPIC leverages FMs, including large language models (LLMs) and vision-language models (VLMs), to autonomously create learnable and interesting tasks for open-ended learning (Figure 1). OMNI-EPIC maintains a growing task archive (Section 3.1) that catalogs successfully learned and completed tasks, as well as unsuccessfully attempted ones. The task generator (Section 3.2) uses information from the archive about what has been learned and what has not, proposing the next *interestingly new* task, described in natural language, for the agent to attempt. Because the model has distilled a sense of what is interesting from training on internet data, it has a MoI that emulates the human capacity for making nuanced judgments of interestingness in open-ended learning (Zhang et al., 2023). The task generator utilizes this MoI when generating tasks. These tasks are then translated into environment code by an environment generator (Section 3.3), specifying the simulated world and functions required for RL. The newly generated task and its environment code are assessed by a second, post-generation MoI (Section 3.4), to ensure the task is indeed interesting given what has come before (it compares the new task to the most similar tasks already in the archive with retrieval-augmented generation (Lewis et al., 2020). Tasks deemed interesting are then used to train an RL agent (Section 3.5). If deemed uninteresting, the task is discarded, and a new task is generated. After training, a success detector (Section 3.6) assesses whether the agent has successfully completed the task. Successfully completed tasks are added to the archive. Failed tasks are iterated upon a maximum number of times and added to the archive as failed tasks if the RL agents are not able to solve them. Then, the cycle of generating the next task restarts. Each component of OMNI-EPIC is explained in more detail below, and hyperparameters are shown in Appendix K. OMNI-EPIC's iterative process ensures continuous generation and learning of new interesting tasks, forming a potentially never-ending growing collection of environments and learned agents.

## 3.1    TASK ARCHIVE

OMNI-EPIC maintains a continuously expanding archive of tasks, including successfully learned ones and those attempted but failed. Successful tasks serve as stepping stones for creating more complex yet learnable ones, while failed tasks provide insights into generating new tasks within the agent's current capabilities. OMNI-EPIC uses past experiences to generate novel and diverse challenges, continuously pushing the boundaries of what's already learned. The task archive is initialized with a few task description seeds in natural language (Appendix N). This archive is unbounded and can grow indefinitely as new tasks are generated and learned. Given the generality, interpretability, and universality of natural language and programs (Hopcroft, 2001), each task is represented by its natural language description and the corresponding environment, represented as executable code.

## 3.2    TASK GENERATOR

Open-ended algorithms require focusing on tasks that are both learnable (i.e., not too difficult or too easy for the agent to learn) and interesting (i.e., worthwhile and sufficiently novel). Previous attempts have resulted in pathologies when optimizing against definitions and quantifications of interestingness (Zhang et al., 2023). Inspired by Zhang et al. (2023), we harness FMs to model

ineffable human notions of interestingness, gleaned from large text corpora of human-generated data. Here, the task generator is an LLM, which proposes novel task descriptions in natural language that are distinct from those already discovered while remaining learnable (full prompt in Appendix L.1).

To ensure the task generator remains open-ended and continuously suggests new and diverse tasks, it uses the content of the task archive as context. Given the limited context length of current LLMs, we retrieve a predefined number of tasks that are most similar to a randomly selected task from the archive (Appendix J). These retrieved tasks include both those that were successfully completed and those that were attempted but failed. These similar tasks serve as examples and are input into the LLM, which then generates the next learnable and interesting task. We opt for the most similar tasks rather than the most different ones to use previous tasks as stepping stones. As LLMs and FMs improve, we expect that a larger portion of the task archive, or potentially the entire archive, could be used as context. However, partial knowledge of stepping stones might be advantageous for creativity and diversity, much as human scientists and artists benefit from not being aware of everything that has come before. The task generator outputs a natural language description of the next task, crafted to be both achievable and interesting for the agent. This description serves as the basis for the subsequent environment generation step, where the natural language task descriptions are translated into executable code to create learning environments.

## 3.3 ENVIRONMENT GENERATOR

The environment generator, powered by an LLM, translates a given natural language task description into executable (here, Python) code defining the learning environment. Appendix L.2 shows the full prompt. This code includes specifications for creating the simulated world and functions needed for RL (Sutton, 2018) based on the standard API Gymnasium (Towers et al., 2024): `reset`, `step`, `reward`, and `terminated`. The `reset` function resets the environment to an initial state, including setting up the initial positions and orientations of the agent and objects. The `step` function updates the environment according to the simulated physics, the agent's action, and any other dynamic behaviors (e.g., moving platforms or activated doors). The `reward` function returns a scalar number, whose cumulative maximization defines the task. The `terminated` function indicates whether the agent has reached a terminal state.

For example, if the task is to "cross a bridge with moving segments", a bridge should be created in the simulated world. The `reset` function should initialize the agent at the start of the bridge and each segment's position. The `step` function should update the environment based on the agent's actions and each segment's movement. The `reward` function should reward the agent's progress across the bridge, and the `termination` function should indicate when the agent falls off the bridge.

If compilation errors occur when generating the environment code, the errors (with the traceback) are fed back into the environment generator, which then modifies and improves the environment code. Appendix L.3 shows the full prompt. This loop is limited to a maximum of five iterations per task. If the code still fails to compile after these attempts, the uncompiled code is discarded, and the task generator proposes a new task, potentially one that is less complex or differently structured.

## 3.4 POST-GENERATION MODEL OF INTERESTINGNESS

While ideally all tasks proposed by the task generator should be interesting (owing to its MoI), the limited context length of current FMs prevents the task generator from considering the entire task archive at once. Furthermore, it is often easier for an FM to evaluate whether a solution is good rather than generate a good solution (Bradley et al., 2023). OMNI-EPIC draws inspiration from the dynamics of human culture. For example, researchers often study a specific subset of previous works to inspire new ideas. Once a new idea is produced, it is crucial to check the literature to determine whether the contribution is truly novel. If the idea is deemed interestingly new, it is published, adding to the ever-growing archive of human knowledge. This process creates a growing set of stepping stones to leap off of, which is a key ingredient of open-endedness (Stanley & Lehman, 2015). OMNI-EPIC captures these important dynamics of open-ended algorithms, considering a small batch of related stepping stones when creating a new task (Section 3.2) and then verifying its novelty against the task archive. Given a newly generated task and its corresponding environment code, we compare it against a predefined number of the most similar tasks from the archive (Appendix J). The post-generation MoI then evaluates if the new task is interesting (e.g., novel, surprising, diverse, worthwhile) (full prompt in Appendix L.4). If the task is deemed interesting, we proceed to train an RL agent on it. If not, the task is discarded, and a new one is generated.

### 3.5 TRAINING AGENTS WITH REINFORCEMENT LEARNING

A key objective of open-ended learning is to enable agents to master an ever-expanding set of tasks. To achieve this, OMNI-EPIC generates a diverse array of tasks along with their corresponding environments programmed in code. Agents are then trained in these environments using RL to solve the generated tasks. In this work, we utilize the PyBullet physics simulator (Coumans & Bai, 2016). While any RL algorithm could be used, we employ DreamerV3 (Hafner et al., 2023). Appendix K details the hyperparameters and compute resources used. Agents receive proprioceptive (joints positions and velocities) and visual information (images of size $64 \times 64 \times 3$). While OMNI-EPIC can be applied to any robot (Appendix P), due to computational limitations, we demonstrate results using an R2D2 robot with a discrete action space of six actions: do nothing, go forward, go backward, rotate clockwise, rotate counterclockwise, and jump. R2D2's simple action space allows the agent to learn tasks more efficiently, requiring fewer time steps than agents with complex action spaces.

Following Wang et al. (2020), we train one specialist RL agent for each task. For tasks used to initialize the archive, the RL agents are trained from scratch. For newly generated tasks, the agent continue training from an existing policy previously trained on tasks in the archive. The trained policy is selected from a successfully completed task with the closest embedding to the new task (Appendix J). This approach allows the agents to build upon their existing knowledge and adapt more efficiently to challenges presented by the new tasks. Additionally, color variation in the environments is a form of domain randomization, as an RL agent trained on an environment with specific colors may not perform well in an identical environment with different colors (Tobin et al.).

### 3.6 SUCCESS DETECTOR

In this infinite task space of any task describable in natural language, an essential ingredient to training agents to learn any generated task is a universal reward function, which can evaluate if *any* proposed task has been completed or not. We employ a success detector, instantiated as an LLM or VLM, that assesses whether the agent has successfully completed the given task. Since our preliminary testing found that current VLMs are not yet accurate enough to be used as success detectors (Appendix C), we use code generated by LLMs for this purpose instead.

When using an LLM as the success detector, we ask the environment generator to generate an additional success-checking function `get_success`, alongside the environment code, that checks if the agent has successfully completed the task. The success-checking function serves a different purpose from the reward function. The reward function is designed to enable the RL agent to learn efficiently, which often results in a more complex function than the success-checking function. The complexity of the reward function arises from the need to shape rewards to facilitate optimal learning (Krakovna et al., 2020). Meanwhile, the success-checking function aims to evaluate whether the agent has accomplished the task. This function is less susceptible to reward hacking, as it does not affect how the agent learn. For example, consider the task of "run forward". The success-checking function would evaluate whether the agent's x-axis position has exceeded a certain threshold. In contrast, the reward function should encourage efficient learning by rewarding the agent's x-axis velocity and promoting cyclic joint movements that resemble a natural running motion.

If a task is deemed to have been successfully learned by the trained agent, it is added to the archive. If not, the task is returned to the environment generator for modifications to aid in the agent's learning (e.g., changing the reward function or making the physical world less complex). This feedback loop is repeated for a max number of attempts. If the task remains unlearned after these attempts, it is added to the archive as a failed task and a new task is generated.

## 4 LONG RUN WITH SIMULATED LEARNING

To illustrate the creative explosion of generated tasks, we run OMNI-EPIC without training RL agents, assuming all generated tasks can be successfully completed. This allows us to showcase a larger number of generated tasks, as training RL agents on each task is more time-consuming. Excluding tasks that did not generate executable code, Figure 2 shows 200 iterations of OMNI-EPIC with the R2D2 robot. Each node represents a generated task, which includes the natural language task description and environment code. The color of the nodes corresponds to the generation number. For better visualization, each task (natural language description and environment code) is encoded using a pre-trained encoder language model (OpenAI's text-embedding-3-small (OpenAI, 2024))

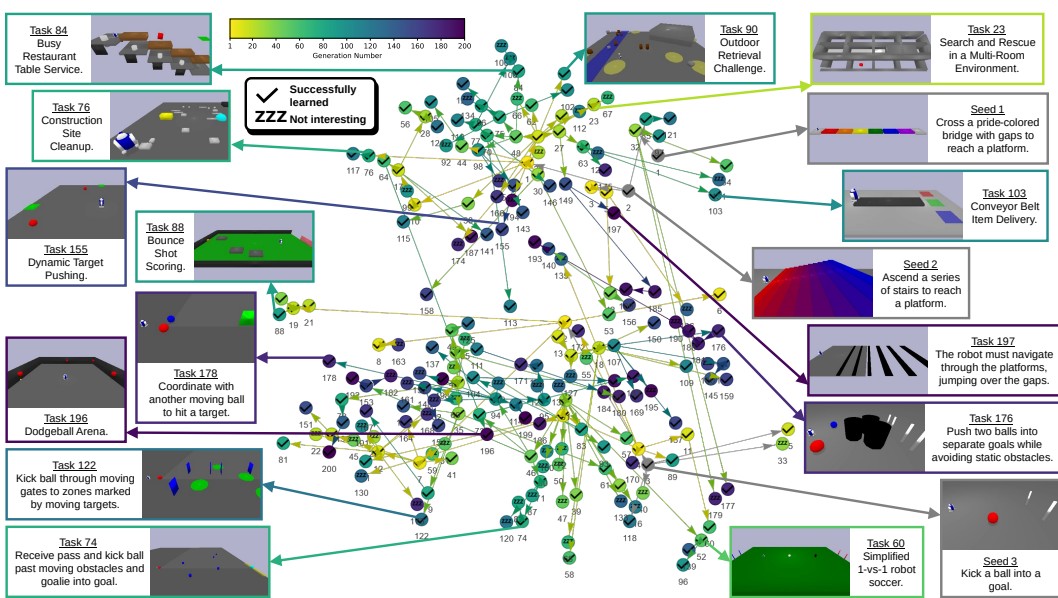

Figure 2: **Long Run with Simulated Learning.** OMNI-EPIC generates a diverse array of tasks, ranging from wildly different objectives to interesting variations of similar overarching tasks. The node color reflects the generation number of the task. A check mark in the node means that the task was successfully learned. A ZZZ symbol means that the task was deemed uninteresting and discarded. The node connections illustrate which tasks were conditioned on when asking an FM to generate a similar yet new and interesting task. Grey nodes show task description seeds that initialized the run.

and then projected into a 2-dimensional space using t-SNE (Van der Maaten & Hinton, 2008). We manually selected a few tasks that are well-distributed across the embedding space to show a sample of the diversity and creativity of OMNI-EPIC. Connections between nodes indicate which tasks were used as stepping stones to generate the next task, meaning they were provided in context to the task generator. For better readability, Figure 2 only displays the parent task that is closest to the child task in the embedding space and the header text of task descriptions. Appendix D shows the full natural language descriptions of the magnified tasks and provides a more comprehensive visualization of all parent-child task connections. Appendix N shows the three task descriptions that seed the run.

OMNI-EPIC creates tasks that evolve and diverge across the embedding space, forming clusters of related and increasingly complex challenges (Figure 2). For example, the bottom-right region features tasks involving kicking a ball into a goal. Moving towards the bottom-left, the tasks gradually include dynamic elements such as moving obstacles or targets, while still focusing on kicking a ball. As we traverse to the top-left region, the focus shifts to tasks that require pushing or delivering objects into designated receptacles or target places. This marks a significant departure from the ball-kicking tasks and showcases the diversity of the generated environments. The top-right portion of the space is characterized by generations that emphasize navigation challenges, such as traversing (often moving) platforms or navigating across varied terrains.

Furthermore, there are niches where some generations appear to be variations of each other (Figure 2). For example, in the top-right corner, the generations surrounding task 90 involve retrieving an object and returning it to a designated location. While the overarching objective surrounding that niche remains consistent, the tasks and their learning environments exhibit notable differences. These variations manifest in several aspects, such as the simulated world in which the task takes place (e.g., task 90 is outdoors, task 24 is in connected rooms, task 27 is in two levels connected by a ramp). Other variations include the number of objects that need to be retrieved (e.g., task 102 requires multiple objects to be retrieved, while task 90 only requires one), and the allotted time for completing the task (e.g., task 27 has a time limit of 3 minutes, while task 24 has a time limit of 5 minutes). These task variations can potentially allow for the development of robust and adaptable agents capable of handling a wide range of scenarios (Adaptive Agent Team et al., 2023).

Overall, OMNI-EPIC generates tasks that significantly diverge from the seed tasks used to initialize the archive (the grey nodes in Figure 2). For example, despite the absence of dynamic objects in the

seed tasks, OMNI-EPIC generates a substantial number of tasks that incorporate dynamic objects and interactions, such as platforms that move horizontally and vertically, buttons and levers to activate, and moving obstacles. OMNI-EPIC not only explores different task niches (e.g., navigating across different terrains vs. retrieving objects) but also generates interesting variations within each niche (e.g., retrieving objects in different simulated world settings).

# 5 SHORT RUN WITH LEARNING

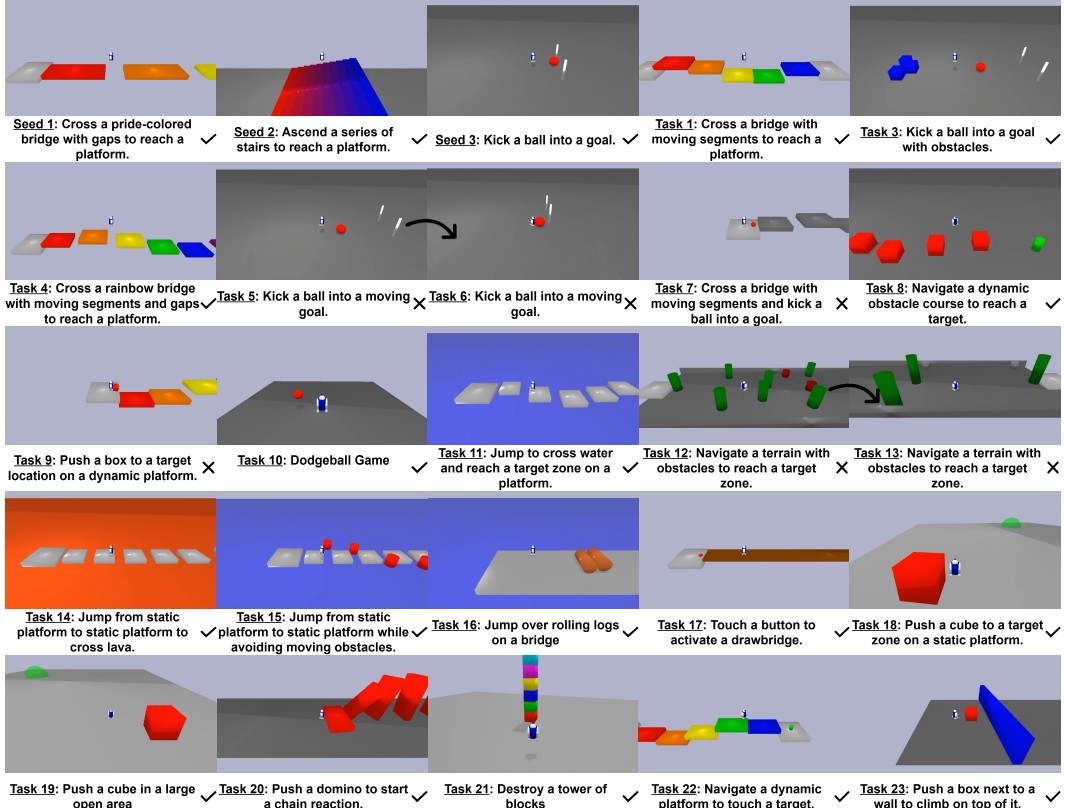

Figure 3: **Short Run with Learning.** OMNI-EPIC adapts to the current capabilities of trained RL agents, generating tasks that are both interesting and learnable. Tasks deemed interesting that are successfully learned are marked by a check and failures by a cross. Uninteresting tasks are not trained on and hence not included here. Arrows between tasks indicate instances where OMNI-EPIC modified a task that the RL agent failed to learn, adjusting the task difficulty to facilitate learning.

To demonstrate OMNI-EPIC's ability to generate tasks of suitable difficulty for training RL agents, we conducted 5 short runs with RL agent training. Due to limited computational resources in our academic lab, the runs are shorter, but still show the creative potential of OMNI-EPIC and its ability to tailor tasks to the agents' abilities. The success detector (Section 3.6) evaluates if the agent has successfully completed each task. All short runs are initialized with 3 task description seeds (Appendix N). We find that OMNI-EPIC can effectively generate tasks and environment code that are not only creative and interesting but also learnable and appropriately challenging for RL agents (Figure 3, Appendix F). In the run shown in Figure 3, the RL agents successfully completed 16 tasks, failed at 6 tasks, and 1 task was deemed uninteresting by the post-generation MoI. We conducted a user study with 50 participants and found a 72.7% alignment rate between human evaluations and the success detector's assessments (Appendix H). Figure 3 shows images of the trained agents and their corresponding tasks, displaying only the header text of the task descriptions for better readability. Appendix E shows the environment code with full task descriptions, images, and a task graph.

OMNI-EPIC leverages previously learned tasks as stepping stones to generate and master more challenging tasks. This iterative process allows RL agents to build upon existing skills to tackle increasingly complex environments. For example, since RL agents in the archive successfully learned

to cross a pride-colored bridge with gaps (seed 1) and cross a bridge with moving segments (task 1), OMNI-EPIC branched off these tasks to generate a more challenging task (task 4) combining these elements. This new task required the agent to cross a rainbow bridge with gaps and moving segments. By continuing training from the policy learned on task 1, the RL agent successfully completed task 4. Furthermore, OMNI-EPIC considers tasks that the RL agents agents failed to learn. For example, since the RL agent failed to learn how to push a box to a target location on a dynamic platform (task 9), future tasks involving crossing platforms did not include pushing objects across them (e.g., tasks 11, 14, 15). This ensures that the generated tasks remain learnable and do not repeatedly incorporate overly difficult challenges. Similarly, when the RL agent failed to navigate through a terrain with obstacles (task 12), OMNI-EPIC generates an easier obstacle course (task 13) that maintains the same objective but reduces the number of obstacles. By using the agent's past experiences (both successes and failures) as building blocks, OMNI-EPIC generates a curriculum of tasks that progressively and interestingly increases in difficulty while remaining learnable (additional ablations in Appendix G). This adaptive task generation process showcases OMNI-EPIC's capacity to create a tailored curriculum that maintains an appropriate level of challenge, ensuring that the generated tasks are neither too simple nor too complex.

## 6 QUANTITATIVE RESULTS

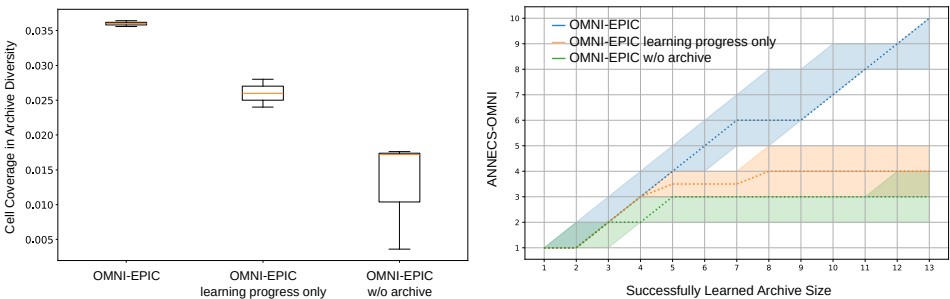

Figure 4: **OMNI-EPIC generates significantly more diverse tasks and continues to innovate throughout the run.** (Left) Cell coverage of archive diversity plots in long runs with simulated learning by OMNI-EPIC and the controls. (Right) ANNECS-OMNI measure of progress for OMNI-EPIC and the controls. Dotted lines are median values, shaded regions are 95% confidence intervals.

To evaluate the impact of having a task archive and the contribution of the notion of interestingness in OMNI-EPIC, we compare against two controls: (1) OMNI-EPIC without the task archive (**OMNI-EPIC w/o archive**), and (2) OMNI-EPIC without the models of interestingness (by removing the request for interesting tasks from the task generator prompts and skipping the post-generation MoI step) (**OMNI-EPIC Learning Progress Only**). The quantitative measures for comparisons are described next, and OMNI-EPIC significantly outperforms the controls on both metrics.

**Task Diversity.** To measure the diversity of generated task archives, we plot the cell coverage of a task archive in a 2D discretized plot. First, we use a pretrained text embedding model (OpenAI's text-embedding-3-small (OpenAI, 2024)) to encode the generated tasks (natural language description and environment code), then reduce the dimensionality to two via PCA (Maćkiewicz & Ratajczak, 1993) (across all tasks from all methods) for easier visualization. We create 2D discretized plots by selecting a discretization level (e.g., 50 in Figure 4) and generating uniform bins across the minimum and maximum values from the PCA embeddings. Each task in the archive is placed in the appropriate bin, and we count the number of unique bins the algorithm fills, a standard measure of diversity (Mouret & Clune, 2015; Pugh et al., 2016). Each method is run with simulated learning, repeated 3 times. OMNI-EPIC achieves significantly higher cell coverage than the controls (p < 0.05, Mann-Whitney U test). This shows that both the task archive and the MoI significantly and quantifiably contribute to OMNI-EPIC's ability to generate more diverse tasks (Figure 4). Appendix I shows the archive diversity plots for each method and that the same results hold for different discretization levels.

**Measure of Progress.** While open-ended systems aim to endlessly create and learn new tasks, the question of how to measure progress in such systems remains. Wang et al. (2020) proposed tracking the Accumulated Number of Novel Environments Created and Solved (ANNECS) throughout the run of an open-ended system. Specifically, ANNECS requires that an environment created at a particular iteration (1) is not too hard or easy for the agent to learn and (2) must eventually be solved by the

system. However, ANNECS lacks a measure of how interesting or novel the newly generated task is. To address this limitation, we introduce a new metric to measure progress in open-ended systems, ANNECS-OMNI. Inspired by ideas in Zhang et al. (2023), ANNECS-OMNI adds a third criterion to ANNECS: the new task must be considered interesting compared to previous tasks (approximated here by asking an FM if the task is interesting given the archive of already-solved tasks).

Each method is run with RL training, repeated 5 times. OMNI-EPIC achieves significantly higher ANNECS-OMNI scores than the controls ($p < 0.05$, Mann-Whitney U test). As the run proceeds, the ANNECS-OMNI metric consistently increases for OMNI-EPIC, indicating that, for as long as we could afford to run it, the algorithm continuously creates meaningfully new and interesting tasks without stagnation (Figure 4). That is a new high watermark in our field's longstanding quest to create open-ended algorithms. The ANNECS metric for the different methods can be found in Appendix I.

## 7 DISCUSSION, FUTURE WORK, CONCLUSION

OMNI-EPIC provides a general recipe for generating a potentially endless stream of learnable and interesting environments. By leveraging FMs to generate tasks and their corresponding environment (and reward) code, OMNI-EPIC eliminates the need for handcrafted parameters or predefined task distributions, unshackling algorithms to have the potential to be truly open-ended. OMNI-EPIC can produce a wide variety of tasks, from simple to complex, and allows for the creation of environments that continuously adapt in response to an agent's developing capabilities. The results demonstrate its effectiveness in generating diverse and creative tasks, highlighting the potential of this approach for training more capable and intelligent agents. By leveraging the generative capabilities of FMs to create a vast array of unique and challenging environments, OMNI-EPIC brings us one step closer to achieving Darwin Completeness, the ability to create *any* possible learning environment.

While we cannot rule out that OMNI-EPIC is creating environments similar to those in its training data, (1) even if it were, this would still be useful for open-endedness; (2) we believe it is generating novel environments, as we were unable to find the same environments when searching online; and (3) with longer runs, we hypothesize it could generate environments endlessly. Its task generator and MoI could generalize well outside the distribution of human data when conditioned on a growing archive of discoveries, though studying this remains a fascinating area for future research.

However, it is important to acknowledge that the current implementation of OMNI-EPIC is not yet Darwin Complete, primarily due to the limitations imposed by the choice of simulator. While OMNI-EPIC can generate a wide variety of tasks and environments, it is constrained by the capabilities and assumptions of the underlying simulation platform. To achieve true Darwin Completeness, OMNI-EPIC could simply be allowed to generate any code, including the ability to download, install, use, or modify any existing simulator, or even write the code for an entirely new simulator from scratch. Since the programming language used here (Python) is Turing Complete, generating code can, in principle, create any computable environment. Enabling the generation of arbitrary code would unlock the full potential of OMNI-EPIC and bring us closer to realizing the vision of Darwin Completeness. Of course, ever smarter FMs are required to take advantage of this opportunity.

The current implementation of OMNI-EPIC trains a population of specialist agents. We do not provide evidence of their generalization across tasks as it is not the focus of the paper. Future research could explore alternative training modalities. One promising direction is to train a single policy across all environments (Adaptive Agent Team et al., 2023), which could encourage the development of more versatile and adaptable agents (Appendix O). Another approach is to prioritize environments based on learning progress (Baranes & Oudeyer, 2013; Kanitscheider et al., 2021), focusing on tasks that provide the most opportunities for improvement. Each of these strategies introduces unique dynamics into the open-ended environment generation process, and understanding their effects on agent performance and generalization represents an exciting avenue for future research.

In conclusion, OMNI-EPIC represents a leap towards open-ended learning by generating a potentially endless stream of learnable and interesting tasks. Intriguingly, it also provides a new way of creating human entertainment and educational resources by offering a limitless supply of engaging challenges (Appendix B). OMNI-EPIC could potentially be applied in myriad ways, covering anything from math problems and poetry challenges to games and virtual worlds. By leveraging FMs to create tasks and environment code, OMNI-EPIC opens up a vast space of possibilities for AI and human agents to explore and master. By combining that expressive power with human notions of interestingness, OMNI-EPIC presents a promising path towards the development of truly open-ended and creative AI.

ACKNOWLEDGMENTS

This research was supported by the Vector Institute, the Canada CIFAR AI Chairs program, a grant from Schmidt Futures, a grant from G-Research, an NSERC Discovery Grant, the Center for AI Safety Compute Cluster, DARPA and a generous donation from Rafael Cosman. Any opinions, findings, and conclusions or recommendations expressed in this material are those of the authors and do not necessarily reflect the views of the sponsors. We also thank Aaron Dharna, Arthur Braida, Ben Norman, Cong Lu, Gabriel Béna, Luca Grillotti, Rach Pradhan, and Shengran Hu, for insightful discussions and feedback.

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

## A    REPRODUCIBILITY

For full transparency and to ensure reproducibility, we have open-sourced all code associated with this work. The code can be found at `https://github.com/maxencefaldor/omni-epic`.

## B    GAME INTERFACE

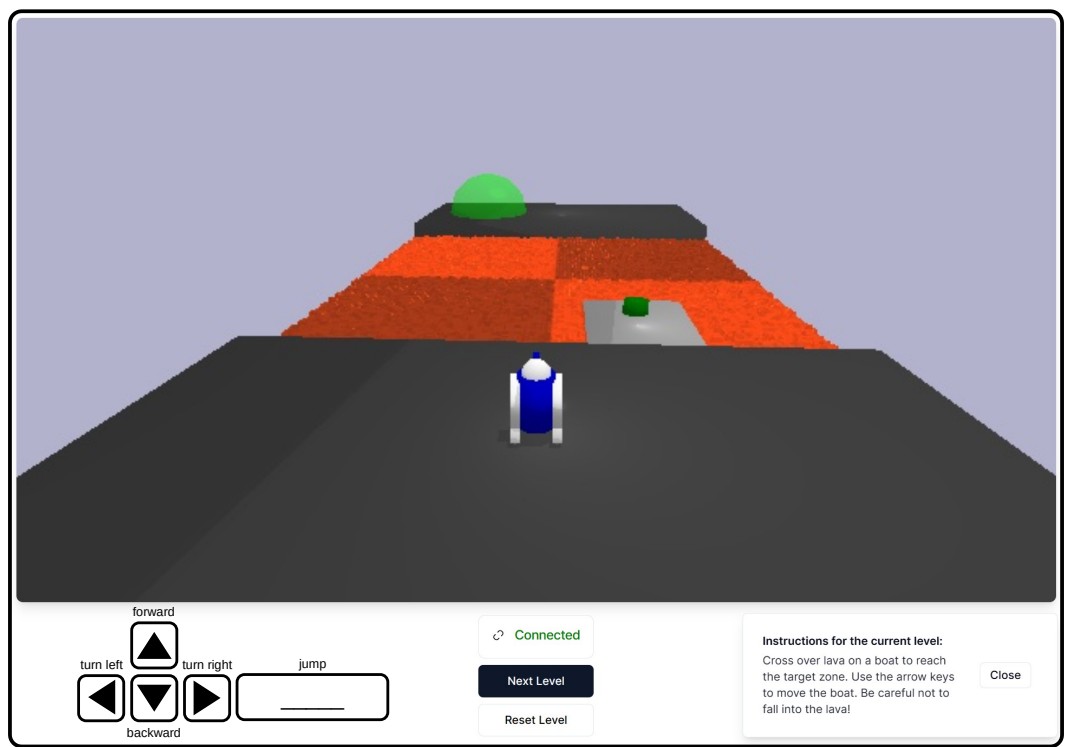

Figure 5: **OMNI-EPIC in a game interface.**

Apart from training agents, OMNI-EPIC can also be a good form of human entertainment! We created a game interface whereby players can control the R2D2 robot with keyboard inputs, and play the generated tasks (Figure 5). OMNI-EPIC dynamically generates the next interesting level for the player, adjusting based on the player's skill level by suggesting tasks that are not too easy or difficult. OMNI-EPIC opens a new era for games, where procedural content is automatically generated and tailored to the player's abilities, ensuring a consistently engaging experience.

## C  VLM AS SUCCESS DETECTOR

The success-checking function can be easily implemented in simulated environments where information can be readily accessed through code (Section 3.6). However, it may face challenges in real-world scenarios, off-the-shelf closed-source video games, or even within simulated environments for tasks that involve visual assessment (e.g., building a castle, arranging boxes to resemble an elephant), where the required information is not directly available or difficult to evaluate using only code. A natural solution is to use VLMs as success detectors (Radford et al., 2021). VLMs can potentially detect success on a wider range of tasks (e.g., tidying a room, doing a backflip) (Du et al., 2023) than code generation. We input snapshots of the agent's behavior every second, the natural language task description, and environment code (see below). Since our preliminary testing found that current VLMs are not yet accurate enough to be used as success detectors, we use code generated by LLMs for this purpose instead. However, we expect VLM capabilities to rapidly improve over time, eventually achieving higher accuracy and making them viable for future use.

**System Prompt:**

```
You are an expert in Python programming and reinforcement learning. Your
    goal is to evaluate if a robot has solved a task. You will be
    provided with the task description, the corresponding environment
    code and an image containing snapshots of the robot attempting to
    complete the task. Your objective is to describe the image, reason
    about whether the task has been completed and determine if the robot
    has solved the task.

Instructions:
— In the description of the image, describe the environment and the
    behavior of the robot.
— In the reasoning, analyze if the environment corresponds to the task
    description and if the behavior of the robot meets the requirements
    for task success.
— The task is considered failed if the environment is constructed in a
    way that makes solving the task impossible.
— If you are unsure, make an educated guess and always provide an answer.
— If you are unsure, say that it has failed.

Robot description:
{ROBOT_DESC}

Desired format:
Description of the image:
<image description>

Reasoning for task success/failure:
<reasoning>

Did the robot solve the task?:
<Yes/No>
```

**User Prompt:**

```
Task description and environment code:
{ENV_CODE}
```

**Image Examples:**

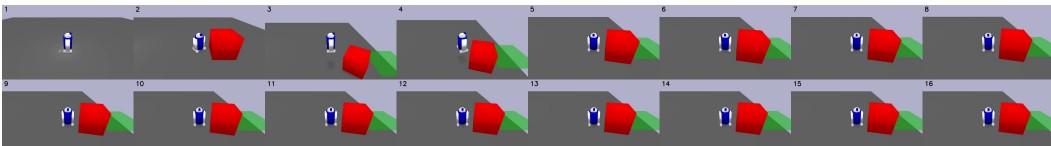

Task Description: Push a box to a target area.

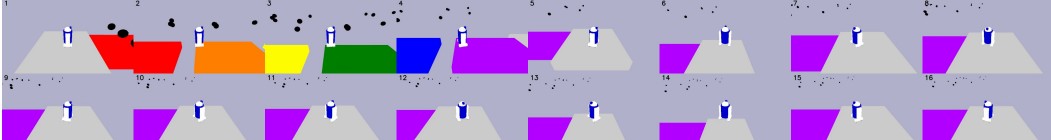

Task Description: Cross a pride-colored bridge with gaps while avoiding moving obstacles to reach a platform.

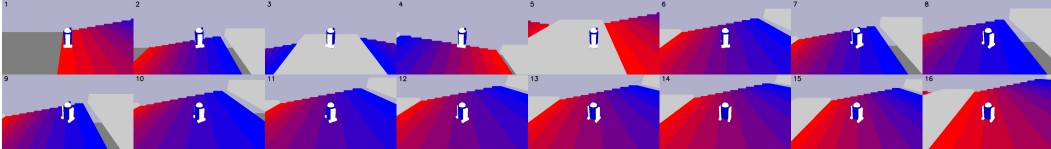

Task Description: Navigate a multi-level environment by ascending and descending multiple staircases.

# D    SUPPLEMENTARY MATERIALS FOR LONG RUN WITH SIMULATED LEARNING

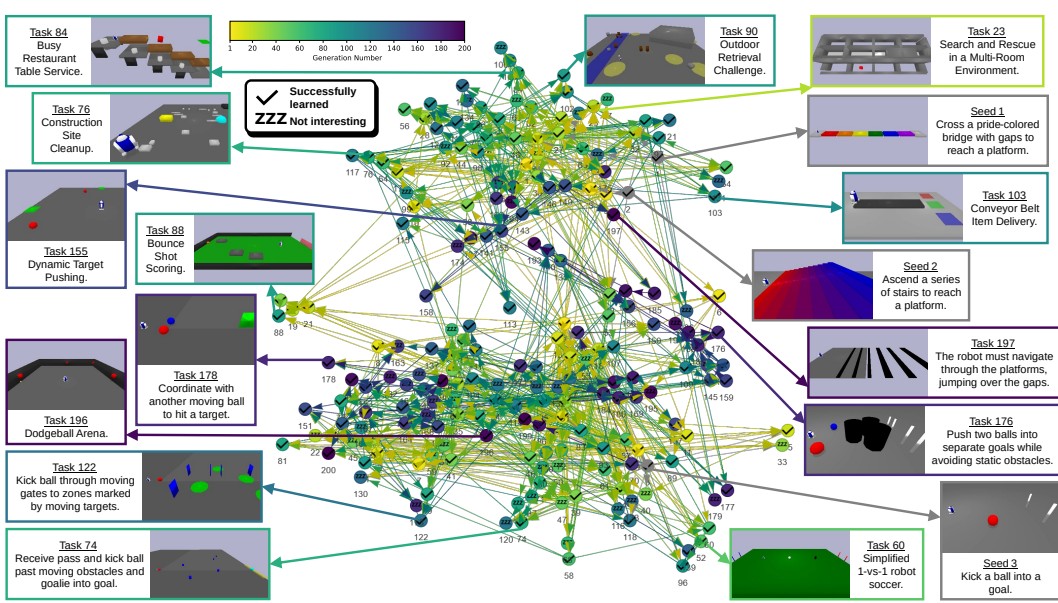

Figure 6: **Long run with simulated learning task graph with all parent-child task connections.** This figure presents the same task graph as Figure 2, but with all parent-child task connections displayed. The node color reflects the generation number of the task. A check mark in the node means that the task was successfully learned. A ZZZ symbol means that the task was deemed uninteresting and discarded. The node connections illustrate which tasks were conditioned on when asking an FM to generate a similar yet new and interesting task. Due to the high density of tasks and connections, visualizing all relationships clearly in a static image is challenging. To better understand and navigate the intricate web of task relationships, an interactive version of the task graph is available at https://dub.sh/omniepic.

The natural language descriptions and environment code for all tasks shown in Figure 2 are available at `https://dub.sh/omniepic`. For conciseness, we present the full natural language descriptions of all magnified tasks, and the environment code of one magnified task (Task 76):

```python
import numpy as np
from oped.envs.r2d2.base import import R2D2Env

class Env(R2D2Env):
    """
    Construction Site Cleanup

    Description:
    — The environment is a construction site with dimensions 10 m x 10 m.
    — There are 3 piles of different types of debris scattered around the
        construction site:
      — 5 bricks (each 0.2 m x 0.2 m x 0.2 m)
      — 5 metal scraps (each 0.5 m x 0.25 m x 0.25 m)
      — 5 wooden planks (each 1 m x 0.2 m x 0.1 m)
    — There are also 2 robotic construction vehicles moving around the
        site in pre—programmed patterns:
      — A bulldozer that pushes dirt piles around in a rectangle, 4 m x 3
          m
      — An excavator that swings its arm and bucket back and forth in a
          180 degree arc, 2 m in radius
    — On one side of the construction site are 3 square receptacle bins,
        each 1 m x 1 m x 0.5 m tall, labeled for each type of debris (
        bricks, metal, wood).
    — The robot starts in one corner of the site.

    Task:
    The robot needs to pick up each piece of debris and place it in the
        correct receptacle, while avoiding collisions with the moving
        construction vehicles. The robot can only carry one piece of
        debris at a time.

    Success Conditions:
    The task is complete when all pieces of debris have been placed in
        their correct bins, and the robot has returned to its starting
        position.

    Time Limit:
    The robot has 10 minutes to complete the cleanup task.

    Rewards:
    — Provide a small reward for each piece of debris successfully picked
        up.
    — Provide a moderate reward for each piece of debris placed in the
        correct bin.
    — Provide a large reward for completing the task and returning to the
        start position.
    — Provide a small penalty for each collision with the construction
        vehicles.

    Termination:
    The episode ends if the robot flips over, or if the time limit is
        exceeded.
    """

    def __init__(self):
        super().__init__()
        self.site_size = [10.0, 10.0, 0.1]
        self.site_position = [0.0, 0.0, 0.0]
        self.site_id = self.create_box(mass=0.0, half_extents=[self.
            site_size[0] / 2, self.site_size[1] / 2, self.site_size[2] /
            2], position=self.site_position, color=[0.5, 0.5, 0.5, 1.0])
```

```python
        self._p.changeDynamics(bodyUniqueId=self.site_id, linkIndex=-1,
            lateralFriction=0.8, restitution=0.5)
        self.brick_size = [0.2, 0.2, 0.2]
        self.metal_size = [0.5, 0.25, 0.25]
        self.wood_size = [1.0, 0.2, 0.1]
        self.debris_sizes = [self.brick_size, self.metal_size, self.
            wood_size]
        self.num_debris_types = len(self.debris_sizes)
        self.num_debris_each = 5
        self.debris_ids = []
        for i in range(self.num_debris_types):
            for _ in range(self.num_debris_each):
                debris_position = [np.random.uniform(-self.site_size[0] /
                    2 + 2.0, self.site_size[0] / 2 - 2.0), np.random.
                    uniform(-self.site_size[1] / 2 + 2.0, self.site_size
                    [1] / 2 - 2.0), self.debris_sizes[i][2] / 2]
                debris_id = self.create_box(mass=1.0, half_extents=[self.
                    debris_sizes[i][0] / 2, self.debris_sizes[i][1] / 2,
                    self.debris_sizes[i][2] / 2], position=
                    debris_position, color=[0.8, 0.8, 0.8, 1.0])
                self.debris_ids.append((debris_id, i))
        self.bulldozer_size = [1.0, 0.5, 0.5]
        self.bulldozer_position_init = [-self.site_size[0] / 4, 0.0, self
            .bulldozer_size[2] / 2]
        self.bulldozer_id = self.create_box(mass=0.0, half_extents=[self.
            bulldozer_size[0] / 2, self.bulldozer_size[1] / 2, self.
            bulldozer_size[2] / 2], position=self.bulldozer_position_init
            , color=[1.0, 1.0, 0.0, 1.0])
        self.excavator_radius = 2.0
        self.excavator_position_init = [self.site_size[0] / 4, 0.0, 0.0]
        self.excavator_id = self.create_sphere(mass=0.0, radius=0.5,
            position=self.excavator_position_init, color=[0.0, 1.0, 1.0,
            1.0])
        self.receptacle_size = [1.0, 1.0, 0.5]
        self.receptacle_positions = [[self.site_size[0] / 2 - self.
            receptacle_size[0] / 2, -self.site_size[1] / 3, self.
            receptacle_size[2] / 2], [self.site_size[0] / 2 - self.
            receptacle_size[0] / 2, 0.0, self.receptacle_size[2] / 2], [
            self.site_size[0] / 2 - self.receptacle_size[0] / 2, self.
            site_size[1] / 3, self.receptacle_size[2] / 2]]
        self.receptacle_ids = []
        for i in range(self.num_debris_types):
            receptacle_id = self.create_box(mass=0.0, half_extents=[self.
                receptacle_size[0] / 2, self.receptacle_size[1] / 2, self
                .receptacle_size[2] / 2], position=self.
                receptacle_positions[i], color=[0.2, 0.2, 0.2, 1.0])
            self.receptacle_ids.append(receptacle_id)
        self.robot_position_init = [-self.site_size[0] / 2 + 2.0, -self.
            site_size[1] / 2 + 2.0, self.site_size[2] + self.robot.links[
            'base'].position_init[2] + 0.1]
        self.robot_orientation_init = self._p.getQuaternionFromEuler
            ([0.0, 0.0, np.pi / 4])
        self.time_limit = 600.0
        self.debris_pick_reward = 1.0
        self.debris_place_reward = 10.0
        self.task_complete_reward = 100.0
        self.collision_penalty = -1.0

    def create_box(self, mass, half_extents, position, color):
        collision_shape_id = self._p.createCollisionShape(shapeType=self.
            _p.GEOM_BOX, halfExtents=half_extents)
        visual_shape_id = self._p.createVisualShape(shapeType=self._p.
            GEOM_BOX, halfExtents=half_extents, rgbaColor=color)
```

```python
        return self._p.createMultiBody(baseMass=mass,
            baseCollisionShapeIndex=collision_shape_id,
            baseVisualShapeIndex=visual_shape_id, basePosition=position)

    def create_sphere(self, mass, radius, position, color):
        collision_shape_id = self._p.createCollisionShape(shapeType=self.
            _p.GEOM_SPHERE, radius=radius)
        visual_shape_id = self._p.createVisualShape(shapeType=self._p.
            GEOM_SPHERE, radius=radius, rgbaColor=color)
        return self._p.createMultiBody(baseMass=mass,
            baseCollisionShapeIndex=collision_shape_id,
            baseVisualShapeIndex=visual_shape_id, basePosition=position)

    def get_object_position(self, object_id):
        return np.asarray(self._p.getBasePositionAndOrientation(object_id
            )[0])

    def reset(self):
        observation = super().reset()
        self.time = 0.0
        self.debris_picked = [False] * len(self.debris_ids)
        self.debris_placed = [False] * len(self.debris_ids)
        self._p.resetBasePositionAndOrientation(self.robot.robot_id, self
            .robot_position_init, self.robot_orientation_init)
        return observation

    def step(self, action):
        observation, reward, terminated, truncated, info = super().step(
            action)
        self.time += self.dt
        bulldozer_position = self.get_object_position(self.bulldozer_id)
        bulldozer_position[0] = self.bulldozer_position_init[0] + 2.0 *
            np.sin(2 * np.pi * self.time / 20.0)
        bulldozer_position[1] = self.bulldozer_position_init[1] + 1.5 *
            np.sin(2 * np.pi * self.time / 30.0)
        self._p.resetBasePositionAndOrientation(self.bulldozer_id,
            bulldozer_position, [0.0, 0.0, 0.0, 1.0])
        excavator_position = self.get_object_position(self.excavator_id)
        excavator_position[0] = self.excavator_position_init[0] + self.
            excavator_radius * np.cos(np.pi * self.time / 10.0)
        excavator_position[1] = self.excavator_position_init[1] + self.
            excavator_radius * np.sin(np.pi * self.time / 10.0)
        self._p.resetBasePositionAndOrientation(self.excavator_id,
            excavator_position, [0.0, 0.0, 0.0, 1.0])
        return (observation, reward, terminated, truncated, info)

    def get_task_rewards(self, action):
        reward_pick = 0.0
        reward_place = 0.0
        reward_complete = 0.0
        penalty_collision = 0.0
        for i, (debris_id, debris_type) in enumerate(self.debris_ids):
            if not self.debris_picked[i] and len(self._p.getContactPoints
                (bodyA=self.robot.robot_id, bodyB=debris_id)) > 0:
                self.debris_picked[i] = True
                reward_pick += self.debris_pick_reward
            if self.debris_picked[i] and (not self.debris_placed[i]) and
                (len(self._p.getContactPoints(bodyA=debris_id, bodyB=self
                .receptacle_ids[debris_type])) > 0):
                self.debris_placed[i] = True
                reward_place += self.debris_place_reward
        if all(self.debris_placed) and np.linalg.norm(self.robot.links['
            base'].position[:2] - np.asarray(self.robot_position_init
            [:2])) < 1.0:
            reward_complete = self.task_complete_reward
```

```
        if len(self._p.getContactPoints(bodyA=self.robot.robot_id, bodyB=
            self.bulldozer_id)) > 0 or len(self._p.getContactPoints(bodyA
            =self.robot.robot_id, bodyB=self.excavator_id)) > 0:
            penalty_collision = self.collision_penalty
        return {'reward_pick': reward_pick, 'reward_place': reward_place,
            'reward_complete': reward_complete, 'penalty_collision':
            penalty_collision}

    def get_terminated(self, action):
        if self.time >= self.time_limit:
            return True
        if np.dot(np.asarray([0, 0, 1]), np.asarray(self._p.
            getMatrixFromQuaternion(self.robot.links['base'].orientation)
            ).reshape(3, 3)[:, 2]) < 0.5:
            return True
        return False

    def get_success(self):
        return all(self.debris_placed) and np.linalg.norm(self.robot.
            links['base'].position[:2] - np.asarray(self.
            robot_position_init[:2])) < 1.0
```

**Task 23**

```
Search and Rescue in a Multi-Room Environment

Description:
- The environment consists of a large, two-story building with multiple
    rooms connected by doors, hallways, and staircases. The layout is
    fixed but contains challenging features:
  - 8 rooms total (4 on each floor), connected by 2 hallways (one per
      floor)
  - Each room is 5 m x 5 m. Hallways are 20 m long and 2 m wide.
      Staircase is 5 m long.
  - Rooms contain various static and movable obstacles like furniture.
  - Doors are 1 m wide and initially closed. Robot must push them open.
- A target object (a 0.5 m x 0.5 m x 1 m box) representing a person is
    located in one of the rooms on the second floor. Its location is
    fixed for now.
- The robot starts at the entrance of the building on the first floor.
- The robot must systematically explore the building, locate the target
    object, attach a virtual tether to it, and guide it back to the
    entrance while avoiding obstacles.

Success Conditions:
The task is considered complete when the robot brings the target object
    to rest within 2 m of the entrance location.

Time Limit:
The robot has a time limit of 10 minutes to complete the task.

Rewards:
- Provide a small reward for exploring new areas of the environment.
- Provide a moderate reward for locating the target object.
- Provide a large reward for successfully bringing the target object to
    the goal location.
- Provide a small penalty for collisions with obstacles to encourage
    careful navigation.

Termination:
The episode terminates if the robot flips over or if the time limit is
    exceeded.
```

**Task 60**

```
Simplified 1—vs—1 robot soccer.

Description:
— The environment consists of a large flat ground representing a soccer
    field.
— Two goals are placed on opposite ends of the field, each defined by two
    posts 2 meters high and 3 meters apart.
— The robot is placed on one side of the field, facing the opponent's
    goal.
— A basic opponent robot is placed on the other side, initially
    stationary.
— A soccer ball is placed in the center of the field.
— The objective is for the robot to navigate to the ball, take possession
    of it, and kick it into the opponent's goal while defending its own
    goal.
— In the first stage, the opponent robot remains stationary. In the
    second stage, the opponent robot also moves to chase after the ball
    and attempt to kick it into the robot's goal.

Success:
The task is completed successfully if the robot is able to kick the ball
    into the opponent's goal while preventing the opponent from scoring.

Rewards:
— The robot is rewarded for possessing the ball, defined as being within
    1 meter of the ball while the opponent is not.
— The robot is rewarded for bringing the ball closer to the opponent's
    goal.
— The robot is rewarded for kicking the ball with a velocity toward the
    opponent's goal.
— The robot is penalized if the opponent takes possession of the ball or
    if the ball goes out of bounds.
— The robot is greatly rewarded for scoring a goal and penalized if the
    opponent scores.

Termination:
The task is terminated if a goal is scored by either side, if the ball
    goes out of bounds, or if a time limit is reached.
```

**Task 74**

```
Receive pass and kick ball past moving obstacles and goalie into goal.

The robot is placed on a large flat ground designed to look like a soccer
    /football pitch with markings.

A teammate robot is positioned 5 meters to the side of the main robot.
    After a fixed delay at the start of the episode, the teammate robot
    passes a ball towards a point 3 meters in front of the main robot at
    a speed of 3 m/s. The pass target point is fixed, but the main robot'
    s initial position has some randomness, so it will need to adjust to
    align itself with the passed ball.

3 cylindrical obstacles of 0.5 meter radius and 1 meter height are placed
    at random positions 3 to 8 meters in front of the robot's initial
    position. The obstacles are colored to resemble soccer players and
    move randomly back and forth over a 2 meter range at speeds between
    0.5 and 1.5 m/s. The obstacles do not actively chase the ball but
    will block it if it comes near them.

A goal area 5 meters wide and 2 meters deep is placed 10 meters in front
    of the robot's initial position, centered along the front line.
```

```
A goalie obstacle 1 meter wide, 0.5 meters deep, and 1.5 meters tall
    moves side to side across the goal line at a speed of 1.5 m/s,
    tracking the ball when it is within 3 meters. The goalie is colored
    differently than the field obstacles.

The task is for the main robot to receive the teammate's pass and then
    kick the moving ball past the field obstacles and goalie to score a
    goal, without touching any of the obstacles.
```

**Task 84**

```
Busy Restaurant Table Service

Description:
— The environment is a busy restaurant dining area, consisting of a 12 m
    x 12 m room with 4 tables, an entrance door, and a kitchen door.
— Restaurant patrons (simulated as cylinders) continuously enter from the
    entrance, move to a table, stay for a period of time, and then exit,
    making the environment dynamic.
— The robot must perform two types of table service tasks:
  1) Table setting: The robot must collect clean dishes and utensils from
      the kitchen and arrange them properly on the tables in preparation
      for new patrons. Dishes should be stacked neatly.
  2) Table busing: After patrons leave, the robot must clear dirty dishes
      and utensils from the tables, load them into a bin, and return the
      bin to the kitchen. The bin has a maximum capacity that the robot
      must respect.
— The robot must avoid colliding with patrons and restaurant furniture as
    it navigates.
— Dishes will break if dropped. The robot must handle them carefully.

Success Conditions:
— For table setting, success means all necessary dishes and utensils are
    properly placed on empty tables prior to new patrons arriving. Dishes
    should be undamaged and neatly stacked.
— For table busing, success involves clearing all dirty dishes from
    unoccupied tables, loading them into the bin without exceeding
    capacity, and delivering the bin to the kitchen. No dishes should be
    broken.
— The robot should avoid any collisions with patrons or furniture
    throughout the episode.

Time Limit:
The episode lasts for 10 minutes of simulated time. The robot must
    continuously perform both table setting and busing tasks during this
    period as needed.

Rewards:
— Provide a moderate reward for each successful table set, with all
    dishes placed neatly and undamaged.
— Provide a moderate reward for each successful table busing, with no
    broken dishes and the bin delivered to the kitchen.
— Give a small reward for carefully handling dishes without breaking them
    .
— Assign a substantial penalty for any collisions between the robot and
    patrons or furniture.
— Assign a small penalty for dropping and breaking a dish.
— Provide a small reward at each timestep for maintaining a tidy dining
    room, with no dirty dishes left on unoccupied tables.

Termination:
The episode ends if the robot crashes into a patron or furniture, or if
    the time limit is reached. The episode is considered successful if
    the robot consistently performs both table setting and busing
    throughout the full time period with no collisions or broken dishes.
```

**Task 88**

```
Bounce Shot Scoring Challenge

Description:
— The environment consists of a walled field measuring 20 m x 10 m. The
    field contains 3 ramps/mounds (triangular prisms) placed randomly,
    each measuring 2 m long x 1 m wide x 0.5 m high.
— A single goal 3 m wide and 1 m high is placed centered on one end of
    the field.
— A ball (0.5 m diameter) is released from a random point along the
    opposite side of the field from the goal. The ball is given an
    initial velocity of 3—5 m/s at a random angle towards the goal end of
    the field.
— Due to the ramps on the field, the ball will bounce and roll
    unpredictably as it crosses the field.
— The robot begins in the center of the field on the goal side.

Task:
The robot must intercept the bouncing ball and kick it into the goal to
    score a point. The robot should dynamically adjust its approach to
    the ball based on the ball's changing trajectory after each bounce
    off a ramp. The robot needs to time its kick to hit the ball at the
    optimal point in its bounce to redirect it into the goal.

Success Conditions:
The task is completed successfully if the robot causes the ball to fully
    enter the goal area.

Rewards:
— Small reward for decreasing distance to the ball, scaled higher as the
    ball gets closer to the robot's side of the field
— Moderate penalty if the ball's velocity drops to zero or it leaves the
    field without scoring
— Large reward for kicking the ball, scaled by the ball's post—kick
    velocity in the direction of the goal
— Large reward for scoring a goal

Termination:
The episode ends if the ball comes to rest, leaves the field, or enters
    the goal area.
```

**Task 90**

```
Outdoor Retrieval Challenge

Description:
— The environment is a large outdoor field 50 m x 50 m, with the robot
    starting in the center.
— The terrain includes various features:
  — A steep hill with a 30 degree incline
  — A shallow stream, 0.5 m deep and 5 m wide
  — Patches of loose sand and gravel
  — Scattered trees and boulders up to 2 m tall
— A 0.5 m diameter soccer ball is placed randomly between 10 m and 20 m
    from the robot's starting position.
— The robot must locate the ball, pick it up or get it balanced on its
    back, and return it to the starting location.
— The ball is heavy enough that it can't simply be pushed the whole way.
    The robot needs to lift/carry it.
— If the robot drops the ball or it rolls more than 5 m away, it is
    replaced at a new random location.
```

```
Success Conditions:
The task is completed when the robot returns to within 2 m of its
    starting position with the ball balanced on its back or in its
    possession.

Time Limit:
The robot has 10 minutes to complete the retrieval.

Rewards:
— Provide a small reward for exploring the environment and locating the
    ball
— Provide a moderate reward for successfully getting the ball balanced on
     the robot's back or lifted off the ground
— Provide a large reward for returning to the start area with the ball
— Provide a small penalty if the ball is dropped or lost, to encourage
    careful handling

Termination:
The episode ends if the robot flips over and can't right itself, or if
    the time limit is exceeded.
```

**Task 103**

```
Conveyor Belt Item Delivery

Description:
— The environment consists of two 5 m x 5 m platforms connected by a 5 m
    long conveyor belt. The conveyor belt moves from left to right at a
    speed of 0.5 m/s.
— On the right platform, there are three target areas marked by 1 m x 1 m
    colored squares (red, green, blue).
— Also on the right platform, there is a dispenser that releases items
    onto the conveyor belt at random intervals between 2—5 seconds. The
    items are 0.5 m cubes colored either red, green, or blue.
— The robot begins on the left platform.

Task:
The robot must jump onto the conveyor belt, pick up the colored items,
    ride the conveyor to the right platform, and deliver each item to the
     target area matching its color. The robot should try to deliver as
    many items as possible to the correct targets within the time limit.

After delivering an item, the robot must jump back onto the conveyor belt
     and return to the left platform before the next item is released.
    The robot should wait on the left platform for the next item.

Rewards:
— Provide a small reward for picking up an item from the conveyor belt.
— Provide a large reward for delivering an item to the correct target
    area based on color.
— Provide a small penalty for delivering an item to the wrong target area
    .
— Provide a moderate penalty if the robot falls off the conveyor belt or
    platforms.

Success Conditions:
The task is considered complete if the robot successfully delivers at
    least 5 items to their correct target areas within the time limit.

Time Limit:
The robot has 3 minutes to deliver as many items as possible.

Termination:
The episode ends if the robot falls off the platforms or conveyor belt,
    or if the time limit is reached.
```

**Task 122**

```
Kick ball through moving gates to zones marked by moving targets.

The environment consists of a large flat ground. A ball is placed 5
    meters directly in front of the robot. After a delay of 1–2 seconds (
    randomized), the ball begins rolling straight ahead at a constant
    velocity of 1–2 m/s (randomized).

Five vertical rectangular gates, each 3 meters tall and 1 meter wide, are
     placed at randomized positions between 5 to 25 meters in front of
    the ball's initial position, at randomized lateral offsets up to 5
    meters on either side. The gates translate laterally with randomized
    constant velocities between 0.2–1 m/s, reversing direction each time
    they move 5 meters from their initial position. The gates' movement
    is triggered at the start of the episode.

Five target zones are marked on the ground at randomized positions
    between 10 to 30 meters in front of the ball's initial position, at
    randomized lateral offsets up to 10 meters on either side. Each
    target zone is a circle with radius 2 meters. At the center of each
    target zone is a flat cylindrical marker, 2 meters in radius and 0.1
    meters tall. The markers rotate in place with randomized constant
    angular velocities between 0.1–0.5 rad/s, switching between clockwise
     and counterclockwise rotation each time the ball passes through a
    gate. The markers' rotation is triggered at the start of the episode.

The robot is initialized 3 meters behind the ball, facing the direction
    of the ball's initial velocity.

The task is for the robot to kick the ball through each gate in order,
    aiming to get the ball to stop within the corresponding target zone
    after it passes through the gate. The robot should decide when and
    how to kick the ball based on the observed motion of the gates and
    markers.

Passing through a gate awards 2 points if the ball subsequently stops in
    the correct target zone, and 1 point otherwise. The episode ends when
     the ball passes through all gates, or a maximum time limit is
    reached. The robot should aim to maximize its total score.
```

**Task 155**

```
Task: Dynamic Target Pushing Challenge

Description:
— The environment is a 15 m x 15 m open area with a flat surface.
— There are 3 dynamic targets (spheres with a 0.5 m diameter) that move
    along predefined linear paths. Each target moves back and forth along
     a 5 m path at a speed of 0.5 m/s.
— The robot starts at the center of the area.
— Three goal zones (1 m x 1 m squares) are located at the edges of the
    area: one at the north edge, one at the east edge, and one at the
    west edge.

Task:
The robot must push each moving target into its corresponding goal zone.
    The targets can be pushed in any order, but each target must come to
    rest fully within its goal zone to be considered successfully
    delivered.

Initial Positions:
— Robot: Center of the area at (0, 0, 0.5)
```

```
— Target 1: Moving along the path from (−2.5, 5, 0.25) to (2.5, 5, 0.25)
— Target 2: Moving along the path from (5, −2.5, 0.25) to (5, 2.5, 0.25)
— Target 3: Moving along the path from (−5, −2.5, 0.25) to (−5, 2.5,
    0.25)
— Goal Zone 1: Centered at (0, 7, 0.05)
— Goal Zone 2: Centered at (7, 0, 0.05)
— Goal Zone 3: Centered at (−7, 0, 0.05)

Success Conditions:
The task is considered complete when all three targets come to rest fully
    within their respective goal zones.

Time Limit:
The robot has a time limit of 5 minutes to complete the task.

Rewards:
— Provide a small reward for moving closer to a target.
— Provide a moderate reward for reaching a target.
— Provide a large reward for pushing a target towards its goal zone.
— Provide a significant reward for delivering a target to its goal zone.
— Apply a small penalty for collisions with the targets or the boundaries
    of the area.

Termination:
The episode terminates if the robot flips over, if a target is pushed out
    of bounds, or if the time limit is exceeded. The episode should
    terminate with success if all targets are delivered to their goal
    zones.
```

**Task 176**

```
Task: Push two balls into separate goals while avoiding static obstacles.

Description:
— The environment consists of a large flat ground.
— Two balls, one red and one blue, are placed at fixed initial positions
    on the ground. The red ball is positioned at [2.0, −2.0, 0.5] and the
    blue ball is positioned at [2.0, 2.0, 0.5].
— Two goal areas are defined by pairs of posts. The red goal is defined
    by posts placed at [10.0, −2.0, 1.0] and [10.0, −1.0, 1.0]. The blue
    goal is defined by posts placed at [10.0, 1.0, 1.0] and [10.0, 2.0,
    1.0].
— Three cylindrical obstacles of 1 meter radius and 2 meters height are
    placed at fixed positions: [5.0, 0.0, 1.0], [7.0, −1.0, 1.0], and
    [7.0, 1.0, 1.0].
— The robot is initialized at [0.0, 0.0, 0.5], facing the positive x−
    direction.
— The task is for the robot to push the red ball into the red goal and
    the blue ball into the blue goal while avoiding the obstacles.

Success:
The task is completed successfully if the robot pushes both balls into
    their respective goals without colliding with any obstacles.

Rewards:
— The robot is rewarded for decreasing its distance to each ball.
— The robot is rewarded for pushing each ball towards its respective goal
    .
— The robot is penalized for colliding with obstacles.
— The robot is greatly rewarded if a ball passes between the goal posts
    of its respective goal.

Termination:
```

The task is terminated if the robot collides with an obstacle, if a ball
    collides with an obstacle, or if a time limit of 60 seconds is
    reached.

**Task 178**

```
Task: Coordinate with another moving ball to hit a target

Description:
— The environment consists of a large flat ground.
— Two balls are placed 5 meters in front of the robot, spaced 2 meters
    apart laterally. Both balls begin rolling in parallel at 1 m/s after
    a short delay of 1 second.
— A target area defined by a 2x2 meter flat vertical target is placed 10
    meters from the balls' initial position. The target moves laterally
    back and forth at 0.5 m/s, covering 3 meters to each side before
    reversing direction.
— The robot is initialized 3 meters behind the balls, facing the
    direction of the target.
— The task is for the robot to coordinate its movements to kick both
    balls such that they hit the moving target simultaneously.

Success:
The task is completed successfully if both balls hit the front face of
    the moving target simultaneously.

Rewards:
— The robot is rewarded for decreasing the distance to each ball as they
    move.
— The robot is rewarded for matching the velocity of each ball as it
    approaches.
— The robot is rewarded for kicking each ball with a velocity that is
    likely to reach the target.
— The robot is greatly rewarded if both balls hit the front face of the
    moving target simultaneously.

Termination:
The task is terminated if either ball stops moving, if the balls do not
    hit the target simultaneously, or if a time limit of 30 seconds is
    reached.
```

**Task 196**

```
Task: Dodgeball Arena

Description:
— The environment is a 15 m x 15 m walled arena with a flat ground
    surface.
— There are 5 automatic ball launchers positioned around the perimeter of
    the arena. Each launcher can shoot a foam ball (0.2 m diameter) at
    varying speeds (up to 5 m/s) and angles.
— The launchers are programmed to shoot balls at random intervals (
    between 1 to 3 seconds) and in random directions.
— The robot starts in the center of the arena.

Task:
The robot must avoid being hit by the foam balls for as long as possible.
    The robot can move around the arena and jump to dodge the balls.

Success Conditions:
The task is considered successful if the robot can avoid being hit by any
    balls for a duration of 2 minutes.

Rewards:
```

```
— Provide a small reward for each second the robot avoids being hit.
— Provide a large reward for successfully avoiding all balls for the
    entire duration.
— Apply a small penalty for each ball that hits the robot.

Termination:
The episode ends if the robot is hit by a ball, or if the time limit of 2
    minutes is exceeded.
```

**Task 197**

```
Task: Jumping Over Gaps

Description:
— The environment consists of a 10 m x 10 m platform with a series of
    gaps that the robot must jump over.
— The platform is divided into three sections:
  1. The first section (3 m x 10 m) has three gaps, each 0.5 m wide,
      spaced 2 m apart.
  2. The second section (3 m x 10 m) has two gaps, each 1 m wide, spaced
      3 m apart.
  3. The third section (4 m x 10 m) has one gap, 1.5 m wide, located 2 m
      from the end of the platform.

— The robot begins at the start of the platform, facing the positive x—
    axis.

Task:
The robot must navigate through the three sections of the platform,
    jumping over the gaps to reach the end of the platform.

Success Conditions:
The task is considered complete when the robot reaches the end of the
    platform without falling into any gaps.

Rewards:
— Provide a small reward for each gap successfully jumped over.
— Provide a large reward for reaching the end of the platform.
— Apply a small penalty for each failed jump (falling into a gap).

Termination:
The episode ends if the robot falls into a gap, flips over, or reaches
    the end of the platform.
```

# E   SUPPLEMENTARY MATERIALS FOR SHORT RUN WITH LEARNING

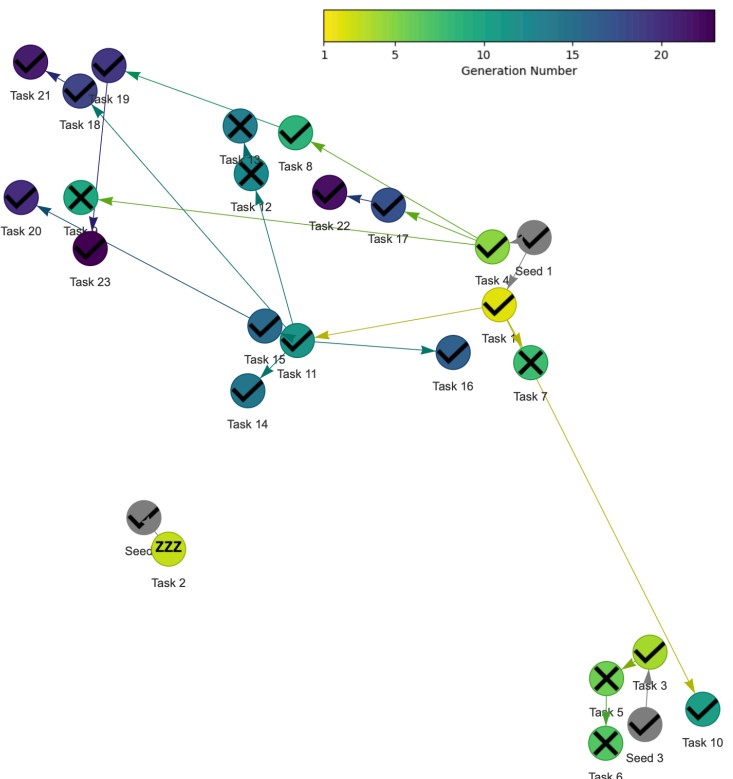

Figure 7: **Short run with Learning task graph.** The node color reflects the generation number of the task. A node with a check mark indicates successful task learning, while a cross mark denotes a task that was attempted but failed to be learned. A ZZZ symbol means that the task was deemed uninteresting and discarded. The node connections illustrate which tasks were conditioned on when asking an FM to generate a similar yet new and interesting task.

**Task 1**

```python
import numpy as np
from oped.envs.r2d2.base import R2D2Env

class Env(R2D2Env):
    """
    Cross a bridge with moving segments to reach a platform.

    Description:
    — A start platform and an end platform (each 3 m in size and 0.5 m in
        thickness) are placed 20 m apart.
    — The two platforms are connected by a bridge (2 m wide) divided into
        5 segments of equal length (3 m each).
    — Each segment moves up and down independently with a sinusoidal
        motion. The amplitude is 1 m and the period is 2 seconds, with
        each segment offset in phase by 0.4 seconds from the previous one
        .
    — The segments are colored red, orange, yellow, green, and blue.
    The robot is initialized on the start platform.
    The task of the robot is to cross the dynamic bridge to reach the end
        platform as fast as possible by timing its jumps between the
        moving segments.
```

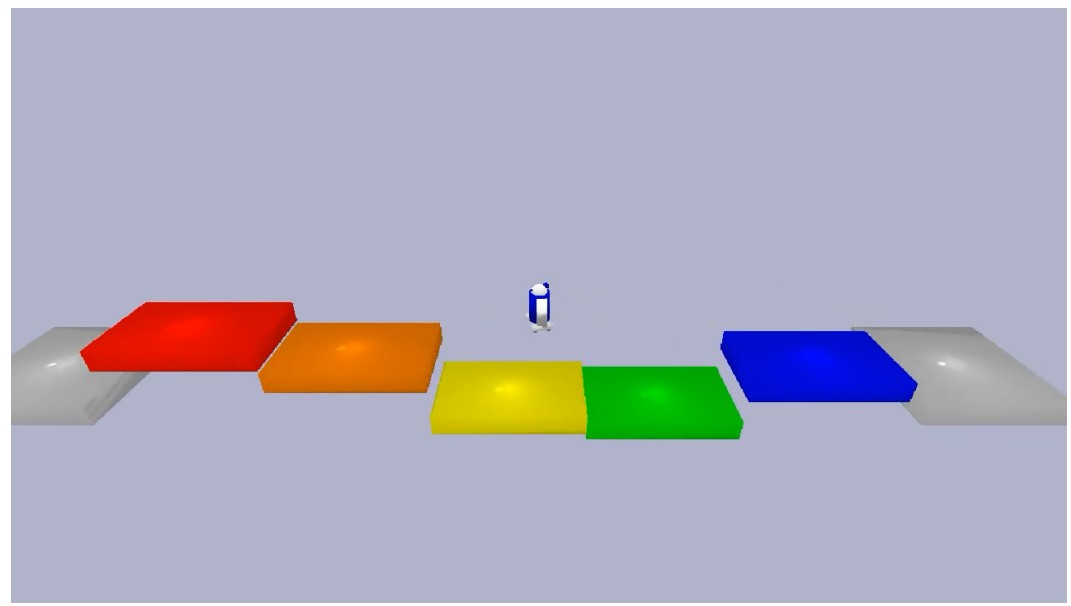

Figure 8: Task 1 of the OMNI-EPIC run presented in Section 5.

```
Success:
The task is successfully completed when the robot reaches the end
    platform.

Rewards:
To help the robot complete the task:
− The robot receives a reward for each time step it remains on the
    bridge or platforms, encouraging steady progress.
− The robot is rewarded based on how much it reduces the distance to
    the end platform, incentivizing swift movement towards the goal.

Termination:
The task terminates immediately if the robot falls off the start
    platform, any segment of the bridge, or the end platform.
"""

def __init__(self):
    super().__init__()
    self.platform_size = [3.0, 3.0, 0.5]
    self.platform_start_position = [0.0, 0.0, 0.0]
    self.platform_end_position = [self.platform_start_position[0] +
        20.0, self.platform_start_position[1], self.
        platform_start_position[2]]
    self.platform_start_id = self.create_box(mass=0.0, half_extents=[
        self.platform_size[0] / 2, self.platform_size[1] / 2, self.
        platform_size[2] / 2], position=self.platform_start_position,
         color=[0.8, 0.8, 0.8, 1.0])
    self.platform_end_id = self.create_box(mass=0.0, half_extents=[
        self.platform_size[0] / 2, self.platform_size[1] / 2, self.
        platform_size[2] / 2], position=self.platform_end_position,
        color=[0.8, 0.8, 0.8, 1.0])
    self.bridge_length = self.platform_end_position[0] − self.
        platform_start_position[0] − self.platform_size[0]
    self.bridge_width = 2.0
    self.num_segments = 5
    self.segment_length = self.bridge_length / self.num_segments
    self.segment_amplitude = 1.0
    self.segment_period = 2.0
```

```python
        self.segment_phase_offset = 0.4
        segment_colors = [[1.0, 0.0, 0.0, 1.0], [1.0, 0.5, 0.0, 1.0],
            [1.0, 1.0, 0.0, 1.0], [0.0, 1.0, 0.0, 1.0], [0.0, 0.0, 1.0,
            1.0]]
        self.segment_ids = []
        for i in range(self.num_segments):
            segment_id = self.create_box(mass=0.0, half_extents=[self.
                segment_length / 2, self.bridge_width / 2, self.
                platform_size[2] / 2], position=[self.
                platform_start_position[0] + self.platform_size[0] / 2 +
                self.segment_length / 2 + i * self.segment_length, self.
                platform_start_position[1], self.platform_start_position
                [2]], color=segment_colors[i])
            self._p.changeDynamics(bodyUniqueId=segment_id, linkIndex=-1,
                lateralFriction=0.8, restitution=0.5)
            self.segment_ids.append(segment_id)

    def create_box(self, mass, half_extents, position, color):
        collision_shape_id = self._p.createCollisionShape(shapeType=self.
            _p.GEOM_BOX, halfExtents=half_extents)
        visual_shape_id = self._p.createVisualShape(shapeType=self._p.
            GEOM_BOX, halfExtents=half_extents, rgbaColor=color)
        return self._p.createMultiBody(baseMass=mass,
            baseCollisionShapeIndex=collision_shape_id,
            baseVisualShapeIndex=visual_shape_id, basePosition=position)

    def get_object_position(self, object_id):
        return np.asarray(self._p.getBasePositionAndOrientation(object_id
            )[0])

    def get_distance_to_object(self, object_id):
        object_position = self.get_object_position(object_id)
        robot_position = self.robot.links['base'].position
        return np.linalg.norm(object_position[:2] - robot_position[:2])

    def reset(self):
        observation = super().reset()
        self.time = 0.0
        self._p.resetBasePositionAndOrientation(self.robot.robot_id, [
            self.platform_start_position[0], self.platform_start_position
            [1], self.platform_start_position[2] + self.platform_size[2]
            / 2 + self.robot.links['base'].position_init[2]], self.robot.
            links['base'].orientation_init)
        return observation

    def step(self, action):
        self.distance_to_platform_end = self.get_distance_to_object(self.
            platform_end_id)
        observation, reward, terminated, truncated, info = super().step(
            action)
        self.time += self.dt
        for i, segment_id in enumerate(self.segment_ids):
            segment_position = self.get_object_position(segment_id)
            new_segment_position = [segment_position[0], segment_position
                [1], self.platform_start_position[2] + self.
                segment_amplitude * np.sin(2 * np.pi * (self.time + i *
                self.segment_phase_offset) / self.segment_period)]
            self._p.resetBasePositionAndOrientation(segment_id,
                new_segment_position, [0.0, 0.0, 0.0, 1.0])
        return (observation, reward, terminated, truncated, info)

    def get_task_rewards(self, action):
        new_distance_to_platform_end = self.get_distance_to_object(self.
            platform_end_id)
```

```python
        on_bridge_or_platforms = 1.0 if self.robot.links['base'].position
            [2] > self.platform_start_position[2] + self.platform_size[2]
            / 2 else -1.0
        reach_platform_end = (self.distance_to_platform_end -
            new_distance_to_platform_end) / self.dt
        return {'on_bridge_or_platforms': on_bridge_or_platforms, '
            reach_platform_end': reach_platform_end}

    def get_terminated(self, action):
        return self.robot.links['base'].position[2] < self.
            platform_start_position[2]

    def get_success(self):
        is_on_platform_end = self.get_distance_to_object(self.
            platform_end_id) < self.platform_size[2] / 2
        return is_on_platform_end
```

**Task 3**

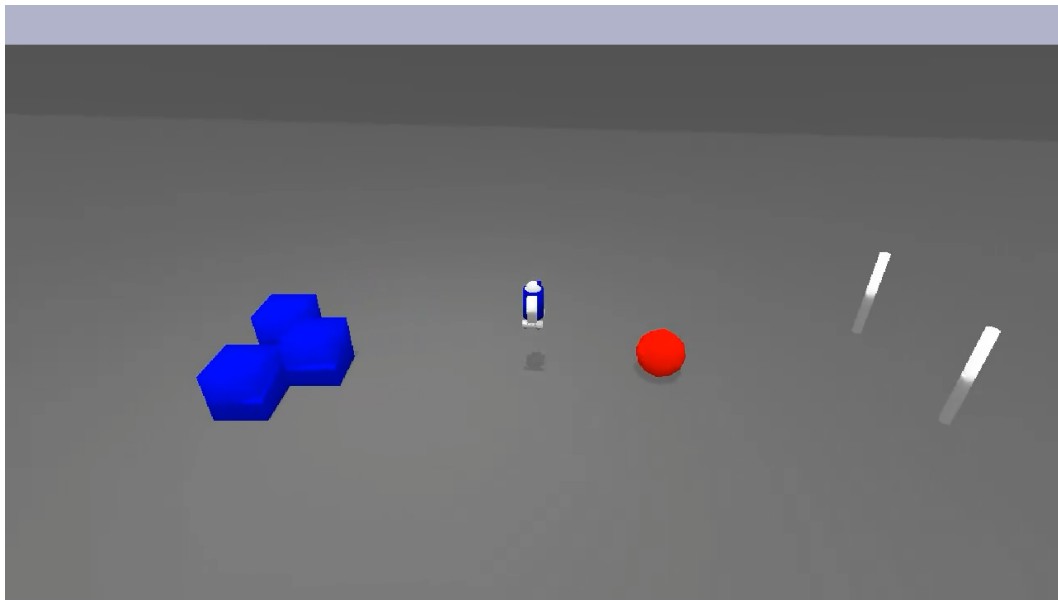

Figure 9: Task 3 of the OMNI-EPIC run presented in Section 5.

```python
import numpy as np
from oped.envs.r2d2.base import R2D2Env

class Env(R2D2Env):
    """
    Kick a ball into a goal with obstacles.

    Description:
    - The environment consists of a large flat ground measuring 1000 x
        1000 x 10 meters.
    - A ball with a radius of 0.5 meters is placed randomly on the ground
        , 5 meters in front of the robot's starting position.
    - The goal is defined by two goal posts, each 2 meters high and
        placed 3 meters apart, forming a goal area. The goal is
        positioned 10 meters in front of the robot's starting position.
    - Three box-shaped obstacles, each measuring 1 x 1 x 1 meters, are
        placed between the robot's starting position and the ball.
        - The first obstacle is placed 2 meters in front and 1 meter to the
            left of the robot.
```

```
            - The second obstacle is placed 3 meters in front of the robot.
            - The third obstacle is placed 2 meters in front and 1 meter to the
                right of the robot.
        - The robot is initialized at a fixed position on the ground, facing
            the goal.
        - The task of the robot is to navigate around the obstacles, reach
            the ball, and kick it into the goal.

    Success:
    The task is successfully completed if the robot kicks the ball so
        that it passes between the two goal posts without colliding with
        any of the obstacles.

    Rewards:
    To help the robot complete the task:
    - The robot is rewarded for survival at each time step.
    - The robot is rewarded for decreasing its distance to the ball while
        avoiding obstacles.
    - The robot is penalized for colliding with obstacles.
    - The robot is rewarded for kicking the ball towards the goal, with
        additional rewards for successfully kicking the ball into the
        goal.

    Termination:
    The task is terminated if the robot collides with any obstacle.
        Otherwise, it does not have a specific termination condition.
    """

    def __init__(self):
        super().__init__()
        self.ground_size = [1000.0, 1000.0, 10.0]
        self.ground_position = [0.0, 0.0, 0.0]
        self.ground_id = self.create_box(mass=0.0, half_extents=[self.
            ground_size[0] / 2, self.ground_size[1] / 2, self.ground_size
            [2] / 2], position=self.ground_position, color=[0.5, 0.5,
            0.5, 1.0])
        self._p.changeDynamics(bodyUniqueId=self.ground_id, linkIndex=-1,
            lateralFriction=0.8, restitution=0.5)
        self.ball_radius = 0.5
        self.ball_id = self.create_sphere(mass=1.0, radius=self.
            ball_radius, position=[0.0, 0.0, 0.0], color=[1.0, 0.0, 0.0,
            1.0])
        self.goal_post_height = 2.0
        self.goal_post_radius = 0.1
        self.goal_width = 3.0
        self.goal_position = [10.0, 0.0, self.ground_position[2] + self.
            ground_size[2] / 2 + self.goal_post_height / 2]
        self.goal_post_left_id = self.create_cylinder(mass=0.0, radius=
            self.goal_post_radius, height=self.goal_post_height, position
            =[self.goal_position[0], self.goal_position[1] - self.
            goal_width / 2, self.goal_position[2]], color=[1.0, 1.0, 1.0,
            1.0])
        self.goal_post_right_id = self.create_cylinder(mass=0.0, radius=
            self.goal_post_radius, height=self.goal_post_height, position
            =[self.goal_position[0], self.goal_position[1] + self.
            goal_width / 2, self.goal_position[2]], color=[1.0, 1.0, 1.0,
            1.0])
        self.obstacle_size = [1.0, 1.0, 1.0]
        self.obstacle_1_id = self.create_box(mass=0.0, half_extents=[self
            .obstacle_size[0] / 2, self.obstacle_size[1] / 2, self.
            obstacle_size[2] / 2], position=[-3.0, -1.0, self.
            ground_position[2] + self.ground_size[2] / 2 + self.
            obstacle_size[2] / 2], color=[0.0, 0.0, 1.0, 1.0])
        self.obstacle_2_id = self.create_box(mass=0.0, half_extents=[self
            .obstacle_size[0] / 2, self.obstacle_size[1] / 2, self.
```

```python
            obstacle_size[2] / 2], position=[-2.0, 0.0, self.
            ground_position[2] + self.ground_size[2] / 2 + self.
            obstacle_size[2] / 2], color=[0.0, 0.0, 1.0, 1.0])
        self.obstacle_3_id = self.create_box(mass=0.0, half_extents=[self
            .obstacle_size[0] / 2, self.obstacle_size[1] / 2, self.
            obstacle_size[2] / 2], position=[-3.0, 1.0, self.
            ground_position[2] + self.ground_size[2] / 2 + self.
            obstacle_size[2] / 2], color=[0.0, 0.0, 1.0, 1.0])
        self.robot_position_init = [0.0, 0.0, self.ground_position[2] +
            self.ground_size[2] / 2 + self.robot.links['base'].
            position_init[2]]
        self.robot_orientation_init = self._p.getQuaternionFromEuler
            ([0.0, 0.0, 0.0])

    def create_box(self, mass, half_extents, position, color):
        collision_shape_id = self._p.createCollisionShape(shapeType=self.
            _p.GEOM_BOX, halfExtents=half_extents)
        visual_shape_id = self._p.createVisualShape(shapeType=self._p.
            GEOM_BOX, halfExtents=half_extents, rgbaColor=color)
        return self._p.createMultiBody(baseMass=mass,
            baseCollisionShapeIndex=collision_shape_id,
            baseVisualShapeIndex=visual_shape_id, basePosition=position)

    def create_sphere(self, mass, radius, position, color):
        collision_shape_id = self._p.createCollisionShape(shapeType=self.
            _p.GEOM_SPHERE, radius=radius)
        visual_shape_id = self._p.createVisualShape(shapeType=self._p.
            GEOM_SPHERE, radius=radius, rgbaColor=color)
        return self._p.createMultiBody(baseMass=mass,
            baseCollisionShapeIndex=collision_shape_id,
            baseVisualShapeIndex=visual_shape_id, basePosition=position)

    def create_cylinder(self, mass, radius, height, position, color):
        collision_shape_id = self._p.createCollisionShape(shapeType=self.
            _p.GEOM_CYLINDER, radius=radius, height=height)
        visual_shape_id = self._p.createVisualShape(shapeType=self._p.
            GEOM_CYLINDER, radius=radius, length=height, rgbaColor=color)
        return self._p.createMultiBody(baseMass=mass,
            baseCollisionShapeIndex=collision_shape_id,
            baseVisualShapeIndex=visual_shape_id, basePosition=position)

    def get_object_position(self, object_id):
        return np.asarray(self._p.getBasePositionAndOrientation(object_id
            )[0])

    def get_distance_to_object(self, object_id):
        object_position = self.get_object_position(object_id)
        robot_position = self.robot.links['base'].position
        return np.linalg.norm(object_position[:2] - robot_position[:2])

    def reset(self):
        observation = super().reset()
        self._p.resetBasePositionAndOrientation(self.robot.robot_id, self
            .robot_position_init, self.robot_orientation_init)
        ball_y_init = np.random.uniform(-2.0, 2.0)
        self._p.resetBasePositionAndOrientation(self.ball_id, [5.0,
            ball_y_init, self.ground_position[2] + self.ground_size[2] /
            2 + self.ball_radius], [0.0, 0.0, 0.0, 1.0])
        return observation

    def step(self, action):
        self.distance_to_ball = self.get_distance_to_object(self.ball_id)
        self.ball_position = self.get_object_position(self.ball_id)
        observation, reward, terminated, truncated, info = super().step(
            action)
```

```python
        return (observation, reward, terminated, truncated, info)

    def get_task_rewards(self, action):
        new_distance_to_ball = self.get_distance_to_object(self.ball_id)
        new_ball_position = self.get_object_position(self.ball_id)
        survival = 1.0
        reach_ball = (self.distance_to_ball - new_distance_to_ball) /
            self.dt
        collision_obstacle_1 = len(self._p.getContactPoints(bodyA=self.
            robot.robot_id, bodyB=self.obstacle_1_id)) > 0
        collision_obstacle_2 = len(self._p.getContactPoints(bodyA=self.
            robot.robot_id, bodyB=self.obstacle_2_id)) > 0
        collision_obstacle_3 = len(self._p.getContactPoints(bodyA=self.
            robot.robot_id, bodyB=self.obstacle_3_id)) > 0
        collision_penalty = -10.0 if collision_obstacle_1 or
            collision_obstacle_2 or collision_obstacle_3 else 0.0
        kick_ball = (new_ball_position[0] - self.ball_position[0]) / self
            .dt
        ball_in_goal = 0.0
        if self.goal_position[1] - self.goal_width / 2 <
            new_ball_position[1] < self.goal_position[1] + self.
            goal_width / 2 and new_ball_position[0] > self.goal_position
            [0]:
            ball_in_goal = 10.0
        return {'survival': survival, 'reach_ball': reach_ball, '
            collision_penalty': collision_penalty, 'kick_ball': kick_ball
            , 'ball_in_goal': ball_in_goal}

    def get_terminated(self, action):
        collision_obstacle_1 = len(self._p.getContactPoints(bodyA=self.
            robot.robot_id, bodyB=self.obstacle_1_id)) > 0
        collision_obstacle_2 = len(self._p.getContactPoints(bodyA=self.
            robot.robot_id, bodyB=self.obstacle_2_id)) > 0
        collision_obstacle_3 = len(self._p.getContactPoints(bodyA=self.
            robot.robot_id, bodyB=self.obstacle_3_id)) > 0
        return collision_obstacle_1 or collision_obstacle_2 or
            collision_obstacle_3

    def get_success(self):
        ball_position = self.get_object_position(self.ball_id)
        return self.goal_position[1] - self.goal_width / 2 <
            ball_position[1] < self.goal_position[1] + self.goal_width /
            2 and ball_position[0] > self.goal_position[0]
```

**Task 4**

```python
import numpy as np
from oped.envs.r2d2.base import R2D2Env

class Env(R2D2Env):
    """
    Cross a rainbow bridge with moving segments and gaps to reach a
        platform.

    Description:
    - A start platform and an end platform (each 3 m in size and 0.5 m in
        thickness) are placed 25 m apart.
    - The two platforms are connected by a bridge (2 m wide) divided into
        6 segments of equal length (3 m each).
    - Each segment moves up and down independently with a sinusoidal
        motion. The amplitude is 1 m and the period is 2 seconds, with
        each segment offset in phase by 1/3 seconds from the previous one
        .
    - The segments are colored red, orange, yellow, green, blue, and
        purple, like a rainbow.
```

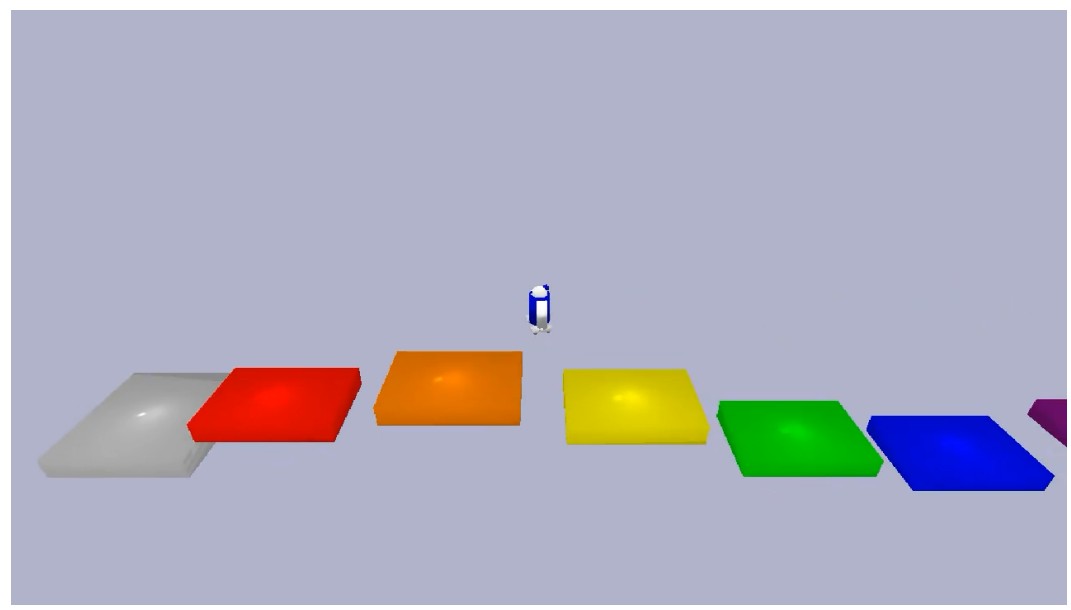

Figure 10: Task 4 of the OMNI-EPIC run presented in Section 5.

```
— The segments have 1 m gaps between them.
The robot is initialized on the start platform.
The task of the robot is to cross the dynamic bridge to reach the end
    platform as fast as possible by timing its jumps between the
    moving segments and over the gaps.

Success:
The task is successfully completed when the robot reaches the end
    platform.

Rewards:
To help the robot complete the task:
— The robot receives a reward for each time step it remains on the
    bridge or platforms, encouraging steady progress.
— The robot is rewarded based on how much it reduces the distance to
    the end platform, incentivizing swift movement towards the goal.

Termination:
The task terminates immediately if the robot falls off the start
    platform, any segment of the bridge, or the end platform.
"""

def __init__(self):
    super().__init__()
    self.platform_size = [3.0, 3.0, 0.5]
    self.platform_start_position = [0.0, 0.0, 0.0]
    self.platform_end_position = [self.platform_start_position[0] +
        25.0, self.platform_start_position[1], self.
        platform_start_position[2]]
    self.platform_start_id = self.create_box(mass=0.0, half_extents=[
        self.platform_size[0] / 2, self.platform_size[1] / 2, self.
        platform_size[2] / 2], position=self.platform_start_position,
         color=[0.8, 0.8, 0.8, 1.0])
    self.platform_end_id = self.create_box(mass=0.0, half_extents=[
        self.platform_size[0] / 2, self.platform_size[1] / 2, self.
        platform_size[2] / 2], position=self.platform_end_position,
        color=[0.8, 0.8, 0.8, 1.0])
```

```python
        self.bridge_length = self.platform_end_position[0] - self.
            platform_start_position[0] - self.platform_size[0]
        self.bridge_width = 2.0
        self.num_segments = 6
        self.segment_length = 3.0
        self.gap_length = 1.0
        self.segment_amplitude = 1.0
        self.segment_period = 2.0
        self.segment_phase_offset = 1.0 / 3.0
        segment_colors = [[1.0, 0.0, 0.0, 1.0], [1.0, 0.5, 0.0, 1.0],
            [1.0, 1.0, 0.0, 1.0], [0.0, 1.0, 0.0, 1.0], [0.0, 0.0, 1.0,
            1.0], [0.5, 0.0, 0.5, 1.0]]
        self.segment_ids = []
        for i in range(self.num_segments):
            segment_id = self.create_box(mass=0.0, half_extents=[self.
                segment_length / 2, self.bridge_width / 2, self.
                platform_size[2] / 2], position=[self.
                platform_start_position[0] + self.platform_size[0] / 2 +
                self.segment_length / 2 + i * (self.segment_length + self
                .gap_length), self.platform_start_position[1], self.
                platform_start_position[2]], color=segment_colors[i])
            self._p.changeDynamics(bodyUniqueId=segment_id, linkIndex=-1,
                lateralFriction=0.8, restitution=0.5)
            self.segment_ids.append(segment_id)

    def create_box(self, mass, half_extents, position, color):
        collision_shape_id = self._p.createCollisionShape(shapeType=self.
            _p.GEOM_BOX, halfExtents=half_extents)
        visual_shape_id = self._p.createVisualShape(shapeType=self._p.
            GEOM_BOX, halfExtents=half_extents, rgbaColor=color)
        return self._p.createMultiBody(baseMass=mass,
            baseCollisionShapeIndex=collision_shape_id,
            baseVisualShapeIndex=visual_shape_id, basePosition=position)

    def get_object_position(self, object_id):
        return np.asarray(self._p.getBasePositionAndOrientation(object_id
            )[0])

    def get_distance_to_object(self, object_id):
        object_position = self.get_object_position(object_id)
        robot_position = self.robot.links['base'].position
        return np.linalg.norm(object_position[:2] - robot_position[:2])

    def reset(self):
        observation = super().reset()
        self.time = 0.0
        self._p.resetBasePositionAndOrientation(self.robot.robot_id, [
            self.platform_start_position[0], self.platform_start_position
            [1], self.platform_start_position[2] + self.platform_size[2]
            / 2 + self.robot.links['base'].position_init[2]], self.robot.
            links['base'].orientation_init)
        return observation

    def step(self, action):
        self.distance_to_platform_end = self.get_distance_to_object(self.
            platform_end_id)
        observation, reward, terminated, truncated, info = super().step(
            action)
        self.time += self.dt
        for i, segment_id in enumerate(self.segment_ids):
            segment_position = self.get_object_position(segment_id)
            new_segment_position = [segment_position[0], segment_position
                [1], self.platform_start_position[2] + self.
                segment_amplitude * np.sin(2 * np.pi * (self.time + i *
                self.segment_phase_offset) / self.segment_period)]
```

```
            self._p.resetBasePositionAndOrientation(segment_id,
                new_segment_position, [0.0, 0.0, 0.0, 1.0])
        return (observation, reward, terminated, truncated, info)

    def get_task_rewards(self, action):
        new_distance_to_platform_end = self.get_distance_to_object(self.
            platform_end_id)
        on_bridge_or_platforms = 1.0 if self.robot.links['base'].position
            [2] > self.platform_start_position[2] + self.platform_size[2]
            / 2 else -1.0
        reach_platform_end = (self.distance_to_platform_end -
            new_distance_to_platform_end) / self.dt
        return {'on_bridge_or_platforms': on_bridge_or_platforms, '
            reach_platform_end': reach_platform_end}

    def get_terminated(self, action):
        return self.robot.links['base'].position[2] < self.
            platform_start_position[2]

    def get_success(self):
        is_on_platform_end = self.get_distance_to_object(self.
            platform_end_id) < self.platform_size[2] / 2
        return is_on_platform_end
```

**Task 5**

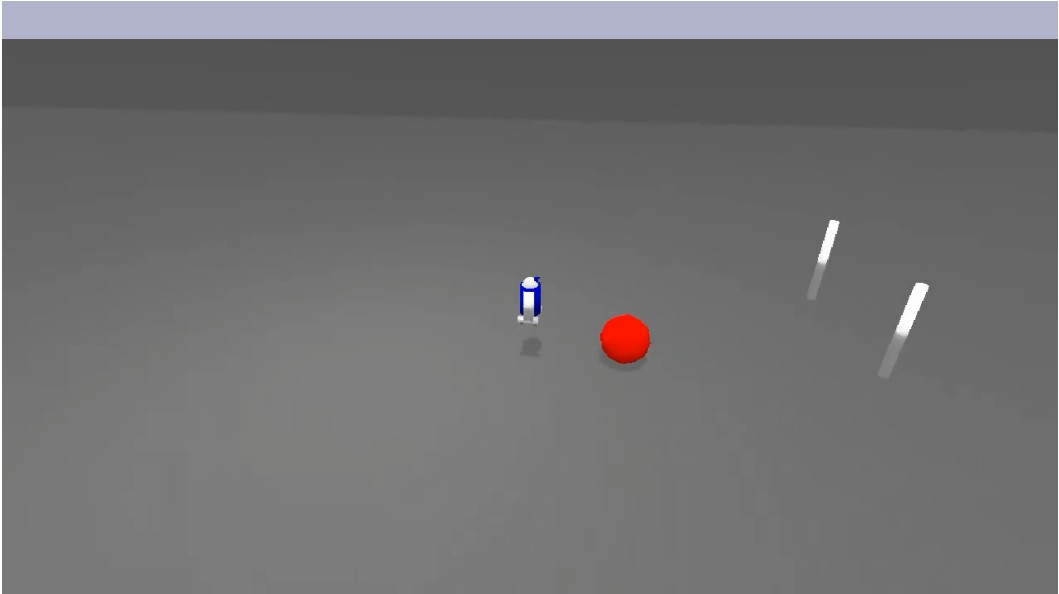

Figure 11: Task 5 of the OMNI-EPIC run presented in Section 5.

```
import numpy as np
from oped.envs.r2d2.base import R2D2Env

class Env(R2D2Env):
    """
    Kick a ball into a moving goal.

    Description:
    - The environment consists of a large flat ground measuring 1000 x
        1000 x 10 meters.
    - A ball with a radius of 0.5 meters is placed on the ground, 5
        meters directly in front of the robot's starting position.
```

```
    - The goal is defined by two goal posts, each 2 meters high and
        placed 3 meters apart, forming a goal area.
    - The goal moves side-to-side (along the y-axis) in a sinusoidal
        motion with an amplitude of 2 meters and a period of 10 seconds.
        The center of its motion is 10 meters in front of the robot's
        starting position.
    - The robot is initialized at a fixed position on the ground, facing
        the goal.
    - The task of the robot is to move towards the ball, time its kick
        correctly, and kick the ball into the moving goal.

    Success:
    The task is successfully completed if the robot kicks the ball so
        that it passes between the two goal posts.

    Rewards:
    To help the robot complete the task:
    - The robot is rewarded for survival at each time step.
    - The robot is rewarded for decreasing its distance to the ball.
    - The robot is rewarded for kicking the ball towards the goal, with
        additional rewards for successfully kicking the ball into the
        goal.

    Termination:
    The task does not have a specific termination condition. The episode
        ends after a fixed amount of time.
    """

    def __init__(self):
        super().__init__()
        self.ground_size = [1000.0, 1000.0, 10.0]
        self.ground_position = [0.0, 0.0, 0.0]
        self.ground_id = self.create_box(mass=0.0, half_extents=[self.
            ground_size[0] / 2, self.ground_size[1] / 2, self.ground_size
            [2] / 2], position=self.ground_position, color=[0.5, 0.5,
            0.5, 1.0])
        self._p.changeDynamics(bodyUniqueId=self.ground_id, linkIndex=-1,
            lateralFriction=0.8, restitution=0.5)
        self.ball_radius = 0.5
        self.ball_id = self.create_sphere(mass=1.0, radius=self.
            ball_radius, position=[0.0, 0.0, 0.0], color=[1.0, 0.0, 0.0,
            1.0])
        self.goal_post_height = 2.0
        self.goal_post_radius = 0.1
        self.goal_width = 3.0
        self.goal_position_init = [10.0, 0.0, self.ground_position[2] +
            self.ground_size[2] / 2 + self.goal_post_height / 2]
        self.goal_post_left_id = self.create_cylinder(mass=0.0, radius=
            self.goal_post_radius, height=self.goal_post_height, position
            =[self.goal_position_init[0], self.goal_position_init[1] -
            self.goal_width / 2, self.goal_position_init[2]], color=[1.0,
             1.0, 1.0, 1.0])
        self.goal_post_right_id = self.create_cylinder(mass=0.0, radius=
            self.goal_post_radius, height=self.goal_post_height, position
            =[self.goal_position_init[0], self.goal_position_init[1] +
            self.goal_width / 2, self.goal_position_init[2]], color=[1.0,
             1.0, 1.0, 1.0])
        self.goal_amplitude = 2.0
        self.goal_period = 10.0
        self.robot_position_init = [self.ground_position[0], self.
            ground_position[1], self.ground_position[2] + self.
            ground_size[2] / 2 + self.robot.links['base'].position_init
            [2]]

    def create_box(self, mass, half_extents, position, color):
```

```python
        collision_shape_id = self._p.createCollisionShape(shapeType=self.
            _p.GEOM_BOX, halfExtents=half_extents)
        visual_shape_id = self._p.createVisualShape(shapeType=self._p.
            GEOM_BOX, halfExtents=half_extents, rgbaColor=color)
        return self._p.createMultiBody(baseMass=mass,
            baseCollisionShapeIndex=collision_shape_id,
            baseVisualShapeIndex=visual_shape_id, basePosition=position)

    def create_sphere(self, mass, radius, position, color):
        collision_shape_id = self._p.createCollisionShape(shapeType=self.
            _p.GEOM_SPHERE, radius=radius)
        visual_shape_id = self._p.createVisualShape(shapeType=self._p.
            GEOM_SPHERE, radius=radius, rgbaColor=color)
        return self._p.createMultiBody(baseMass=mass,
            baseCollisionShapeIndex=collision_shape_id,
            baseVisualShapeIndex=visual_shape_id, basePosition=position)

    def create_cylinder(self, mass, radius, height, position, color):
        collision_shape_id = self._p.createCollisionShape(shapeType=self.
            _p.GEOM_CYLINDER, radius=radius, height=height)
        visual_shape_id = self._p.createVisualShape(shapeType=self._p.
            GEOM_CYLINDER, radius=radius, length=height, rgbaColor=color)
        return self._p.createMultiBody(baseMass=mass,
            baseCollisionShapeIndex=collision_shape_id,
            baseVisualShapeIndex=visual_shape_id, basePosition=position)

    def get_object_position(self, object_id):
        return np.asarray(self._p.getBasePositionAndOrientation(object_id
            )[0])

    def get_distance_to_object(self, object_id):
        object_position = self.get_object_position(object_id)
        robot_position = self.robot.links['base'].position
        return np.linalg.norm(object_position[:2] − robot_position[:2])

    def reset(self):
        observation = super().reset()
        self.time = 0.0
        self._p.resetBasePositionAndOrientation(self.robot.robot_id, self
            .robot_position_init, self.robot.links['base'].
            orientation_init)
        self._p.resetBasePositionAndOrientation(self.ball_id, [self.
            robot_position_init[0] + 5.0, self.robot_position_init[1],
            self.ground_position[2] + self.ground_size[2] / 2 + self.
            ball_radius], [0.0, 0.0, 0.0, 1.0])
        return observation

    def step(self, action):
        self.distance_to_ball = self.get_distance_to_object(self.ball_id)
        self.ball_position = self.get_object_position(self.ball_id)
        observation, reward, terminated, truncated, info = super().step(
            action)
        self.time += self.dt
        goal_y = self.goal_amplitude * np.sin(2 * np.pi * self.time /
            self.goal_period)
        self._p.resetBasePositionAndOrientation(self.goal_post_left_id, [
            self.goal_position_init[0], self.goal_position_init[1] − self
            .goal_width / 2 + goal_y, self.goal_position_init[2]], [0.0,
            0.0, 0.0, 1.0])
        self._p.resetBasePositionAndOrientation(self.goal_post_right_id,
            [self.goal_position_init[0], self.goal_position_init[1] +
            self.goal_width / 2 + goal_y, self.goal_position_init[2]],
            [0.0, 0.0, 0.0, 1.0])
        return (observation, reward, terminated, truncated, info)
```

```python
    def get_task_rewards(self, action):
        new_distance_to_ball = self.get_distance_to_object(self.ball_id)
        new_ball_position = self.get_object_position(self.ball_id)
        survival = 1.0
        reach_ball = (self.distance_to_ball - new_distance_to_ball) /
            self.dt
        kick_ball = (new_ball_position[0] - self.ball_position[0]) / self
            .dt
        goal_y = self.goal_amplitude * np.sin(2 * np.pi * self.time /
            self.goal_period)
        ball_in_goal = 0.0
        if self.goal_position_init[0] - self.goal_post_radius <
            new_ball_position[0] < self.goal_position_init[0] + self.
            goal_post_radius and self.goal_position_init[1] - self.
            goal_width / 2 + goal_y < new_ball_position[1] < self.
            goal_position_init[1] + self.goal_width / 2 + goal_y:
            ball_in_goal = 10.0
        return {'survival': survival, 'reach_ball': reach_ball, '
            kick_ball': kick_ball, 'ball_in_goal': ball_in_goal}

    def get_terminated(self, action):
        return False

    def get_success(self):
        ball_position = self.get_object_position(self.ball_id)
        goal_y = self.goal_amplitude * np.sin(2 * np.pi * self.time /
            self.goal_period)
        return self.goal_position_init[0] - self.goal_post_radius <
            ball_position[0] < self.goal_position_init[0] + self.
            goal_post_radius and self.goal_position_init[1] - self.
            goal_width / 2 + goal_y < ball_position[1] < self.
            goal_position_init[1] + self.goal_width / 2 + goal_y
```

**Task 6**

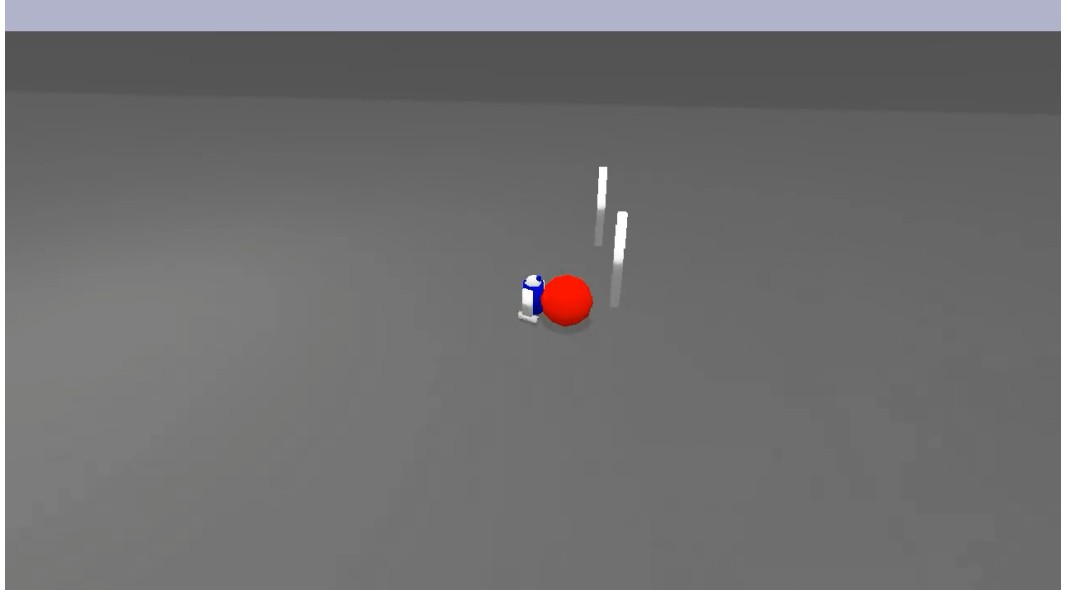

Figure 12: Task 6 of the OMNI-EPIC run presented in Section 5.

```python
import numpy as np
from oped.envs.r2d2.base import R2D2Env
```

```python
class Env(R2D2Env):
    """
    Kick a ball into a moving goal.

    Description:
    - The environment consists of a large flat ground measuring 1000 x
        1000 x 10 meters.
    - A ball with a radius of 0.5 meters is placed on the ground, 5
        meters directly in front of the robot's starting position.
    - The goal is defined by two goal posts, each 2 meters high and
        placed 3 meters apart, forming a goal area.
    - The goal moves side-to-side (along the y-axis) in a sinusoidal
        motion with an amplitude of 2 meters and a period of 10 seconds.
        The center of its motion is 10 meters in front of the robot's
        starting position.
    - The robot is initialized at a fixed position on the ground, facing
        the goal.
    - The task of the robot is to move towards the ball, time its kick
        correctly, and kick the ball into the moving goal.

    Success:
    The task is successfully completed if the robot kicks the ball so
        that it passes between the two goal posts.

    Rewards:
    To help the robot complete the task:
    - The robot is rewarded for survival at each time step.
    - The robot is rewarded for decreasing its distance to the ball.
    - The robot is rewarded for kicking the ball towards the goal, with
        additional rewards for successfully kicking the ball into the
        goal.

    Termination:
    The task terminates if the ball goes out of bounds or if the agent
        fails to kick the ball within a certain time frame.
    """

    def __init__(self):
        super().__init__()
        self.ground_size = [1000.0, 1000.0, 10.0]
        self.ground_position = [0.0, 0.0, 0.0]
        self.ground_id = self.create_box(mass=0.0, half_extents=[self.
            ground_size[0] / 2, self.ground_size[1] / 2, self.ground_size
            [2] / 2], position=self.ground_position, color=[0.5, 0.5,
            0.5, 1.0])
        self._p.changeDynamics(bodyUniqueId=self.ground_id, linkIndex=-1,
            lateralFriction=0.8, restitution=0.5)
        self.ball_radius = 0.5
        self.ball_id = self.create_sphere(mass=1.0, radius=self.
            ball_radius, position=[0.0, 0.0, 0.0], color=[1.0, 0.0, 0.0,
            1.0])
        self.goal_post_height = 2.0
        self.goal_post_radius = 0.1
        self.goal_width = 3.0
        self.goal_position_init = [10.0, 0.0, self.ground_position[2] +
            self.ground_size[2] / 2 + self.goal_post_height / 2]
        self.goal_post_left_id = self.create_cylinder(mass=0.0, radius=
            self.goal_post_radius, height=self.goal_post_height, position
            =[self.goal_position_init[0], self.goal_position_init[1] -
            self.goal_width / 2, self.goal_position_init[2]], color=[1.0,
            1.0, 1.0, 1.0])
        self.goal_post_right_id = self.create_cylinder(mass=0.0, radius=
            self.goal_post_radius, height=self.goal_post_height, position
            =[self.goal_position_init[0], self.goal_position_init[1] +
```

```python
            self.goal_width / 2, self.goal_position_init[2]], color=[1.0,
                1.0, 1.0, 1.0])
        self.goal_amplitude = 2.0
        self.goal_period = 10.0
        self.robot_position_init = [self.ground_position[0], self.
            ground_position[1], self.ground_position[2] + self.
            ground_size[2] / 2 + self.robot.links['base'].position_init
            [2]]

    def create_box(self, mass, half_extents, position, color):
        collision_shape_id = self._p.createCollisionShape(shapeType=self.
            _p.GEOM_BOX, halfExtents=half_extents)
        visual_shape_id = self._p.createVisualShape(shapeType=self._p.
            GEOM_BOX, halfExtents=half_extents, rgbaColor=color)
        return self._p.createMultiBody(baseMass=mass,
            baseCollisionShapeIndex=collision_shape_id,
            baseVisualShapeIndex=visual_shape_id, basePosition=position)

    def create_sphere(self, mass, radius, position, color):
        collision_shape_id = self._p.createCollisionShape(shapeType=self.
            _p.GEOM_SPHERE, radius=radius)
        visual_shape_id = self._p.createVisualShape(shapeType=self._p.
            GEOM_SPHERE, radius=radius, rgbaColor=color)
        return self._p.createMultiBody(baseMass=mass,
            baseCollisionShapeIndex=collision_shape_id,
            baseVisualShapeIndex=visual_shape_id, basePosition=position)

    def create_cylinder(self, mass, radius, height, position, color):
        collision_shape_id = self._p.createCollisionShape(shapeType=self.
            _p.GEOM_CYLINDER, radius=radius, height=height)
        visual_shape_id = self._p.createVisualShape(shapeType=self._p.
            GEOM_CYLINDER, radius=radius, length=height, rgbaColor=color)
        return self._p.createMultiBody(baseMass=mass,
            baseCollisionShapeIndex=collision_shape_id,
            baseVisualShapeIndex=visual_shape_id, basePosition=position)

    def get_object_position(self, object_id):
        return np.asarray(self._p.getBasePositionAndOrientation(object_id
            )[0])

    def get_distance_to_object(self, object_id):
        object_position = self.get_object_position(object_id)
        robot_position = self.robot.links['base'].position
        return np.linalg.norm(object_position[:2] − robot_position[:2])

    def reset(self):
        observation = super().reset()
        self.time = 0.0
        self._p.resetBasePositionAndOrientation(self.robot.robot_id, self
            .robot_position_init, self.robot.links['base'].
            orientation_init)
        self._p.resetBasePositionAndOrientation(self.ball_id, [self.
            robot_position_init[0] + 5.0, self.robot_position_init[1],
            self.ground_position[2] + self.ground_size[2] / 2 + self.
            ball_radius], [0.0, 0.0, 0.0, 1.0])
        return observation

    def step(self, action):
        self.distance_to_ball = self.get_distance_to_object(self.ball_id)
        self.ball_position = self.get_object_position(self.ball_id)
        observation, reward, terminated, truncated, info = super().step(
            action)
        self.time += self.dt
        goal_y = self.goal_amplitude ∗ np.sin(2 ∗ np.pi ∗ self.time /
            self.goal_period)
```

```
            self._p.resetBasePositionAndOrientation(self.goal_post_left_id, [
                self.goal_position_init[0], self.goal_position_init[1] - self
                .goal_width / 2 + goal_y, self.goal_position_init[2]], [0.0,
                0.0, 0.0, 1.0])
            self._p.resetBasePositionAndOrientation(self.goal_post_right_id,
                [self.goal_position_init[0], self.goal_position_init[1] +
                self.goal_width / 2 + goal_y, self.goal_position_init[2]],
                [0.0, 0.0, 0.0, 1.0])
            return (observation, reward, terminated, truncated, info)

    def get_task_rewards(self, action):
        new_distance_to_ball = self.get_distance_to_object(self.ball_id)
        new_ball_position = self.get_object_position(self.ball_id)
        survival = 1.0
        reach_ball = (self.distance_to_ball - new_distance_to_ball) /
            self.dt
        kick_ball = (new_ball_position[0] - self.ball_position[0]) / self
            .dt
        goal_y = self.goal_amplitude * np.sin(2 * np.pi * self.time /
            self.goal_period)
        ball_in_goal = 0.0
        if self.goal_position_init[0] - self.goal_post_radius <
            new_ball_position[0] < self.goal_position_init[0] + self.
            goal_post_radius and self.goal_position_init[1] - self.
            goal_width / 2 + goal_y < new_ball_position[1] < self.
            goal_position_init[1] + self.goal_width / 2 + goal_y:
            ball_in_goal = 10.0
        return {'survival': survival, 'reach_ball': reach_ball, '
            kick_ball': kick_ball, 'ball_in_goal': ball_in_goal}

    def get_terminated(self, action):
        ball_position = self.get_object_position(self.ball_id)
        out_of_bounds = ball_position[0] < 0 or ball_position[0] > self.
            ground_size[0] or ball_position[1] < 0 or (ball_position[1] >
             self.ground_size[1])
        return out_of_bounds

    def get_success(self):
        ball_position = self.get_object_position(self.ball_id)
        goal_y = self.goal_amplitude * np.sin(2 * np.pi * self.time /
            self.goal_period)
        return self.goal_position_init[0] - self.goal_post_radius <
            ball_position[0] < self.goal_position_init[0] + self.
            goal_post_radius and self.goal_position_init[1] - self.
            goal_width / 2 + goal_y < ball_position[1] < self.
            goal_position_init[1] + self.goal_width / 2 + goal_y
```

**Task 7**

```
import numpy as np
from oped.envs.r2d2.base import R2D2Env

class Env(R2D2Env):
    """
    Cross a bridge with moving segments and kick a ball into a goal.

    Description:
    - A start platform and an end platform (each 3 m in size and 0.5 m in
        thickness) are placed 25 m apart.
    - The two platforms are connected by a bridge (2 m wide) divided into
        6 segments of equal length (3 m each).
    - Each segment moves up and down independently with a sinusoidal
        motion. The amplitude is 1 m and the period is 2 seconds, with
        each segment offset in phase by 1/3 seconds from the previous one
        .
```

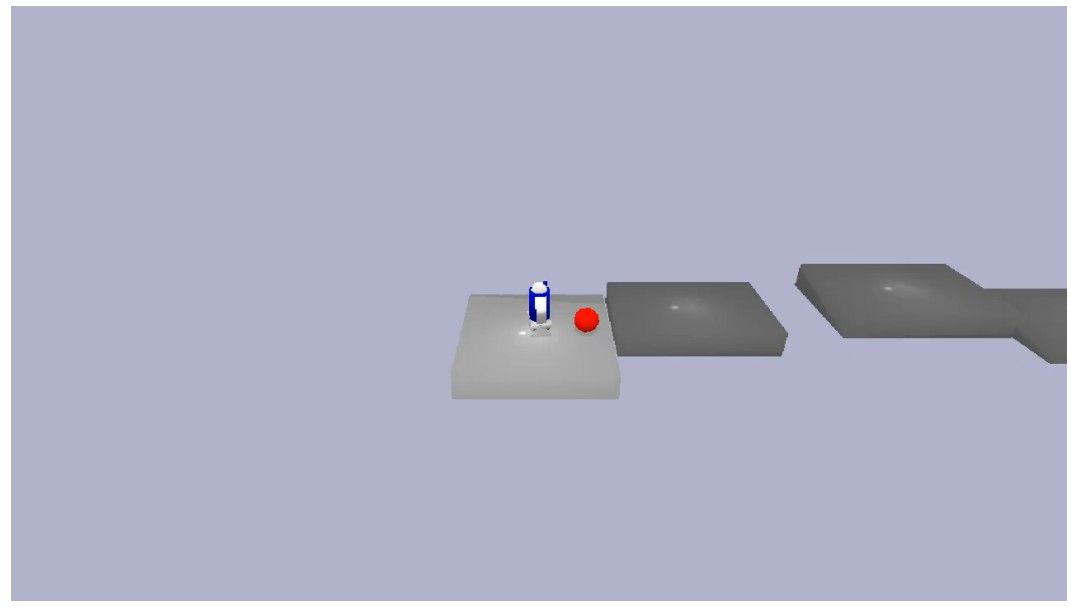

Figure 13: Task 7 of the OMNI-EPIC run presented in Section 5.

```
    — The segments have 1 m gaps between them.
    — A spherical ball (0.5 m diameter) is placed on the start platform,
        1 m in front of the robot's initial position.
    — A goal area (3 m wide and indicated by two posts) is placed on the
        end platform.
The robot is initialized on the start platform, facing the ball and
        bridge.
The task is for the robot to kick or push the ball across the dynamic
        bridge segments, navigating the gaps, to get the ball into the
        goal area on the other side. The robot must also successfully
        cross the bridge and reach the end platform.

Success:
The task is successfully completed when the ball reaches the goal
        area on the end platform and the robot also reaches the end
        platform intact.

Rewards:
    — Reward for decreasing distance between robot and ball, encouraging
        the robot to approach and interact with the ball.
    — Reward for increasing the forward progress of the ball across the
        bridge.
    — Large reward for getting the ball into the goal area.
    — Reward for robot staying on the bridge segments or platforms and
        avoiding falling.
    — Reward for robot reaching the end platform.

Termination:
The episode terminates if the robot or ball falls off the bridge or
        platforms. It also terminates if the ball goes off the sides of
        the bridge or platforms.
    """

    def __init__(self):
        super().__init__()
        self.platform_size = [3.0, 3.0, 0.5]
        self.platform_start_position = [0.0, 0.0, 0.0]
```

```python
        self.platform_end_position = [self.platform_start_position[0] +
            25.0, self.platform_start_position[1], self.
            platform_start_position[2]]
        self.platform_start_id = self.create_box(mass=0.0, half_extents=[
            self.platform_size[0] / 2, self.platform_size[1] / 2, self.
            platform_size[2] / 2], position=self.platform_start_position,
             color=[0.8, 0.8, 0.8, 1.0])
        self.platform_end_id = self.create_box(mass=0.0, half_extents=[
            self.platform_size[0] / 2, self.platform_size[1] / 2, self.
            platform_size[2] / 2], position=self.platform_end_position,
            color=[0.8, 0.8, 0.8, 1.0])
        self.bridge_length = self.platform_end_position[0] - self.
            platform_start_position[0] - self.platform_size[0]
        self.bridge_width = 2.0
        self.num_segments = 6
        self.segment_length = 3.0
        self.gap_length = 1.0
        self.segment_amplitude = 1.0
        self.segment_period = 2.0
        self.segment_phase_offset = 1.0 / 3.0
        self.segment_ids = []
        for i in range(self.num_segments):
            segment_id = self.create_box(mass=0.0, half_extents=[self.
                segment_length / 2, self.bridge_width / 2, self.
                platform_size[2] / 2], position=[self.
                platform_start_position[0] + self.platform_size[0] / 2 +
                self.segment_length / 2 + i * (self.segment_length + self
                .gap_length), self.platform_start_position[1], self.
                platform_start_position[2]], color=[0.5, 0.5, 0.5, 1.0])
            self._p.changeDynamics(bodyUniqueId=segment_id, linkIndex=-1,
                lateralFriction=0.8, restitution=0.5)
            self.segment_ids.append(segment_id)
        self.ball_radius = 0.25
        self.ball_position_init = [self.platform_start_position[0] + 1.0,
             self.platform_start_position[1], self.
            platform_start_position[2] + self.platform_size[2] / 2 + self
            .ball_radius]
        self.ball_id = self.create_sphere(mass=1.0, radius=self.
            ball_radius, position=self.ball_position_init, color=[1.0,
            0.0, 0.0, 1.0])
        self.goal_width = 3.0
        self.goal_position = [self.platform_end_position[0], self.
            platform_end_position[1], self.platform_end_position[2] +
            self.platform_size[2] / 2 + self.ball_radius]
        self.goal_left_post_id = self.create_cylinder(mass=0.0, radius
            =0.1, height=1.0, position=[self.goal_position[0], self.
            goal_position[1] - self.goal_width / 2, self.goal_position
            [2]], color=[0.0, 1.0, 0.0, 1.0])
        self.goal_right_post_id = self.create_cylinder(mass=0.0, radius
            =0.1, height=1.0, position=[self.goal_position[0], self.
            goal_position[1] + self.goal_width / 2, self.goal_position
            [2]], color=[0.0, 1.0, 0.0, 1.0])

    def create_box(self, mass, half_extents, position, color):
        collision_shape_id = self._p.createCollisionShape(shapeType=self.
            _p.GEOM_BOX, halfExtents=half_extents)
        visual_shape_id = self._p.createVisualShape(shapeType=self._p.
            GEOM_BOX, halfExtents=half_extents, rgbaColor=color)
        return self._p.createMultiBody(baseMass=mass,
            baseCollisionShapeIndex=collision_shape_id,
            baseVisualShapeIndex=visual_shape_id, basePosition=position)

    def create_sphere(self, mass, radius, position, color):
        collision_shape_id = self._p.createCollisionShape(shapeType=self.
            _p.GEOM_SPHERE, radius=radius)
```

```python
        visual_shape_id = self._p.createVisualShape(shapeType=self._p.
            GEOM_SPHERE, radius=radius, rgbaColor=color)
        return self._p.createMultiBody(baseMass=mass,
            baseCollisionShapeIndex=collision_shape_id,
            baseVisualShapeIndex=visual_shape_id, basePosition=position)

    def create_cylinder(self, mass, radius, height, position, color):
        collision_shape_id = self._p.createCollisionShape(shapeType=self.
            _p.GEOM_CYLINDER, radius=radius, height=height)
        visual_shape_id = self._p.createVisualShape(shapeType=self._p.
            GEOM_CYLINDER, radius=radius, length=height, rgbaColor=color)
        return self._p.createMultiBody(baseMass=mass,
            baseCollisionShapeIndex=collision_shape_id,
            baseVisualShapeIndex=visual_shape_id, basePosition=position)

    def get_object_position(self, object_id):
        return np.asarray(self._p.getBasePositionAndOrientation(object_id
            )[0])

    def get_distance_to_object(self, object_id):
        object_position = self.get_object_position(object_id)
        robot_position = self.robot.links['base'].position
        return np.linalg.norm(object_position[:2] - robot_position[:2])

    def reset(self):
        observation = super().reset()
        self.time = 0.0
        self._p.resetBasePositionAndOrientation(self.ball_id, self.
            ball_position_init, [0.0, 0.0, 0.0, 1.0])
        self._p.resetBasePositionAndOrientation(self.robot.robot_id, [
            self.platform_start_position[0], self.platform_start_position
            [1], self.platform_start_position[2] + self.platform_size[2]
            / 2 + self.robot.links['base'].position_init[2]], self.robot.
            links['base'].orientation_init)
        return observation

    def step(self, action):
        self.distance_to_ball = self.get_distance_to_object(self.ball_id)
        self.ball_position = self.get_object_position(self.ball_id)
        self.distance_to_platform_end = self.get_distance_to_object(self.
            platform_end_id)
        observation, reward, terminated, truncated, info = super().step(
            action)
        self.time += self.dt
        for i, segment_id in enumerate(self.segment_ids):
            segment_position = self.get_object_position(segment_id)
            new_segment_position = [segment_position[0], segment_position
                [1], self.platform_start_position[2] + self.
                segment_amplitude * np.sin(2 * np.pi * (self.time + i *
                self.segment_phase_offset) / self.segment_period)]
            self._p.resetBasePositionAndOrientation(segment_id,
                new_segment_position, [0.0, 0.0, 0.0, 1.0])
        return (observation, reward, terminated, truncated, info)

    def get_task_rewards(self, action):
        new_distance_to_ball = self.get_distance_to_object(self.ball_id)
        new_ball_position = self.get_object_position(self.ball_id)
        new_distance_to_platform_end = self.get_distance_to_object(self.
            platform_end_id)
        approach_ball = (self.distance_to_ball - new_distance_to_ball) /
            self.dt
        ball_forward_velocity = (new_ball_position[0] - self.
            ball_position[0]) / self.dt
        ball_in_goal = 10.0 if self.goal_position[1] - self.goal_width /
            2 < new_ball_position[1] < self.goal_position[1] + self.
```

```
                goal_width / 2 and new_ball_position[0] > self.goal_position
                    [0] else 0.0
            on_bridge_or_platforms = 1.0 if self.robot.links['base'].position
                [2] > self.platform_start_position[2] + self.platform_size[2]
                 / 2 else −1.0
            reach_platform_end = (self.distance_to_platform_end −
                new_distance_to_platform_end) / self.dt
            return {'approach_ball': approach_ball, 'ball_forward_velocity':
                ball_forward_velocity, 'ball_in_goal': ball_in_goal, '
                on_bridge_or_platforms': on_bridge_or_platforms, '
                reach_platform_end': reach_platform_end}

    def get_terminated(self, action):
        is_robot_on_bridge_or_platforms = self.robot.links['base'].
            position[2] > self.platform_start_position[2]
        is_ball_on_bridge_or_platforms = self.get_object_position(self.
            ball_id)[2] > self.platform_start_position[2]
        is_ball_off_sides = np.abs(self.get_object_position(self.ball_id)
            [1]) > self.bridge_width / 2 + self.platform_size[1] / 2
        return not is_robot_on_bridge_or_platforms or not
            is_ball_on_bridge_or_platforms or is_ball_off_sides

    def get_success(self):
        ball_position = self.get_object_position(self.ball_id)
        is_ball_in_goal = self.goal_position[1] − self.goal_width / 2 <
            ball_position[1] < self.goal_position[1] + self.goal_width /
            2 and ball_position[0] > self.goal_position[0]
        is_robot_on_platform_end = self.get_distance_to_object(self.
            platform_end_id) < self.platform_size[2] / 2
        return is_ball_in_goal and is_robot_on_platform_end
```

**Task 8**

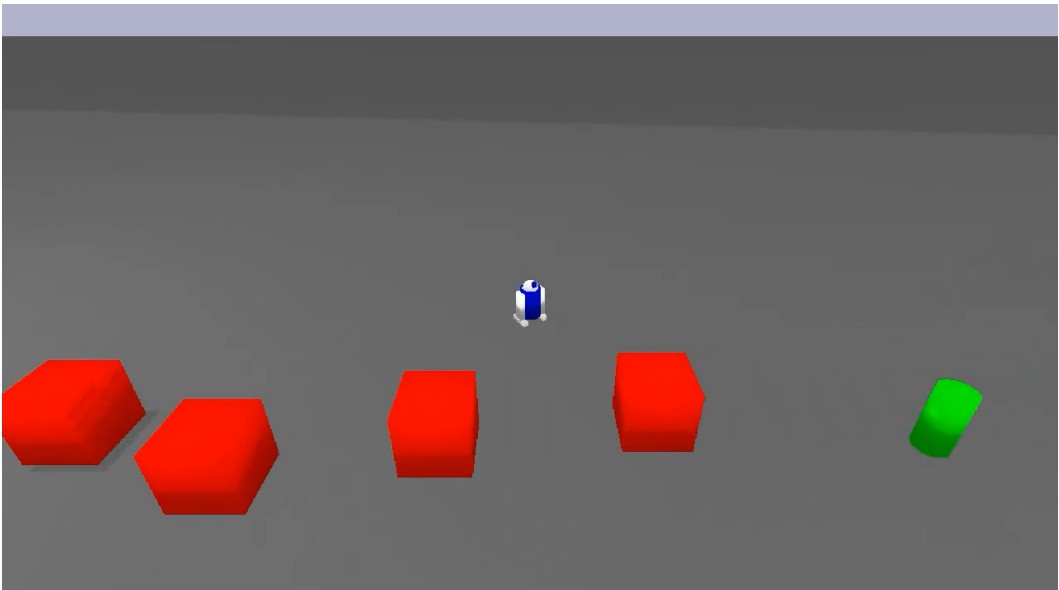

Figure 14: Task 8 of the OMNI-EPIC run presented in Section 5.

```
import numpy as np
from oped.envs.r2d2.base import R2D2Env

class Env(R2D2Env):
    """
```

```
Navigate a dynamic obstacle course to reach a target.

Description:
- The environment consists of a large flat ground measuring 1000 x
    1000 x 10 meters.
- The robot starts at a fixed position on the ground, facing the
    positive x-axis.
- The target is a cylindrical object (0.5 meters in diameter and 1
    meter in height) placed 20 meters directly in front of the robot'
    s starting position.
- The obstacle course consists of 5 dynamic obstacles, each moving in
     a predictable pattern:
  - Obstacle 1: A box (1 x 1 x 1 meters) moving side-to-side (along
      the y-axis) with an amplitude of 2 meters and a period of 4
      seconds. It is placed 5 meters in front of the robot's starting
       position.
  - Obstacle 2: A box (1 x 1 x 1 meters) moving up and down (along
      the z-axis) with an amplitude of 1 meter and a period of 3
      seconds. It is placed 8 meters in front of the robot's starting
       position.
  - Obstacle 3: A box (1 x 1 x 1 meters) rotating around its center
      with a radius of 1 meter and a period of 5 seconds. It is
      placed 11 meters in front of the robot's starting position.
  - Obstacle 4: A box (1 x 1 x 1 meters) moving diagonally (along
      both x and y axes) with an amplitude of 1 meter and a period of
       6 seconds. It is placed 14 meters in front of the robot's
      starting position.
  - Obstacle 5: A box (1 x 1 x 1 meters) moving in a circular pattern
       (along the x-y plane) with a radius of 1 meter and a period of
       7 seconds. It is placed 17 meters in front of the robot's
      starting position.
- The task of the robot is to navigate through the dynamic obstacle
    course and reach the target as quickly as possible.

Success:
The task is successfully completed when the robot reaches the target.

Rewards:
To help the robot complete the task:
- The robot receives a reward for each time step it remains in the
    environment, encouraging steady progress.
- The robot is rewarded based on how much it reduces the distance to
    the target, incentivizing swift movement towards the goal.
- The robot is penalized for colliding with any obstacles.

Termination:
The task terminates immediately if the robot collides with any
    obstacle or reaches the target.
"""

def __init__(self):
    super().__init__()
    self.ground_size = [1000.0, 1000.0, 10.0]
    self.ground_position = [0.0, 0.0, 0.0]
    self.ground_id = self.create_box(mass=0.0, half_extents=[self.
        ground_size[0] / 2, self.ground_size[1] / 2, self.ground_size
        [2] / 2], position=self.ground_position, color=[0.5, 0.5,
        0.5, 1.0])
    self._p.changeDynamics(bodyUniqueId=self.ground_id, linkIndex=-1,
        lateralFriction=0.8, restitution=0.5)
    self.target_radius = 0.25
    self.target_height = 1.0
    self.target_position = [20.0, 0.0, self.ground_position[2] + self
        .ground_size[2] / 2 + self.target_height / 2]
```

```python
        self.target_id = self.create_cylinder(mass=0.0, radius=self.
            target_radius, height=self.target_height, position=self.
            target_position, color=[0.0, 1.0, 0.0, 1.0])
        self.obstacle_size = [1.0, 1.0, 1.0]
        self.obstacle_1_position_init = [5.0, 0.0, self.ground_position
            [2] + self.ground_size[2] / 2 + self.obstacle_size[2] / 2]
        self.obstacle_2_position_init = [8.0, 0.0, self.ground_position
            [2] + self.ground_size[2] / 2 + self.obstacle_size[2] / 2]
        self.obstacle_3_position_init = [11.0, 0.0, self.ground_position
            [2] + self.ground_size[2] / 2 + self.obstacle_size[2] / 2]
        self.obstacle_4_position_init = [14.0, 0.0, self.ground_position
            [2] + self.ground_size[2] / 2 + self.obstacle_size[2] / 2]
        self.obstacle_5_position_init = [17.0, 0.0, self.ground_position
            [2] + self.ground_size[2] / 2 + self.obstacle_size[2] / 2]
        self.obstacle_1_id = self.create_box(mass=0.0, half_extents=[self
            .obstacle_size[0] / 2, self.obstacle_size[1] / 2, self.
            obstacle_size[2] / 2], position=self.obstacle_1_position_init
            , color=[1.0, 0.0, 0.0, 1.0])
        self.obstacle_2_id = self.create_box(mass=0.0, half_extents=[self
            .obstacle_size[0] / 2, self.obstacle_size[1] / 2, self.
            obstacle_size[2] / 2], position=self.obstacle_2_position_init
            , color=[1.0, 0.0, 0.0, 1.0])
        self.obstacle_3_id = self.create_box(mass=0.0, half_extents=[self
            .obstacle_size[0] / 2, self.obstacle_size[1] / 2, self.
            obstacle_size[2] / 2], position=self.obstacle_3_position_init
            , color=[1.0, 0.0, 0.0, 1.0])
        self.obstacle_4_id = self.create_box(mass=0.0, half_extents=[self
            .obstacle_size[0] / 2, self.obstacle_size[1] / 2, self.
            obstacle_size[2] / 2], position=self.obstacle_4_position_init
            , color=[1.0, 0.0, 0.0, 1.0])
        self.obstacle_5_id = self.create_box(mass=0.0, half_extents=[self
            .obstacle_size[0] / 2, self.obstacle_size[1] / 2, self.
            obstacle_size[2] / 2], position=self.obstacle_5_position_init
            , color=[1.0, 0.0, 0.0, 1.0])
        self.robot_position_init = [0.0, 0.0, self.ground_position[2] +
            self.ground_size[2] / 2 + self.robot.links['base'].
            position_init[2]]
        self.robot_orientation_init = self._p.getQuaternionFromEuler
            ([0.0, 0.0, 0.0])

    def create_box(self, mass, half_extents, position, color):
        collision_shape_id = self._p.createCollisionShape(shapeType=self.
            _p.GEOM_BOX, halfExtents=half_extents)
        visual_shape_id = self._p.createVisualShape(shapeType=self._p.
            GEOM_BOX, halfExtents=half_extents, rgbaColor=color)
        return self._p.createMultiBody(baseMass=mass,
            baseCollisionShapeIndex=collision_shape_id,
            baseVisualShapeIndex=visual_shape_id, basePosition=position)

    def create_cylinder(self, mass, radius, height, position, color):
        collision_shape_id = self._p.createCollisionShape(shapeType=self.
            _p.GEOM_CYLINDER, radius=radius, height=height)
        visual_shape_id = self._p.createVisualShape(shapeType=self._p.
            GEOM_CYLINDER, radius=radius, length=height, rgbaColor=color)
        return self._p.createMultiBody(baseMass=mass,
            baseCollisionShapeIndex=collision_shape_id,
            baseVisualShapeIndex=visual_shape_id, basePosition=position)

    def get_object_position(self, object_id):
        return np.asarray(self._p.getBasePositionAndOrientation(object_id
            )[0])

    def get_distance_to_object(self, object_id):
        object_position = self.get_object_position(object_id)
        robot_position = self.robot.links['base'].position
```

```python
        return np.linalg.norm(object_position[:2] − robot_position[:2])

    def reset(self):
        observation = super().reset()
        self.time = 0.0
        self._p.resetBasePositionAndOrientation(self.robot.robot_id, self
            .robot_position_init, self.robot_orientation_init)
        return observation

    def step(self, action):
        self.distance_to_target = self.get_distance_to_object(self.
            target_id)
        observation, reward, terminated, truncated, info = super().step(
            action)
        self.time += self.dt
        obstacle_1_new_y = self.obstacle_1_position_init[1] + 2.0 * np.
            sin(2 * np.pi * self.time / 4.0)
        self._p.resetBasePositionAndOrientation(self.obstacle_1_id, [self
            .obstacle_1_position_init[0], obstacle_1_new_y, self.
            obstacle_1_position_init[2]], [0.0, 0.0, 0.0, 1.0])
        obstacle_2_new_z = self.obstacle_2_position_init[2] + 1.0 * np.
            sin(2 * np.pi * self.time / 3.0)
        self._p.resetBasePositionAndOrientation(self.obstacle_2_id, [self
            .obstacle_2_position_init[0], self.obstacle_2_position_init
            [1], obstacle_2_new_z], [0.0, 0.0, 0.0, 1.0])
        obstacle_3_new_x = self.obstacle_3_position_init[0] + 1.0 * np.
            cos(2 * np.pi * self.time / 5.0)
        obstacle_3_new_y = self.obstacle_3_position_init[1] + 1.0 * np.
            sin(2 * np.pi * self.time / 5.0)
        self._p.resetBasePositionAndOrientation(self.obstacle_3_id, [
            obstacle_3_new_x, obstacle_3_new_y, self.
            obstacle_3_position_init[2]], [0.0, 0.0, 0.0, 1.0])
        obstacle_4_new_x = self.obstacle_4_position_init[0] + 1.0 * np.
            cos(2 * np.pi * self.time / 6.0)
        obstacle_4_new_y = self.obstacle_4_position_init[1] + 1.0 * np.
            sin(2 * np.pi * self.time / 6.0)
        self._p.resetBasePositionAndOrientation(self.obstacle_4_id, [
            obstacle_4_new_x, obstacle_4_new_y, self.
            obstacle_4_position_init[2]], [0.0, 0.0, 0.0, 1.0])
        obstacle_5_new_x = self.obstacle_5_position_init[0] + 1.0 * np.
            cos(2 * np.pi * self.time / 7.0)
        obstacle_5_new_y = self.obstacle_5_position_init[1] + 1.0 * np.
            sin(2 * np.pi * self.time / 7.0)
        self._p.resetBasePositionAndOrientation(self.obstacle_5_id, [
            obstacle_5_new_x, obstacle_5_new_y, self.
            obstacle_5_position_init[2]], [0.0, 0.0, 0.0, 1.0])
        return (observation, reward, terminated, truncated, info)

    def get_task_rewards(self, action):
        new_distance_to_target = self.get_distance_to_object(self.
            target_id)
        survival = 1.0
        reach_target = (self.distance_to_target − new_distance_to_target)
            / self.dt
        return {'survival': survival, 'reach_target': reach_target}

    def get_terminated(self, action):
        collision_obstacle_1 = len(self._p.getContactPoints(bodyA=self.
            robot.robot_id, bodyB=self.obstacle_1_id)) > 0
        collision_obstacle_2 = len(self._p.getContactPoints(bodyA=self.
            robot.robot_id, bodyB=self.obstacle_2_id)) > 0
        collision_obstacle_3 = len(self._p.getContactPoints(bodyA=self.
            robot.robot_id, bodyB=self.obstacle_3_id)) > 0
        collision_obstacle_4 = len(self._p.getContactPoints(bodyA=self.
            robot.robot_id, bodyB=self.obstacle_4_id)) > 0
```

```
        collision_obstacle_5 = len(self._p.getContactPoints(bodyA=self.
            robot.robot_id, bodyB=self.obstacle_5_id)) > 0
        collision_target = len(self._p.getContactPoints(bodyA=self.robot.
            robot_id, bodyB=self.target_id)) > 0
        return collision_obstacle_1 or collision_obstacle_2 or
            collision_obstacle_3 or collision_obstacle_4 or
            collision_obstacle_5 or collision_target

    def get_success(self):
        contact_points_target = self._p.getContactPoints(bodyA=self.robot
            .robot_id, bodyB=self.target_id)
        return len(contact_points_target) > 0
```

**Task 9**

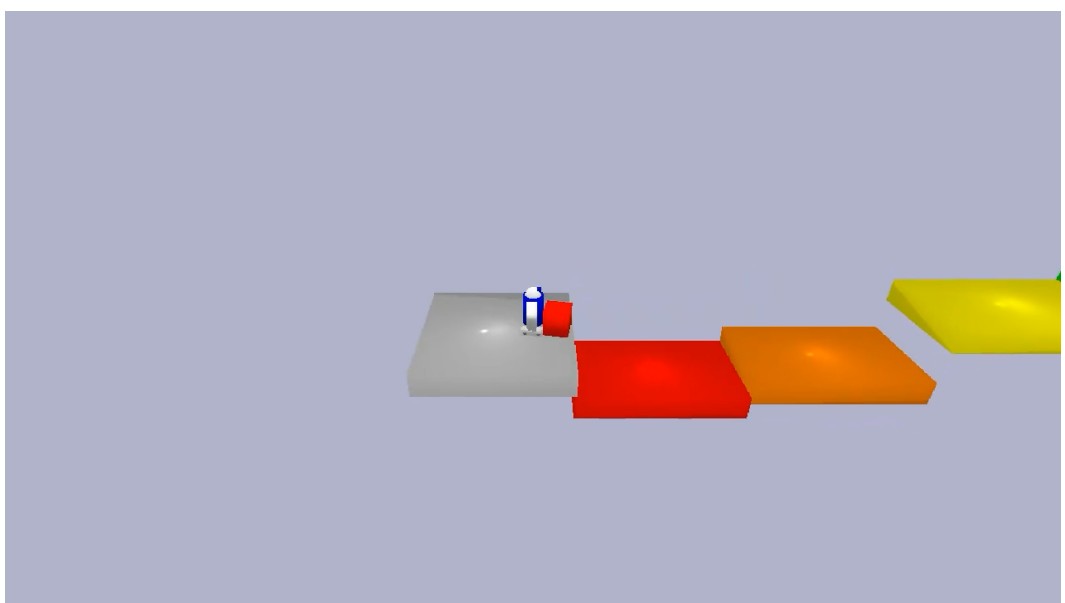

Figure 15: Task 9 of the OMNI-EPIC run presented in Section 5.

```
import numpy as np
from oped.envs.r2d2.base import R2D2Env

class Env(R2D2Env):
    """
    Push a box to a target location on a dynamic platform.

    Description:
    — The environment consists of a start platform and an end platform (
        each 3 m in size and 0.5 m in thickness) placed 20 m apart.
    — The two platforms are connected by a dynamic bridge (2 m wide)
        divided into 5 segments of equal length (3 m each).
    — Each segment moves up and down independently with a sinusoidal
        motion. The amplitude is 1 m and the period is 2 seconds, with
        each segment offset in phase by 0.4 seconds from the previous one
        .
    — A box (0.5 m in size) is placed on the start platform, 1 m in front
         of the robot's initial position.
    — The robot is initialized on the start platform, facing the box and
        bridge.
    — The task of the robot is to push the box across the dynamic bridge
        segments to reach a target location on the end platform.
```

```
    Success:
    The task is successfully completed when the box reaches the target
        location on the end platform.

    Rewards:
    - Reward for decreasing the distance between the robot and the box,
        encouraging the robot to approach and interact with the box.
    - Reward for increasing the forward progress of the box across the
        bridge.
    - Large reward for getting the box to the target location on the end
        platform.
    - Reward for the robot staying on the bridge segments or platforms
        and avoiding falling.
    - Reward for the robot reaching the end platform.

    Termination:
    The episode terminates if the robot or box falls off the bridge or
        platforms.
    """

    def __init__(self):
        super().__init__()
        self.platform_size = [3.0, 3.0, 0.5]
        self.platform_start_position = [0.0, 0.0, 0.0]
        self.platform_end_position = [self.platform_start_position[0] +
            20.0, self.platform_start_position[1], self.
            platform_start_position[2]]
        self.platform_start_id = self.create_box(mass=0.0, half_extents=[
            self.platform_size[0] / 2, self.platform_size[1] / 2, self.
            platform_size[2] / 2], position=self.platform_start_position,
             color=[0.8, 0.8, 0.8, 1.0])
        self.platform_end_id = self.create_box(mass=0.0, half_extents=[
            self.platform_size[0] / 2, self.platform_size[1] / 2, self.
            platform_size[2] / 2], position=self.platform_end_position,
            color=[0.8, 0.8, 0.8, 1.0])
        self.bridge_length = self.platform_end_position[0] - self.
            platform_start_position[0] - self.platform_size[0]
        self.bridge_width = 2.0
        self.num_segments = 5
        self.segment_length = self.bridge_length / self.num_segments
        self.segment_amplitude = 1.0
        self.segment_period = 2.0
        self.segment_phase_offset = 0.4
        segment_colors = [[1.0, 0.0, 0.0, 1.0], [1.0, 0.5, 0.0, 1.0],
            [1.0, 1.0, 0.0, 1.0], [0.0, 1.0, 0.0, 1.0], [0.0, 0.0, 1.0,
            1.0]]
        self.segment_ids = []
        for i in range(self.num_segments):
            segment_id = self.create_box(mass=0.0, half_extents=[self.
                segment_length / 2, self.bridge_width / 2, self.
                platform_size[2] / 2], position=[self.
                platform_start_position[0] + self.platform_size[0] / 2 +
                self.segment_length / 2 + i * self.segment_length, self.
                platform_start_position[1], self.platform_start_position
                [2]], color=segment_colors[i])
            self._p.changeDynamics(bodyUniqueId=segment_id, linkIndex=-1,
                lateralFriction=0.8, restitution=0.5)
            self.segment_ids.append(segment_id)
        self.box_size = [0.5, 0.5, 0.5]
        self.box_position_init = [self.platform_start_position[0] + 1.0,
            self.platform_start_position[1], self.platform_start_position
            [2] + self.platform_size[2] / 2 + self.box_size[2] / 2]
        self.box_id = self.create_box(mass=1.0, half_extents=[self.
            box_size[0] / 2, self.box_size[1] / 2, self.box_size[2] / 2],
             position=self.box_position_init, color=[1.0, 0.0, 0.0, 1.0])
```

```python
        self.target_position = [self.platform_end_position[0], self.
            platform_end_position[1], self.platform_end_position[2] +
            self.platform_size[2] / 2 + self.box_size[2] / 2]

    def create_box(self, mass, half_extents, position, color):
        collision_shape_id = self._p.createCollisionShape(shapeType=self.
            _p.GEOM_BOX, halfExtents=half_extents)
        visual_shape_id = self._p.createVisualShape(shapeType=self._p.
            GEOM_BOX, halfExtents=half_extents, rgbaColor=color)
        return self._p.createMultiBody(baseMass=mass,
            baseCollisionShapeIndex=collision_shape_id,
            baseVisualShapeIndex=visual_shape_id, basePosition=position)

    def get_object_position(self, object_id):
        return np.asarray(self._p.getBasePositionAndOrientation(object_id
            )[0])

    def get_distance_to_object(self, object_id):
        object_position = self.get_object_position(object_id)
        robot_position = self.robot.links['base'].position
        return np.linalg.norm(object_position[:2] - robot_position[:2])

    def reset(self):
        observation = super().reset()
        self.time = 0.0
        self._p.resetBasePositionAndOrientation(self.robot.robot_id, [
            self.platform_start_position[0], self.platform_start_position
            [1], self.platform_start_position[2] + self.platform_size[2]
            / 2 + self.robot.links['base'].position_init[2]], self.robot.
            links['base'].orientation_init)
        self._p.resetBasePositionAndOrientation(self.box_id, self.
            box_position_init, [0.0, 0.0, 0.0, 1.0])
        return observation

    def step(self, action):
        self.distance_to_box = self.get_distance_to_object(self.box_id)
        self.box_position = self.get_object_position(self.box_id)
        observation, reward, terminated, truncated, info = super().step(
            action)
        self.time += self.dt
        for i, segment_id in enumerate(self.segment_ids):
            segment_position = self.get_object_position(segment_id)
            new_segment_position = [segment_position[0], segment_position
                [1], self.platform_start_position[2] + self.
                segment_amplitude * np.sin(2 * np.pi * (self.time + i *
                self.segment_phase_offset) / self.segment_period)]
            self._p.resetBasePositionAndOrientation(segment_id,
                new_segment_position, [0.0, 0.0, 0.0, 1.0])
        return (observation, reward, terminated, truncated, info)

    def get_task_rewards(self, action):
        new_distance_to_box = self.get_distance_to_object(self.box_id)
        new_box_position = self.get_object_position(self.box_id)
        approach_box = (self.distance_to_box - new_distance_to_box) /
            self.dt
        forward_progress_box = (new_box_position[0] - self.box_position
            [0]) / self.dt
        on_bridge_or_platforms = 1.0 if self.robot.links['base'].position
            [2] > self.platform_start_position[2] + self.platform_size[2]
            / 2 else -1.0
        reach_end_platform = 1.0 if self.robot.links['base'].position[0]
            >= self.platform_end_position[0] else 0.0
        reach_target_location = 10.0 if np.linalg.norm(new_box_position
            [:2] - self.target_position[:2]) < self.box_size[0] / 2 else
            0.0
```

```
        return {'approach_box': approach_box, 'forward_progress_box':
            forward_progress_box, 'on_bridge_or_platforms':
            on_bridge_or_platforms, 'reach_end_platform':
            reach_end_platform, 'reach_target_location':
            reach_target_location}

    def get_terminated(self, action):
        robot_fell = self.robot.links['base'].position[2] < self.
            platform_start_position[2]
        box_fell = self.get_object_position(self.box_id)[2] < self.
            platform_start_position[2]
        return robot_fell or box_fell

    def get_success(self):
        box_position = self.get_object_position(self.box_id)
        return np.linalg.norm(box_position[:2] - self.target_position
            [:2]) < self.box_size[0] / 2
```

**Task 10**

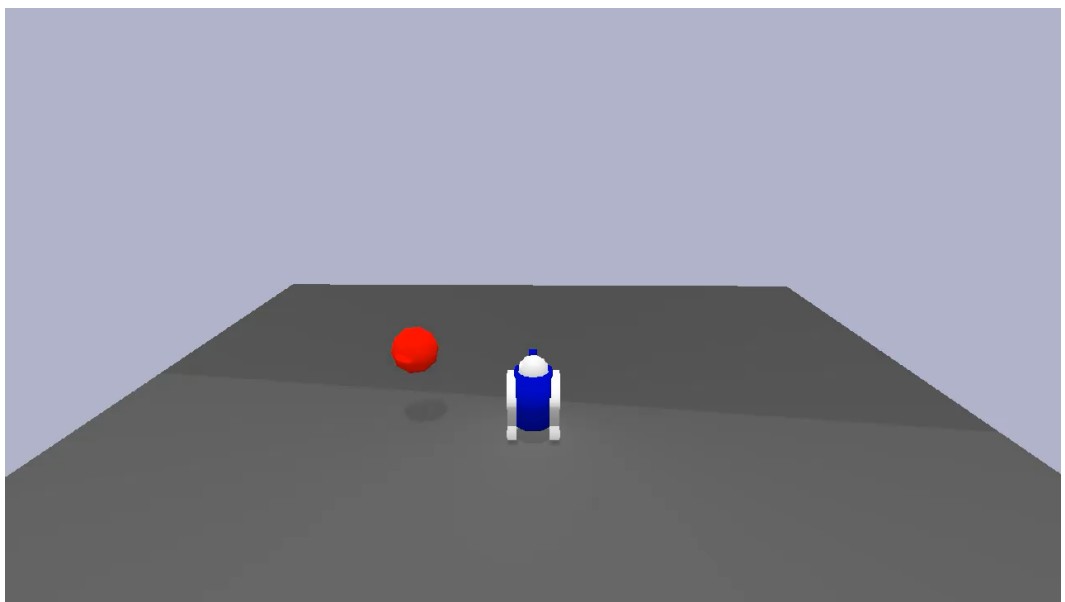

Figure 16: Task 10 of the OMNI-EPIC run presented in Section 5.

```
import numpy as np
from oped.envs.r2d2.base import R2D2Env

class Env(R2D2Env):
    """
    Dodgeball Game

    Description:
    - The robot spawns in a square arena measuring 10 meters by 10 meters
        .
    - Balls with a radius of 0.25 meters spawn randomly within the arena
        at a height of 1 meter and are launched toward the robot with an
        initial velocity of 5 meters per second.
    - The balls spawn at a rate of one ball every 2 seconds.
    - The robot's task is to dodge the balls for as long as possible.

    Success:
```

```
    The task is successfully completed if the robot survives for a
        predefined duration of 60 seconds without being hit by any balls.

    Rewards:
    - The robot receives a reward for each time step it remains standing
        and avoids being hit by a ball, encouraging it to dodge
        effectively.
    - A small survival reward is given for each second the robot remains
        in the game, incentivizing prolonged survival.
    - A penalty is applied if the robot is hit by a ball, encouraging it
        to avoid collisions.

    Termination:
    - The task terminates immediately if the robot is hit by a ball.
    - The task also terminates if the robot survives for 60 seconds
        without being hit by any balls.
    """

    def __init__(self):
        super().__init__()
        self.arena_size = [10.0, 10.0]
        self.arena_position = [0.0, 0.0, 0.0]
        self.arena_id = self.create_box(mass=0.0, half_extents=[self.
            arena_size[0] / 2, self.arena_size[1] / 2, 0.1], position=
            self.arena_position, color=[0.5, 0.5, 0.5, 1.0])
        self._p.changeDynamics(bodyUniqueId=self.arena_id, linkIndex=-1,
            lateralFriction=0.8, restitution=0.5)
        self.ball_radius = 0.25
        self.ball_velocity = 5.0
        self.ball_spawn_interval = 2.0
        self.ball_ids = []
        self.robot_position_init = [0.0, 0.0, self.arena_position[2] +
            0.1 + self.robot.links['base'].position_init[2]]
        self.robot_orientation_init = self._p.getQuaternionFromEuler
            ([0.0, 0.0, 0.0])
        self.survival_duration = 16.0

    def create_box(self, mass, half_extents, position, color):
        collision_shape_id = self._p.createCollisionShape(shapeType=self.
            _p.GEOM_BOX, halfExtents=half_extents)
        visual_shape_id = self._p.createVisualShape(shapeType=self._p.
            GEOM_BOX, halfExtents=half_extents, rgbaColor=color)
        return self._p.createMultiBody(baseMass=mass,
            baseCollisionShapeIndex=collision_shape_id,
            baseVisualShapeIndex=visual_shape_id, basePosition=position)

    def create_sphere(self, mass, radius, position, velocity, color):
        collision_shape_id = self._p.createCollisionShape(shapeType=self.
            _p.GEOM_SPHERE, radius=radius)
        visual_shape_id = self._p.createVisualShape(shapeType=self._p.
            GEOM_SPHERE, radius=radius, rgbaColor=color)
        ball_id = self._p.createMultiBody(baseMass=mass,
            baseCollisionShapeIndex=collision_shape_id,
            baseVisualShapeIndex=visual_shape_id, basePosition=position)
        self._p.resetBaseVelocity(objectUniqueId=ball_id, linearVelocity=
            velocity)
        return ball_id

    def reset(self):
        observation = super().reset()
        self.time = 0.0
        self._p.resetBasePositionAndOrientation(self.robot.robot_id, self
            .robot_position_init, self.robot_orientation_init)
        for ball_id in self.ball_ids:
            self._p.removeBody(ball_id)
```

```python
        self.ball_ids = []
        return observation

    def step(self, action):
        self.hit_by_ball = False
        observation, reward, terminated, truncated, info = super().step(
            action)
        self.time += self.dt
        if self.time % self.ball_spawn_interval < self.dt:
            ball_position = [np.random.uniform(low=-self.arena_size[0] /
                2 + self.ball_radius, high=self.arena_size[0] / 2 - self.
                ball_radius), np.random.uniform(low=-self.arena_size[1] /
                 2 + self.ball_radius, high=self.arena_size[1] / 2 - self
                .ball_radius), 1.0]
            ball_velocity = [np.random.uniform(low=-1.0, high=1.0), np.
                random.uniform(low=-1.0, high=1.0), 0.0]
            ball_velocity = self.ball_velocity * np.array(ball_velocity)
                / np.linalg.norm(ball_velocity)
            ball_id = self.create_sphere(mass=1.0, radius=self.
                ball_radius, position=ball_position, velocity=
                ball_velocity, color=[1.0, 0.0, 0.0, 1.0])
            self.ball_ids.append(ball_id)
        for ball_id in self.ball_ids:
            contact_points = self._p.getContactPoints(bodyA=self.robot.
                robot_id, bodyB=ball_id)
            if len(contact_points) > 0:
                self.hit_by_ball = True
                break
        return (observation, reward, terminated, truncated, info)

    def get_task_rewards(self, action):
        survival = 1.0
        hit_penalty = -10.0 if self.hit_by_ball else 0.0
        return {'survival': survival, 'hit_penalty': hit_penalty}

    def get_terminated(self, action):
        if self.hit_by_ball:
            return True
        if self.time >= self.survival_duration:
            return True
        return False

    def get_success(self):
        return self.time >= self.survival_duration and (not self.
            hit_by_ball)
```

**Task 11**

```python
import numpy as np
from oped.envs.r2d2.base import R2D2Env

class Env(R2D2Env):
    """
    Jump from moving platform to moving platform to cross water and reach
        a target zone on a platform.

    Description:
    - The environment consists of a start platform (3 m x 3 m x 0.5 m)
        and an end platform (3 m x 3 m x 0.5 m) placed 20 meters apart.
    - The two platforms are connected by a series of 5 moving platforms
        (2 m x 2 m x 0.5 m) placed 3 meters apart.
    - Each moving platform follows a sinusoidal motion along the y-axis
        with an amplitude of 1 meter and a period of 3 seconds. The
        motion of each platform is offset by 0.6 seconds from the
        previous one.
```

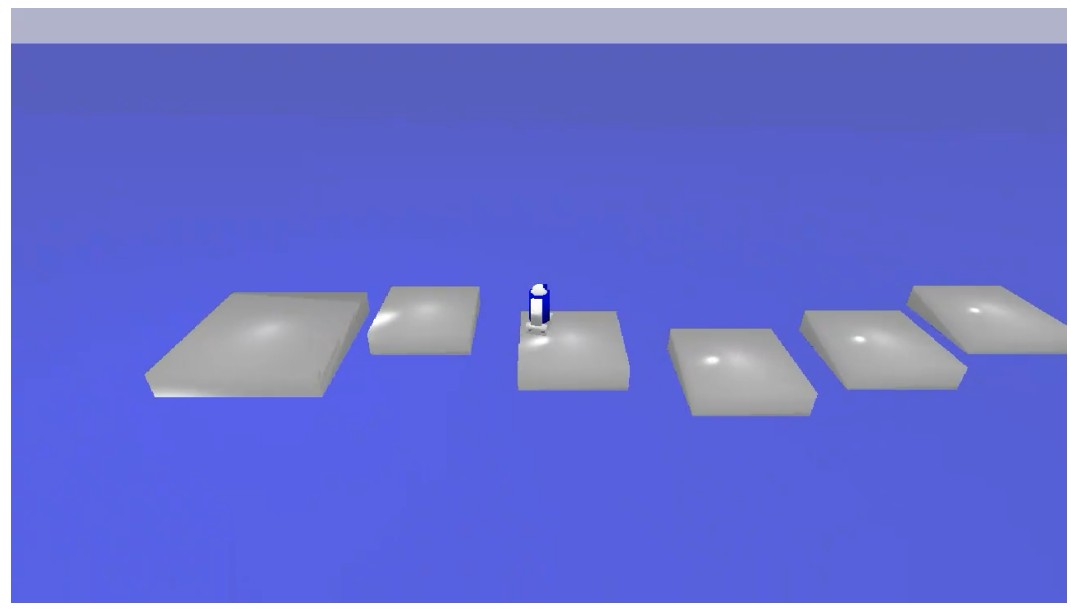

Figure 17: Task 11 of the OMNI-EPIC run presented in Section 5.

```
    — The platforms are placed above a water surface, and the robot must
        jump from one moving platform to the next to reach the end
        platform.

Success:
The task is successfully completed when the robot reaches the end
    platform.

Rewards:
— The robot receives a reward for each time step it remains on the
    platforms, encouraging steady progress.
— The robot is rewarded based on how much it reduces the distance to
    the end platform, incentivizing swift movement towards the goal.
— The robot is penalized for falling into the water.

Termination:
The task terminates immediately if the robot falls into the water or
    reaches the end platform.
"""

def __init__(self):
    super().__init__()
    self.water_size = [1000.0, 1000.0, 10.0]
    self.water_position = [0.0, 0.0, 0.0]
    self.water_id = self.create_box(mass=0.0, half_extents=[self.
        water_size[0] / 2, self.water_size[1] / 2, self.water_size[2]
         / 2], position=self.water_position, color=[0.0, 0.0, 1.0,
        0.5])
    self.platform_start_size = [3.0, 3.0, 0.5]
    self.platform_start_position = [0.0, 0.0, self.water_position[2]
        + self.water_size[2] / 2 + self.platform_start_size[2] / 2]
    self.platform_start_id = self.create_box(mass=0.0, half_extents=[
        self.platform_start_size[0] / 2, self.platform_start_size[1]
        / 2, self.platform_start_size[2] / 2], position=self.
        platform_start_position, color=[0.8, 0.8, 0.8, 1.0])
    self.platform_end_size = [3.0, 3.0, 0.5]
```

```python
        self.platform_end_position = [self.platform_start_position[0] +
            20.0, self.platform_start_position[1], self.
            platform_start_position[2]]
        self.platform_end_id = self.create_box(mass=0.0, half_extents=[
            self.platform_end_size[0] / 2, self.platform_end_size[1] / 2,
             self.platform_end_size[2] / 2], position=self.
            platform_end_position, color=[0.8, 0.8, 0.8, 1.0])
        self.num_moving_platforms = 5
        self.moving_platform_size = [2.0, 2.0, 0.5]
        self.moving_platform_amplitude = 1.0
        self.moving_platform_period = 3.0
        self.moving_platform_phase_offset = 0.6
        self.moving_platform_ids = []
        for i in range(self.num_moving_platforms):
            moving_platform_position = [self.platform_start_position[0] +
                 (i + 1) * 3.0, self.platform_start_position[1], self.
                platform_start_position[2]]
            moving_platform_id = self.create_box(mass=0.0, half_extents=[
                self.moving_platform_size[0] / 2, self.
                moving_platform_size[1] / 2, self.moving_platform_size[2]
                 / 2], position=moving_platform_position, color=[0.8,
                0.8, 0.8, 1.0])
            self.moving_platform_ids.append(moving_platform_id)

    def create_box(self, mass, half_extents, position, color):
        collision_shape_id = self._p.createCollisionShape(shapeType=self.
            _p.GEOM_BOX, halfExtents=half_extents)
        visual_shape_id = self._p.createVisualShape(shapeType=self._p.
            GEOM_BOX, halfExtents=half_extents, rgbaColor=color)
        return self._p.createMultiBody(baseMass=mass,
            baseCollisionShapeIndex=collision_shape_id,
            baseVisualShapeIndex=visual_shape_id, basePosition=position)

    def get_object_position(self, object_id):
        return np.asarray(self._p.getBasePositionAndOrientation(object_id
            )[0])

    def get_distance_to_object(self, object_id):
        object_position = self.get_object_position(object_id)
        robot_position = self.robot.links['base'].position
        return np.linalg.norm(object_position[:2] - robot_position[:2])

    def reset(self):
        observation = super().reset()
        self.time = 0.0
        self._p.resetBasePositionAndOrientation(self.robot.robot_id, [
            self.platform_start_position[0], self.platform_start_position
            [1], self.platform_start_position[2] + self.
            platform_start_size[2] / 2 + self.robot.links['base'].
            position_init[2]], self.robot.links['base'].orientation_init)
        return observation

    def step(self, action):
        self.distance_to_platform_end = self.get_distance_to_object(self.
            platform_end_id)
        observation, reward, terminated, truncated, info = super().step(
            action)
        self.time += self.dt
        for i, moving_platform_id in enumerate(self.moving_platform_ids):
            moving_platform_position = self.get_object_position(
                moving_platform_id)
            new_moving_platform_position = [moving_platform_position[0],
                self.platform_start_position[1] + self.
                moving_platform_amplitude * np.sin(2 * np.pi * (self.time
```

```
                + i * self.moving_platform_phase_offset) / self.
                moving_platform_period), moving_platform_position[2]]
            self._p.resetBasePositionAndOrientation(moving_platform_id,
                new_moving_platform_position, [0.0, 0.0, 0.0, 1.0])
        return (observation, reward, terminated, truncated, info)

    def get_task_rewards(self, action):
        new_distance_to_platform_end = self.get_distance_to_object(self.
            platform_end_id)
        on_platform = 1.0 if self.robot.links['base'].position[2] > self.
            platform_start_position[2] else -1.0
        reach_platform_end = (self.distance_to_platform_end -
            new_distance_to_platform_end) / self.dt
        return {'on_platform': on_platform, 'reach_platform_end':
            reach_platform_end}

    def get_terminated(self, action):
        is_in_water = self.robot.links['base'].position[2] < self.
            water_position[2] + self.water_size[2] / 2
        is_on_platform_end = self.get_distance_to_object(self.
            platform_end_id) < self.platform_end_size[0] / 2
        return is_in_water or is_on_platform_end

    def get_success(self):
        is_on_platform_end = self.get_distance_to_object(self.
            platform_end_id) < self.platform_end_size[0] / 2
        return is_on_platform_end
```

**Task 12**

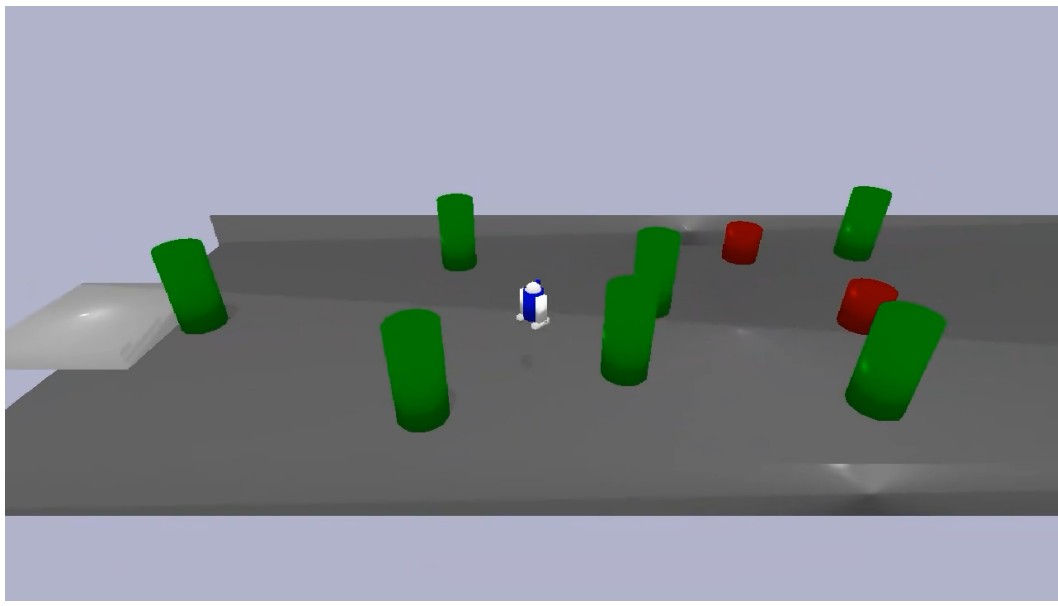

Figure 18: Task 12 of the OMNI-EPIC run presented in Section 5.

```
import numpy as np
from oped.envs.r2d2.base import R2D2Env

class Env(R2D2Env):
    """
    Navigate a terrain with obstacles to reach a target zone.

    Description:
```

```
    — The environment consists of a start platform (3 m x 3 m x 0.5 m)
        and an end platform (3 m x 3 m x 0.5 m) placed 30 meters apart.
    — The terrain between the platforms is filled with static obstacles
        such as rocks (cylinders) and trees (cylinders with a height of 3
         meters).
    — The obstacles are randomly placed with a minimum distance of 1
        meter between them, ensuring the robot cannot go around them
        easily.
    — The robot is initialized on the start platform, facing the end
        platform.

    Success:
    The task is successfully completed when the robot reaches the end
        platform.

    Rewards:
    — The robot receives a reward for each time step it remains on the
        platforms or the terrain, encouraging steady progress.
    — The robot is rewarded based on how much it reduces the distance to
        the end platform, incentivizing swift movement towards the goal.
    — The robot is penalized for collisions with obstacles.

    Termination:
    The task terminates immediately if the robot falls off the start
        platform, the terrain, or the end platform, or if the robot
        collides with an obstacle.
    """

    def __init__(self):
        super().__init__()
        self.platform_size = [3.0, 3.0, 0.5]
        self.platform_start_position = [0.0, 0.0, 0.0]
        self.platform_end_position = [self.platform_start_position[0] +
            30.0, self.platform_start_position[1], self.
            platform_start_position[2]]
        self.platform_start_id = self.create_box(mass=0.0, half_extents=[
            self.platform_size[0] / 2, self.platform_size[1] / 2, self.
            platform_size[2] / 2], position=self.platform_start_position,
             color=[0.8, 0.8, 0.8, 1.0])
        self.platform_end_id = self.create_box(mass=0.0, half_extents=[
            self.platform_size[0] / 2, self.platform_size[1] / 2, self.
            platform_size[2] / 2], position=self.platform_end_position,
            color=[0.8, 0.8, 0.8, 1.0])
        self._p.changeDynamics(bodyUniqueId=self.platform_start_id,
            linkIndex=-1, lateralFriction=0.8, restitution=0.5)
        self._p.changeDynamics(bodyUniqueId=self.platform_end_id,
            linkIndex=-1, lateralFriction=0.8, restitution=0.5)
        self.terrain_size = [self.platform_end_position[0] - self.
            platform_start_position[0], 10.0, 0.1]
        self.terrain_position = [self.platform_start_position[0] + self.
            terrain_size[0] / 2, self.platform_start_position[1], self.
            platform_start_position[2] - self.platform_size[2] / 2 - self
            .terrain_size[2] / 2]
        self.terrain_id = self.create_box(mass=0.0, half_extents=[self.
            terrain_size[0] / 2, self.terrain_size[1] / 2, self.
            terrain_size[2] / 2], position=self.terrain_position, color
            =[0.5, 0.5, 0.5, 1.0])
        self._p.changeDynamics(bodyUniqueId=self.terrain_id, linkIndex
            =-1, lateralFriction=0.8, restitution=0.5)
        self.wall_size = [self.terrain_size[0], 0.1, 1.0]
        self.wall_left_position = [self.terrain_position[0], self.
            terrain_position[1] - self.terrain_size[1] / 2 - self.
            wall_size[1] / 2, self.terrain_position[2] + self.
            terrain_size[2] / 2 + self.wall_size[2] / 2]
```

```python
        self.wall_right_position = [self.terrain_position[0], self.
            terrain_position[1] + self.terrain_size[1] / 2 + self.
            wall_size[1] / 2, self.terrain_position[2] + self.
            terrain_size[2] / 2 + self.wall_size[2] / 2]
        self.wall_left_id = self.create_box(mass=0.0, half_extents=[self.
            wall_size[0] / 2, self.wall_size[1] / 2, self.wall_size[2] /
            2], position=self.wall_left_position, color=[0.5, 0.5, 0.5,
            1.0])
        self.wall_right_id = self.create_box(mass=0.0, half_extents=[self
            .wall_size[0] / 2, self.wall_size[1] / 2, self.wall_size[2] /
             2], position=self.wall_right_position, color=[0.5, 0.5, 0.5,
             1.0])
        self.num_obstacles = 10
        self.obstacle_radius = 0.5
        self.obstacle_height = 1.0
        self.tree_height = 3.0
        self.obstacle_ids = []
        for _ in range(self.num_obstacles):
            while True:
                obstacle_position = [np.random.uniform(self.
                    platform_start_position[0] + self.platform_size[0] /
                    2 + self.obstacle_radius, self.platform_end_position
                    [0] − self.platform_size[0] / 2 − self.
                    obstacle_radius), np.random.uniform(self.
                    terrain_position[1] − self.terrain_size[1] / 2 + self
                    .obstacle_radius, self.terrain_position[1] + self.
                    terrain_size[1] / 2 − self.obstacle_radius), self.
                    terrain_position[2] + self.terrain_size[2] / 2 + self
                    .obstacle_height / 2]
                if all((np.linalg.norm(np.array(obstacle_position[:2]) −
                    np.array(other_obstacle_position[:2])) > 2 ∗ self.
                    obstacle_radius + 1.0 for other_obstacle_position in
                    [self.get_object_position(obstacle_id)[:2] for
                    obstacle_id in self.obstacle_ids])):
                    break
            if np.random.rand() < 0.5:
                obstacle_id = self.create_cylinder(mass=0.0, radius=self.
                    obstacle_radius, height=self.obstacle_height,
                    position=obstacle_position, orientation=[0.0, 0.0,
                    0.0, 1.0], color=[0.5, 0.0, 0.0, 1.0])
            else:
                obstacle_id = self.create_cylinder(mass=0.0, radius=self.
                    obstacle_radius, height=self.tree_height, position=
                    obstacle_position, orientation=[0.0, 0.0, 0.0, 1.0],
                    color=[0.0, 0.5, 0.0, 1.0])
            self.obstacle_ids.append(obstacle_id)

    def create_box(self, mass, half_extents, position, color):
        collision_shape_id = self._p.createCollisionShape(shapeType=self.
            _p.GEOM_BOX, halfExtents=half_extents)
        visual_shape_id = self._p.createVisualShape(shapeType=self._p.
            GEOM_BOX, halfExtents=half_extents, rgbaColor=color)
        return self._p.createMultiBody(baseMass=mass,
            baseCollisionShapeIndex=collision_shape_id,
            baseVisualShapeIndex=visual_shape_id, basePosition=position)

    def create_cylinder(self, mass, radius, height, position, orientation
        , color):
        collision_shape_id = self._p.createCollisionShape(shapeType=self.
            _p.GEOM_CYLINDER, radius=radius, height=height)
        visual_shape_id = self._p.createVisualShape(shapeType=self._p.
            GEOM_CYLINDER, radius=radius, length=height, rgbaColor=color)
        return self._p.createMultiBody(baseMass=mass,
            baseCollisionShapeIndex=collision_shape_id,
```

```python
            baseVisualShapeIndex=visual_shape_id, basePosition=position,
            baseOrientation=orientation)

    def get_object_position(self, object_id):
        return np.asarray(self._p.getBasePositionAndOrientation(object_id
            )[0])

    def get_distance_to_object(self, object_id):
        object_position = self.get_object_position(object_id)
        robot_position = self.robot.links['base'].position
        return np.linalg.norm(object_position[:2] - robot_position[:2])

    def reset(self):
        observation = super().reset()
        self._p.resetBasePositionAndOrientation(self.robot.robot_id, [
            self.platform_start_position[0], self.platform_start_position
            [1], self.platform_start_position[2] + self.platform_size[2]
            / 2 + self.robot.links['base'].position_init[2]], self.robot.
            links['base'].orientation_init)
        return observation

    def step(self, action):
        self.distance_to_platform_end = self.get_distance_to_object(self.
            platform_end_id)
        observation, reward, terminated, truncated, info = super().step(
            action)
        return (observation, reward, terminated, truncated, info)

    def get_task_rewards(self, action):
        new_distance_to_platform_end = self.get_distance_to_object(self.
            platform_end_id)
        survival = 1.0
        reach_platform_end = (self.distance_to_platform_end -
            new_distance_to_platform_end) / self.dt
        collision = -1.0 if any((len(self._p.getContactPoints(bodyA=self.
            robot.robot_id, bodyB=obstacle_id)) > 0 for obstacle_id in
            self.obstacle_ids)) else 0.0
        return {'survival': survival, 'reach_platform_end':
            reach_platform_end, 'collision': collision}

    def get_terminated(self, action):
        is_fall_off = self.robot.links['base'].position[2] < self.
            terrain_position[2]
        is_collision = any((len(self._p.getContactPoints(bodyA=self.robot
            .robot_id, bodyB=obstacle_id)) > 0 for obstacle_id in self.
            obstacle_ids))
        return is_fall_off or is_collision

    def get_success(self):
        is_on_platform_end = self.get_distance_to_object(self.
            platform_end_id) < self.platform_size[0] / 2
        return is_on_platform_end
```

**Task 13**

```python
import numpy as np
from oped.envs.r2d2.base import R2D2Env

class Env(R2D2Env):
    """
    Navigate a terrain with obstacles to reach a target zone.

    Description:
    - The environment consists of a start platform (3 m x 3 m x 0.5 m)
        and an end platform (3 m x 3 m x 0.5 m) placed 30 meters apart.
```

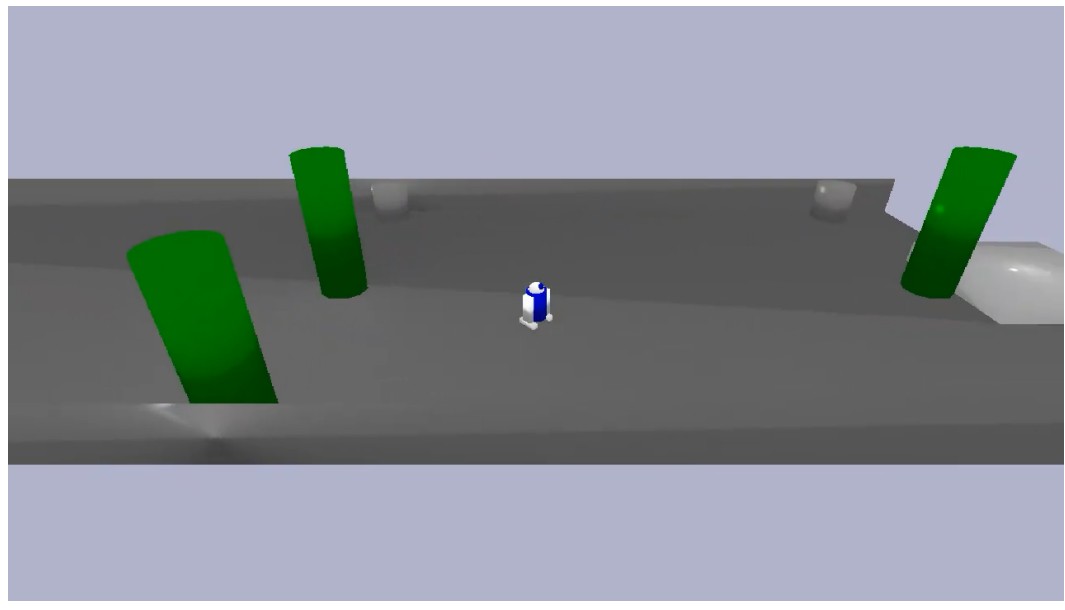

Figure 19: Task 13 of the OMNI-EPIC run presented in Section 5.

```
    — The terrain between the platforms is filled with static obstacles
        such as rocks (cylinders) and trees (cylinders with a height of 3
        meters).
    — The obstacles are placed in a grid pattern with enough space
        between them to ensure a feasible path.
    — The robot is initialized on the start platform, facing the end
        platform.

    Success:
    The task is successfully completed when the robot reaches the end
        platform.

    Rewards:
    — The robot receives a reward for each time step it remains on the
        platforms or the terrain, encouraging steady progress.
    — The robot is rewarded based on how much it reduces the distance to
        the end platform, incentivizing swift movement towards the goal.
    — The robot is penalized for collisions with obstacles.

    Termination:
    The task terminates if the robot falls off the start platform, the
        terrain, or the end platform.
    """

    def __init__(self):
        super().__init__()
        self.platform_size = [3.0, 3.0, 0.5]
        self.platform_start_position = [0.0, 0.0, 0.0]
        self.platform_end_position = [self.platform_start_position[0] +
            30.0, self.platform_start_position[1], self.
            platform_start_position[2]]
        self.platform_start_id = self.create_box(mass=0.0, half_extents=[
            self.platform_size[0] / 2, self.platform_size[1] / 2, self.
            platform_size[2] / 2], position=self.platform_start_position,
             color=[0.8, 0.8, 0.8, 1.0])
        self.platform_end_id = self.create_box(mass=0.0, half_extents=[
            self.platform_size[0] / 2, self.platform_size[1] / 2, self.
```

```
                    platform_size[2] / 2], position=self.platform_end_position,
                    color=[0.8, 0.8, 0.8, 1.0])
            self._p.changeDynamics(bodyUniqueId=self.platform_start_id,
                    linkIndex=-1, lateralFriction=0.8, restitution=0.5)
            self._p.changeDynamics(bodyUniqueId=self.platform_end_id,
                    linkIndex=-1, lateralFriction=0.8, restitution=0.5)
            self.terrain_size = [self.platform_end_position[0] - self.
                    platform_start_position[0], 10.0, 0.1]
            self.terrain_position = [self.platform_start_position[0] + self.
                    terrain_size[0] / 2, self.platform_start_position[1], self.
                    platform_start_position[2] - self.platform_size[2] / 2 - self
                    .terrain_size[2] / 2]
            self.terrain_id = self.create_box(mass=0.0, half_extents=[self.
                    terrain_size[0] / 2, self.terrain_size[1] / 2, self.
                    terrain_size[2] / 2], position=self.terrain_position, color
                    =[0.5, 0.5, 0.5, 1.0])
            self._p.changeDynamics(bodyUniqueId=self.terrain_id, linkIndex
                    =-1, lateralFriction=0.8, restitution=0.5)
            self.wall_size = [self.terrain_size[0], 0.1, 1.0]
            self.wall_left_position = [self.terrain_position[0], self.
                    terrain_position[1] - self.terrain_size[1] / 2 - self.
                    wall_size[1] / 2, self.terrain_position[2] + self.
                    terrain_size[2] / 2 + self.wall_size[2] / 2]
            self.wall_right_position = [self.terrain_position[0], self.
                    terrain_position[1] + self.terrain_size[1] / 2 + self.
                    wall_size[1] / 2, self.terrain_position[2] + self.
                    terrain_size[2] / 2 + self.wall_size[2] / 2]
            self.wall_left_id = self.create_box(mass=0.0, half_extents=[self.
                    wall_size[0] / 2, self.wall_size[1] / 2, self.wall_size[2] /
                    2], position=self.wall_left_position, color=[0.5, 0.5, 0.5,
                    1.0])
            self.wall_right_id = self.create_box(mass=0.0, half_extents=[self
                    .wall_size[0] / 2, self.wall_size[1] / 2, self.wall_size[2] /
                     2], position=self.wall_right_position, color=[0.5, 0.5, 0.5,
                     1.0])
            self.num_obstacles = 10
            self.obstacle_radius = 0.5
            self.obstacle_height = 1.0
            self.tree_height = 3.0
            self.obstacle_ids = []
            self.place_obstacles()

    def create_box(self, mass, half_extents, position, color):
        collision_shape_id = self._p.createCollisionShape(shapeType=self.
                _p.GEOM_BOX, halfExtents=half_extents)
        visual_shape_id = self._p.createVisualShape(shapeType=self._p.
                GEOM_BOX, halfExtents=half_extents, rgbaColor=color)
        return self._p.createMultiBody(baseMass=mass,
                baseCollisionShapeIndex=collision_shape_id,
                baseVisualShapeIndex=visual_shape_id, basePosition=position)

    def create_cylinder(self, mass, radius, height, position, orientation
        , color):
        collision_shape_id = self._p.createCollisionShape(shapeType=self.
                _p.GEOM_CYLINDER, radius=radius, height=height)
        visual_shape_id = self._p.createVisualShape(shapeType=self._p.
                GEOM_CYLINDER, radius=radius, length=height, rgbaColor=color)
        return self._p.createMultiBody(baseMass=mass,
                baseCollisionShapeIndex=collision_shape_id,
                baseVisualShapeIndex=visual_shape_id, basePosition=position,
                baseOrientation=orientation)

    def get_object_position(self, object_id):
        return np.asarray(self._p.getBasePositionAndOrientation(object_id
                )[0])
```

```python
    def get_distance_to_object(self, object_id):
        object_position = self.get_object_position(object_id)
        robot_position = self.robot.links['base'].position
        return np.linalg.norm(object_position[:2] - robot_position[:2])

    def place_obstacles(self):
        grid_size = int(np.sqrt(self.num_obstacles))
        x_positions = np.linspace(self.platform_start_position[0] + self.
            platform_size[0] / 2 + self.obstacle_radius, self.
            platform_end_position[0] - self.platform_size[0] / 2 - self.
            obstacle_radius, grid_size)
        y_positions = np.linspace(self.terrain_position[1] - self.
            terrain_size[1] / 2 + self.obstacle_radius, self.
            terrain_position[1] + self.terrain_size[1] / 2 - self.
            obstacle_radius, grid_size)
        for x in x_positions:
            for y in y_positions:
                if np.random.rand() < 0.5:
                    obstacle_id = self.create_cylinder(mass=0.0, radius=
                        self.obstacle_radius, height=self.obstacle_height
                        , position=[x, y, self.terrain_position[2] + self
                        .terrain_size[2] / 2 + self.obstacle_height / 2],
                         orientation=[0.0, 0.0, 0.0, 1.0], color=[0.5,
                        0.5, 0.5, 1.0])
                else:
                    obstacle_id = self.create_cylinder(mass=0.0, radius=
                        self.obstacle_radius, height=self.tree_height,
                        position=[x, y, self.terrain_position[2] + self.
                        terrain_size[2] / 2 + self.tree_height / 2],
                        orientation=[0.0, 0.0, 0.0, 1.0], color=[0.0,
                        0.5, 0.0, 1.0])
                self.obstacle_ids.append(obstacle_id)

    def reset(self):
        observation = super().reset()
        self._p.resetBasePositionAndOrientation(self.robot.robot_id, [
            self.platform_start_position[0], self.platform_start_position
            [1], self.platform_start_position[2] + self.platform_size[2]
            / 2 + self.robot.links['base'].position_init[2]], self.robot.
            links['base'].orientation_init)
        return observation

    def step(self, action):
        self.distance_to_platform_end = self.get_distance_to_object(self.
            platform_end_id)
        observation, reward, terminated, truncated, info = super().step(
            action)
        return (observation, reward, terminated, truncated, info)

    def get_task_rewards(self, action):
        new_distance_to_platform_end = self.get_distance_to_object(self.
            platform_end_id)
        survival = 1.0
        reach_platform_end = (self.distance_to_platform_end -
            new_distance_to_platform_end) / self.dt
        collision = -1.0 if any((len(self._p.getContactPoints(bodyA=self.
            robot.robot_id, bodyB=obstacle_id)) > 0 for obstacle_id in
            self.obstacle_ids)) else 0.0
        return {'survival': survival, 'reach_platform_end':
            reach_platform_end, 'collision': collision}

    def get_terminated(self, action):
        is_fall_off = self.robot.links['base'].position[2] < self.
            terrain_position[2]
```

```
        is_collision = any((len(self._p.getContactPoints(bodyA=self.robot
            .robot_id, bodyB=obstacle_id)) > 0 for obstacle_id in self.
            obstacle_ids))
        return is_fall_off or is_collision

    def get_success(self):
        is_on_platform_end = self.get_distance_to_object(self.
            platform_end_id) < self.platform_size[0] / 2
        return is_on_platform_end
```

**Task 14**

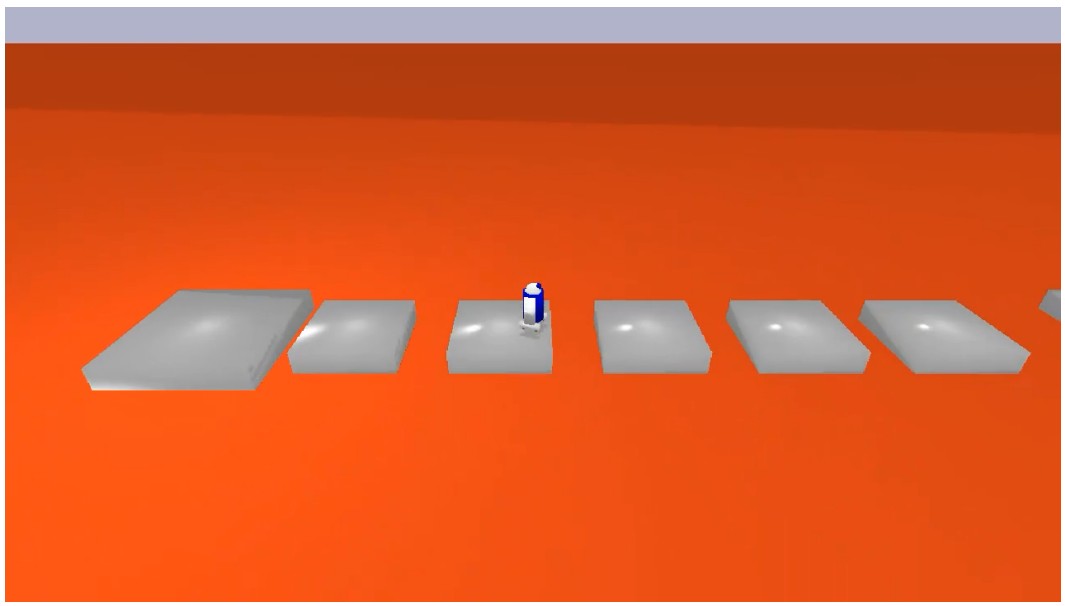

Figure 20: Task 14 of the OMNI-EPIC run presented in Section 5.

```
import numpy as np
from oped.envs.r2d2.base import R2D2Env

class Env(R2D2Env):
    """
    Jump from static platform to static platform to cross lava.

    Description:
    - The environment consists of a start platform (3 m x 3 m x 0.5 m)
        and an end platform (3 m x 3 m x 0.5 m) placed 20 meters apart.
    - The two platforms are connected by a series of 5 static platforms
        (2 m x 2 m x 0.5 m) placed 3 meters apart.
    - The platforms are placed above a lava surface, and the robot must
        jump from one static platform to the next to reach the end
        platform.

    Success:
    The task is successfully completed when the robot reaches the end
        platform.

    Rewards:
    - The robot receives a reward for each time step it remains on the
        platforms, encouraging steady progress.
    - The robot is rewarded based on how much it reduces the distance to
        the end platform, incentivizing swift movement towards the goal.
    - The robot is penalized for falling into the lava.
```

```
    Termination:
    The task terminates immediately if the robot falls into the lava or
        reaches the end platform.
    """

    def __init__(self):
        super().__init__()
        self.lava_size = [1000.0, 1000.0, 10.0]
        self.lava_position = [0.0, 0.0, 0.0]
        self.lava_id = self.create_box(mass=0.0, half_extents=[self.
            lava_size[0] / 2, self.lava_size[1] / 2, self.lava_size[2] /
            2], position=self.lava_position, color=[1.0, 0.3, 0.1, 1.0])
        self.platform_start_size = [3.0, 3.0, 0.5]
        self.platform_start_position = [0.0, 0.0, self.lava_position[2] +
             self.lava_size[2] / 2 + self.platform_start_size[2] / 2]
        self.platform_start_id = self.create_box(mass=0.0, half_extents=[
            self.platform_start_size[0] / 2, self.platform_start_size[1]
            / 2, self.platform_start_size[2] / 2], position=self.
            platform_start_position, color=[0.8, 0.8, 0.8, 1.0])
        self.platform_end_size = [3.0, 3.0, 0.5]
        self.platform_end_position = [self.platform_start_position[0] +
            20.0, self.platform_start_position[1], self.
            platform_start_position[2]]
        self.platform_end_id = self.create_box(mass=0.0, half_extents=[
            self.platform_end_size[0] / 2, self.platform_end_size[1] / 2,
             self.platform_end_size[2] / 2], position=self.
            platform_end_position, color=[0.8, 0.8, 0.8, 1.0])
        self.num_static_platforms = 5
        self.static_platform_size = [2.0, 2.0, 0.5]
        self.static_platform_ids = []
        for i in range(self.num_static_platforms):
            static_platform_position = [self.platform_start_position[0] +
                 (i + 1) * 3.0, self.platform_start_position[1], self.
                platform_start_position[2]]
            static_platform_id = self.create_box(mass=0.0, half_extents=[
                self.static_platform_size[0] / 2, self.
                static_platform_size[1] / 2, self.static_platform_size[2]
                 / 2], position=static_platform_position, color=[0.8,
                0.8, 0.8, 1.0])
            self.static_platform_ids.append(static_platform_id)

    def create_box(self, mass, half_extents, position, color):
        collision_shape_id = self._p.createCollisionShape(shapeType=self.
            _p.GEOM_BOX, halfExtents=half_extents)
        visual_shape_id = self._p.createVisualShape(shapeType=self._p.
            GEOM_BOX, halfExtents=half_extents, rgbaColor=color)
        return self._p.createMultiBody(baseMass=mass,
            baseCollisionShapeIndex=collision_shape_id,
            baseVisualShapeIndex=visual_shape_id, basePosition=position)

    def get_object_position(self, object_id):
        return np.asarray(self._p.getBasePositionAndOrientation(object_id
            )[0])

    def get_distance_to_object(self, object_id):
        object_position = self.get_object_position(object_id)
        robot_position = self.robot.links['base'].position
        return np.linalg.norm(object_position[:2] - robot_position[:2])

    def reset(self):
        observation = super().reset()
        self._p.resetBasePositionAndOrientation(self.robot.robot_id, [
            self.platform_start_position[0], self.platform_start_position
            [1], self.platform_start_position[2] + self.
```

```python
            platform_start_size[2] / 2 + self.robot.links['base'].
                position_init[2]], self.robot.links['base'].orientation_init)
        return observation

    def step(self, action):
        self.distance_to_platform_end = self.get_distance_to_object(self.
            platform_end_id)
        observation, reward, terminated, truncated, info = super().step(
            action)
        return (observation, reward, terminated, truncated, info)

    def get_task_rewards(self, action):
        new_distance_to_platform_end = self.get_distance_to_object(self.
            platform_end_id)
        on_platform = 1.0 if self.robot.links['base'].position[2] > self.
            platform_start_position[2] else −1.0
        reach_platform_end = (self.distance_to_platform_end −
            new_distance_to_platform_end) / self.dt
        return {'on_platform': on_platform, 'reach_platform_end':
            reach_platform_end}

    def get_terminated(self, action):
        is_in_lava = self.robot.links['base'].position[2] < self.
            lava_position[2] + self.lava_size[2] / 2
        is_on_platform_end = self.get_distance_to_object(self.
            platform_end_id) < self.platform_end_size[0] / 2
        return is_in_lava or is_on_platform_end

    def get_success(self):
        is_on_platform_end = self.get_distance_to_object(self.
            platform_end_id) < self.platform_end_size[0] / 2
        return is_on_platform_end
```

**Task 15**

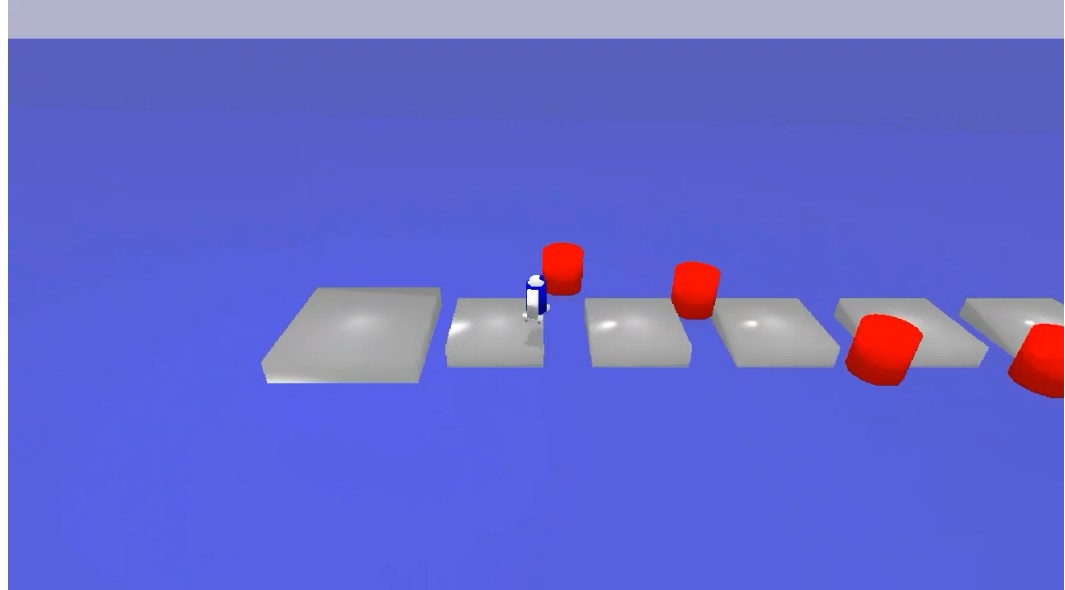

Figure 21: Task 15 of the OMNI-EPIC run presented in Section 5.

```python
import numpy as np
from oped.envs.r2d2.base import R2D2Env
```

```python
class Env(R2D2Env):
    """
    Jump from static platform to static platform while avoiding moving
        obstacles.

    Description:
    - The environment consists of a start platform (3 m x 3 m x 0.5 m)
        and an end platform (3 m x 3 m x 0.5 m) placed 20 meters apart.
    - The two platforms are connected by a series of 5 static platforms
        (2 m x 2 m x 0.5 m) placed 3 meters apart.
    - Between these static platforms, moving obstacles (cylinders with a
        radius of 0.5 m and height of 1 m) move along the y-axis with a
        sinusoidal motion. The amplitude of the motion is 2 meters, and
        the period is 4 seconds. The motion of each obstacle is offset by
        0.8 seconds from the previous one.
    - The platforms and obstacles are placed above a water surface, and
        the robot must jump from one static platform to the next while
        avoiding the moving obstacles to reach the end platform.

    Success:
    The task is successfully completed when the robot reaches the end
        platform.

    Rewards:
    - The robot receives a reward for each time step it remains on the
        platforms, encouraging steady progress.
    - The robot is rewarded based on how much it reduces the distance to
        the end platform, incentivizing swift movement towards the goal.
    - The robot is penalized for collisions with moving obstacles.

    Termination:
    The task terminates immediately if the robot falls into the water,
        collides with an obstacle, or reaches the end platform.
    """

    def __init__(self):
        super().__init__()
        self.water_size = [1000.0, 1000.0, 10.0]
        self.water_position = [0.0, 0.0, 0.0]
        self.water_id = self.create_box(mass=0.0, half_extents=[self.
            water_size[0] / 2, self.water_size[1] / 2, self.water_size[2]
             / 2], position=self.water_position, color=[0.0, 0.0, 1.0,
            0.5])
        self.platform_start_size = [3.0, 3.0, 0.5]
        self.platform_start_position = [0.0, 0.0, self.water_position[2]
            + self.water_size[2] / 2 + self.platform_start_size[2] / 2]
        self.platform_start_id = self.create_box(mass=0.0, half_extents=[
            self.platform_start_size[0] / 2, self.platform_start_size[1]
            / 2, self.platform_start_size[2] / 2], position=self.
            platform_start_position, color=[0.8, 0.8, 0.8, 1.0])
        self.platform_end_size = [3.0, 3.0, 0.5]
        self.platform_end_position = [self.platform_start_position[0] +
            20.0, self.platform_start_position[1], self.
            platform_start_position[2]]
        self.platform_end_id = self.create_box(mass=0.0, half_extents=[
            self.platform_end_size[0] / 2, self.platform_end_size[1] / 2,
             self.platform_end_size[2] / 2], position=self.
            platform_end_position, color=[0.8, 0.8, 0.8, 1.0])
        self.num_static_platforms = 5
        self.static_platform_size = [2.0, 2.0, 0.5]
        self.static_platform_ids = []
        for i in range(self.num_static_platforms):
            static_platform_position = [self.platform_start_position[0] +
                (i + 1) * 3.0, self.platform_start_position[1], self.
                platform_start_position[2]]
```

```python
            static_platform_id = self.create_box(mass=0.0, half_extents=[
                self.static_platform_size[0] / 2, self.
                static_platform_size[1] / 2, self.static_platform_size[2]
                 / 2], position=static_platform_position, color=[0.8,
                0.8, 0.8, 1.0])
            self.static_platform_ids.append(static_platform_id)
        self.num_moving_obstacles = 4
        self.moving_obstacle_radius = 0.5
        self.moving_obstacle_height = 1.0
        self.moving_obstacle_amplitude = 2.0
        self.moving_obstacle_period = 4.0
        self.moving_obstacle_phase_offset = 0.8
        self.moving_obstacle_ids = []
        for i in range(self.num_moving_obstacles):
            moving_obstacle_position = [self.platform_start_position[0] +
                (i + 1) * 3.0 + 1.5, self.platform_start_position[1],
                self.platform_start_position[2] + self.
                moving_obstacle_height / 2]
            moving_obstacle_id = self.create_cylinder(mass=0.0, radius=
                self.moving_obstacle_radius, height=self.
                moving_obstacle_height, position=moving_obstacle_position
                , color=[1.0, 0.0, 0.0, 1.0])
            self.moving_obstacle_ids.append(moving_obstacle_id)

    def create_box(self, mass, half_extents, position, color):
        collision_shape_id = self._p.createCollisionShape(shapeType=self.
            _p.GEOM_BOX, halfExtents=half_extents)
        visual_shape_id = self._p.createVisualShape(shapeType=self._p.
            GEOM_BOX, halfExtents=half_extents, rgbaColor=color)
        return self._p.createMultiBody(baseMass=mass,
            baseCollisionShapeIndex=collision_shape_id,
            baseVisualShapeIndex=visual_shape_id, basePosition=position)

    def create_cylinder(self, mass, radius, height, position, color):
        collision_shape_id = self._p.createCollisionShape(shapeType=self.
            _p.GEOM_CYLINDER, radius=radius, height=height)
        visual_shape_id = self._p.createVisualShape(shapeType=self._p.
            GEOM_CYLINDER, radius=radius, length=height, rgbaColor=color)
        return self._p.createMultiBody(baseMass=mass,
            baseCollisionShapeIndex=collision_shape_id,
            baseVisualShapeIndex=visual_shape_id, basePosition=position)

    def get_object_position(self, object_id):
        return np.asarray(self._p.getBasePositionAndOrientation(object_id
            )[0])

    def get_distance_to_object(self, object_id):
        object_position = self.get_object_position(object_id)
        robot_position = self.robot.links['base'].position
        return np.linalg.norm(object_position[:2] - robot_position[:2])

    def reset(self):
        observation = super().reset()
        self.time = 0.0
        self._p.resetBasePositionAndOrientation(self.robot.robot_id, [
            self.platform_start_position[0], self.platform_start_position
            [1], self.platform_start_position[2] + self.
            platform_start_size[2] / 2 + self.robot.links['base'].
            position_init[2]], self.robot.links['base'].orientation_init)
        return observation

    def step(self, action):
        self.distance_to_platform_end = self.get_distance_to_object(self.
            platform_end_id)
```

```python
        observation, reward, terminated, truncated, info = super().step(
            action)
        self.time += self.dt
        for i, moving_obstacle_id in enumerate(self.moving_obstacle_ids):
            moving_obstacle_position = self.get_object_position(
                moving_obstacle_id)
            new_moving_obstacle_position = [moving_obstacle_position[0],
                self.platform_start_position[1] + self.
                moving_obstacle_amplitude * np.sin(2 * np.pi * (self.time
                 + i * self.moving_obstacle_phase_offset) / self.
                moving_obstacle_period), moving_obstacle_position[2]]
            self._p.resetBasePositionAndOrientation(moving_obstacle_id,
                new_moving_obstacle_position, [0.0, 0.0, 0.0, 1.0])
        return (observation, reward, terminated, truncated, info)

    def get_task_rewards(self, action):
        new_distance_to_platform_end = self.get_distance_to_object(self.
            platform_end_id)
        on_platform = 1.0 if self.robot.links['base'].position[2] > self.
            platform_start_position[2] else -1.0
        reach_platform_end = (self.distance_to_platform_end -
            new_distance_to_platform_end) / self.dt
        collision_with_obstacle = -1.0 if len(self._p.getContactPoints(
            bodyA=self.robot.robot_id, bodyB=self.moving_obstacle_ids[0])
            ) > 0 or len(self._p.getContactPoints(bodyA=self.robot.
            robot_id, bodyB=self.moving_obstacle_ids[1])) > 0 or len(self
            ._p.getContactPoints(bodyA=self.robot.robot_id, bodyB=self.
            moving_obstacle_ids[2])) > 0 or (len(self._p.getContactPoints
            (bodyA=self.robot.robot_id, bodyB=self.moving_obstacle_ids
            [3])) > 0) else 0.0
        return {'on_platform': on_platform, 'reach_platform_end':
            reach_platform_end, 'collision_with_obstacle':
            collision_with_obstacle}

    def get_terminated(self, action):
        is_in_water = self.robot.links['base'].position[2] < self.
            water_position[2] + self.water_size[2] / 2
        is_on_platform_end = self.get_distance_to_object(self.
            platform_end_id) < self.platform_end_size[0] / 2
        collision_with_obstacle = len(self._p.getContactPoints(bodyA=self
            .robot.robot_id, bodyB=self.moving_obstacle_ids[0])) > 0 or
            len(self._p.getContactPoints(bodyA=self.robot.robot_id, bodyB
            =self.moving_obstacle_ids[1])) > 0 or len(self._p.
            getContactPoints(bodyA=self.robot.robot_id, bodyB=self.
            moving_obstacle_ids[2])) > 0 or (len(self._p.getContactPoints
            (bodyA=self.robot.robot_id, bodyB=self.moving_obstacle_ids
            [3])) > 0)
        return is_in_water or is_on_platform_end or
            collision_with_obstacle

    def get_success(self):
        is_on_platform_end = self.get_distance_to_object(self.
            platform_end_id) < self.platform_end_size[0] / 2
        return is_on_platform_end
```

**Task 16**

```python
import numpy as np
from oped.envs.r2d2.base import R2D2Env

class Env(R2D2Env):
    """
    Task: Jump over rolling logs on a bridge

    Description:
```

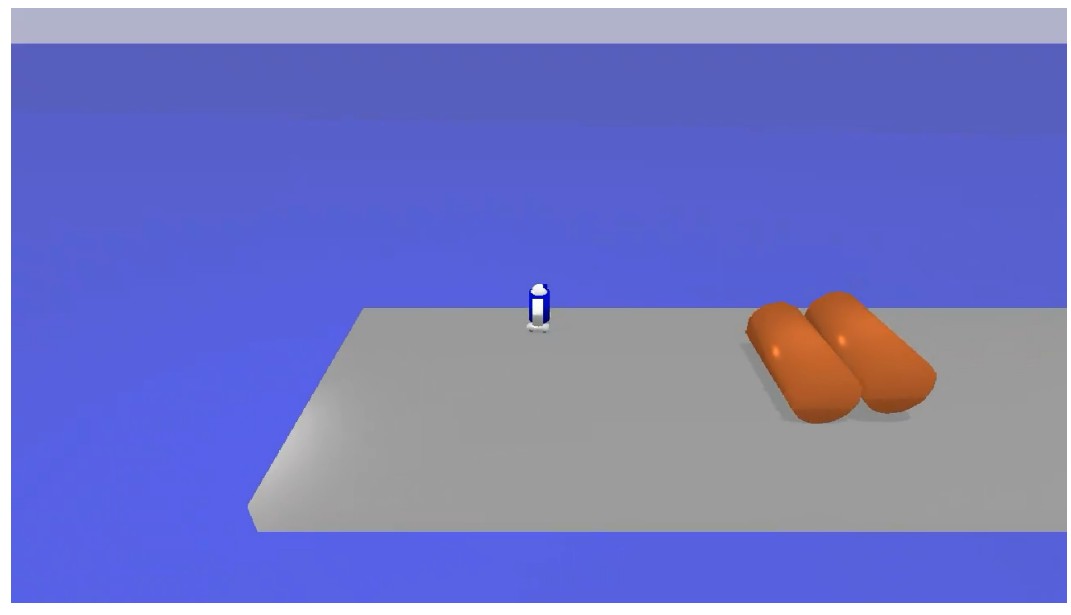

Figure 22: Task 16 of the OMNI-EPIC run presented in Section 5.

```
— The environment consists of a wide static bridge that is 50 meters
    long and 5 meters wide, elevated 10 meters above a water surface.
— The robot starts at the beginning of the bridge, facing the
    positive x-axis.
— Logs (cylinders) with a radius of 0.5 meters are spawned 2 meters
    above the bridge and 10 meters ahead of the robot, rolling
    towards the robot with an initial velocity of 5 m/s along the
    negative x-axis.
— The robot must move forward on the bridge while jumping over the
    rolling logs to reach the end of the bridge.

Success:
The task is successfully completed when the robot reaches the end of
    the bridge.

Rewards:
— The robot receives a survival reward for each time step it remains
    on the bridge, encouraging steady progress.
— The robot is rewarded based on how much it reduces the distance to
    the end of the bridge, incentivizing swift movement towards the
    goal.
— The robot is penalized for colliding with the logs.

Termination:
The task terminates immediately if the robot falls off the bridge,
    collides with a log, or reaches the end of the bridge.
"""

def __init__(self):
    super().__init__()
    self.water_size = [1000.0, 1000.0, 10.0]
    self.water_position = [0.0, 0.0, 0.0]
    self.water_id = self.create_box(mass=0.0, half_extents=[self.
        water_size[0] / 2, self.water_size[1] / 2, self.water_size[2]
         / 2], position=self.water_position, color=[0.0, 0.0, 1.0,
        0.5])
    self.bridge_length = 50.0
    self.bridge_width = 5.0
```

```python
        self.bridge_height = 0.5
        self.bridge_position = [self.water_position[0] + self.
            bridge_length / 2, self.water_position[1], self.
            water_position[2] + self.water_size[2] / 2 + self.
            bridge_height / 2]
        self.bridge_id = self.create_box(mass=0.0, half_extents=[self.
            bridge_length / 2, self.bridge_width / 2, self.bridge_height
            / 2], position=self.bridge_position, color=[0.8, 0.8, 0.8,
            1.0])
        self.log_radius = 0.5
        self.log_height = 2.0
        self.log_spawn_distance = 10.0
        self.log_spawn_height = 2.0
        self.log_velocity = -5.0
        self.log_ids = []
        self.robot_position_init = [self.bridge_position[0] - self.
            bridge_length / 2 + 1.0, self.bridge_position[1], self.
            bridge_position[2] + self.bridge_height / 2 + self.robot.
            links['base'].position_init[2]]
        self.robot_orientation_init = self._p.getQuaternionFromEuler
            ([0.0, 0.0, 0.0])

    def create_box(self, mass, half_extents, position, color):
        collision_shape_id = self._p.createCollisionShape(shapeType=self.
            _p.GEOM_BOX, halfExtents=half_extents)
        visual_shape_id = self._p.createVisualShape(shapeType=self._p.
            GEOM_BOX, halfExtents=half_extents, rgbaColor=color)
        return self._p.createMultiBody(baseMass=mass,
            baseCollisionShapeIndex=collision_shape_id,
            baseVisualShapeIndex=visual_shape_id, basePosition=position)

    def create_cylinder(self, mass, radius, height, position, orientation
        , color):
        collision_shape_id = self._p.createCollisionShape(shapeType=self.
            _p.GEOM_CYLINDER, radius=radius, height=height)
        visual_shape_id = self._p.createVisualShape(shapeType=self._p.
            GEOM_CYLINDER, radius=radius, length=height, rgbaColor=color)
        return self._p.createMultiBody(baseMass=mass,
            baseCollisionShapeIndex=collision_shape_id,
            baseVisualShapeIndex=visual_shape_id, basePosition=position,
            baseOrientation=orientation)

    def get_object_position(self, object_id):
        return np.asarray(self._p.getBasePositionAndOrientation(object_id
            )[0])

    def get_distance_to_object(self, object_id):
        object_position = self.get_object_position(object_id)
        robot_position = self.robot.links['base'].position
        return np.linalg.norm(object_position[:2] - robot_position[:2])

    def reset(self):
        observation = super().reset()
        self._p.resetBasePositionAndOrientation(self.robot.robot_id, self
            .robot_position_init, self.robot_orientation_init)
        for log_id in self.log_ids:
            self._p.removeBody(log_id)
        self.log_ids = []
        return observation

    def step(self, action):
        self.distance_to_bridge_end = self.bridge_length - (self.robot.
            links['base'].position[0] - (self.bridge_position[0] - self.
            bridge_length / 2))
```

```python
        observation, reward, terminated, truncated, info = super().step(
            action)
        if np.random.rand() < 0.05:
            log_position = [self.robot.links['base'].position[0] + self.
                log_spawn_distance, self.bridge_position[1], self.
                bridge_position[2] + self.bridge_height / 2 + self.
                log_spawn_height]
            log_orientation = self._p.getQuaternionFromEuler([np.pi / 2,
                0.0, 0.0])
            log_id = self.create_cylinder(mass=10.0, radius=self.
                log_radius, height=self.log_height, position=log_position
                , orientation=log_orientation, color=[0.8, 0.4, 0.2,
                1.0])
            self._p.resetBaseVelocity(log_id, linearVelocity=[self.
                log_velocity, 0.0, 0.0])
            self.log_ids.append(log_id)
        for log_id in self.log_ids:
            log_position = self.get_object_position(log_id)
            if log_position[0] < self.bridge_position[0] - self.
                bridge_length / 2:
                self._p.removeBody(log_id)
                self.log_ids.remove(log_id)
        return (observation, reward, terminated, truncated, info)

    def get_task_rewards(self, action):
        new_distance_to_bridge_end = self.bridge_length - (self.robot.
            links['base'].position[0] - (self.bridge_position[0] - self.
            bridge_length / 2))
        survival = 1.0 if self.robot.links['base'].position[2] > self.
            bridge_position[2] else -1.0
        reach_bridge_end = (self.distance_to_bridge_end -
            new_distance_to_bridge_end) / self.dt
        collision_with_log = -1.0 if any((len(self._p.getContactPoints(
            bodyA=self.robot.robot_id, bodyB=log_id)) > 0 for log_id in
            self.log_ids)) else 0.0
        return {'survival': survival, 'reach_bridge_end':
            reach_bridge_end, 'collision_with_log': collision_with_log}

    def get_terminated(self, action):
        is_off_bridge = self.robot.links['base'].position[2] < self.
            bridge_position[2]
        is_at_bridge_end = self.robot.links['base'].position[0] > self.
            bridge_position[0] + self.bridge_length / 2
        collision_with_log = any((len(self._p.getContactPoints(bodyA=self
            .robot.robot_id, bodyB=log_id)) > 0 for log_id in self.
            log_ids))
        return is_off_bridge or is_at_bridge_end or collision_with_log

    def get_success(self):
        is_at_bridge_end = self.robot.links['base'].position[0] > self.
            bridge_position[0] + self.bridge_length / 2
        return is_at_bridge_end
```

**Task 17**

```python
import numpy as np
from oped.envs.r2d2.base import R2D2Env

class Env(R2D2Env):
    """
    Touch a lever or a button to activate a drawbridge. The robot is on a
        start platform and needs to reach a target platform 25 m away. A
        25-meter long drawbridge is on the end platform in an open (
        upright) position. A button or lever can be activated to lower
        the drawbridge to connect the two platforms.
```

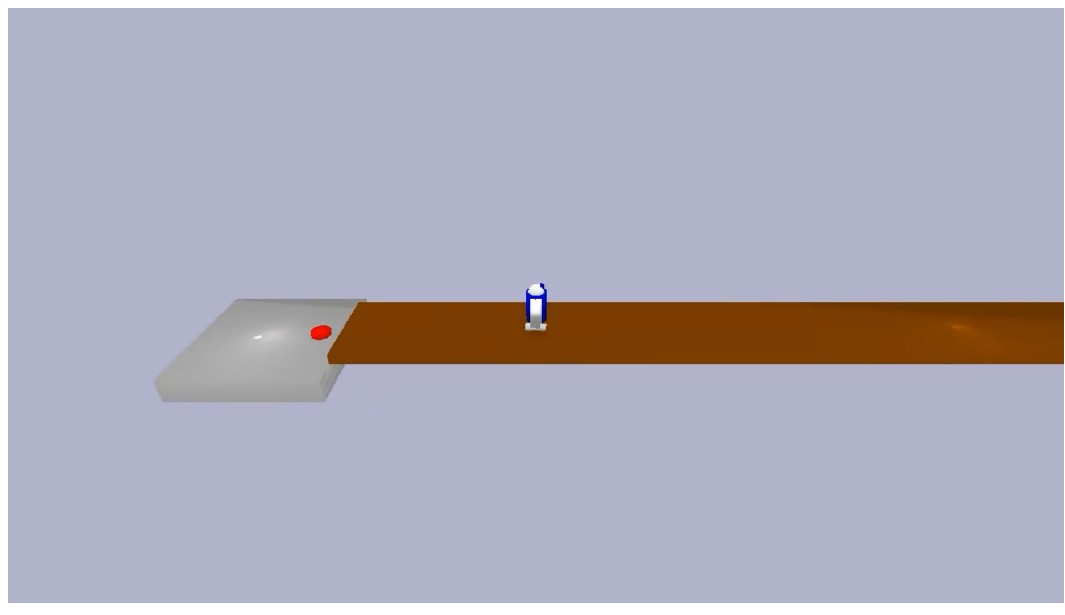

Figure 23: Task 17 of the OMNI-EPIC run presented in Section 5.

```
Description:
— The environment consists of a start platform (3 m x 3 m x 0.5 m)
    and an end platform (3 m x 3 m x 0.5 m) placed 25 meters apart.
— A drawbridge (25 m long and 2 m wide) is positioned at the edge of
    the end platform in an upright position.
— A button or lever is placed on the start platform.
— The robot must touch the button or lever to lower the drawbridge,
    allowing it to cross to the end platform.

Success:
The task is successfully completed when the robot reaches the end
    platform.

Rewards:
— The robot receives a reward for each time step it remains on the
    platforms, encouraging steady progress.
— The robot is rewarded for touching the button or lever.
— The robot is rewarded based on how much it reduces the distance to
    the end platform, incentivizing swift movement towards the goal.

Termination:
The task terminates immediately if the robot falls off the platforms
    or reaches the end platform.
"""

def __init__(self):
    super().__init__()
    self.platform_size = [3.0, 3.0, 0.5]
    self.platform_start_position = [0.0, 0.0, 0.0]
    self.platform_end_position = [self.platform_start_position[0] +
        25.0, self.platform_start_position[1], self.
        platform_start_position[2]]
    self.platform_start_id = self.create_box(mass=0.0, half_extents=[
        self.platform_size[0] / 2, self.platform_size[1] / 2, self.
        platform_size[2] / 2], position=self.platform_start_position,
         color=[0.8, 0.8, 0.8, 1.0])
```

```python
        self.platform_end_id = self.create_box(mass=0.0, half_extents=[
            self.platform_size[0] / 2, self.platform_size[1] / 2, self.
            platform_size[2] / 2], position=self.platform_end_position,
            color=[0.8, 0.8, 0.8, 1.0])
        self.drawbridge_length = 25.0
        self.drawbridge_width = 2.0
        self.drawbridge_thickness = 0.2
        self.drawbridge_position_lowered = [self.platform_start_position
            [0] + self.platform_size[0] / 2 + self.drawbridge_length / 2,
             self.platform_start_position[1], self.
            platform_start_position[2] + self.platform_size[2] / 2 + self
            .drawbridge_thickness / 2]
        self.drawbridge_position_raised = [self.platform_end_position[0]
            - self.platform_size[0] / 2, self.platform_end_position[1],
            self.platform_end_position[2] + self.platform_size[2] / 2 +
            self.drawbridge_length / 2]
        self.drawbridge_id = self.create_box(mass=0.0, half_extents=[self
            .drawbridge_length / 2, self.drawbridge_width / 2, self.
            drawbridge_thickness / 2], position=self.
            drawbridge_position_raised, color=[0.6, 0.3, 0.0, 1.0])
        self.button_radius = 0.2
        self.button_height = 0.1
        self.button_position = [self.platform_start_position[0] + self.
            platform_size[0] / 2 - 0.5, self.platform_start_position[1],
            self.platform_start_position[2] + self.platform_size[2] / 2 +
             self.button_height / 2]
        self.button_id = self.create_cylinder(mass=0.0, radius=self.
            button_radius, height=self.button_height, position=self.
            button_position, color=[1.0, 0.0, 0.0, 1.0])
        self.drawbridge_activated = False

    def create_box(self, mass, half_extents, position, color):
        collision_shape_id = self._p.createCollisionShape(shapeType=self.
            _p.GEOM_BOX, halfExtents=half_extents)
        visual_shape_id = self._p.createVisualShape(shapeType=self._p.
            GEOM_BOX, halfExtents=half_extents, rgbaColor=color)
        return self._p.createMultiBody(baseMass=mass,
            baseCollisionShapeIndex=collision_shape_id,
            baseVisualShapeIndex=visual_shape_id, basePosition=position)

    def create_cylinder(self, mass, radius, height, position, color):
        collision_shape_id = self._p.createCollisionShape(shapeType=self.
            _p.GEOM_CYLINDER, radius=radius, height=height)
        visual_shape_id = self._p.createVisualShape(shapeType=self._p.
            GEOM_CYLINDER, radius=radius, length=height, rgbaColor=color)
        return self._p.createMultiBody(baseMass=mass,
            baseCollisionShapeIndex=collision_shape_id,
            baseVisualShapeIndex=visual_shape_id, basePosition=position)

    def get_object_position(self, object_id):
        return np.asarray(self._p.getBasePositionAndOrientation(object_id
            )[0])

    def get_distance_to_object(self, object_id):
        object_position = self.get_object_position(object_id)
        robot_position = self.robot.links['base'].position
        return np.linalg.norm(object_position[:2] - robot_position[:2])

    def reset(self):
        observation = super().reset()
        self._p.resetBasePositionAndOrientation(self.robot.robot_id, [
            self.platform_start_position[0], self.platform_start_position
            [1], self.platform_start_position[2] + self.platform_size[2]
            / 2 + self.robot.links['base'].position_init[2]], self.robot.
            links['base'].orientation_init)
```

```python
        self.drawbridge_activated = False
        self._p.resetBasePositionAndOrientation(self.drawbridge_id, self.
            drawbridge_position_raised, [0.0, 0.0, 0.0, 1.0])
        return observation

    def step(self, action):
        self.distance_to_platform_end = self.get_distance_to_object(self.
            platform_end_id)
        observation, reward, terminated, truncated, info = super().step(
            action)
        if not self.drawbridge_activated and len(self._p.getContactPoints
            (bodyA=self.robot.robot_id, bodyB=self.button_id)) > 0:
            self.drawbridge_activated = True
            self._p.resetBasePositionAndOrientation(self.drawbridge_id,
                self.drawbridge_position_lowered, [0.0, 0.0, 0.0, 1.0])
        return (observation, reward, terminated, truncated, info)

    def get_task_rewards(self, action):
        new_distance_to_platform_end = self.get_distance_to_object(self.
            platform_end_id)
        on_platforms = 1.0 if self.robot.links['base'].position[2] > self
            .platform_start_position[2] + self.platform_size[2] / 2 else
            -1.0
        activate_drawbridge = 10.0 if not self.drawbridge_activated and
            len(self._p.getContactPoints(bodyA=self.robot.robot_id, bodyB
            =self.button_id)) > 0 else 0.0
        reach_platform_end = (self.distance_to_platform_end -
            new_distance_to_platform_end) / self.dt
        return {'on_platforms': on_platforms, 'activate_drawbridge':
            activate_drawbridge, 'reach_platform_end': reach_platform_end
            }

    def get_terminated(self, action):
        is_off_platforms = self.robot.links['base'].position[2] < self.
            platform_start_position[2]
        is_on_platform_end = self.get_distance_to_object(self.
            platform_end_id) < self.platform_size[0] / 2
        return is_off_platforms or is_on_platform_end

    def get_success(self):
        is_on_platform_end = self.get_distance_to_object(self.
            platform_end_id) < self.platform_size[0] / 2
        return is_on_platform_end
```

**Task 18**

```python
import numpy as np
from oped.envs.r2d2.base import R2D2Env

class Env(R2D2Env):
    """
    Task: Push a cube to a target zone on a static platform

    Description:
    - The environment consists of a large static platform (50 m x 50 m).
    - A cube (2 meters in size and 5 kg in mass) is placed at a random
        location on the platform.
    - A target zone (3 meters in radius) is also placed at a random
        location on the platform. The collision for the target zone is
        set to False.
    - The robot is initialized at a fixed position on the platform.

    The task of the robot is to push the cube to the target zone as
        quickly as possible.
```

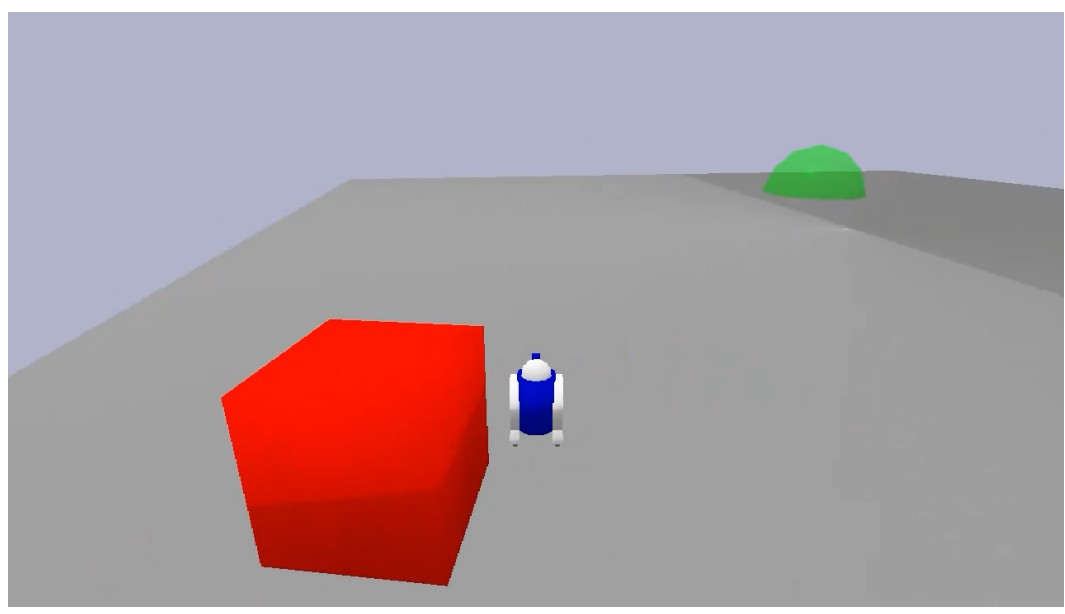

Figure 24: Task 18 of the OMNI-EPIC run presented in Section 5.

```
Success:
The task is successfully completed when the cube is entirely within
    the target zone.

Rewards:
- The robot receives a survival reward for each time step it remains
    on the platform, encouraging steady progress.
- The robot is rewarded based on how much it reduces the distance to
    the cube, incentivizing interaction with the cube.
- The robot is rewarded based on how much it reduces the distance
    between the cube and the target zone, incentivizing pushing the
    cube towards the goal.
- A large reward is given when the cube reaches the target zone.

Termination:
The episode terminates if the robot falls off the platform or if the
    cube reaches the target zone.
"""

def __init__(self):
    super().__init__()
    self.platform_size = [50.0, 50.0, 0.1]
    self.platform_position = [0.0, 0.0, 0.0]
    self.platform_id = self.create_box(mass=0.0, half_extents=[self.
        platform_size[0] / 2, self.platform_size[1] / 2, self.
        platform_size[2] / 2], position=self.platform_position, color
        =[0.8, 0.8, 0.8, 1.0])
    self._p.changeDynamics(bodyUniqueId=self.platform_id, linkIndex
        =-1, lateralFriction=0.8, restitution=0.5)
    self.cube_size = [2.0, 2.0, 2.0]
    self.cube_mass = 5.0
    self.cube_id = self.create_box(mass=self.cube_mass, half_extents
        =[self.cube_size[0] / 2, self.cube_size[1] / 2, self.
        cube_size[2] / 2], position=[0.0, 0.0, 0.0], color=[1.0, 0.0,
        0.0, 1.0])
    self.target_zone_radius = 3.0
```

```python
        self.target_zone_id = self.create_sphere(mass=0.0, radius=self.
            target_zone_radius, collision=False, position=[0.0, 0.0,
            0.0], color=[0.0, 1.0, 0.0, 0.5])
        self.robot_position_init = [0.0, 0.0, self.platform_position[2] +
            self.platform_size[2] / 2 + self.robot.links['base'].
            position_init[2]]

    def create_box(self, mass, half_extents, position, color):
        collision_shape_id = self._p.createCollisionShape(shapeType=self.
            _p.GEOM_BOX, halfExtents=half_extents)
        visual_shape_id = self._p.createVisualShape(shapeType=self._p.
            GEOM_BOX, halfExtents=half_extents, rgbaColor=color)
        return self._p.createMultiBody(baseMass=mass,
            baseCollisionShapeIndex=collision_shape_id,
            baseVisualShapeIndex=visual_shape_id, basePosition=position)

    def create_sphere(self, mass, radius, collision, position, color):
        if collision:
            collision_shape_id = self._p.createCollisionShape(shapeType=
                self._p.GEOM_SPHERE, radius=radius)
            visual_shape_id = self._p.createVisualShape(shapeType=self._p
                .GEOM_SPHERE, radius=radius, rgbaColor=color)
            return self._p.createMultiBody(baseMass=mass,
                baseCollisionShapeIndex=collision_shape_id,
                baseVisualShapeIndex=visual_shape_id, basePosition=
                position)
        else:
            visual_shape_id = self._p.createVisualShape(shapeType=self._p
                .GEOM_SPHERE, radius=radius, rgbaColor=color)
            return self._p.createMultiBody(baseMass=mass,
                baseVisualShapeIndex=visual_shape_id, basePosition=
                position)

    def get_object_position(self, object_id):
        return np.asarray(self._p.getBasePositionAndOrientation(object_id
            )[0])

    def get_distance_between_objects(self, object1_id, object2_id):
        object1_position = self.get_object_position(object1_id)
        object2_position = self.get_object_position(object2_id)
        return np.linalg.norm(object1_position[:2] - object2_position
            [:2])

    def reset(self):
        observation = super().reset()
        self._p.resetBasePositionAndOrientation(self.robot.robot_id, self
            .robot_position_init, self.robot.links['base'].
            orientation_init)
        cube_x = np.random.uniform(self.platform_position[0] - self.
            platform_size[0] / 2 + self.cube_size[0] / 2, self.
            platform_position[0] + self.platform_size[0] / 2 - self.
            cube_size[0] / 2)
        cube_y = np.random.uniform(self.platform_position[1] - self.
            platform_size[1] / 2 + self.cube_size[1] / 2, self.
            platform_position[1] + self.platform_size[1] / 2 - self.
            cube_size[1] / 2)
        self._p.resetBasePositionAndOrientation(self.cube_id, [cube_x,
            cube_y, self.platform_position[2] + self.platform_size[2] / 2
             + self.cube_size[2] / 2], [0.0, 0.0, 0.0, 1.0])
        target_zone_x = np.random.uniform(self.platform_position[0] -
            self.platform_size[0] / 2 + self.target_zone_radius, self.
            platform_position[0] + self.platform_size[0] / 2 - self.
            target_zone_radius)
        target_zone_y = np.random.uniform(self.platform_position[1] -
            self.platform_size[1] / 2 + self.target_zone_radius, self.
```

```python
                    platform_position[1] + self.platform_size[1] / 2 - self.
                        target_zone_radius)
                self._p.resetBasePositionAndOrientation(self.target_zone_id, [
                    target_zone_x, target_zone_y, self.platform_position[2] +
                    self.platform_size[2] / 2], [0.0, 0.0, 0.0, 1.0])
                return observation

    def step(self, action):
        self.distance_robot_to_cube = self.get_distance_between_objects(
            self.robot.robot_id, self.cube_id)
        self.distance_cube_to_target_zone = self.
            get_distance_between_objects(self.cube_id, self.
            target_zone_id)
        observation, reward, terminated, truncated, info = super().step(
            action)
        return (observation, reward, terminated, truncated, info)

    def get_task_rewards(self, action):
        new_distance_robot_to_cube = self.get_distance_between_objects(
            self.robot.robot_id, self.cube_id)
        new_distance_cube_to_target_zone = self.
            get_distance_between_objects(self.cube_id, self.
            target_zone_id)
        survival = 1.0
        reach_cube = (self.distance_robot_to_cube -
            new_distance_robot_to_cube) / self.dt
        push_cube_to_target_zone = (self.distance_cube_to_target_zone -
            new_distance_cube_to_target_zone) / self.dt
        cube_in_target_zone = 10.0 if new_distance_cube_to_target_zone <
            self.target_zone_radius else 0.0
        return {'survival': survival, 'reach_cube': reach_cube, '
            push_cube_to_target_zone': push_cube_to_target_zone, '
            cube_in_target_zone': cube_in_target_zone}

    def get_terminated(self, action):
        is_off_platform = self.robot.links['base'].position[2] < self.
            platform_position[2]
        is_cube_in_target_zone = self.get_distance_between_objects(self.
            cube_id, self.target_zone_id) < self.target_zone_radius
        return is_off_platform or is_cube_in_target_zone

    def get_success(self):
        is_cube_in_target_zone = self.get_distance_between_objects(self.
            cube_id, self.target_zone_id) < self.target_zone_radius
        return is_cube_in_target_zone
```

**Task 19**

```python
import numpy as np
from oped.envs.r2d2.base import R2D2Env

class Env(R2D2Env):
    """
    Push a cube into a target zone in a large open arena.

    Description:
    - The environment consists of a large flat arena measuring 50 x 50
        meters.
    - A cube with dimensions of 2 meters in size and a mass of 5 kg is
        placed at a random location within the arena.
    - A target zone with a radius of 3 meters is also randomly placed
        within the arena. The target zone has no collision, allowing the
        cube to be pushed into it without obstruction.
    - The robot is initialized at a random position within the arena,
        facing the positive x-axis.
```

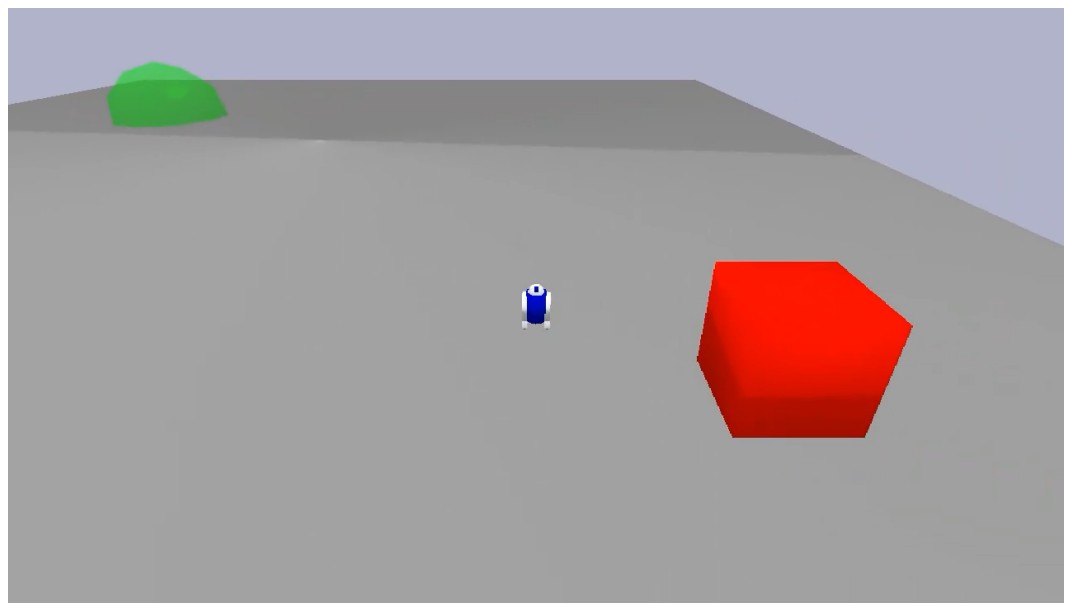

Figure 25: Task 19 of the OMNI-EPIC run presented in Section 5.

```
Success:
The task is successfully completed when the robot pushes the cube
    into the target zone.

Rewards:
— The robot receives a survival reward at each time step to encourage
    steady progress.
— The robot is rewarded for decreasing its distance to the cube,
    encouraging it to approach and interact with the cube.
— The robot is rewarded for pushing the cube towards the target zone,
    with additional rewards for getting the cube into the target
    zone.
— The robot is rewarded for remaining within the arena, avoiding
    falling off the platform.

Termination:
The episode terminates if the robot falls off the platform. The
    episode does not terminate if the cube is in the target zone.
"""

def __init__(self):
    super().__init__()
    self.arena_size = [50.0, 50.0, 0.1]
    self.arena_position = [0.0, 0.0, 0.0]
    self.arena_id = self.create_box(mass=0.0, half_extents=[self.
        arena_size[0] / 2, self.arena_size[1] / 2, self.arena_size[2]
        / 2], position=self.arena_position, color=[0.5, 0.5, 0.5,
        1.0])
    self._p.changeDynamics(bodyUniqueId=self.arena_id, linkIndex=-1,
        lateralFriction=0.8, restitution=0.5)
    self.cube_size = [2.0, 2.0, 2.0]
    self.cube_mass = 5.0
    self.cube_id = self.create_box(mass=self.cube_mass, half_extents
        =[self.cube_size[0] / 2, self.cube_size[1] / 2, self.
        cube_size[2] / 2], position=[0.0, 0.0, 0.0], color=[1.0, 0.0,
        0.0, 1.0])
    self.target_zone_radius = 3.0
```

```python
        self.target_zone_id = self.create_sphere(mass=0.0, radius=self.
            target_zone_radius, collision=False, position=[0.0, 0.0,
            0.0], color=[0.0, 1.0, 0.0, 0.5])

    def create_box(self, mass, half_extents, position, color):
        collision_shape_id = self._p.createCollisionShape(shapeType=self.
            _p.GEOM_BOX, halfExtents=half_extents)
        visual_shape_id = self._p.createVisualShape(shapeType=self._p.
            GEOM_BOX, halfExtents=half_extents, rgbaColor=color)
        return self._p.createMultiBody(baseMass=mass,
            baseCollisionShapeIndex=collision_shape_id,
            baseVisualShapeIndex=visual_shape_id, basePosition=position)

    def create_sphere(self, mass, radius, collision, position, color):
        if collision:
            collision_shape_id = self._p.createCollisionShape(shapeType=
                self._p.GEOM_SPHERE, radius=radius)
            visual_shape_id = self._p.createVisualShape(shapeType=self._p
                .GEOM_SPHERE, radius=radius, rgbaColor=color)
            return self._p.createMultiBody(baseMass=mass,
                baseCollisionShapeIndex=collision_shape_id,
                baseVisualShapeIndex=visual_shape_id, basePosition=
                position)
        else:
            visual_shape_id = self._p.createVisualShape(shapeType=self._p
                .GEOM_SPHERE, radius=radius, rgbaColor=color)
            return self._p.createMultiBody(baseMass=mass,
                baseVisualShapeIndex=visual_shape_id, basePosition=
                position)

    def get_object_position(self, object_id):
        return np.asarray(self._p.getBasePositionAndOrientation(object_id
            )[0])

    def get_distance_to_object(self, object_id):
        object_position = self.get_object_position(object_id)
        robot_position = self.robot.links['base'].position
        return np.linalg.norm(object_position[:2] − robot_position[:2])

    def reset(self):
        observation = super().reset()
        cube_x_init = np.random.uniform(low=−self.arena_size[0] / 2 +
            self.cube_size[0] / 2, high=self.arena_size[0] / 2 − self.
            cube_size[0] / 2)
        cube_y_init = np.random.uniform(low=−self.arena_size[1] / 2 +
            self.cube_size[1] / 2, high=self.arena_size[1] / 2 − self.
            cube_size[1] / 2)
        self._p.resetBasePositionAndOrientation(self.cube_id, [
            cube_x_init, cube_y_init, self.arena_position[2] + self.
            arena_size[2] / 2 + self.cube_size[2] / 2], [0.0, 0.0, 0.0,
            1.0])
        target_zone_x = np.random.uniform(low=−self.arena_size[0] / 2 +
            self.target_zone_radius, high=self.arena_size[0] / 2 − self.
            target_zone_radius)
        target_zone_y = np.random.uniform(low=−self.arena_size[1] / 2 +
            self.target_zone_radius, high=self.arena_size[1] / 2 − self.
            target_zone_radius)
        self.target_zone_position = [target_zone_x, target_zone_y, self.
            arena_position[2] + self.arena_size[2] / 2]
        self._p.resetBasePositionAndOrientation(self.target_zone_id, self
            .target_zone_position, [0.0, 0.0, 0.0, 1.0])
        robot_x_init = np.random.uniform(low=−self.arena_size[0] / 2 +
            self.robot.links['base'].position_init[0], high=self.
            arena_size[0] / 2 − self.robot.links['base'].position_init
            [0])
```

```python
        robot_y_init = np.random.uniform(low=-self.arena_size[1] / 2 +
            self.robot.links['base'].position_init[1], high=self.
            arena_size[1] / 2 - self.robot.links['base'].position_init
            [1])
        self._p.resetBasePositionAndOrientation(self.robot.robot_id, [
            robot_x_init, robot_y_init, self.arena_position[2] + self.
            arena_size[2] / 2 + self.robot.links['base'].position_init
            [2]], self.robot.links['base'].orientation_init)
        return observation

    def step(self, action):
        self.distance_to_cube = self.get_distance_to_object(self.cube_id)
        self.distance_cube_to_target_zone = self.get_distance_to_object(
            self.target_zone_id)
        self.cube_position = self.get_object_position(self.cube_id)
        observation, reward, terminated, truncated, info = super().step(
            action)
        return (observation, reward, terminated, truncated, info)

    def get_task_rewards(self, action):
        new_distance_to_cube = self.get_distance_to_object(self.cube_id)
        new_distance_cube_to_target_zone = self.get_distance_to_object(
            self.target_zone_id)
        new_cube_position = self.get_object_position(self.cube_id)
        survival = 1.0
        reach_cube = (self.distance_to_cube - new_distance_to_cube) /
            self.dt
        push_cube = (self.distance_cube_to_target_zone -
            new_distance_cube_to_target_zone) / self.dt
        if new_distance_cube_to_target_zone < self.target_zone_radius:
            push_cube += 5.0
        in_arena = 1.0 if abs(self.robot.links['base'].position[0]) <
            self.arena_size[0] / 2 and abs(self.robot.links['base'].
            position[1]) < self.arena_size[1] / 2 else -1.0
        return {'survival': survival, 'reach_cube': reach_cube, '
            push_cube': push_cube, 'in_arena': in_arena}

    def get_terminated(self, action):
        return abs(self.robot.links['base'].position[0]) > self.
            arena_size[0] / 2 or abs(self.robot.links['base'].position
            [1]) > self.arena_size[1] / 2

    def get_success(self):
        cube_distance_to_target_zone = self.get_distance_to_object(self.
            target_zone_id)
        return cube_distance_to_target_zone < self.target_zone_radius
```

**Task 21**

```python
import numpy as np
from oped.envs.r2d2.base import R2D2Env

class Env(R2D2Env):
    """
    Push a domino to start a chain reaction.

    Description:
    - The environment consists of a large platform measuring 1000 x 10 x
        0.1 meters.
    - The robot is initialized at a fixed position on the platform.
    - A domino with dimensions 0.5 x 2 x 4 meters and a mass of 5 kg is
        positioned on the platform, 5 meters away from the robot.
    - The dominos are spaced by 3 meters, and positioned to create a
        chain reaction.
```

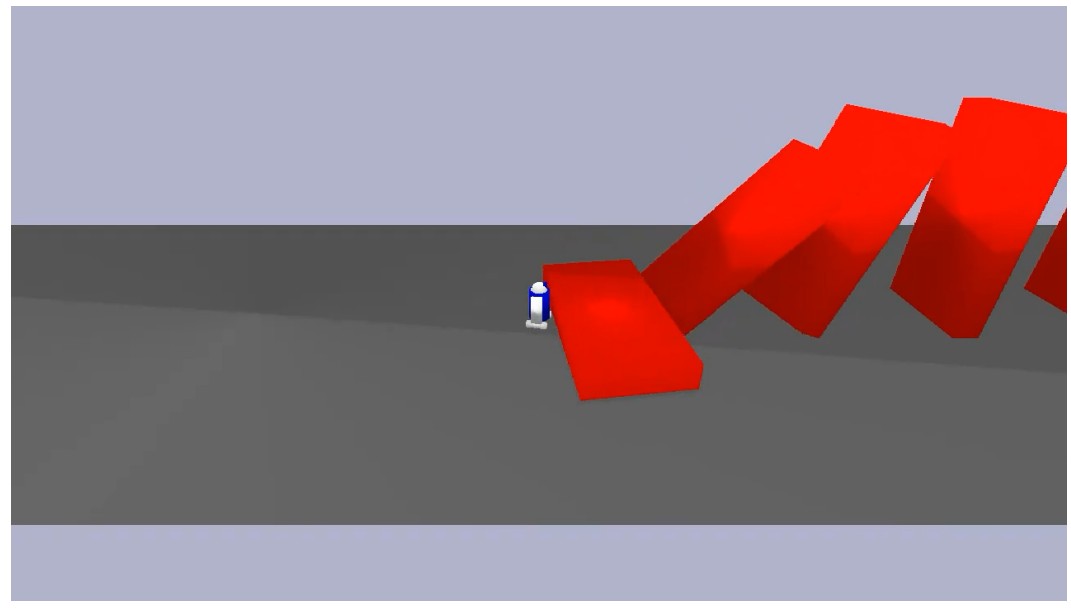

Figure 26: Task 21 of the OMNI-EPIC run presented in Section 5.

```
— The task of the robot is to push the domino to start a chain
    reaction.

Success:
The task is successfully completed when the robot pushes the domino
    and starts the chain reaction.

Rewards:
— The robot is rewarded for forward progress towards the domino,
    encouraging it to approach and push the domino.
— The robot is rewarded for the number of dominos that fall,
    incentivizing it to initiate the chain reaction.
— The robot is penalized for falling off the platform, ensuring it
    stays on the platform.

Termination:
The task terminates immediately if the robot falls off the platform.
"""

def __init__(self):
    super().__init__()
    self.platform_size = [1000.0, 10.0, 0.1]
    self.platform_position = [0.0, 0.0, 0.0]
    self.platform_id = self.create_box(mass=0.0, half_extents=[self.
        platform_size[0] / 2, self.platform_size[1] / 2, self.
        platform_size[2] / 2], position=self.platform_position, color
        =[0.5, 0.5, 0.5, 1.0])
    self._p.changeDynamics(bodyUniqueId=self.platform_id, linkIndex
        =-1, lateralFriction=0.8, restitution=0.5)
    self.domino_size = [0.5, 2.0, 4.0]
    self.domino_mass = 5.0
    self.domino_spacing = 3.0
    self.num_dominos = 10
    self.domino_ids = []
    for i in range(self.num_dominos):
        domino_position = [self.platform_position[0] + 5.0 + i * self
            .domino_spacing, self.platform_position[1], self.
```

```python
                platform_position[2] + self.platform_size[2] / 2 + self.
                    domino_size[2] / 2]
            domino_id = self.create_box(mass=self.domino_mass,
                half_extents=[self.domino_size[0] / 2, self.domino_size
                [1] / 2, self.domino_size[2] / 2], position=
                domino_position, color=[1.0, 0.0, 0.0, 1.0])
            self.domino_ids.append(domino_id)
        self.robot_position_init = [self.platform_position[0], self.
            platform_position[1], self.platform_position[2] + self.
            platform_size[2] / 2 + self.robot.links['base'].position_init
            [2]]

    def create_box(self, mass, half_extents, position, color):
        collision_shape_id = self._p.createCollisionShape(shapeType=self.
            _p.GEOM_BOX, halfExtents=half_extents)
        visual_shape_id = self._p.createVisualShape(shapeType=self._p.
            GEOM_BOX, halfExtents=half_extents, rgbaColor=color)
        return self._p.createMultiBody(baseMass=mass,
            baseCollisionShapeIndex=collision_shape_id,
            baseVisualShapeIndex=visual_shape_id, basePosition=position)

    def get_object_position(self, object_id):
        return np.asarray(self._p.getBasePositionAndOrientation(object_id
            )[0])

    def get_distance_to_object(self, object_id):
        object_position = self.get_object_position(object_id)
        robot_position = self.robot.links['base'].position
        return np.linalg.norm(object_position[:2] − robot_position[:2])

    def reset(self):
        observation = super().reset()
        self._p.resetBasePositionAndOrientation(self.robot.robot_id, self
            .robot_position_init, self.robot.links['base'].
            orientation_init)
        for i, domino_id in enumerate(self.domino_ids):
            domino_position = [self.platform_position[0] + 5.0 + i ∗ self
                .domino_spacing, self.platform_position[1], self.
                platform_position[2] + self.platform_size[2] / 2 + self.
                domino_size[2] / 2]
            self._p.resetBasePositionAndOrientation(domino_id,
                domino_position, [0.0, 0.0, 0.0, 1.0])
        return observation

    def step(self, action):
        self.distance_to_first_domino = self.get_distance_to_object(self.
            domino_ids[0])
        self.num_fallen_dominos = sum([self._p.getBaseVelocity(domino_id)
            [0][2] < −0.1 for domino_id in self.domino_ids])
        observation, reward, terminated, truncated, info = super().step(
            action)
        return (observation, reward, terminated, truncated, info)

    def get_task_rewards(self, action):
        new_distance_to_first_domino = self.get_distance_to_object(self.
            domino_ids[0])
        new_num_fallen_dominos = sum([self._p.getBaseVelocity(domino_id)
            [0][2] < −0.1 for domino_id in self.domino_ids])
        forward_progress = (self.distance_to_first_domino −
            new_distance_to_first_domino) / self.dt
        domino_fall_reward = (new_num_fallen_dominos − self.
            num_fallen_dominos) ∗ 10.0
        return {'forward_progress': forward_progress, 'domino_fall_reward
            ': domino_fall_reward}
```

```python
    def get_terminated(self, action):
        return self.robot.links['base'].position[2] < self.
            platform_position[2]

    def get_success(self):
        return any([self._p.getBaseVelocity(domino_id)[0][2] < −0.1 for
            domino_id in self.domino_ids])
```

**Task 23**

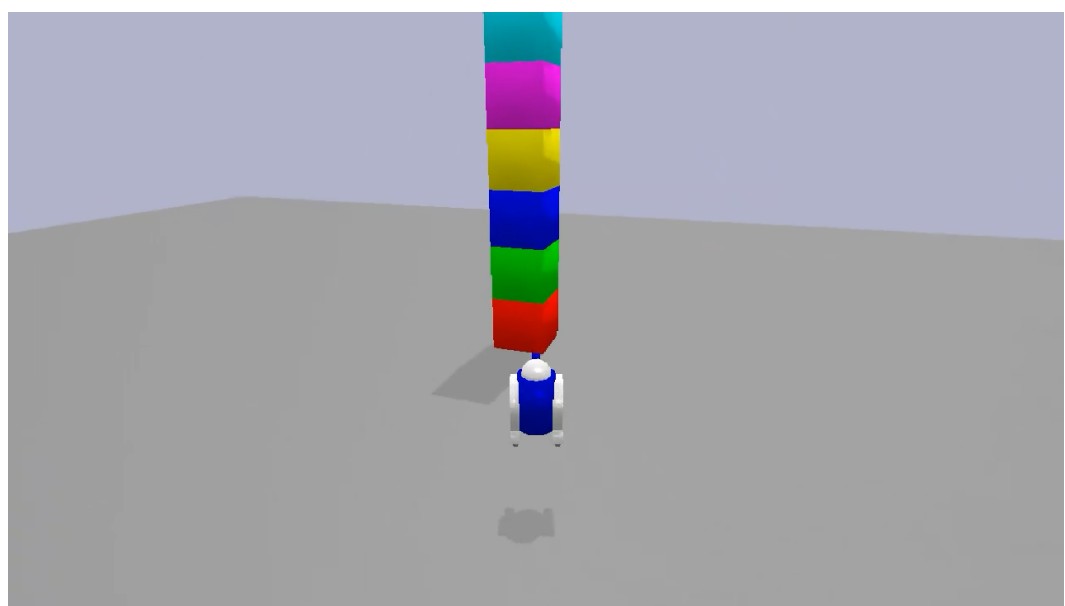

Figure 27: Task 23 of the OMNI-EPIC run presented in Section 5.

```python
import numpy as np
from oped.envs.r2d2.base import R2D2Env

class Env(R2D2Env):
    """
    Task: Destroy a tower of blocks

    Description:
    — The environment consists of a large flat platform measuring 50
        meters by 50 meters.
    — A tower of 10 blocks is placed at a random location on the platform
        . Each block is a cube measuring 1 meter by 1 meter by 1 meter
        and weighs 5 kg. The blocks are of different colors to make the
        environment visually rich.
    — The robot is initialized at a fixed position on the platform,
        facing the tower.

    The task of the robot is to destroy the tower by pushing the blocks
        until all blocks are knocked over and no longer stacked.

    Success:
    The task is successfully completed when all blocks are knocked over
        and none are stacked on top of each other.

    Rewards:
    — The robot receives a survival reward for each time step it remains
        on the platform, encouraging steady progress.
```

```python
        - The robot is rewarded for reducing the height of the tower,
            incentivizing the robot to knock over the blocks.
        - A large reward is given when all blocks are knocked over and none
            are stacked.

    Termination:
    The episode terminates if the robot falls off the platform or if all
        blocks are knocked over and none are stacked.
    """

    def __init__(self):
        super().__init__()
        self.platform_size = [50.0, 50.0, 0.1]
        self.platform_position = [0.0, 0.0, 0.0]
        self.platform_id = self.create_box(mass=0.0, half_extents=[self.
            platform_size[0] / 2, self.platform_size[1] / 2, self.
            platform_size[2] / 2], position=self.platform_position, color
            =[0.8, 0.8, 0.8, 1.0])
        self._p.changeDynamics(bodyUniqueId=self.platform_id, linkIndex
            =-1, lateralFriction=0.8, restitution=0.5)
        self.block_size = [1.0, 1.0, 1.0]
        self.block_mass = 5.0
        self.num_blocks = 10
        self.block_ids = []
        self.block_colors = [[1.0, 0.0, 0.0, 1.0], [0.0, 1.0, 0.0, 1.0],
            [0.0, 0.0, 1.0, 1.0], [1.0, 1.0, 0.0, 1.0], [1.0, 0.0, 1.0,
            1.0], [0.0, 1.0, 1.0, 1.0], [0.5, 0.5, 0.5, 1.0], [1.0, 0.5,
            0.0, 1.0], [0.5, 0.0, 0.5, 1.0], [0.0, 0.5, 0.5, 1.0]]
        for i in range(self.num_blocks):
            block_id = self.create_box(mass=self.block_mass, half_extents
                =[self.block_size[0] / 2, self.block_size[1] / 2, self.
                block_size[2] / 2], position=[0.0, 0.0, 0.0], color=self.
                block_colors[i])
            self.block_ids.append(block_id)
        self.robot_position_init = [0.0, 0.0, self.platform_position[2] +
             self.platform_size[2] / 2 + self.robot.links['base'].
            position_init[2]]

    def create_box(self, mass, half_extents, position, color):
        collision_shape_id = self._p.createCollisionShape(shapeType=self.
            _p.GEOM_BOX, halfExtents=half_extents)
        visual_shape_id = self._p.createVisualShape(shapeType=self._p.
            GEOM_BOX, halfExtents=half_extents, rgbaColor=color)
        return self._p.createMultiBody(baseMass=mass,
            baseCollisionShapeIndex=collision_shape_id,
            baseVisualShapeIndex=visual_shape_id, basePosition=position)

    def get_object_position(self, object_id):
        return np.asarray(self._p.getBasePositionAndOrientation(object_id
            )[0])

    def get_distance_between_objects(self, object1_id, object2_id):
        object1_position = self.get_object_position(object1_id)
        object2_position = self.get_object_position(object2_id)
        return np.linalg.norm(object1_position[:2] - object2_position
            [:2])

    def reset(self):
        observation = super().reset()
        self._p.resetBasePositionAndOrientation(self.robot.robot_id, self
            .robot_position_init, self.robot.links['base'].
            orientation_init)
        tower_x = np.random.uniform(self.platform_position[0] - self.
            platform_size[0] / 2 + self.block_size[0] / 2, self.
```

```
                    platform_position[0] + self.platform_size[0] / 2 - self.
                        block_size[0] / 2)
                tower_y = np.random.uniform(self.platform_position[1] - self.
                    platform_size[1] / 2 + self.block_size[1] / 2, self.
                    platform_position[1] + self.platform_size[1] / 2 - self.
                    block_size[1] / 2)
                for i, block_id in enumerate(self.block_ids):
                    self._p.resetBasePositionAndOrientation(block_id, [tower_x,
                        tower_y, self.platform_position[2] + self.platform_size
                        [2] / 2 + (i + 0.5) * self.block_size[2]], [0.0, 0.0,
                        0.0, 1.0])
                return observation

    def step(self, action):
        self.tower_height = self.calculate_tower_height()
        observation, reward, terminated, truncated, info = super().step(
            action)
        return (observation, reward, terminated, truncated, info)

    def calculate_tower_height(self):
        heights = [self.get_object_position(block_id)[2] for block_id in
            self.block_ids]
        return max(heights) - min(heights)

    def get_task_rewards(self, action):
        new_tower_height = self.calculate_tower_height()
        survival = 1.0
        reduce_tower_height = (self.tower_height - new_tower_height) /
            self.dt
        all_blocks_knocked_over = 10.0 if new_tower_height <= self.
            block_size[2] else 0.0
        return {'survival': survival, 'reduce_tower_height':
            reduce_tower_height, 'all_blocks_knocked_over':
            all_blocks_knocked_over}

    def get_terminated(self, action):
        is_off_platform = self.robot.links['base'].position[2] < self.
            platform_position[2]
        all_blocks_knocked_over = self.calculate_tower_height() <= self.
            block_size[2]
        return is_off_platform or all_blocks_knocked_over

    def get_success(self):
        all_blocks_knocked_over = self.calculate_tower_height() <= self.
            block_size[2]
        return all_blocks_knocked_over
```

**Task 25**

```
import numpy as np
from oped.envs.r2d2.base import R2D2Env

class Env(R2D2Env):
    """
    Push a box next to a wall to climb on top of it.

    Description:
    - The environment consists of a large platform measuring 1000 x 10 x
        0.1 meters.
    - The robot is initialized at a fixed position on the platform.
    - A wall with dimensions 0.5 x 10 x 2 meters is positioned on the
        platform, 5 meters away from the robot.
    - A box with dimensions 1 x 1 x 1 meters (5 kg in mass) is positioned
        on the platform, 3 meters away from the robot.
```

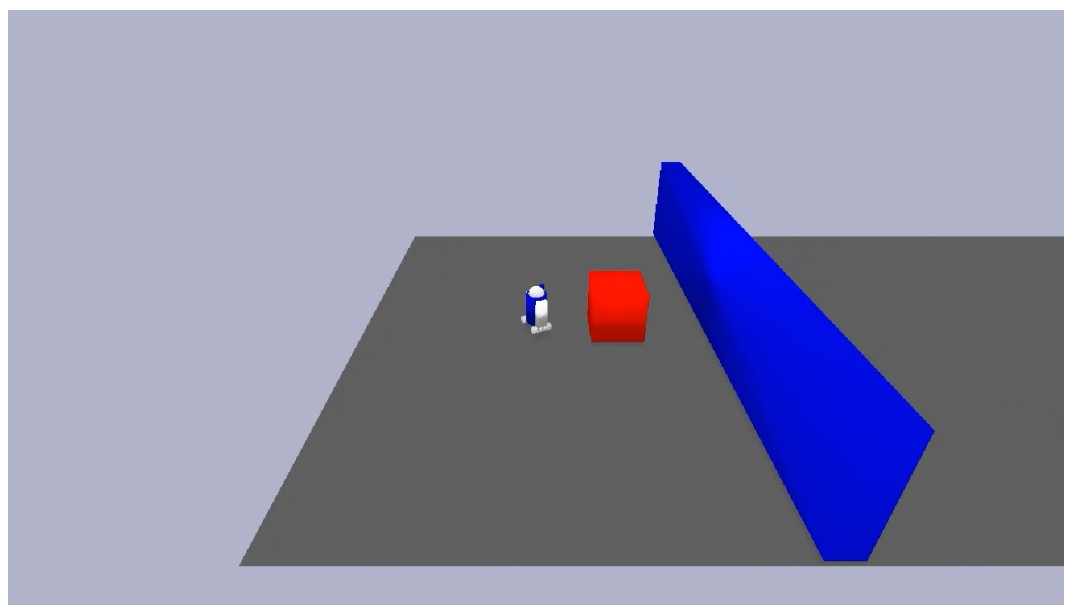

Figure 28: Task 25 of the OMNI-EPIC run presented in Section 5.

```
  — The task of the robot is first to push the box against the wall.
    Then, the robot must climb on top of the box and jump on top of
    the wall.

Success:
The task is successfully completed if the robot climbs on top of the
    box and then jumps on top of the wall.

Rewards:
— The robot receives a reward for each time step it remains on the
    platform.
— The robot is rewarded for decreasing its distance to the box,
    encouraging it to approach and interact with the box.
— The robot is rewarded for pushing the box towards the wall.
— The robot is rewarded for climbing on top of the box and an
    additional reward for jumping on top of the wall.

Termination:
The episode terminates if the robot falls off the platform. The
    episode does not terminate if the robot successfully jumps on top
    of the wall.
"""

def __init__(self):
    super().__init__()
    self.robot_position_init = [0.0, 0.0, 0.0]
    self.platform_size = [1000.0, 10.0, 0.1]
    self.platform_position = [self.robot_position_init[0] + self.
        platform_size[0] / 2 — 2.0, self.robot_position_init[1], self
        .robot_position_init[2] — self.platform_size[2] / 2]
    self.platform_id = self.create_box(mass=0.0, half_extents=[self.
        platform_size[0] / 2, self.platform_size[1] / 2, self.
        platform_size[2] / 2], position=self.platform_position, color
        =[0.5, 0.5, 0.5, 1.0])
    self._p.changeDynamics(bodyUniqueId=self.platform_id, linkIndex
        =—1, lateralFriction=0.8, restitution=0.5)
    self.wall_size = [0.5, 10.0, 2.0]
```

```python
        self.wall_position = [self.robot_position_init[0] + 5.0, self.
            platform_position[1], self.platform_position[2] + self.
            platform_size[2] / 2 + self.wall_size[2] / 2]
        self.wall_id = self.create_box(mass=0.0, half_extents=[self.
            wall_size[0] / 2, self.wall_size[1] / 2, self.wall_size[2] /
            2], position=self.wall_position, color=[0.0, 0.0, 1.0, 1.0])
        self.box_size = [1.0, 1.0, 1.0]
        self.box_mass = 5.0
        self.box_position_init = [self.robot_position_init[0] + 3.0, self
            .platform_position[1], self.platform_position[2] + self.
            platform_size[2] / 2 + self.box_size[2] / 2]
        self.box_id = self.create_box(mass=self.box_mass, half_extents=[
            self.box_size[0] / 2, self.box_size[1] / 2, self.box_size[2]
            / 2], position=self.box_position_init, color=[1.0, 0.0, 0.0,
            1.0])

    def create_box(self, mass, half_extents, position, color):
        collision_shape_id = self._p.createCollisionShape(shapeType=self.
            _p.GEOM_BOX, halfExtents=half_extents)
        visual_shape_id = self._p.createVisualShape(shapeType=self._p.
            GEOM_BOX, halfExtents=half_extents, rgbaColor=color)
        return self._p.createMultiBody(baseMass=mass,
            baseCollisionShapeIndex=collision_shape_id,
            baseVisualShapeIndex=visual_shape_id, basePosition=position)

    def get_object_position(self, object_id):
        return np.asarray(self._p.getBasePositionAndOrientation(object_id
            )[0])

    def get_distance_to_object(self, object_id):
        object_position = self.get_object_position(object_id)
        robot_position = self.robot.links['base'].position
        return np.linalg.norm(object_position[:2] - robot_position[:2])

    def reset(self):
        observation = super().reset()
        self._p.resetBasePositionAndOrientation(self.box_id, self.
            box_position_init, [0.0, 0.0, 0.0, 1.0])
        self._p.resetBasePositionAndOrientation(self.robot.robot_id, [
            self.robot_position_init[0], self.robot_position_init[1],
            self.robot_position_init[2] + self.robot.links['base'].
            position_init[2]], self.robot.links['base'].orientation_init)
        return observation

    def step(self, action):
        self.distance_to_box = self.get_distance_to_object(self.box_id)
        self.box_position = self.get_object_position(self.box_id)
        self.robot_on_box = len(self._p.getContactPoints(bodyA=self.robot
            .robot_id, bodyB=self.box_id)) > 0
        self.robot_on_wall = len(self._p.getContactPoints(bodyA=self.
            robot.robot_id, bodyB=self.wall_id)) > 0
        observation, reward, terminated, truncated, info = super().step(
            action)
        return (observation, reward, terminated, truncated, info)

    def get_task_rewards(self, action):
        new_distance_to_box = self.get_distance_to_object(self.box_id)
        new_box_position = self.get_object_position(self.box_id)
        new_robot_on_box = len(self._p.getContactPoints(bodyA=self.robot.
            robot_id, bodyB=self.box_id)) > 0
        new_robot_on_wall = len(self._p.getContactPoints(bodyA=self.robot
            .robot_id, bodyB=self.wall_id)) > 0
        survival = 1.0
        reach_box = (self.distance_to_box - new_distance_to_box) / self.
            dt
```

```python
        push_box = (new_box_position[0] - self.box_position[0]) / self.dt
        climb_on_box = 5.0 if new_robot_on_box and (not self.robot_on_box
            ) else 0.0
        jump_on_wall = 10.0 if new_robot_on_wall and (not self.
            robot_on_wall) else 0.0
        return {'survival': survival, 'reach_box': reach_box, 'push_box':
             push_box, 'climb_on_box': climb_on_box, 'jump_on_wall':
            jump_on_wall}

    def get_terminated(self, action):
        return self.robot.links['base'].position[2] < self.
            platform_position[2]

    def get_success(self):
        return len(self._p.getContactPoints(bodyA=self.robot.robot_id,
            bodyB=self.wall_id)) > 0
```

# F    ADDITIONAL SHORT RUNS WITH LEARNING

Although the initial seed tasks in the task archive remain the same across repeated runs, we observe significant differences in the generated environments. This suggests that OMNI-EPIC's ability to generate diverse learnable tasks is not heavily influenced by the initial set of environments.

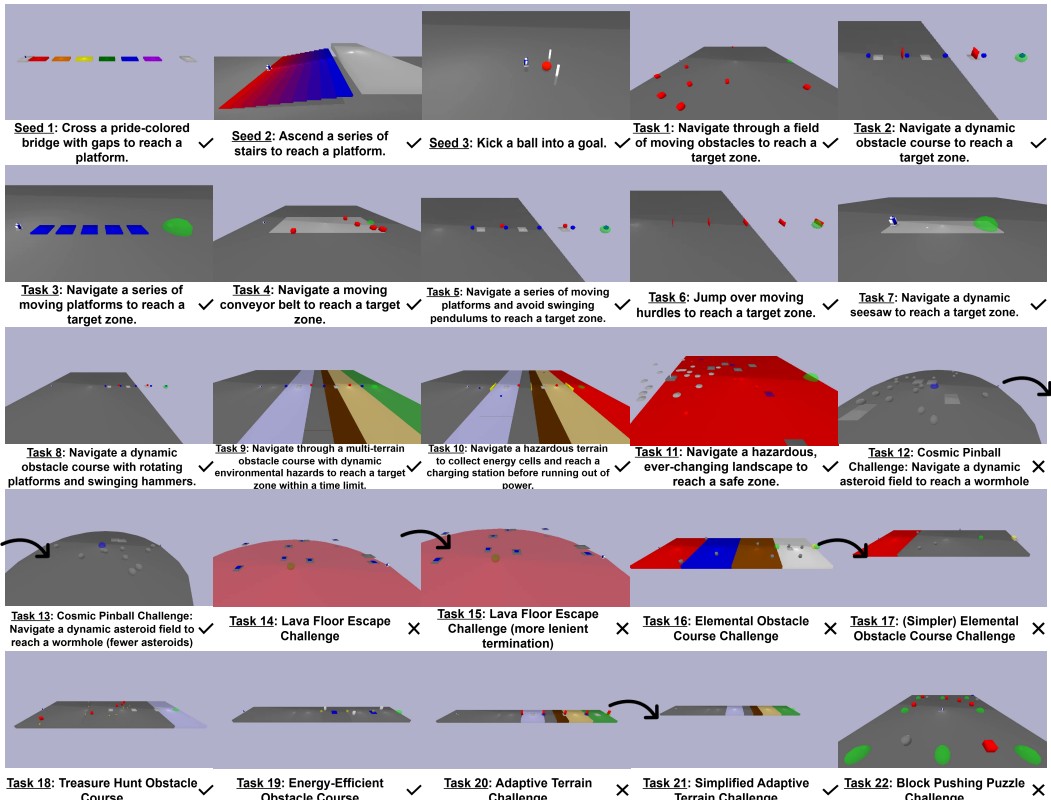

Figure 29: **Additional Short Run with Learning (Run 2).** This figure shows another instance of the short run experiment under the same settings as in Figure 3. It demonstrates the progression of tasks generated by OMNI-EPIC and the learning outcomes of the RL agent. Checkmarks indicate successful learning, crosses indicate failures, and arrows show iterations on failed tasks.

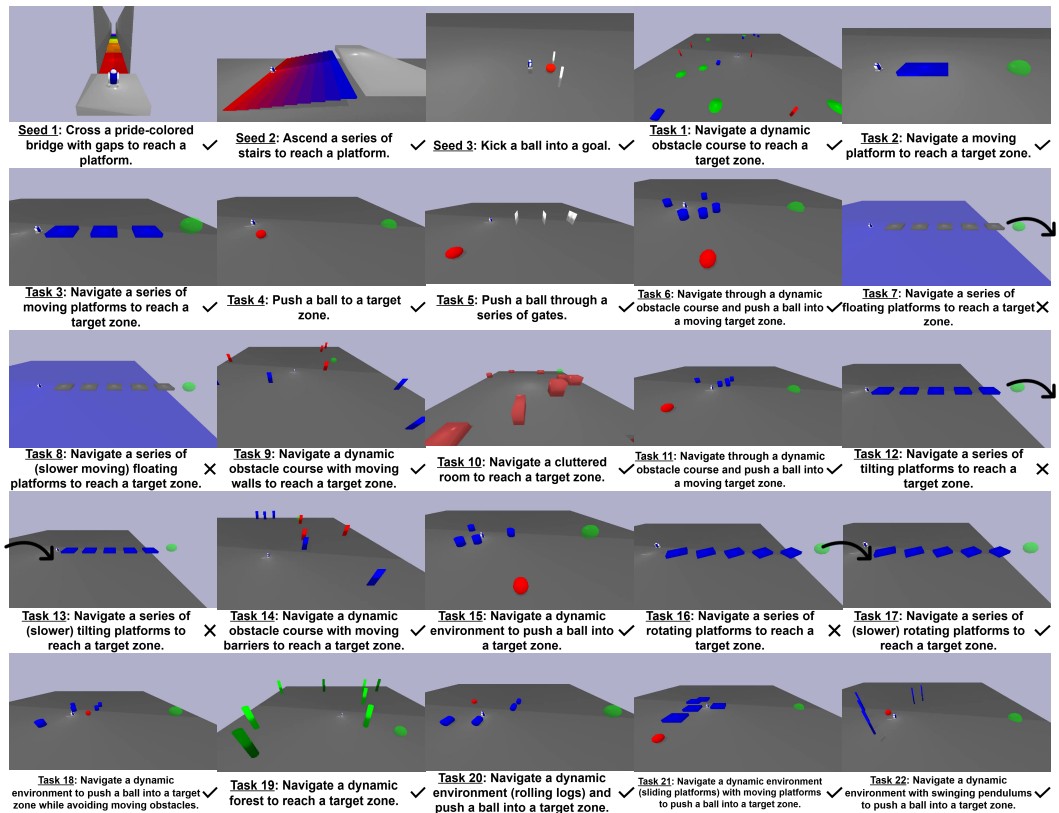

Figure 30: **Additional Short Run with Learning (Run 3).** This figure shows another instance of the short run experiment under the same settings as in Figure 3. It demonstrates the progression of tasks generated by OMNI-EPIC and the learning outcomes of the RL agent. Checkmarks indicate successful learning, crosses indicate failures, and arrows show iterations on failed tasks.

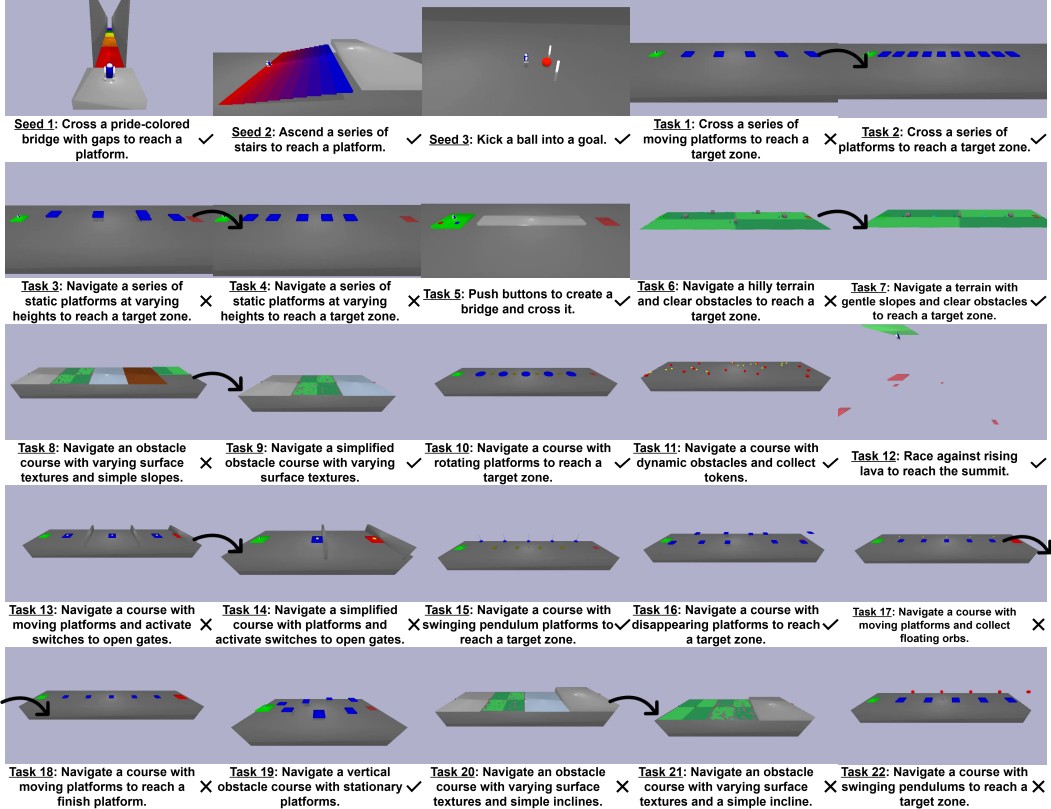

Figure 31: **Additional Short Run with Learning (Run 4).** This figure shows another instance of the short run experiment under the same settings as in Figure 3. It demonstrates the progression of tasks generated by OMNI-EPIC and the learning outcomes of the RL agent. Checkmarks indicate successful learning, crosses indicate failures, and arrows show iterations on failed tasks.

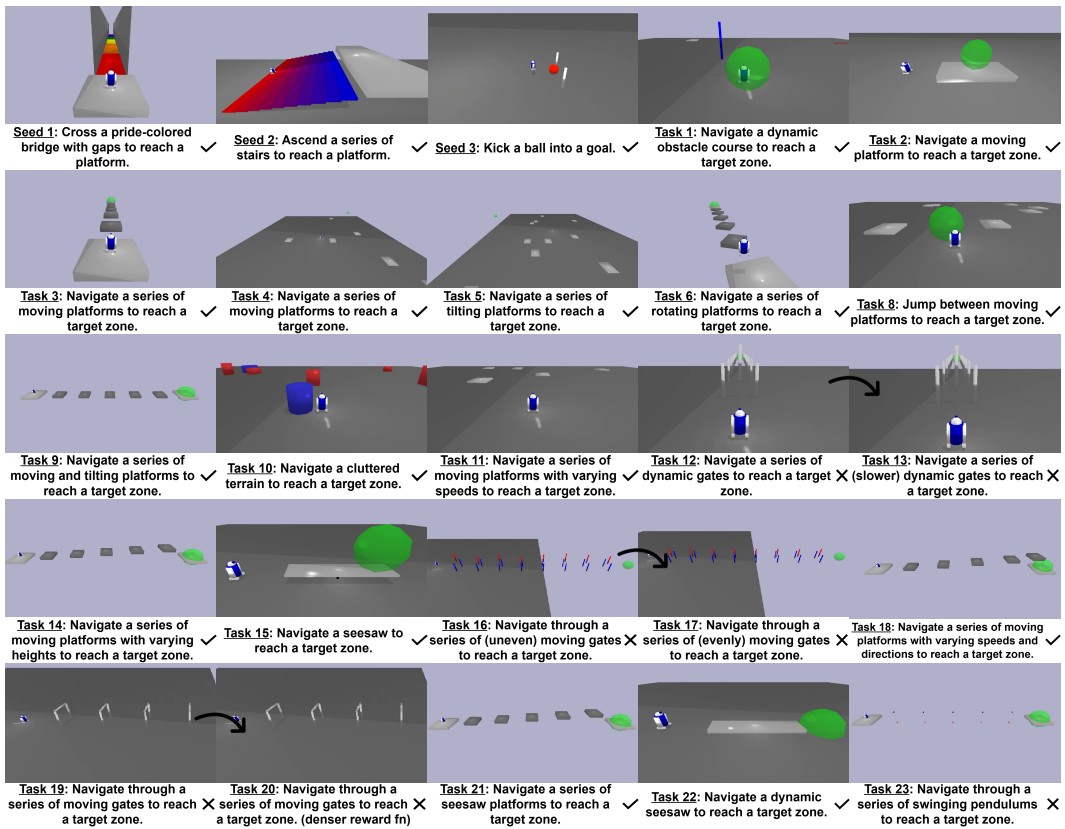

Figure 32: **Additional Short Run with Learning (Run 5).** This figure shows another instance of the short run experiment under the same settings as in Figure 3. It demonstrates the progression of tasks generated by OMNI-EPIC and the learning outcomes of the RL agent. Checkmarks indicate successful learning, crosses indicate failures, and arrows show iterations on failed tasks.

# G ABLATIONS ON SHORT RUNS WITH LEARNING

## G.1 WITHOUT TRANSFER LEARNING

We conducted an experiment in which we ran OMNI-EPIC (i.e., with transfer learning between tasks) on 25 tasks with 5 replications and compared it to training policies from scratch (i.e., without transfer learning between tasks). We allocated the same total number of training steps (2 million) to the RL agent for each task. The only difference between the two settings was the policy initialization for each new task: in the transfer learning scenario, the RL policy was initialized from a checkpoint (a policy that had trained on another environment), whereas in the non-transfer learning scenario, it was initialized randomly (i.e., trained from scratch). OMNI-EPIC achieves higher success rate, with a median of 63.2% (CI: 42.1 - 69.5) than training from scratch, with a median of 36.8% (CI: 31.6 - 53.7). This demonstrates that the OMNI-EPIC agents build upon previously learned skills, creating a curriculum of increasing difficulty.

## G.2 UNIFORM SAMPLING EXAMPLES FROM THE ARCHIVE

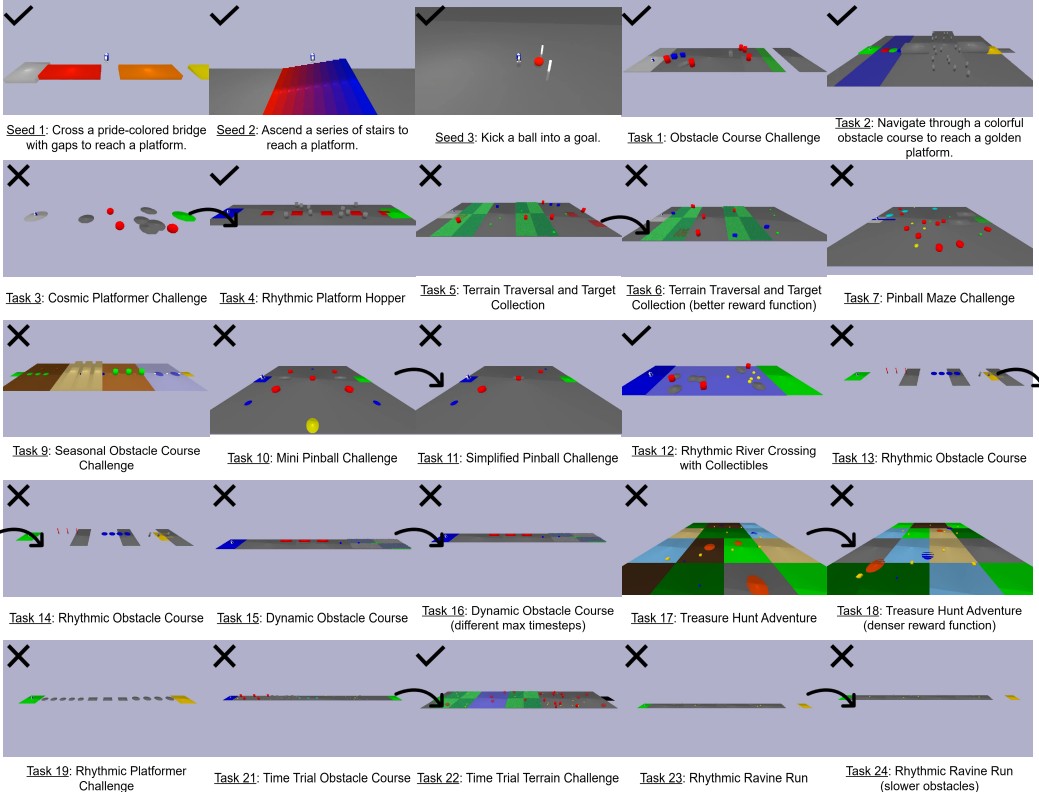

Figure 33: **Ablation of short run with learning, by sampling in-context examples uniformly.** This figure shows an ablation of the short run with learning experiment. The same settings as Figure 3 are used, except that, instead of using the most similar tasks as in-context examples for the task generator, we uniformly sample from the archive. Checkmarks indicate successful learning, crosses indicate failures, and arrows show iterations on failed tasks. Using uniform sampling for in-context examples results in learning only 5 of the generated tasks, much fewer than in Figure 3, where the most similar tasks are used as in-context examples.

### G.3 WITHOUT FAILED EXAMPLES

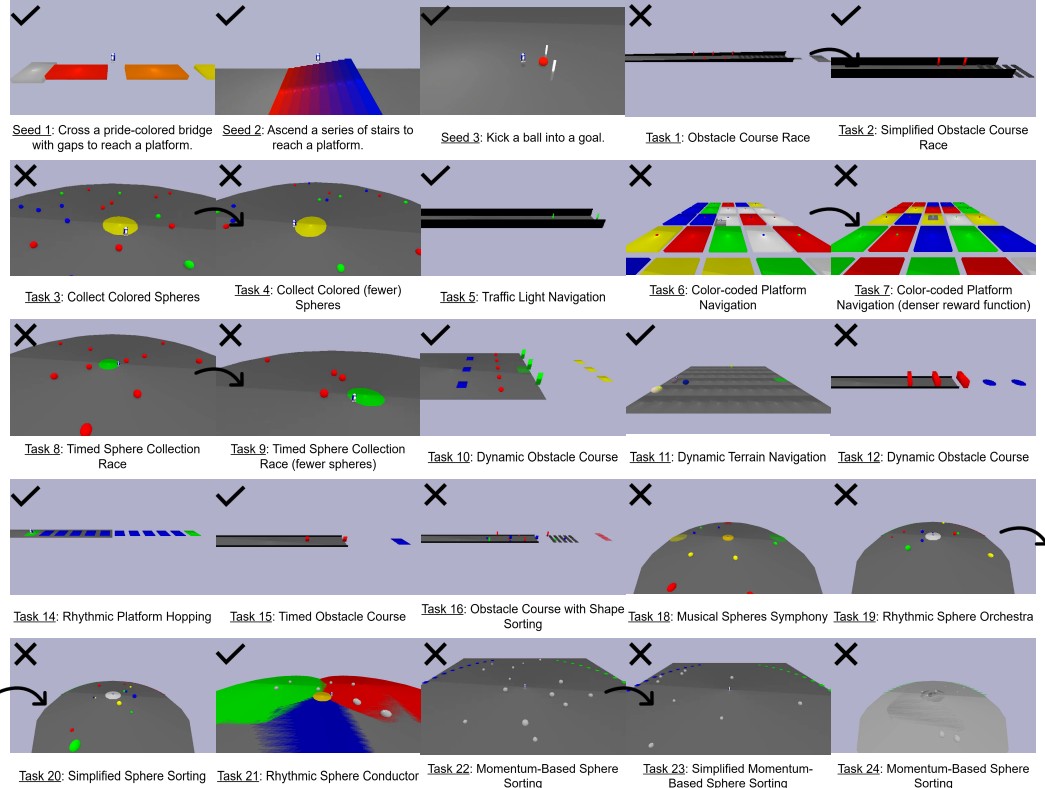

Figure 34: **Ablation of short run with learning, without failed examples input to the task generator.** This figure shows an ablation of the short run with learning experiment. The same settings as Figure 3 are used, except that no failed examples (only successful ones) are given to the task generator. Checkmarks indicate successful learning, crosses indicate failures, and arrows show iterations on failed tasks. We see that similar tasks are regenerated, even though the RL agent previously failed to learn them due to unsuitable reward function or incorrect environment configuration. Not giving the task generator examples of tasks that were attempted but failed results in learning only 7 of the generated tasks, much fewer than in Figure 3, where both successful and failed examples are used as input.

## G.4 WITHOUT LEARNING PROGRESS

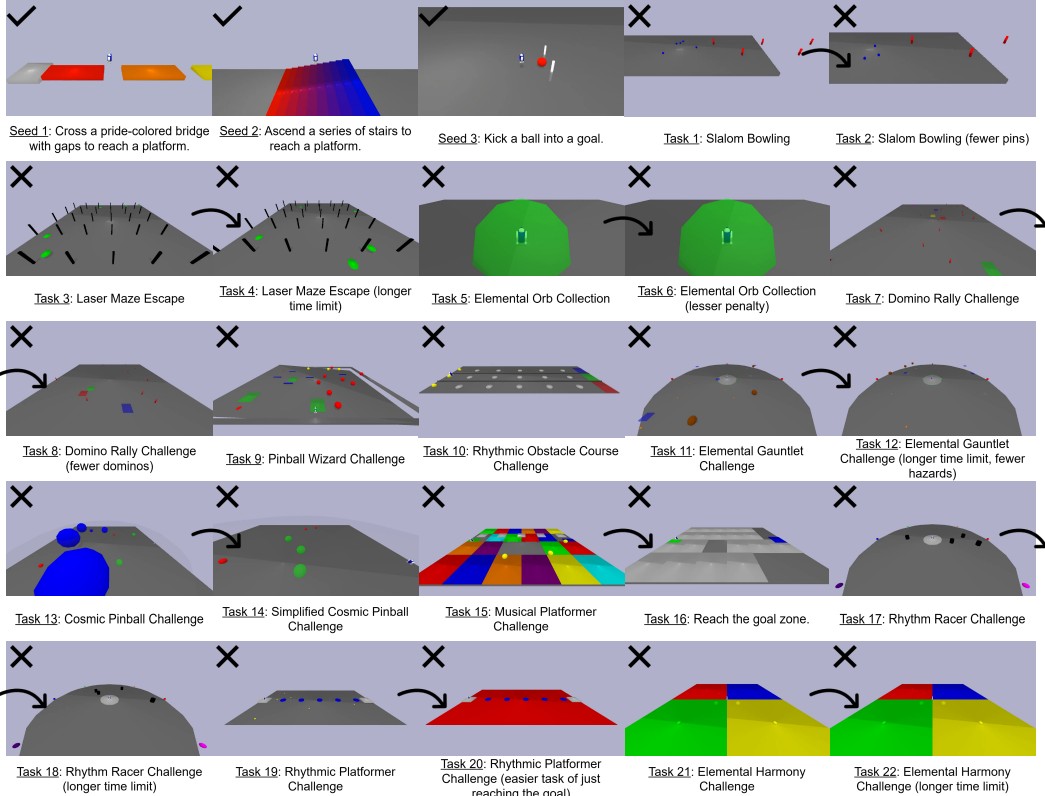

Figure 35: **Ablation of short run with learning, without learning progress notions in the task generator's prompt.** This figure shows an ablation of the short run with learning experiment. The same settings as Figure 3 are used, except that the task generator's prompt does not include any notion of learning progress. Checkmarks indicate successful learning, crosses indicate failures, and arrows show iterations on failed tasks. Without incorporating a notion of learning progress in the task generator's prompt, no tasks were successfully learned, which is much fewer than in Figure 3, where the task generator's prompt includes both notions of interestingness and learning progress.

**System Prompt:**

```
You are an expert in reinforcement learning. Your goal is to help a robot
    master a diverse set of interesting tasks in simulation using
    PyBullet. You will be provided with the list of tasks that the robot
    has successfully learned, along with their corresponding environment
    code, and the list of tasks that the robot has attempted but failed
    to learn, along with their corresponding environment code. Your
    objective is to decide the next task for the robot, selecting one
    that is interesting and novel.

Instructions:
— The next task should be interesting:
    — Novel and creative compared to the tasks the robot has already
        tried.
    — Useful according to humans.
    — Design rich environments with a large number of diverse objects and
        terrains, and with a clear task for the robot to execute.
    — The task should be fun or engaging to watch. You can draw
        inspiration from real—world tasks or video games. Be creative!
— Be specific in the task description:
    — State clearly what the task of the robot is.
```

```
            — Define clearly what the success condition is.
            — Define clearly what are the different reward and penalty components
                .
            — Define clearly what the termination conditions are. If the reward
                components include a survival reward, ensure the episode only
                terminates when the agent fails the task.
    — The task should not take too long to complete.
    — The robot can push objects around but lacks the ability to grab, pick
        up, carry, or stack objects. Don't suggest tasks that involve these
        skills.
    — Don't suggest tasks that require the robot to navigate through a maze.
    — If the task involves navigating a terrain with obstacles, make sure
        that the robot can not go around the obstacles.
    — If the task involves a target zone, make sure that the collision of the
        target zone is set to False.
    — Return only the task description, not the environment code.
    — Ensure that the designs pose no harm to humans and align with human
        values and ethics.

Robot description:
{ROBOT_DESC}

Desired format:
Reasoning for what the next task should be:
<reasoning>

Next task description:
\"\"\"
<task description>
\"\"\"
```

## H    HUMAN EVALUATION SETUP

To evaluate the alignment between human judgment and the automated success detector, we conducted a study with 50 participants. The goal was to assess whether human evaluators agreed with the success detector's assessments of task completion by the robot. Each participant reviewed videos of a robot attempting various tasks, alongside the corresponding task descriptions. Below are the detailed instructions and setup used for the evaluation.

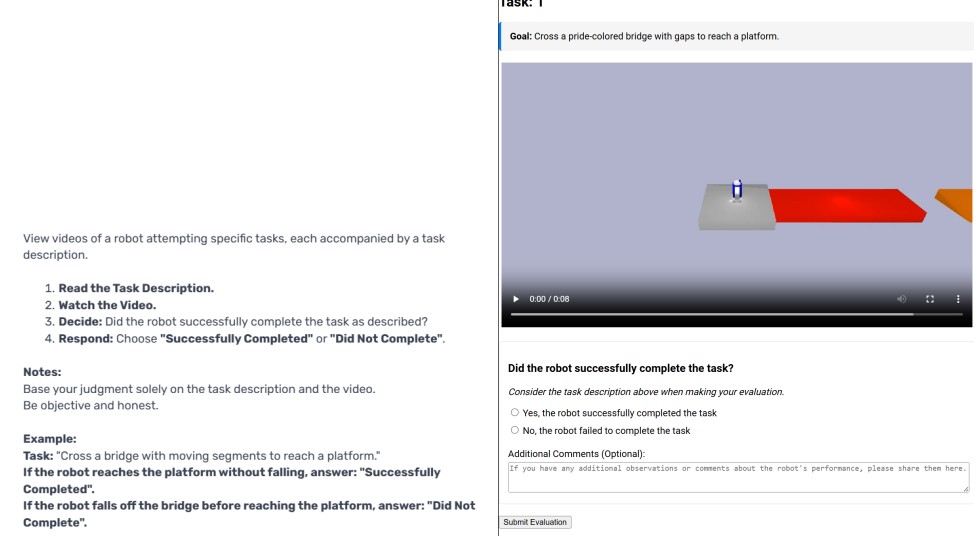

Figure 36: **Human evaluation setup.** (Left) Instructions given to participants. (Right) Annotation interface used by participants to annotate the data.

The human evaluation study was conducted using the `cloudresearch.com` platform. Participants were compensated $0.25 for each response (estimated hourly rate of $15.00/hour, as each response took less than 1 minute), ensuring fair and ethical payment for their time and effort. The study aimed to collect accurate and unbiased assessments to compare with the success detector's automated evaluations.

# I Supplementary Materials for Quantitative Results

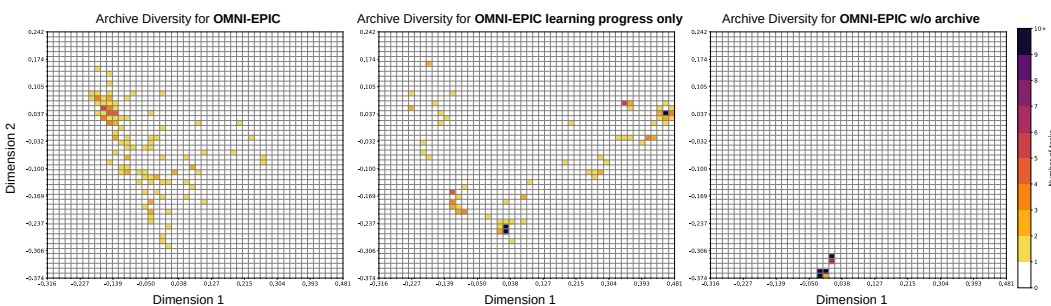

Figure 37: **Archive diversity results.** Archive diversity plots of long runs with simulated learning of OMNI-EPIC and the controls. Fewer tasks fall into the same discretized cells for OMNI-EPIC than OMNI-EPIC Learning Progress only or OMNI-EPIC w/o archive. The substantial difference between the left and center plots is more easily observed in Figure 4.

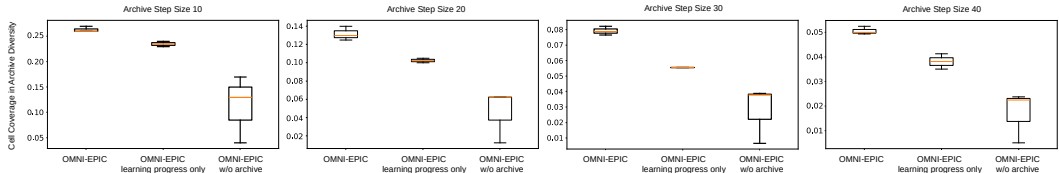

Figure 38: **Cell coverage of archive diversity plots with different discretization levels for long runs with simulated learning.** This figure is similar to Figure 4, which uses an archive discretization level of 50, but here we present cell coverage results for archive diversity plots with discretization levels of [10, 20, 30, 40] across methods on long runs with simulated learning. OMNI-EPIC consistently achieves significantly higher cell coverage compared to the controls, even across different archive discretization levels (p < 0.05, Mann-Whitney U test).

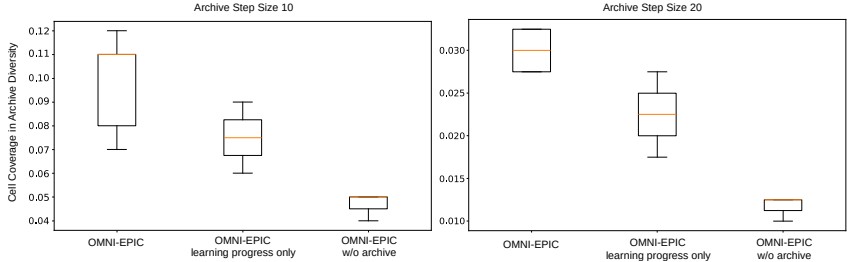

Figure 39: **Cell coverage of archive diversity plots with different discretization levels for short runs with learning.** This figure is similar to Figure 4, which uses an archive discretization level of 50 for long run with simulated learning, but here we present cell coverage results for archive diversity plots with discretization levels of [10, 20] across methods on short runs with learning. While we see similar trends as Figure 4 and Figure 38, the differences between methods are not always statistically significant (not all p < 0.05, Mann-Whitney U test). This is due to the shorter training runs, as the effects of OMNI-EPIC become more pronounced over longer runs.

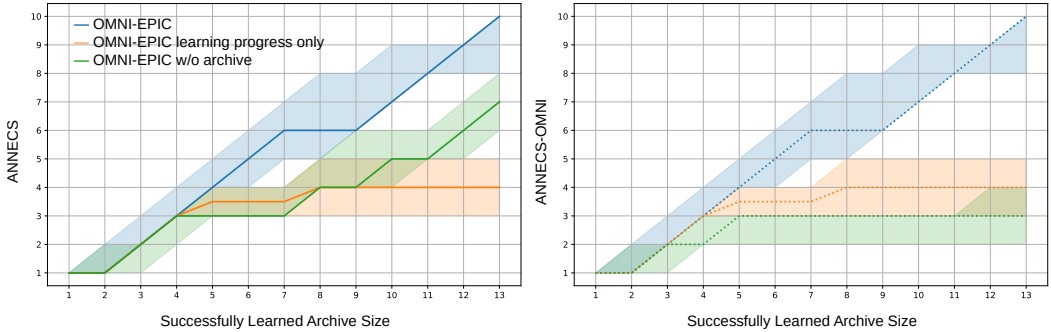

Figure 40: **(Left) ANNECS and (Right) ANNECS-OMNI results.** Short runs with RL training by OMNI-EPIC and the controls are repeated five times. Darker lines are median values, shaded regions are 95% confidence intervals. OMNI-EPIC significantly outperforms the controls on both metrics. There is no difference between ANNECS and ANNECS-OMNI for OMNI-EPIC, indicating that all tasks learned by OMNI-EPIC are considered interesting.

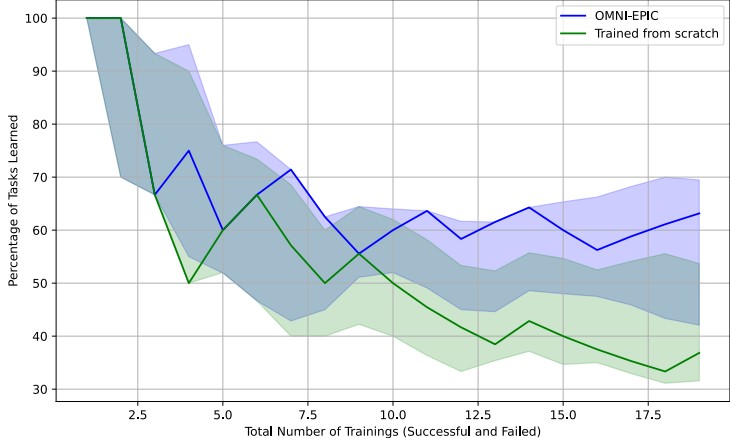

Figure 41: **Percentage of tasks learned over attempted tasks in short runs with learning experiments.** Attempted tasks include all tasks that were attempted with RL training. Short runs with RL training by OMNI-EPIC are repeated five times. The blue line represents OMNI-EPIC, while the green line represents using the same generated tasks from OMNI-EPIC and training each task from scratch. The darker lines indicate median values, and the shaded regions represent 95% confidence intervals. The decreasing percentage of learned tasks, along with the widening gap between tasks learned by OMNI-EPIC and those trained from scratch, shows that OMNI-EPIC generates increasingly difficult tasks over time.

## J  SELECTING THE MOST SIMILAR TASKS

To generate new tasks that are both relevant and challenging, it is essential to select tasks from the archive as a reference. This selection process ensures that the generated tasks build upon the agent's prior knowledge and skills. OMNI-EPIC's process of selecting the most similar tasks, given a query task, takes inspiration from Lewis et al. (2020). The first step is to embed all tasks (both their natural language descriptions and environment code) within the archive into a high-dimensional vector space using a pretrained language encoder. This embedding allows us to represent each task as a dense vector that captures its semantic meaning and characteristics. Once all tasks are embedded, we can efficiently compare the similarity between tasks using cosine similarity. For a given query task, we retrieve a predefined number of tasks from the archive that exhibit the closest cosine similarity to the query's embedding, ensuring that the selected tasks are the most relevant and similar to the current learning context.

A potential limitation of our approach is the possibility of cyclic behavior where the task generator alternates between generating task types that are consistently rejected due to their similarity to existing tasks in the archive. While this scenario is unlikely in practice due to the stochastic nature of task generation and the diversity of the environment distribution, it cannot be completely ruled out. Furthermore, as context lengths increase with advancements in FM capabilities, the likelihood of such cyclic behavior would diminish even more.

## K  HYPERPARAMETERS

Table 1: OMNI-EPIC hyperparameters

| Component | Parameter | Value |
|---|---|---|
| General | Max number of iterations | 5 |
| | Embedding method | OpenAI (text-embedding-3-small) |
| Task Generator | Number of learned examples | 5 |
| | Number of failed examples | 5 |
| | Client | Anthropic |
| | Model | Claude 3 Opus (claude-3-opus-20240229) |
| | Temperature | 0 |
| Environment Generator | Number of few-shot examples | 5 |
| | Client | Anthropic |
| | Model | Claude 3 Opus (claude-3-opus-20240229) |
| | Temperature | 0 |
| Post-Generation MoI | Number of similar tasks | 10 |
| | Client | OpenAI |
| | Model | GPT-4o (gpt-4o-2024-05-13) |
| | Temperature | 0 |
| Success Detector | Client | OpenAI |
| | Model | GPT-4o (gpt-4o-2024-05-13) |
| | Temperature | 0 |
| Task Reflection | Max number of iterations | 1 |
| | Number of few-shot examples | 5 |
| | Client | OpenAI |
| | Model | GPT-4o (gpt-4o-2024-05-13) |
| | Temperature | 0 |

Table 2: DreamerV3 hyperparameters

| Parameter | Value |
|---|---|
| Total time steps | $2 \times 10^6$ |
| Replay buffer size | $10^6$ |
| Batch size | 16 |
| Batch length | 64 |
| Discount factor | 0.997 |
| Learning rate | $3 \times 10^{-4}$ |

**Compute Resources**
For each task, the R2D2 agent is trained for approximately 1 hour using 2 NVIDIA RTX 6000 Ada Generation GPUs and 32 CPU cores.

# L PROMPTS

## L.1 TASK GENERATOR

**System Prompt:**

```
You are an expert in curriculum learning and reinforcement learning. Your
    goal is to help a robot master a diverse set of interesting tasks in
    simulation using PyBullet. You will be provided with the list of
    tasks that the robot has successfully learned, along with their
    corresponding environment code, and the list of tasks that the robot
    has attempted but failed to learn, along with their corresponding
    environment code. Your objective is to decide the next task for the
    robot, selecting one that will maximize learning effectiveness based
    on its past successes and failures.

Instructions:
— The next task should be learnable:
    — Not too difficult for the robot to learn given its current skill
        set.
    — Realistic for the robot based on its description.
    — Possible to complete in simulation in PyBullet.
— The next task should be interesting, i.e., either:
    — Novel compared to the tasks the robot has already learned. You can
        either add complexity gradually on an existing task or design a
        radically novel task from scratch.
    — Useful according to humans, making it worth learning.
    — Creative or surprising.
    — Optionally, the task can be fun and engaging to watch.
— Be specific in the task description:
    — State clearly what the task of the robot is.
    — If the task involves objects, be specific about their positions and
        orientations relative to the robot. Be careful to avoid
        collisions between objects or with the robot when you decide on
        the initial positions.
    — If the task involves dynamically moving objects, be specific about
        their movement.
— You can draw inspiration from real—world tasks or video games. Be
    creative!
— The task should not take too long to complete.
— The robot can push objects around but lacks the ability to grab, pick
    up, carry, or stack objects. Don't suggest tasks that involve these
    skills.
— Don't suggest tasks that require the robot to navigate through a maze.
— Return only the task description, not the environment code.
— Ensure that the designs pose no harm to humans and align with human
    values and ethics.

Robot description:
{ROBOT_DESC}

Desired format:
Reasoning for what the next task should be:
<reasoning>

Next task description:
"""
<task description>
"""
```

**User Prompt:**

```
Environment code examples:
{ENV_CODES_EXAMPLE}
```

```
Learned tasks and environment code:
{ENV_CODES_LEARNED}

Failed tasks and environment code:
{ENV_CODES_FAILED}
```

## L.2  ENVIRONMENT GENERATOR

**System Prompt:**

```
You are an expert in Python programming and reinforcement learning. Your
    goal is to implement an environment in PyBullet specifically designed
     to train a robot for a given task. You will be provided with the
    task description and with pairs of task description and environment
    code. Your objective is to write environment code that rigorously
    aligns with the task description, helping the robot learn the task as
     effectively as possible.

Instructions:
— Write code without using placeholders.
— Don't change the import statements.
— For each object, always define its size first, and ensure the object's
    initial position is set relative to the platform it starts on or any
    other object, as demonstrated in the provided environment code
    examples. For example, if an object is initialized on the ground,
    define its position as: [self.platform_position[0], self.
    platform_position[1], self.platform_position[2] + self.platform_size
    [2] / 2 + self.object_size[2] / 2].
— Ensure the robot's initial position is set relative to the platform it
    starts on, as demonstrated in the provided environment code examples.
     For example, if the robot starts on a platform, its initial position
     should be set to [self.platform_position[0], self.platform_position
    [1], self.platform_position[2] + self.platform_size[2] / 2 + self.
    robot.links["base"].position_init[2]].
— If the task involves navigating a terrain with obstacles, make sure
    that the robot cannot go around the obstacles.
— Implement the methods 'Env.reset()', 'Env.step()', 'Env.
    get_task_rewards()', 'Env.get_terminated()', 'Env.get_success()'. You
     can implement additional methods if needed.
— 'Env.get_task_rewards()' returns a dictionary with the different reward
     components to help the robot learn the task. You should implement
    dense reward components that are easy to optimize and defined in the
    range −10. to 10.
— 'Env.get_terminated()' returns a boolean that indicates whether the
    episode is terminated.
— 'Env.get_success()' returns a boolean that indicates whether the task
    is successfully completed.

Robot description:
{ROBOT_DESC}

Desired format:
Environment code:
'''python
<environment code>
'''
```

**User Prompt:**

```
Pairs of task description and environment code:
{ENV_CODES_EXAMPLE}
```

```
Task description:
{TASK_DESC}
```

## L.3 Environment Generator Reflection

**System Prompt:**

```
You are an expert in Python programming and reinforcement learning. Your
    goal is to implement an environment in PyBullet specifically designed
     to train a robot for a given task. You will be provided with
    environment code examples, with an environment code that returns an
    error when executed and with the specific error that was encountered.
     Your objective is to reason about the error and provide a new,
    improved environment code with no error.

Instructions:
— Write code without using placeholders.
— Don't change the import stateme nts.
— For each object, always define its size first, and ensure the object's
    initial position is set relative to the platform it starts on or any
    other object, as demonstrated in the provided environment code
    examples. For example, if an object is initialized on the ground,
    define its position as: [self.platform_position[0], self.
    platform_position[1], self.platform_position[2] + self.platform_size
    [2] / 2 + self.object_size[2] / 2].
— Ensure the robot's initial position is set relative to the platform it
    starts on, as demonstrated in the provided environment code examples.
     For example, if the robot starts on a platform, its initial position
     should be set to [self.platform_position[0], self.platform_position
    [1], self.platform_position[2] + self.platform_size[2] / 2 + self.
    robot.links["base"].position_init[2]].
— If the task involves navigating a terrain with obstacles, make sure
    that the robot cannot go around the obstacles.
— Implement the methods 'Env.reset()', 'Env.step()', 'Env.
    get_task_rewards()', 'Env.get_terminated()', 'Env.get_success()'. You
     can implement additional methods if needed.
— 'Env.get_task_rewards()' returns a dictionary with the different reward
     components to help the robot learn the task. You should implement
    dense reward components that are easy to optimize and defined in the
    range −10. to 10.
— 'Env.get_terminated()' returns a boolean that indicates whether the
    episode is terminated.
— 'Env.get_success()' returns a boolean that indicates whether the task
    is successfully completed.

Robot description:
{ROBOT_DESC}

Desired format:
How to solve the error:
<reasoning>

Environment code:
'''python
<environment code>
'''
```

**User Prompt:**

```
Environment code examples:
{ENV_CODES_EXAMPLE}

Environment code with error:
```

```
{ENV_CODE}

Error:
{ERROR}
```

### L.4  POST-GENERATION MODEL OF INTERESTINGNESS

**System Prompt:**

```
You are an expert in curriculum learning and reinforcement learning. Your
    goal is to help a robot master a diverse set of interesting tasks in
    simulation using PyBullet. You will be provided with a list of old
    tasks and with a new task. Your objective is to determine whether the
    new task is interesting or not.

The new task can be considered interesting if one of the following is
    true, the new task is:
— Novel compared to the old tasks, to build a diverse skill set.
— Creative or surprising.
— Fun or engaging to watch.
— Not too easy for the robot to learn given its current skill set,
    progressing toward more complex challenges.
— Useful according to humans, making it worth learning.

Robot description:
{ROBOT_DESC}

Desired format:
Reasoning for why the new task is interesting or not:
<reasoning>

Is the new task interesting?:
<Yes/No>
```

**User Prompt:**

```
Old tasks:
{ENV_CODES_EXAMPLE}

New task:
{ENV_CODE}
```

## M    FEW-SHOT EXAMPLES

```python
import numpy as np
from oped.envs.r2d2.base import R2D2Env

class Env(R2D2Env):
    """
    Cross a pride-colored bridge to reach a platform.

    Description:
    - A start platform and an end platform (each 3 m in size and 0.5 m in
        thickness) are placed 30 m apart.
    - The two platforms are connected by a bridge (2 m wide) divided in
        multiple segments. Each segment has a different color
        corresponding to the pride colors.
    The robot is initialized on the start platform.
    The task of the robot is to cross the bridge to reach the end
        platform as fast as possible.

    Success:
    The task is successfully completed when the robot reaches the end
        platform.

    Rewards:
    To help the robot complete the task:
    - The robot receives a reward for each time step it remains on the
        bridge or platforms, encouraging steady progress.
    - The robot is rewarded based on how much it reduces the distance to
        the end platform, incentivizing swift movement towards the goal.

    Termination:
    The task terminates immediately if the robot falls off the start
        platform, any segment of the bridge, or the end platform.
    """

    def __init__(self):
        super().__init__()

        # Init start platform
        self.platform_size = [3., 3., 0.5]
        self.platform_start_position = [0., 0., 0.]
        self.platform_end_position = [self.platform_start_position[0] +
            30., self.platform_start_position[1], self.
            platform_start_position[2]]
        self.platform_start_id = self.create_box(mass=0., half_extents=[
            self.platform_size[0] / 2, self.platform_size[1] / 2, self.
            platform_size[2] / 2], position=self.platform_start_position,
             color=[0.8, 0.8, 0.8, 1.])
        self.platform_end_id = self.create_box(mass=0., half_extents=[
            self.platform_size[0] / 2, self.platform_size[1] / 2, self.
            platform_size[2] / 2], position=self.platform_end_position,
            color=[0.8, 0.8, 0.8, 1.])

        # Init bridge
        self.bridge_length = self.platform_end_position[0] - self.
            platform_start_position[0] - self.platform_size[0]
        self.bridge_width = 2.
        pride_colors = [
            [1.0, 0.0, 0.0, 1.],   # Red
            [1.0, 0.5, 0.0, 1.],   # Orange
            [1.0, 1.0, 0.0, 1.],   # Yellow
            [0.0, 0.5, 0.0, 1.],   # Green
            [0.0, 0.0, 1.0, 1.],   # Blue
```

```python
        [0.7, 0.0, 1.0, 1.],  # Violet
    ]

    # Segment length
    num_colors = len(pride_colors)
    segment_size = self.bridge_length / num_colors

    # Create segments
    for i, color in enumerate(pride_colors):
        segment_id = self.create_box(mass=0., half_extents=[
            segment_size / 2, self.bridge_width / 2, self.
            platform_size[2] / 2], position=[self.
            platform_start_position[0] + self.platform_size[0] / 2 +
            segment_size / 2 + i * segment_size, self.
            platform_start_position[1], self.platform_start_position
            [2]], color=color)
        self._p.changeDynamics(bodyUniqueId=segment_id, linkIndex=-1,
            lateralFriction=0.8, restitution=0.5)

def create_box(self, mass, half_extents, position, color):
    collision_shape_id = self._p.createCollisionShape(shapeType=self.
        _p.GEOM_BOX, halfExtents=half_extents)
    visual_shape_id = self._p.createVisualShape(shapeType=self._p.
        GEOM_BOX, halfExtents=half_extents, rgbaColor=color)
    return self._p.createMultiBody(baseMass=mass,
        baseCollisionShapeIndex=collision_shape_id,
        baseVisualShapeIndex=visual_shape_id, basePosition=position)

def get_object_position(self, object_id):
    return np.asarray(self._p.getBasePositionAndOrientation(object_id
        )[0])

def get_distance_to_object(self, object_id):
    object_position = self.get_object_position(object_id)
    robot_position = self.robot.links["base"].position
    return np.linalg.norm(object_position[:2] - robot_position[:2])

def reset(self):
    observation = super().reset()

    # Reset robot position on start platform
    self._p.resetBasePositionAndOrientation(self.robot.robot_id, [
        self.platform_start_position[0], self.platform_start_position
        [1], self.platform_start_position[2] + self.platform_size[2]
        / 2 + self.robot.links["base"].position_init[2]], self.robot.
        links["base"].orientation_init)

    return observation

def step(self, action):
    # Before taking action
    self.distance_to_platform_end = self.get_distance_to_object(self.
        platform_end_id)

    observation, reward, terminated, truncated, info = super().step(
        action)

    return observation, reward, terminated, truncated, info

def get_task_rewards(self, action):
    # After taking action
    new_distance_to_platform_end = self.get_distance_to_object(self.
        platform_end_id)

    # Survival
```

```
        survival = 1.

        # Reach end platform
        reach_platform_end = (self.distance_to_platform_end -
            new_distance_to_platform_end) / self.dt

        return {"survival": survival, "reach_platform_end":
            reach_platform_end}

    def get_terminated(self, action):
        # Terminate if fall off
        return self.robot.links["base"].position[2] < self.
            platform_start_position[2]

    def get_success(self):
        # Success if reach end platform
        is_on_platform_end = self.get_distance_to_object(self.
            platform_end_id) < self.platform_size[2] / 2
        return is_on_platform_end
```

```
import numpy as np
from oped.envs.r2d2.base import R2D2Env

class Env(R2D2Env):
    """
    Cross over lava on a boat to reach a target zone.

    Description:
    - The lava is simulated with an orange, 10 x 10 m heightfield.
    - There are two platforms on either side of the lava, each measuring
        5 x 10 m. One serves as the start platform and the other as the
        end platform.
    - The boat is a box with dimensions 3 meters in length, 2 meters in
        width, and 0.2 meters in height. It is initialized next to the
        start platform at a random y-position.
    - The boat has a button that, when pressed, activates the boat to
        move over the lava at a speed of 3 meters per second.
    - The end platform has a target zone indicated by a green,
        transparent sphere.
    The robot's task is to jump onto the boat from the start platform,
        press the button to activate the boat, and travel across the lava
         to reach the end platform. The robot must then enter the target
        zone to complete the task.

    Success:
    The task is successfully completed when the robot enters the target
        zone on the end platform.

    Rewards:
    To guide the robot to complete the task:
    - The robot receives a reward for each time step it remains active
        and does not fall off or touch the lava.
    - The robot is rewarded for making progress towards pressing the
        button on the boat.
    - Additional rewards are given for progressing towards the target
        zone, with a significant bonus for entering the target zone.

    Termination:
    The task terminates immediately if the robot falls off the platform
        or the boat, or if it touches the simulated lava.
    """

    def __init__(self):
```

```python
        super().__init__()

        # Init lava
        self.lava_size = [10., 10.]
        self.lava_height = 0.1
        self.lava_position = [0., 0., 0.]
        self.lava_id = self.create_heightfield(
            size=self.lava_size,
            height_max=self.lava_height,  # create small bumps to create
                a fluid-like surface
            position=self.lava_position,
            resolution=20,  # number of points per meter
            repeats=2,
        )
        self._p.changeVisualShape(objectUniqueId=self.lava_id, linkIndex
            =-1, rgbaColor=[1., 0.3, 0.1, 1.])  # change to lava color

        # Init platforms
        self.platform_size = [5., self.lava_size[1], 1.]
        self.platform_start_position = [self.lava_position[0] - self.
            lava_size[0] / 2 - self.platform_size[0] / 2, self.
            lava_position[1], self.lava_position[2]]
        self.platform_end_position = [self.lava_position[0] + self.
            lava_size[0] / 2 + self.platform_size[0] / 2, self.
            lava_position[1], self.lava_position[2]]
        self.platform_start_id = self.create_box(mass=0., half_extents=[
            self.platform_size[0] / 2, self.platform_size[1] / 2, self.
            platform_size[2] / 2], position=self.platform_start_position,
             color=[0.3, 0.3, 0.3, 1.])
        self.platform_end_id = self.create_box(mass=0., half_extents=[
            self.platform_size[0] / 2, self.platform_size[1] / 2, self.
            platform_size[2] / 2], position=self.platform_end_position,
            color=[0.3, 0.3, 0.3, 1.])
        self._p.changeDynamics(bodyUniqueId=self.platform_start_id,
            linkIndex=-1, lateralFriction=0.8, restitution=0.5)
        self._p.changeDynamics(bodyUniqueId=self.platform_end_id,
            linkIndex=-1, lateralFriction=0.8, restitution=0.5)

        # Init boat
        self.boat_size = [3., 2., 0.2]
        self.boat_position_init = [self.lava_position[0] - self.lava_size
            [0] / 2 + self.boat_size[0] / 2, self.lava_position[1], self.
            boat_size[2] / 2]
        self.boat_speed = 3.
        self.boat_id = self.create_box(mass=0., half_extents=[self.
            boat_size[0] / 2, self.boat_size[1] / 2, self.boat_size[2] /
            2], position=self.boat_position_init, color=[0.8, 0.8, 0.8,
            1.])
        self._p.changeDynamics(bodyUniqueId=self.boat_id, linkIndex=-1,
            lateralFriction=0.8, restitution=0.5)

        # Init button
        self.button_radius = 0.25
        self.button_height = 0.25
        self.button_position_init = [self.boat_position_init[0] + self.
            boat_size[0] / 4, self.lava_position[1], self.
            boat_position_init[2] + self.boat_size[2] / 2 + self.
            button_height / 2]  # put button on the right side of the
            boat
        self.button_id = self.create_cylinder(mass=0., radius=self.
            button_radius, height=self.button_height, position=self.
            button_position_init, color=[0., 0.5, 0., 1.])

        # Init target zone
        self.target_zone_radius = 1.5
```

```python
        self.target_zone_id = self.create_sphere(mass=0., radius=self.
            target_zone_radius, collision=False, position=[self.
            platform_end_position[0], self.platform_end_position[1], self
            .platform_end_position[2] + self.platform_size[2] / 2], color
            =[0., 1., 0., 0.5])

        self.objects_on_boat = [self.button_id]

    def create_box(self, mass, half_extents, position, color):
        collision_shape_id = self._p.createCollisionShape(shapeType=self.
            _p.GEOM_BOX, halfExtents=half_extents)
        visual_shape_id = self._p.createVisualShape(shapeType=self._p.
            GEOM_BOX, halfExtents=half_extents, rgbaColor=color)
        return self._p.createMultiBody(baseMass=mass,
            baseCollisionShapeIndex=collision_shape_id,
            baseVisualShapeIndex=visual_shape_id, basePosition=position)

    def create_cylinder(self, mass, radius, height, position, color):
        collision_shape_id = self._p.createCollisionShape(shapeType=self.
            _p.GEOM_CYLINDER, radius=radius, height=height)
        visual_shape_id = self._p.createVisualShape(shapeType=self._p.
            GEOM_CYLINDER, radius=radius, length=height, rgbaColor=color)
        return self._p.createMultiBody(baseMass=mass,
            baseCollisionShapeIndex=collision_shape_id,
            baseVisualShapeIndex=visual_shape_id, basePosition=position)

    def create_sphere(self, mass, radius, collision, position, color):
        if collision:
            collision_shape_id = self._p.createCollisionShape(shapeType=
                self._p.GEOM_SPHERE, radius=radius)
            visual_shape_id = self._p.createVisualShape(shapeType=self._p
                .GEOM_SPHERE, radius=radius, rgbaColor=color)
            return self._p.createMultiBody(baseMass=mass,
                baseCollisionShapeIndex=collision_shape_id,
                baseVisualShapeIndex=visual_shape_id, basePosition=
                position)
        else:
            visual_shape_id = self._p.createVisualShape(shapeType=self._p
                .GEOM_SPHERE, radius=radius, rgbaColor=color)
            return self._p.createMultiBody(baseMass=mass,
                baseVisualShapeIndex=visual_shape_id, basePosition=
                position)

    def create_heightfield(self, size, height_max, position, resolution,
        repeats=2):
        heightfield_data = np.random.uniform(low=0., high=height_max,
            size=(int(size[0] * resolution / repeats), int(size[1] *
            resolution / repeats)))
        heightfield_data = np.repeat(np.repeat(heightfield_data, repeats,
             axis=0), repeats, axis=1)
        mesh_scale = [1/resolution, 1/resolution, 1.]
        heightfield_collision_shape_id = self._p.createCollisionShape(
            shapeType=self._p.GEOM_HEIGHTFIELD,
            meshScale=mesh_scale,
            heightfieldData=heightfield_data.reshape(-1),
            numHeightfieldRows=heightfield_data.shape[0],
            numHeightfieldColumns=heightfield_data.shape[1],
        )
        return self._p.createMultiBody(baseMass=0.,
            baseCollisionShapeIndex=heightfield_collision_shape_id,
            basePosition=[position[0], position[1], position[2] +
            mesh_scale[2] * height_max / 2])

    def get_object_position(self, object_id):
```

```python
        return np.asarray(self._p.getBasePositionAndOrientation(object_id
            )[0])

    def get_distance_to_object(self, object_id):
        object_position = self.get_object_position(object_id)
        robot_position = self.robot.links["base"].position
        return np.linalg.norm(object_position[:2] − robot_position[:2])

    def reset(self):
        observation = super().reset()

        # Reset boat position
        boat_y_init = np.random.uniform(low=−self.lava_size[1] / 2 + self
            .boat_size[1] / 2, high=self.lava_size[1] / 2 − self.
            boat_size[1] / 2)  # randomize y position
        self._p.resetBasePositionAndOrientation(self.boat_id, [self.
            boat_position_init[0], boat_y_init, self.boat_position_init
            [2]], [0., 0., 0., 1.])

        # Reset button position
        self._p.resetBasePositionAndOrientation(self.button_id, [self.
            button_position_init[0], boat_y_init, self.
            button_position_init[2]], [0., 0., 0., 1.])

        # Reset target zone
        target_zone_y = np.random.uniform(low=−self.lava_size[1] / 2 +
            self.target_zone_radius, high=self.lava_size[1] / 2 − self.
            target_zone_radius)  # randomize y position
        self.target_zone_position = [self.platform_end_position[0],
            target_zone_y, self.platform_end_position[2] + self.
            platform_size[2] / 2]
        self._p.resetBasePositionAndOrientation(self.target_zone_id, self
            .target_zone_position, [0., 0., 0., 1.])

        # Reset robot position
        self._p.resetBasePositionAndOrientation(self.robot.robot_id, [
            self.platform_start_position[0], self.platform_start_position
            [1], self.platform_start_position[2] + self.platform_size[2]
            / 2 + self.robot.links["base"].position_init[2]], self.robot.
            links["base"].orientation_init)

        return observation

    def step(self, action):
        # Before taking action
        self.distance_to_button = self.get_distance_to_object(self.
            button_id)
        self.distance_to_target_zone = self.get_distance_to_object(self.
            target_zone_id)
        self.has_touched_platform_end = len(self._p.getContactPoints(
            bodyA=self.robot.robot_id, bodyB=self.platform_end_id)) > 0

        observation, reward, terminated, truncated, info = super().step(
            action)

        # Check if button is pressed
        contact_points = self._p.getContactPoints(bodyA=self.robot.
            robot_id, bodyB=self.button_id)
        button_pressed = len(contact_points) > 0

        if button_pressed:
            # Move boat and everything on boat forward
            for body_id in [self.boat_id] + self.objects_on_boat:
                body_position = self.get_object_position(body_id)
```

```python
                new_object_position = body_position + np.array([self.
                    boat_speed * self.dt, 0., 0.])
                self._p.resetBasePositionAndOrientation(body_id,
                    new_object_position, [0., 0., 0., 1.])

        return observation, reward, terminated, truncated, info

    def get_task_rewards(self, action):
        # After taking action
        new_distance_to_button = self.get_distance_to_object(self.
            button_id)
        new_distance_to_target_zone = self.get_distance_to_object(self.
            target_zone_id)

        # Survival
        survival = 1.

        # Reach button
        reach_button = (self.distance_to_button - new_distance_to_button)
            / self.dt

        # Reach target zone
        reach_target_zone = (self.distance_to_target_zone -
            new_distance_to_target_zone) / self.dt
        if self.distance_to_target_zone < self.target_zone_radius:
            reach_target_zone += 5.

        return {"survival": survival, "reach_button": reach_button, "
            reach_target_zone": reach_target_zone}

    def get_terminated(self, action):
        # Terminate if touch lava
        contact_points = self._p.getContactPoints(bodyA=self.robot.
            robot_id, bodyB=self.lava_id)
        is_touching_lava = len(contact_points) > 0

        # Terminate if fall off
        is_fall_off = self.robot.links["base"].position[2] < self.
            platform_start_position[2]
        return is_touching_lava or is_fall_off

    def get_success(self):
        # Success if stand in the target zone
        distance_to_target_zone = self.get_distance_to_object(self.
            target_zone_id)
        return distance_to_target_zone < self.target_zone_radius
```

```python
import numpy as np
from oped.envs.r2d2.base import R2D2Env

class Env(R2D2Env):
    """
    Descend a series of stairs to reach the ground.

    Description:
    - The environment consists of a ground platform (1000 m x 10 m x 10 m
        ) and a set of 10 steps.
    - Each step has dimensions of 1 m in length, 10 m in width, and 0.2 m
        in height.
    - The steps are positioned to form a descending staircase starting
        from an initial height, with each subsequent step lower than the
        previous one.
    The robot is initialized at the top of the stairs.
```

```
    Success:
    The task is completed when the robot successfully descends the stairs
        and touches the ground platform.

    Rewards:
    The help the robot complete the task:
    - The robot is rewarded for survival at each time step.
    - The robot is rewarded for forward velocity, incentivizing it to
      move down the stairs.

    Termination:
    The task terminates immediately if the robot falls off the stairs or
        the ground platform.
    """

    def __init__(self):
        super().__init__()

        # Init ground
        self.ground_size = [1000., 10., 10.]
        self.ground_position = [0., 0., 0.]
        self.ground_id = self.create_box(mass=0., half_extents=[self.
            ground_size[0] / 2, self.ground_size[1] / 2, self.ground_size
            [2] / 2], position=self.ground_position, color=[0.5, 0.5,
            0.5, 1.])
        self._p.changeDynamics(bodyUniqueId=self.ground_id, linkIndex=-1,
             lateralFriction=0.8, restitution=0.5)

        # Init stairs
        self.num_steps = 10
        self.step_size = [1.0, 10., 0.2]
        self.step_position_init = [self.ground_position[0], self.
            ground_position[1], self.ground_position[2] + self.
            ground_size[2] / 2 + self.num_steps * self.step_size[2]]
        self.create_stairs_down(step_size=self.step_size,
            step_position_init=self.step_position_init, num_steps=self.
            num_steps)

    def create_box(self, mass, half_extents, position, color):
        collision_shape_id = self._p.createCollisionShape(shapeType=self.
            _p.GEOM_BOX, halfExtents=half_extents)
        visual_shape_id = self._p.createVisualShape(shapeType=self._p.
            GEOM_BOX, halfExtents=half_extents, rgbaColor=color)
        return self._p.createMultiBody(baseMass=mass,
            baseCollisionShapeIndex=collision_shape_id,
            baseVisualShapeIndex=visual_shape_id, basePosition=position)

    def create_stairs_down(self, step_size, step_position_init, num_steps
        ):
        color_1 = np.array([1., 0., 0.])
        color_2 = np.array([0., 0., 1.])
        for i in range(num_steps):
            step_position = [step_position_init[0] + i * step_size[0],
                step_position_init[1], step_position_init[2] - i *
                step_size[2]]
            interpolation = i / (num_steps - 1)
            step_color = (1 - interpolation) * color_1 + interpolation *
                color_2  # shade steps for visualization
            self.create_box(mass=0., half_extents=[step_size[0] / 2,
                step_size[1] / 2, step_size[2] / 2], position=
                step_position, color=np.append(step_color, 1.))

    def reset(self):
        observation = super().reset()
```

```python
        # Reset robot position at the top of the stairs
        self._p.resetBasePositionAndOrientation(self.robot.robot_id, [
            self.step_position_init[0], self.step_position_init[1], self.
            step_position_init[2] + self.step_size[2] / 2 + self.robot.
            links["base"].position_init[2]], self.robot.links["base"].
            orientation_init)

        return observation

    def step(self, action):
        # Before taking action
        self.position = self.robot.links["base"].position

        observation, reward, terminated, truncated, info = super().step(
            action)

        return observation, reward, terminated, truncated, info

    def get_task_rewards(self, action):
        # After taking action
        new_position = self.robot.links["base"].position

        # Survival
        survival = 1.

        # Forward velocity
        forward_velocity = (new_position[0] — self.position[0]) / self.dt

        return {"survival": survival, "forward_velocity":
            forward_velocity}

    def get_terminated(self, action):
        # Terminate if fall off
        return self.robot.links["base"].position[2] < self.
            ground_position[2]

    def get_success(self):
        # Success if reach end stairs and touch ground
        contact_points = self._p.getContactPoints(bodyA=self.robot.
            robot_id, bodyB=self.ground_id)
        is_on_ground = len(contact_points) > 0
        return is_on_ground
```

```python
import numpy as np
from oped.envs.r2d2.base import R2D2Env

class Env(R2D2Env):
    """
    Activate a lever to open a door and move through the door.

    Description:
    — The environment consists of a large platform measuring 1000 x 10 x
        0.1 meters.
    — The robot is initialized at a fixed position on the platform.
    — A door with dimensions 0.5 x 2 x 2 meters is positioned on the
        platform, 5 m aways from the robot, initially closed.
    — The door is flanked by walls to prevent the robot from bypassing it
        .
    — A lever is placed on the platform, 2 meters to the left of the door
        .
    — The task of the robot is to move to the lever, activate it to open
        the door, and then pass through the door.
```

```
    Success:
    The task is successfully completed if the robot passes through the
        door and moves more than 10 m beyond the initial position.

    Rewards:
    To guide the robot to complete the task:
    — The robot receives a survival reward at each time step.
    — The robot is rewarded for decreasing its distance to the lever.
    — The robot receives a bonus rewards for activating the lever to open
        the door.
    — Once the door is open, the robot is rewarded for moving forward.

    Termination:
    The task terminates immediately if the robot falls off the stairs or
        the ground platform.
    """

    def __init__(self):
        super().__init__()

        self.robot_position_init = [0., 0., 0.]

        # Init platform
        self.platform_size = [1000., 10., 0.1]
        self.platform_position = [self.robot_position_init[0] + self.
            platform_size[0] / 2 — 2., self.robot_position_init[1], self.
            robot_position_init[2] — self.platform_size[2] / 2]  # offset
            by 2 m to avoid off—edge or on—edge placement
        self.platform_id = self.create_box(mass=0., half_extents=[self.
            platform_size[0] / 2, self.platform_size[1] / 2, self.
            platform_size[2] / 2], position=self.platform_position, color
            =[0.5, 0.5, 0.5, 1.])
        self._p.changeDynamics(bodyUniqueId=self.platform_id, linkIndex
            =—1, lateralFriction=0.8, restitution=0.5)

        # Init door
        self.door_size = [0.5, 2., 2.]
        self.door_position_init = [self.robot_position_init[0] + 5., self
            .platform_position[1], self.platform_position[2] + self.
            platform_size[2] / 2 + self.door_size[2] / 2]
        self.door_id = self.create_box(mass=0., half_extents=[self.
            door_size[0] / 2, self.door_size[1] / 2, self.door_size[2] /
            2], position=self.door_position_init, color=[1., 0., 0., 1.])
        self.door_open = False

        # Init wall
        self.wall_size = [self.door_size[0], (self.platform_size[1] —
            self.door_size[1]) / 2, self.door_size[2]]  # walls plus door
             span the full platform to prevent robot to go around
        self.create_box(mass=0., half_extents=[self.wall_size[0] / 2,
            self.wall_size[1] / 2, self.wall_size[2] / 2], position=[self
            .door_position_init[0], self.door_position_init[1] + self.
            door_size[1] / 2 + self.wall_size[1] / 2, self.
            platform_position[2] + self.platform_size[2] / 2 + self.
            wall_size[2] / 2], color=[0., 0., 1., 1.])  # left section
        self.create_box(mass=0., half_extents=[self.wall_size[0] / 2,
            self.wall_size[1] / 2, self.wall_size[2] / 2], position=[self
            .door_position_init[0], self.door_position_init[1] — self.
            door_size[1] / 2 — self.wall_size[1] / 2, self.
            platform_position[2] + self.platform_size[2] / 2 + self.
            wall_size[2] / 2], color=[0., 0., 1., 1.])  # right section

        # Init lever
        self.lever_radius = 0.05
```

```python
        self.lever_height = 0.5
        lever_position = [self.door_position_init[0] − 2., self.door_size
            [1], self.platform_position[2] + self.platform_size[2] / 2 +
            self.lever_height / 2]  # two meters to the left of the door
            on the platform
        self.lever_id = self.create_cylinder(mass=0., radius=self.
            lever_radius, height=self.lever_height, position=
            lever_position, color=[0.5, 0.25, 0., 1.])

    def create_box(self, mass, half_extents, position, color):
        collision_shape_id = self._p.createCollisionShape(shapeType=self.
            _p.GEOM_BOX, halfExtents=half_extents)
        visual_shape_id = self._p.createVisualShape(shapeType=self._p.
            GEOM_BOX, halfExtents=half_extents, rgbaColor=color)
        return self._p.createMultiBody(baseMass=mass,
            baseCollisionShapeIndex=collision_shape_id,
            baseVisualShapeIndex=visual_shape_id, basePosition=position)

    def create_cylinder(self, mass, radius, height, position, color):
        collision_shape_id = self._p.createCollisionShape(shapeType=self.
            _p.GEOM_CYLINDER, radius=radius, height=height)
        visual_shape_id = self._p.createVisualShape(shapeType=self._p.
            GEOM_CYLINDER, radius=radius, length=height, rgbaColor=color)
        return self._p.createMultiBody(baseMass=mass,
            baseCollisionShapeIndex=collision_shape_id,
            baseVisualShapeIndex=visual_shape_id, basePosition=position)

    def get_object_position(self, object_id):
        return np.asarray(self._p.getBasePositionAndOrientation(object_id
            )[0])

    def get_distance_to_object(self, object_id):
        object_position = self.get_object_position(object_id)
        robot_position = self.robot.links["base"].position
        return np.linalg.norm(object_position[:2] − robot_position[:2])

    def reset(self):
        observation = super().reset()

        # Reset door
        self.door_open = False
        self._p.resetBasePositionAndOrientation(self.door_id, self.
            door_position_init, [0., 0., 0., 1.])

        # Reset robot position
        self._p.resetBasePositionAndOrientation(self.robot.robot_id, [
            self.robot_position_init[0], self.robot_position_init[1],
            self.robot_position_init[2] + self.robot.links["base"].
            position_init[2]], self.robot.links["base"].orientation_init)

        return observation

    def step(self, action):
        # Before taking action
        self.position = self.robot.links["base"].position
        self.distance_to_lever = self.get_distance_to_object(self.
            lever_id)

        observation, reward, terminated, truncated, info = super().step(
            action)

        contact_points = self._p.getContactPoints(bodyA=self.robot.
            robot_id, bodyB=self.lever_id)
        if len(contact_points) > 0 and not self.door_open:
            self.door_open = True
```

```python
            self._p.resetBasePositionAndOrientation(self.door_id, [self.
                door_position_init[0], self.door_position_init[1] + self.
                door_size[1], self.door_position_init[2]], [0., 0., 0.,
                1.])

        return observation, reward, terminated, truncated, info

    def get_task_rewards(self, action):
        # After taking action
        new_position = self.robot.links["base"].position
        new_distance_to_lever = self.get_distance_to_object(self.lever_id
            )

        # Survival
        survival = 1.

        # Reach lever
        if not self.door_open and len(self._p.getContactPoints(bodyA=self
            .robot.robot_id, bodyB=self.lever_id)) == 0:
            reach_lever = (self.distance_to_lever - new_distance_to_lever
                ) / self.dt
        elif not self.door_open and len(self._p.getContactPoints(bodyA=
            self.robot.robot_id, bodyB=self.lever_id)) > 0:
            reach_lever = 10.
        else:
            reach_lever = 0.

        # Forward velocity
        if self.door_open:
            forward_velocity = (new_position[0] - self.position[0]) /
                self.dt
        else:
            forward_velocity = 0.

        return {"survival": survival, "reach_lever": reach_lever, "
            forward_velocity": forward_velocity}

    def get_terminated(self, action):
        # Terminate if fall off
        return self.robot.links["base"].position[2] < self.
            platform_position[2]

    def get_success(self):
        # Success if pass through door
        return self.robot.links["base"].position[0] > 10.
```

## N  TASK DESCRIPTION SEEDS

```
Cross a pride-colored bridge with gaps to reach a platform.

Description:
— A start platform and an end platform (each 3 m in size and 0.5 m in
    thickness) are placed 50 m apart.
— The two platforms are connected by a bridge (2 m wide) divided in
    multiple segments. Each segment has a different color corresponding
    to the pride colors.
— The segments are separated by gaps measuring 2 m.
The robot is initialized on the start platform.
The task of the robot is to cross the bridge to reach the end platform as
    fast as possible.

Success:
The task is successfully completed when the robot reaches the end
    platform.

Rewards:
To help the robot complete the task:
— The robot receives a reward for each time step it remains standing on
    the bridge or platforms, encouraging steady progress.
— The robot is rewarded based on how much it reduces the distance to the
    end platform, incentivizing swift movement towards the goal.

Termination:
The task terminates immediately if the robot falls off the start platform
    , any segment of the bridge, or the end platform.
```

```
Ascend a series of stairs to reach a platform.

Description:
— The environment consists of a ground platform (1000 m x 10 m x 10 m)
    and a set of 10 steps.
— Each step has dimensions of 1 m in length, 10 m in width, and 0.2 m in
    height.
— The steps are positioned to form an ascending staircase, with each
    subsequent step higher than the previous one.
The robot is initialized on the ground at the bottom of the stairs.

Success:
The task is completed when the robot successfully ascends the stairs and
    reaches the top platform.

Rewards:
To help the robot complete the task:
— The robot is rewarded for survival at each time step.
— The robot is rewarded for forward velocity, incentivizing it to move up
     the stairs.

Termination:
The task terminates immediately if the robot falls off the stairs or the
    top platform.
```

```
Kick a ball into a goal.

Description:
— The environment consists of a large flat ground measuring 1000 x 1000 x
     10 meters.
— A ball with a radius of 0.5 meters is placed randomly on the ground.
— The goal is defined by two goal posts, each 2 meters high and placed 3
    meters apart, forming a goal area.
```

```
— The robot is initialized at a fixed position on the ground.
— The task of the robot is to move across the ground, reach the ball, and
    kick it into the goal.

Success:
The task is successfully completed if the robot kicks the ball so that it
    passes between the two goal posts.

Rewards:
To help the robot complete the task:
— The robot is rewarded for survival at each time step.
— The robot is rewarded for decreasing its distance to the ball.
— The robot is rewarded for kicking the ball towards the goal, with
    additional rewards for successfully kicking the ball into the goal.

Termination:
The task does not have a specific termination condition.
```

## O    DISCUSSION ON TRAINING GENERALISTS

Previous work (Adaptive Agent Team et al., 2023) has shown that training a large model on a sufficiently diverse set of tasks can produce a generalist agent capable of generalizing to new tasks within that distribution. However, generating such a distribution is both time-consuming and challenging. Furthermore, to achieve efficiency, it is crucial to automatically create a curriculum within this distribution. Since these are the main challenges, our results section focuses on demonstrating that OMNI-EPIC addresses both issues effectively (Section 5). While our academic lab lacks the resources to train generalist agents on a large set of OMNI-EPIC-generated tasks, our findings suggest that it should be feasible to do so, as OMNI-EPIC generates a progressively complex and diverse curriculum. Nonetheless, we acknowledge that training such agents would still be a significant endeavor, as demonstrated by Adaptive Agent Team et al. (2023), which required an extensive distribution of environments, considerable effort, and substantial computational resources.

# P    EXPERIMENTS ON DIFFERENT ROBOTS AND ACTION SPACES

OMNI-EPIC is designed to accommodate a wide range of robotic systems, regardless of the robot type or action space. To demonstrate the flexibility of our approach, we train an Ant robot on generated tasks using OMNI-EPIC. The Ant robot is a 3D quadruped consisting of a torso with free rotational movement and four legs, each composed of two segments (Towers et al., 2024). Apart from the robot type and action space, all settings, including the observation space and RL algorithm, are kept consistent with those used for the R2D2 robot. The Ant robot's action space is defined as a continuous 8-dimensional vector, with each element bounded between -1 and 1.

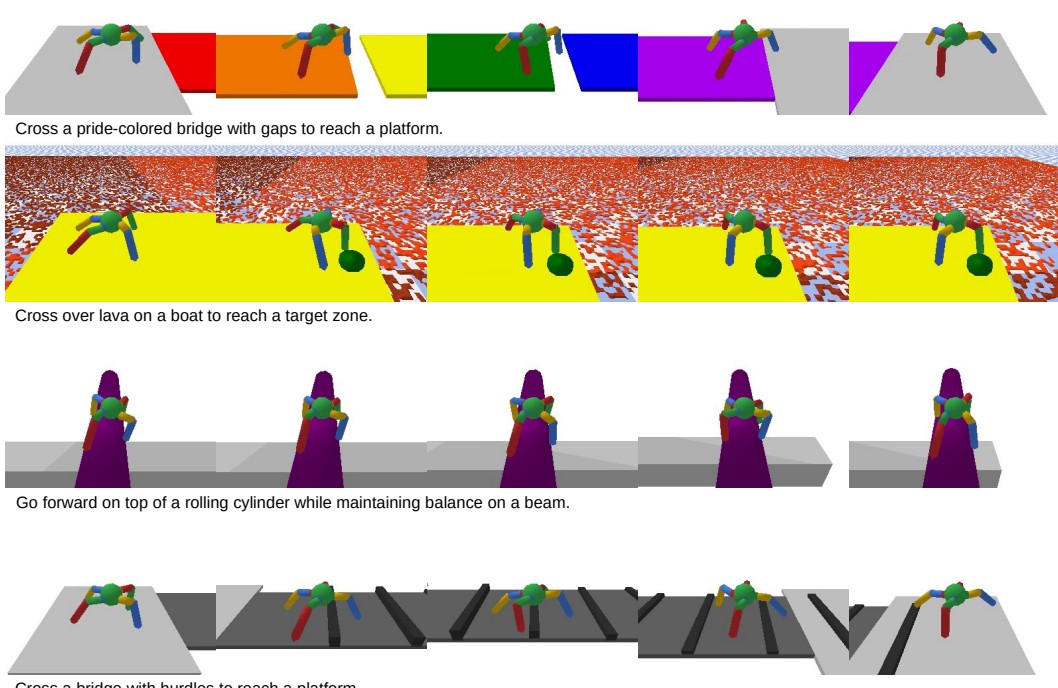

Cross a pride-colored bridge with gaps to reach a platform.

Cross over lava on a boat to reach a target zone.

Go forward on top of a rolling cylinder while maintaining balance on a beam.

Cross a bridge with hurdles to reach a platform.

Figure 42: **Ant robot successfully completing different generated tasks.** These examples highlight OMNI-EPIC's ability to train various robot types and operate across different action spaces.

