# OpenReview forum: "OMNI-EPIC: Open-endedness via Models of human Notions of Interestingness with Environments Programmed in Code"
_ICLR.cc/2025/Conference — ICLR 2025 Poster_

### Official Review · Reviewer_xWEx · 2024-10-24

**Soundness:** 2
**Presentation:** 4
**Contribution:** 3
**Rating:** 8
**Confidence:** 5

**Summary:**

OMNI-EPIC is a novel framework that aims to create truly open-ended learning environments by combining foundation models with reinforcement learning. The system generates new tasks (descriptions + code) with potential applications for open-ended learning.

Overall I really like the ambition of this paper, and I think it would be good to show new directions to the community. It's tackling a very important problem and proposes a sensible approach to tackle it.  This said, I think the claims made in the paper are strongly exaggerated and mostly unsupported. Below I will list ideas of follow-up experiments that could help support the claims.

I feel like the computational constraints of the project associated to its ambition caused a necessary tradeoff between running the main experiments of this paper and running the sufficient set of side experiments (variations, ablations, baselines).

I will assign a score of 6 as I think this paper is already valuable in its current state (if the claims were to be toned down). I'm willing to raise the score if new experiments are shown to support the claims.

I updated my score in light of the discussion (see below)

**Strengths:**

* The idea is original and forward-looking: this is making first steps towards the future of autotelic learning / open-ended learning by extending the space of tasks an agent can generate to drive its own learning.
* The paper is well written and well motivated, the methods are well described with many details in the appendix.
* Generated tasks are fun and I appreciate the accompanying website showing task descriptions and codes.

**Weaknesses:**

* The paper doesn't make an explicit list of contributions. Is it about showing open-ended learning? Is it about extending task generation from goals and levels to full environments? Is it about this and tailoring task distributions to the agent's learning capabilities? I would appreciate a clear statement about the contributions of the paper.
* The paper makes a lot of strong claims about interestingness, diversity, curriculum learning and open-endedness, but I would argue that the experiments don't support most of them (see more below).

**Questions:**

This section includes questions and comments.

The "never-ending curriculum" claim:
* The approach does not train a single generalist agent to solve all tasks, why not? Was it tried and did it fail? I assume there would be issues with continual learning / catastrophic forgetting, did the authors find this was a problem?
* Is there transfer learning across tasks? Here, there is a different agent for each new task, but an agent for a new task is initialized to the weights of the agent for the nearest solved task. This supposedly enables some form of transfer learning, but I don't think this is empirically tested anywhere?
  * Do agents initialized this way empirically learn faster than agents trained from scratch?
  * Is training an agent in a given task generally faster when this task is invented later, than earlier? When invented later, there should be a closer nearest neighbor and if there is transfer learning (previous point), then agents should learn faster and faster as the run progresses.  This would indicate that the overall agent (collection of agents) is actually learning something on the way (getting better at achieving tasks).
* To show evidence for a curriculum effect driven by task generation, one would also need to show that the difficulty of tasks is increasing with time, i.e. by showing that the success rate of agents trained from scratch is decreasing as the experiment advances. In POET for instance, the latest tasks are shown to be out of reach for agents trained from scratch, is this the case here? Showing this would be strong evidence to support the "curriculum" claim (not sure about the "never-ending" bit).
* The claim that the task generator optimizes for learning progress is also not empirically verified, is it really shifting the distribution towards harder tasks as the experiment advances (point above)? Here one would need to show a baseline that doesn't leverage the success/failure feedback or simply is not prompted to generate tasks of intermediate difficulty and compare the relative abilities of the two algorithms to solve new tasks in the end (by taking the nearest neighbors in the respective archive and training from them). Is the "learning progress" bit actually helping training more competent agents?
* Since the agents don't seem to be trained again on past tasks, the archive does not contain any information about the current capabilities of the meta agent (population of agents). As a result, the task generator cannot generate "tasks of intermediate difficulty" for a progressing agent (it doesn't get information about the current agent). It can only generate tasks of intermediate difficulty for the "average" of all agents since the start of the experiment.
* Without evidence for transfer learning across tasks, agent (population) improvement over time and task difficulty increase over time, the claim of a "never-ending curriculum learning" is not supported and should thus be removed.

Measuring success / performance
* Success is measured by a VLM, but this VLM is not validated. Does it agree with people? I see human judgements in the video on the website, is there an evaluation of how good the VLM is using these ground truth feedbacks?
* If this signal is not reliable (as suggested early in the paper), then the final metric measuring the number of different tasks solved is also not reliable. Having an unreliable fitness metric is fine during training, but becomes a problem when it's used for evaluation and algorithms comparison.
* The "long run without learning" experiment is interesting to get a sense of the diversity of the tasks, but it would be good to know how many of these tasks are actually learnable? I feel like there is a high chance that while the code is executable, the task might be purely impossible to solve. Maybe this could be obtained by subsampling generated tasks and estimating the proportion of tasks that can be solved? The second experiment does test for solvability (through learning and the solved/failed label), but it would be interesting to know if the in-context learning of the task generator (prompted to generate tasks with intermediate difficulty) is actually working, ie if the ratio of solvable tasks is higher in the second versus the first condition?


Measuring novelty, interestingness, diversity:
* The diversity metric in Figure 4 is computed on the simulated learning variants, which means that it's not filtered for solvable tasks and it's not clear which of these are actually solvable. I feel like we care more about the diversity of solved tasks (so the diversity of learned behaviors).
* Is the PCA computed over all tasks from all algorithms? The discretization size is very important here and I'm not sure these results would hold for different step sizes?
* Wouldn't sampling in context examples uniformly in the archive generate even more diversity by increasing the possibilities for recombinations?
* Is the MoI really doing its job? In Figure 3, tasks are considered novel enough if they just change the surface form of the goal, eg "jumping across platform" is considered new if it's above water, or above lava, although these don't change anything to the task, only the color of the background. More generally, this model is not validated against human judgements but is here also used for algorithm evaluations.

Open-endedness claim:
* As far as I understand, this claim is only supported by the graph in Figure 4. I appreciate the extra criterion on whether a new problem is "interesting" but this decision mechanism is probably quite unstable, e.g. the order in which tasks are presented probably influences the judgment of whether a new task is deemed novel.
* Stating that we don't see it plateauing considering there are only 13 environments generated is true but also not a strong marker for open-endedness and shouldn't be used to make this claim.
* Support for actual curriculum learning would help support the claim for open-endedness, although the "never-ending" aspect would probably require much longer runs of the algorithm.

Related Work: The related work mentions key research in environment design but does not acknowledge the older and wider field of intrinsic motivations, which I feel is foundational for the presented work.

* Learning progress has an older history than Kanitscheider et al. LP in prediction networks dates from Schmidhuber's 91 work on curiosity (see history in https://ieeexplore.ieee.org/stamp/stamp.jsp?arnumber=5508364), and was formalized in 2007 by Oudeyer and Kaplan (https://ieeexplore.ieee.org/stamp/stamp.jsp?arnumber=4141061, https://pmc.ncbi.nlm.nih.gov/articles/PMC2533589/)
Around 2013 Oudeyer's lab has been using the "competence progress" form, measuring progress in goal-achievement / task completion to drive exploration and skill learning (https://www.sciencedirect.com/science/article/pii/S0921889012000644 and others in 2013-2014). Since 2018, it's been using with intrinsically motivated deep RL approaches too (http://proceedings.mlr.press/v97/colas19a/colas19a.pdf, https://proceedings.neurips.cc/paper_files/paper/2019/file/b6f97e6f0fd175613910d613d574d0cb-Paper.pdf)
* "Focusing on interesting tasks" is basically what intrinsic motivations are about. In Oudeyer and Kaplan's initial framework on IM, there is already the "competence-based IM" aiming at selecting the most useful goals for exploration and skill learning (https://pmc.ncbi.nlm.nih.gov/articles/PMC2533589/). Since 2018, it's been combined with deep RL and there is now a whole field of methods focusing on that (see review in https://arxiv.org/pdf/2012.09830).
* VLM success detectors: https://arxiv.org/pdf/2303.07280, I don't think Christiano 2017 was a VLM? There is no language there and the model is for a single task.



Clarification questions and other suggestions:
* Do the authors plan on releasing the code? This approach has many components and many parameters, probably making it hard to reproduce. Releasing the code would ensure reproducibility.
* "Previous attempts have resulted in pathologies when optimizing against definitions and quantifications of interestingness" → which ones? How is the current approach overcoming these issues?
* Are the task embeddings computed from the linguistic descriptions? the code? the agent trajectories? combinations of these?
* It's not so clear what happens to tasks who keep on failing to compile, are they added as failed to archive? If not added to the archive, how is this information routed back to the task generator? If added as failed, shouldn't we differentiate "failed because of compilation" and "failed because the agent couldn't solve it"?
* It would be good to show somewhere the success rates at various stages, e.g. the % of tasks that are successfully turned into executable environment code, the % of these that are accepted by the post-hoc MOI checker, the % of the latter that are successfully solved. In the long run without training, how many of the 200 iterations led to executable environment code? And how is executability checked here? By running a random policy maybe?
* I think it would generally be good to conduct experiments for variants/ablations of the task generator:
  * does the LP prompt actually generate tasks more tailored to the agent?
  * does combining it with success/failure actually help generate tasks that are more solvable and not trivial?
  * does conditioning on nearby tasks help the LP modeling and the effects above compared to sampling tasks uniformly from the archive?


Typos
* L17: "narrow distributions of environment" → +s
* L175: "OMNI-EPIC continuously generate and solve new, interesting tasks in simulation." → generates, solves

---

> ### Author Response · Authors · 2024-11-23
>
> Thank you for your detailed feedback. We now address each of your concerns and questions.
>
> ---
>
> > The paper doesn't make an explicit list of contributions… I would appreciate a clear statement about the contributions of the paper.
>
> We will make this more salient in the revised manuscript. The key contributions of this work are as follows:
> - We introduce OMNI-EPIC, which uses FMs to automatically generate learnable and interesting environments, eliminating the need for hand-crafted curricula or environments.
> - Our approach creates environments in code, including simulation environment, reward, and termination functions, marking a significant step toward Darwin Completeness by progressing toward the ability to generate virtually any computable task.
> - We show OMNI-EPIC can generate an extremely diverse set of tasks.
> - We show that RL agents trained with OMNI-EPIC learn increasingly complex tasks over time, showcasing the potential of our framework to drive self-improving AI systems.
>
> > The approach does not train a single generalist agent to solve all tasks, why not?
>
> In our paper, we aim to generate a diverse task distribution and create an effective curriculum within that distribution for either specialist or generalist agents. As mentioned in our results section, OMNI-EPIC successfully accomplished those goals, which are critical prerequisites for training specialist or generalist agents in an open-ended manner. Training generalist agents requires much more computational resources, so we chose to demonstrate OMNI-EPIC’s capabilities by training specialist agents. OMNI-EPIC generates a progressively complex and diverse set of tasks (Section 4, Section 5, new Appendix I), which, in turn, train a growing population of specialist agents. Furthermore, based on prior work by AdA Team et al. (2023), we believe our approach would also be highly beneficial for training generalist agents, albeit acknowledging the significant resources required for such an extensive evaluation. But the value of OMNI-EPIC is not just for training generalist agents, and doing so is not the problem we are trying to solve, and we thus do not believe it should be a prerequisite for publication.
>
> > Is there transfer learning across tasks?
> > Do agents initialized this way empirically learn faster than agents trained from scratch?
> > To show evidence for a curriculum effect driven by task generation, one would also need to show that the difficulty of tasks is increasing with time, i.e. by showing that the success rate of agents trained from scratch is decreasing as the experiment advances.
> > Support for actual curriculum learning would help support the claim for open-endedness
>
> We conducted an experiment in which we ran OMNI-EPIC (i.e., with transfer learning between tasks) on 25 tasks with 5 replications and compared it to training policies from scratch on the same tasks in each run (i.e., without transfer learning between tasks). OMNI-EPIC achieves a significantly higher success rate (median: 70.6%,  CI: 61.4 - 74.1) than training from scratch (median: 57.9%, CI: 44.6 - 65.5) (p < 0.05, Mann-Whitney U test) (new Appendix G.1). This demonstrates that the OMNI-EPIC agents build upon previously learned skills, creating a curriculum of increasing difficulty.
>
> > Is training an agent in a given task generally faster when this task is invented later, than earlier? When invented later, there should be a closer nearest neighbor and if there is transfer learning (previous point), then agents should learn faster and faster as the run progresses.
>
> In OMNI-EPIC, each RL agent is trained for an additional 2 million steps per new task. OMNI-EPIC dynamically generates a curriculum designed to maintain the level of learnability, ensuring that tasks become progressively more challenging. Although OMNI-EPIC leverages transfer learning from previous experiences to accelerate learning, the RL agent trained on a new, more complex task might not necessarily learn faster because the difficulty could be that much higher to keep the learning rate roughly the same.

---

> > ### Author Response · Authors · 2024-11-23
> >
> > > Here one would need to show a baseline that doesn't leverage the success/failure feedback
> > > does combining it with success/failure actually help generate tasks that are more solvable and not trivial?
> >
> > We demonstrate the importance of using both success and failure feedback through our ablation studies. Without the archive (i.e., without both success and failure examples), OMNI-EPIC w/o archive learns significantly fewer tasks and achieves substantially lower scores on the ANNECS-OMNI metric compared to OMNI-EPIC (Section 6).
> >
> > We performed a new ablation study using only success examples (omitting failure examples) (new Appendix G.3). When the task generator does not receive failed examples, the agent manages to learn only 7 out of the 23 generated tasks. In contrast, OMNI-EPIC, which uses both successful and failed examples, successfully learns a median of 15 out of 23 tasks. This demonstrates that incorporating both success and failure feedback enhances the generation of more learnable and non-trivial tasks.
> >
> > > is it really shifting the distribution towards harder tasks as the experiment advances (point above)? Here one would need to show a baseline that … is not prompted to generate tasks of intermediate difficulty
> > > does the LP prompt actually generate tasks more tailored to the agent?
> > > it would be interesting to know if the in-context learning of the task generator (prompted to generate tasks with intermediate difficulty) is actually working
> >
> > We performed an additional ablation study without learning progress in the task generator’s prompt (new Appendix G.4). In this setup, no tasks were successfully learned. In contrast, OMNI-EPIC, which uses both notions of interestingness and learning progress in the task generator’s prompt, successfully learned a median of 15/23 tasks. This highlights the importance of including learning progress in the prompt.
> >
> > > Since the agents don't seem to be trained again on past tasks, the archive does not contain any information about the current capabilities of the meta agent (population of agents). As a result, the task generator cannot generate "tasks of intermediate difficulty" for a progressing agent (it doesn't get information about the current agent). It can only generate tasks of intermediate difficulty for the "average" of all agents since the start of the experiment.
> >
> > Thank you for highlighting this limitation. We agree that selecting examples from the archive provides the task generator with only partial information about the population of agents. However, as suggested, we conducted additional ablations and experiments (see above) demonstrating that the generated environments do become progressively more complex and challenging (Appendix F). Addressing how to provide OMNI-EPIC with more detailed and up-to-date information about the agents' current capabilities remains an interesting open research question, which we plan to explore in future work. One simple solution, which we like, but is computationally expensive, is to train a generalist agent on all tasks generated so far, and provide the current performance level (and possibly recent performance levels too) to the task generator. Previous work from AdA Team et al. (2023) suggests that approach would work well.
> >
> > > Without evidence for transfer learning across tasks, agent (population) improvement over time and task difficulty increase over time, the claim of a "never-ending curriculum learning" is not supported and should thus be removed.
> >
> > We have toned down our rhetoric and hope that the evidence provided is sufficient to demonstrate that transfer learning across tasks is effective and that the task difficulty generated by OMNI-EPIC increases over time. Our point was to say that OMNI-EPIC has the chance to produce an endless stream of tasks, but we now make clear that whether it fulfills that promise is an open question worthy of future study.
> >
> > > Success is measured by a VLM, but this VLM is not validated. Does it agree with people?
> >
> > We clarified in the revised manuscript that, since our preliminary testing found that current VLMs are not yet accurate enough to serve as success detectors, we use code generated by LLMs for this purpose instead in our experiments. However, we anticipate that VLM capabilities will rapidly improve, eventually reaching the accuracy needed for effective use. Our paper only suggests that one could use VLMs as success detectors, and this will be especially helpful as they improve in the future.
> >
> > Additionally, to address your question, we conducted a new user study with 50 participants, and found a 72.7% alignment rate between human evaluations and the LLM success detector's assessments (Section 5 and new Appendix G).

---

> > > ### Author Response · Authors · 2024-11-23
> > >
> > > > Maybe this could be obtained by subsampling generated tasks and estimating the proportion of tasks that can be solved?
> > > > it would be interesting to know if the in-context learning of the task generator (prompted to generate tasks with intermediate difficulty) is actually working
> > >
> > > Following your suggestion, we sampled the first 100 of the 200 generated environments in the long run without learning for human evaluation and found that 71.5% were solvable.
> > >
> > > > I feel like we care more about the diversity of solved tasks (so the diversity of learned behaviors).
> > >
> > > We have included additional diversity plots of solved tasks in the short runs with learning (new Appendix I). While these plots show a similar qualitative trend (of OMNI-EPIC generating more diverse tasks than the controls) to the long runs without learning, the limited number of data points means the differences are not statistically significant (not all p-values < 0.05, Mann-Whitney U test). By presenting the diversity plots from the long runs without learning, we demonstrate that if OMNI-EPIC is run for a longer duration, the diversity of generated tasks becomes significantly higher compared to the controls.
> > >
> > > > Is the PCA computed over all tasks from all algorithms?
> > >
> > > Yes. We have included a clarification in the revised manuscript.
> > >
> > > > The discretization size is very important here and I'm not sure these results would hold for different step sizes?
> > >
> > > We included additional diversity plots for different step sizes (new Appendix I). We evaluated step sizes of [10, 20, 30, 40, 50], and all results show the same qualitative results and are  statistically significant (p-values < 0.05).
> > >
> > > > Wouldn't sampling in context examples uniformly in the archive generate even more diversity by increasing the possibilities for recombinations?
> > > > does conditioning on nearby tasks help the LP modeling and the effects above compared to sampling tasks uniformly from the archive?
> > >
> > > We conducted an additional ablation study in which in-context examples for the task generator were sampled uniformly (new Appendix G.2). Using uniform sampling resulted in learning only 5 out of 23 generated tasks, compared to 16 out of 23 in OMNI-EPIC, where the most similar tasks were used as in-context examples. This demonstrates that sampling in-context examples based on similarity improves the generation of learnable tasks.
> > >
> > > However, we acknowledge that uniform sampling from the archive could potentially increase task diversity. Exploring the trade-offs between learnability and diversity is an interesting area for future research.
> > >
> > > > Is the MoI really doing its job? In Figure 3, tasks are considered novel enough if they just change the surface form of the goal
> > >
> > > Thank you for raising this point. We would like to clarify that changes in surface forms, such as color, are not merely superficial in the context of RL. In RL, variations in visual attributes like color serve as a form of data augmentation, which can significantly impact the agent’s learning process and generalization capabilities. These variations can introduce new challenges and improve the robustness of the agent by requiring it to adapt to different observation inputs.
> > >
> > > > the order in which tasks are presented probably influences the judgment of whether a new task is deemed novel.
> > >
> > > Yes, the order in which tasks are generated does influence the judgment of whether a new task is considered novel, and we believe this is appropriate. For example, in scientific discovery, the novelty of a proposed concept or finding depends on the sequence of prior discoveries (e.g., proposing a smart phone in 2024 would not count as a novel idea). Similarly, in our framework, a task's novelty is evaluated relative to the previously generated tasks.
> > >
> > > > Stating that we don't see it plateauing considering there are only 13 environments generated is true but also not a strong marker for open-endedness and shouldn't be used to make this claim.
> > >
> > > Thank you. We agree, and have changed the language.
> > >
> > > > The related work mentions key research in environment design but does not acknowledge the older and wider field of intrinsic motivations, which I feel is foundational for the presented work.
> > >
> > > We apologize for the oversight, and have added references to the suggested relevant works.
> > >
> > > > Do the authors plan on releasing the code?
> > >
> > > Yes! We clarified this in the revised manuscript (new Appendix A).

---

> > > > ### Author Response · Authors · 2024-11-23
> > > >
> > > > > "Previous attempts have resulted in pathologies when optimizing against definitions and quantifications of interestingness" → which ones? How is the current approach overcoming these issues?
> > > >
> > > > Previous attempts to quantify interestingness in open-ended learning have encountered significant challenges. Methods that optimize for novelty or diversity often lead to pathologies where agents exploit trivial variations of tasks without progressing to meaningful challenges. For instance, in the minimal criterion coevolution framework, removing walls from a maze to maximize novelty undermines the task's intended complexity, ultimately preventing the maze from being solved (Brant & Stanley, 2017). Similarly, approaches like POET (Wang et al., 2019) aim to generate environments through random mutation. However, POET can produce redundant or trivial tasks that fail to contribute to the development of genuinely interesting environments or behaviors. Intrinsic motivation methods that encourage exploration to new states can suffer from the "noisy TV" problem, where agents become fixated on stochastic or uninformative aspects of the environment (that still technically result in new observations), leading to meaningless activities (Burda et al., 2018). These issues highlight the difficulty of quantifying what we mean by "interestingness" into fixed equations or metrics.
> > > >
> > > > OMNI-EPIC addresses these challenges by leveraging FMs, which inherently possess human notions of interestingness, rather than relying on fixed formulas or handcrafted metrics. Additionally, it generates diverse environments and employs post-generation MoI filtering to ensure that the tasks created are indeed novel. This approach enables OMNI-EPIC to focus on tasks that are genuinely interesting and contribute to open-ended learning, overcoming the limitations of previous methods.
> > > >
> > > > > Are the task embeddings computed from the linguistic descriptions? the code? the agent trajectories? combinations of these?
> > > >
> > > > The task embeddings are computed from the entire environment code, which includes the natural language task description. We have clarified this in the revised manuscript (Section4, Section 6, Appendix I).
> > > >
> > > > > It's not so clear what happens to tasks who keep on failing to compile, are they added as failed to archive? If not added to the archive, how is this information routed back to the task generator? If added as failed, shouldn't we differentiate "failed because of compilation" and "failed because the agent couldn't solve it"?
> > > >
> > > > Tasks that fail to compile after the maximum number of iterations are discarded, and the compilation errors are not used when generating the next task. Instead, the task generator creates an entirely new task. However, your idea of including it could improve task generation efficiency, and is an interesting thing to try. We have made this clearer in the revised manuscript.
> > > >
> > > > > It would be good to show somewhere the success rates at various stages, e.g. the % of tasks that are successfully turned into executable environment code, the % of these that are accepted by the post-hoc MOI checker, the % of the latter that are successfully solved. In the long run without training, how many of the 200 iterations led to executable environment code? And how is executability checked here? By running a random policy maybe?
> > > >
> > > > Across 5 repeated runs with learning:
> > > > 96.4% (CI: 93.1 - 100.0) of tasks were successfully turned into executable environment code.
> > > > 96.2% (CI: 67.9 - 100.0) of these were accepted by the post-generation MoI.
> > > > 73.7% (CI: 48.2 - 81.5) of the MoI-validated tasks were successfully solved.
> > > >
> > > > Across 3 repeated long runs without learning:
> > > > 85.1% (CI: 83.1 - 96.5) of generations led to executable environment code.
> > > >
> > > > First, we check if the code can be compiled by running each of the environment functions (e.g., `reset`, `step`, `get_success`). Next, we use a random action policy. At the start of the episode, we ensure that the robot is not colliding with any obstacles, as such collisions could cause the robot to "fly." Then, over 100 steps of random actions, we verify that the objects are not flying across the space at a high velocity  (indicating stable physics, where the objects are not “exploding”) and that the robot is not in free fall (ensuring it is initialized with a platform beneath it).
> > > >
> > > > > typos
> > > >
> > > > We've corrected the typos.
> > > >
> > > > ---
> > > >
> > > > > I'm willing to raise the score if new experiments are shown to support the claims.
> > > >
> > > > Thank you for recognizing the contribution of our work. Have we addressed all of your concerns? If not, please let us know and we will endeavor to do so. If so, we greatly appreciate you saying you are willing to update your score.

---

> > > > > ### Comment · Reviewer_xWEx · 2024-11-25
> > > > >
> > > > > I thank the authors for their thorough answers and the efforts they put in providing additional experiments to answer my questions.
> > > > >
> > > > > **Curriculum:**
> > > > > * "We show that RL agents trained with OMNI-EPIC learn increasingly complex tasks over time"
> > > > > * "We aim to generate a diverse task distribution and create an effective curriculum within that distribution for either specialist or generalist agents. As mentioned in our results section, OMNI-EPIC successfully accomplished those goals, which are critical prerequisites for training specialist or generalist agents in an open-ended manner."
> > > > >
> > > > > I appreciate the new experiments to support the curriculum learning claim (Appendix G). I think these are crucial as they're the only piece of evidence supporting one of the core claim of the paper, that OMNI-EPIC generates a diversity of tasks that is useful to train agents. Currently, they are pointed to with a non-specific pointer "additional ablations in Appendix G", but I think stating this result in the main paper would make the claim stronger.
> > > > >
> > > > > About the experiment specifically, I'd like to hear more details (given how important they are to support the claim): how were the computational (nb of environment samples, nb of updates) matched across the two conditions? Do we see an increase in difficulty wrt time where later tasks are harder to solve than earlier tasks? Is it that agents from scratch can't solve these tasks or is it that the budget they were given is a bit too short? These results should also be given in conjunction with the diversity measures / examples as we could imagine a run where after the first task, the 24 following ones are very close to the first one which would benefit the transfer agents (reusing weights from the first agent) vs agents from scratch.
> > > > >
> > > > >
> > > > > ---
> > > > >
> > > > > **Conclusion**
> > > > >
> > > > > I appreciate the new ablation studies and extra experiments conducted in appendix F and G. I think they provide a more solid support for the claims made in the paper about the ability of OMNI to tailor its task generation to the agent (to some extent) and the claim that the series of generated tasks form a curriculum. I also appreciate the new human user study to validate the global success/failure signal used to mark task success. I appreciate that they were willing to tone down the statements neither current nor new experiments could support.
> > > > >
> > > > > Overall I think this is an inventive paper that tackles an important problem. The approach is promising, although it is hard to say how OMNI-EPIC may behave when run longer with a more capable learning agent. Open-ended research is hard and there is so far no general method to evaluate such systems. In this context, I think this paper makes good efforts in showcasing possible ways of evaluating and characterizing open-ended systems and I think it will inspire others to build on the approach and refine its evaluation protocols. These are all good reasons to accept this paper today and I will raise my score as a consequence.

---

> > > > > > ### Author Response · Authors · 2024-11-25
> > > > > >
> > > > > > Dear Reviewer xWEx,
> > > > > >
> > > > > > Thank you for your thoughtful feedback and for indicating that you would raise your score based on our response. We deeply appreciate that!
> > > > > >
> > > > > > We will discuss your other comments amongst the authors and get back to you about them very soon.
> > > > > >
> > > > > > We noticed that the score still appears as 6 on OpenReview. Could you kindly verify if the score update was properly registered on the platform?
> > > > > >
> > > > > > Thank you. The OMNI-EPIC authors.

---

> > > > > > > ### Author Response · Authors · 2024-11-26
> > > > > > >
> > > > > > > Thank you for following up on the rebuttal discussion and aim to address the remaining concerns below.
> > > > > > >
> > > > > > > > I appreciate the new experiments to support the curriculum learning claim (Appendix G). I think these are crucial as they're the only piece of evidence supporting one of the core claim of the paper, that OMNI-EPIC generates a diversity of tasks that is useful to train agents. Currently, they are pointed to with a non-specific pointer "additional ablations in Appendix G", but I think stating this result in the main paper would make the claim stronger.
> > > > > > >
> > > > > > > Thank you for this comment. We will change the paper in the way you suggested.
> > > > > > >
> > > > > > > > About the experiment specifically, I'd like to hear more details (given how important they are to support the claim): how were the computational (nb of environment samples, nb of updates) matched across the two conditions?
> > > > > > >
> > > > > > > In the ablation study comparing with and without transfer learning, we allocated the same total number of training steps (2 million) to the RL agent for each task. The only difference between the two settings was the policy initialization for each new environment: in the transfer learning scenario, the RL policy was initialized from a checkpoint (a policy that had trained on another environment), whereas in the non-transfer learning scenario, it was initialized randomly (i.e., trained from scratch). We have updated the manuscript with this detail.
> > > > > > >
> > > > > > > > Do we see an increase in difficulty wrt time where later tasks are harder to solve than earlier tasks?
> > > > > > >
> > > > > > > Yes, we observe an increase in task difficulty over time. To support this, we have included an additional plot (Figure 43 in Appendix I of the just-updated manuscript) that illustrates the percentage of tasks learned out of all attempted tasks during the short run of learning experiments. The plot shows that the percentage of learned tasks decreases over time, with a widening gap between tasks learned by OMNI-EPIC and those trained from scratch. This indicates that OMNI-EPIC generates tasks of increasing difficulty as training progresses, and as time progresses it becomes less and less likely that training from scratch works to solve the (increasingly harder) tasks.
> > > > > > >
> > > > > > > > Is it that agents from scratch can't solve these tasks or is it that the budget they were given is a bit too short?
> > > > > > >
> > > > > > > While it is difficult to definitively know if RL can eventually solve a task (similar to the Halting Problem), we can usually get a sense by looking at the measures of progress. These runs appear to have converged, and we do not believe additional compute would make a significant difference, suggesting that the agents trained from scratch are unable to solve the tasks.
> > > > > > >
> > > > > > > > These results should also be given in conjunction with the diversity measures / examples as we could imagine a run where after the first task, the 24 following ones are very close to the first one which would benefit the transfer agents (reusing weights from the first agent) vs agents from scratch.
> > > > > > >
> > > > > > > We will change the paper in the way you suggested.
> > > > > > >
> > > > > > > ---
> > > > > > >
> > > > > > > Thank you for your continued engagement and for recognizing the merits of OMNI-EPIC. We hope these changes sufficiently address your concerns.

---

### Official Review · Reviewer_PisJ · 2024-10-31

**Soundness:** 2
**Presentation:** 3
**Contribution:** 3
**Rating:** 8
**Confidence:** 4

**Summary:**

The paper introduces OMNI-EPIC, an open-ended system that generates novel and learnable environments and tasks for RL agents. The method leverages current language and vision foundation models to generate the environments (through code) and filter the novel and learnable ones. The authors show how the system generates diverse, interesting, and learnable tasks for RL agents, starting from a very small set of initial environments. Experiments show the potential of the system to produce endless automatic curricula for increasingly complex agents.

**Strengths:**

- The paper is generally well written and the ideas are mostly clearly transmitted. The figures are clean and support the text, helping to transmit the paper's main ideas.

-  I think that the contribution is substantial. The fact that OMNI-EPIC can successfully generate diverse, novel, and learnable environments through code (Python+PyBullet) constitutes a valuable opportunity for the open-endeness community to analyze these systems and advance the field.

- Moreover, OMNI-EPIC successfully employs foundation models to leverage notions of human interestingness. Generating novel, but interesting environments (that are relevant to humans) is a key challenge in unsupervised environment design and the open-endedness field. Although previous works have pointed to the possibility of using foundation models for this purpose [1], this work is an excellent example of how foundation models can be successfully used for this purpose, which is also a valuable contribution by itself.

[1] Hughes, Edward, et al. "Position: Open-Endedness is Essential for Artificial Superhuman Intelligence", ICML24.

**Weaknesses:**

**W1:** One of my main concerns with this work is the gap between the claims made and the actual demonstrations in the paper.
Authors claim multiple times that OMNI-EPIC can generate an endless stream of environments (e.g., L082, L151, L424). See line 424: *"[...] ensuring a progressive, never-ending curriculum for training"*, note that this is a **very** strong claim. However, the experimental results are far from demonstrating this capability. Experiments show the generation of 200 environments, but considering that all are solvable (which I consider a very strong assumption). When considering the performance of RL agents in the generated environments, the authors only show a sequence of 22 tasks.  These are some suggestions to address the issue:

- Use simpler alternatives to Dreamer V3 (e.g., PPO with simpler NN architectures) to reduce the computational cost of training the agent and demonstrate that the novelty and learnability of the tasks generated by OMNNI-EPIC hold for longer environment generation steps.

- Reduce the claims made in the paper. Under the current experimental results, there's no real evidence that OMNI-EPIC could continue to generate novel and learnable environments for very large runs.

**W2:** Although the performance of current LLMs is excellent for many coding and text generation tasks, these also present some limitations, which some relate to difficulties in generating novelty. The fact that OMNI-EPIC completely relies on these models, makes me believe that the presented method would especially suffer from the mentioned novelty issue. This weakness is related to the previous, as no evidence is shown on the capability of OMNI-EPIC to  "endlessly" generate novel and solvable environments.

**W3:** As many deep NN models, LLMs also suffer from brittleness. As OMNI-EPIC relies on multiple LLMs working together, this makes me believe that the presented system can greatly suffer from this brittleness issue, as there is no evidence for the contrary. Further experiments on this topic would greatly help to analyze the actual capabilities and limitations of the proposed method. What is the ratio between successful and failed attempts at generating working/solvable environments? This is only an idea, but many similar experiments could be included that would significantly improve the soundness of the paper.

**Minor issues:**
- Authors mention multiple times that "code is Turing complete" (e.g., L93, L520). This is very vague, please be rigorous, as many types of "code" (i.e., programming languages) exist but not all are Turing complete.
- Python being Turing complete does not guarantee that a foundation model (and by consequence, OMNI-EPIC) can generate any computable environment. There would be many computable environments that would have zero probability of being generated by a specific foundation model. Present rigorous arguments or remove/modify this claim.
- I think that the "cell coverage of archive diversity" metric is not explained in the main text. I think that the overall clarity of Section 6 could be improved.
- Check the coherence of the references. For example, some references include the proceedings with different styles: some use "In Proceedings [...]", others "In [...]", and others just directly mention the specific proceeding.

I want to emphasize that I think that the contribution of the paper is valuable to the field and that I'm willing to update my score if the mentioned issues are addressed.

**Questions:**

**Q1:** Do you have any result or hypothesis on how much the initial set of environments in the task archive conditions the generation of new environments?

**Q2:** How does the task similarity method scale with the number of tasks? If the idea is to endlessly generate tasks, this similarity method should be very computationally efficient.

**Q3:** Does the simplified action space limit the diversity and complexity of the generated environments?

**Q4:** L283 states "We employ a success detector, instantiated as an LLM or VLM", later L304-305 mentions "Although our preliminary testing found that current VLMs are not yet accurate enough to be used as success detectors". Given the second sentence, I find very surprising the usage of VLMs as success detectors in  OMNI-EPIC. Why does OMNI-EPIC employ VLMs for this purpose even though the authors mention that VLMs are still not ready for the task?

---

> ### Author Response · Authors · 2024-11-23
>
> Thank you for your thoughtful comments and the opportunity to address your concerns.
>
> ---
>
> > Although previous works have pointed to the possibility of using foundation models for this purpose [1], this work is an excellent example of how foundation models can be successfully used for this purpose, which is also a valuable contribution by itself.
>
> Thank you for your kind comment. We greatly admire that paper as well, and in fact, it cites OMNI-EPIC to support this very point. We are delighted to be in agreement and appreciate the recognition of our contribution.
>
> >  the gap between the claims made and the actual demonstrations in the paper. Authors claim multiple times that OMNI-EPIC can generate an endless stream of environments … note that this is a very strong claim. However, the experimental results are far from demonstrating this capability.
> > The fact that OMNI-EPIC completely relies on these models, makes me believe that the presented method would especially suffer from the mentioned novelty issue.
> > Reduce the claims made in the paper. Under the current experimental results, there's no real evidence that OMNI-EPIC could continue to generate novel and learnable environments for very large runs.
>
> We have toned down our claims in the revised manuscript. We acknowledge that our assertions about the potential to generate an endless stream of learnable and interesting tasks refer to what could be achieved with significantly more resources and time. Our current results provide only an initial glimpse of the system's capabilities, and we recognize that realizing the full scope of this potential remains an open research question. Moreover, we recognize that it is an open, and fascinating question as to whether OMNI-EPIC could truly generate environments forever, including well beyond the distribution of human data. We think it might, but we recognize that is an open, and not-yet-answered research question. We have made that clear in the paper and more properly phrased our claims throughout. This conversation is part of the larger open question and debate in the community regarding whether LLMs can generate truly new knowledge (e.g. AlphaGo’s move 37).
>
> > Experiments show the generation of 200 environments, but considering that all are solvable.
> > What is the ratio between successful and failed attempts at generating working/solvable environments?
> > many similar experiments could be included that would significantly improve the soundness of the paper
>
> We sampled 100 out of the 200 generated environments for human evaluation and found that 71.5% were solvable. We have also included additional ablation studies (new Appendix G) to demonstrate how various components of OMNI-EPIC contribute to training RL agents.
>
> > Use simpler alternatives to Dreamer V3 (e.g., PPO with simpler NN architectures) to reduce the computational cost of training the agent
>
> We use DreamerV3 because it is more sample efficient, which is particularly important given the slow simulation speed of PyBullet (it thus ends up being substantially computationally cheaper than PPO). This choice helps us maximize learning efficiency despite the computational constraints.
>
> > Authors mention multiple times that "code is Turing complete" (e.g., L93, L520). This is very vague, please be rigorous, as many types of "code" (i.e., programming languages) exist but not all are Turing complete.
>
> Thank you. We have corrected our statement about Turing completeness to specify that not all but “the programming language used here (Python) is Turing complete”.
>
> > There would be many computable environments that would have zero probability of being generated by a specific foundation model. Present rigorous arguments or remove/modify this claim.
>
> We have toned down our claims and clarified in the revised manuscript that, while the goal is to eventually generate any computable environment, OMNI-EPIC represents just one step toward achieving this.
>
> > I think that the "cell coverage of archive diversity" metric is not explained in the main text. I think that the overall clarity of Section 6 could be improved.
>
> We have updated the manuscript to include a more detailed explanation of how the cell coverage of archive diversity is calculated and to improve the overall clarity of Section 6.
>
> > Check the coherence of the references.
>
> We have updated the references, using BibTeX entries sourced directly from Google Scholar (correcting any errors if necessary).
>
> >  Do you have any result or hypothesis on how much the initial set of environments in the task archive conditions the generation of new environments?
>
> Although the initial seed tasks in the task archive are the same across repeated runs, we observe that the generated environments still exhibit significant differences. This suggests that OMNI-EPIC’s ability to generate diverse tasks is not heavily conditioned by the initial set of environments. We have clarified this in the revised manuscript (Appendix F).

---

> > ### Author Response · Authors · 2024-11-23
> >
> > > How does the task similarity method scale with the number of tasks? If the idea is to endlessly generate tasks, this similarity method should be very computationally efficient.
> >
> > Our current method for measuring task similarity is based on retrieval-augmented generation (RAG) techniques, similar to K-nearest neighbors (KNN). There are efficient implementations of these methods, such as approximate nearest neighbor search algorithms, which are designed to scale well with a large number of tasks.
> >
> > > Does the simplified action space limit the diversity and complexity of the generated environments?
> >
> > OMNI-EPIC is not limited by the simplified action space. That was just a design choice to showcase the algorithm’s capabilities within a manageable scope. To demonstrate that OMNI-EPIC is not limited to this specific choice, we conducted additional experiments using an Ant robot with a continuous action space (new Appendix O). These results show that OMNI-EPIC can generate diverse and complex environments even with non-discrete, larger action spaces.
> >
> > We also note that the generated challenges are quite diverse (e.g., obstacle courses, kicking balls through goalposts, dodgeball, cleaning up construction sites, exploring buildings, and clearing dishes from cluttered restaurant tables). While they may superficially appear similar due to the use of the same robot in the same simulator, OMNI-EPIC represents a step toward the vision of generating any environment that can be produced with code. In this initial work, we focus on a single simulator, but future work could explore generating challenges in different simulators (or non-embodied domains such as math, writing, or protein-folding challenges) or for different robots.
> >
> > > Why does OMNI-EPIC employ VLMs for this purpose even though the authors mention that VLMs are still not ready for the task?
> >
> > We clarified in the revised manuscript that, since our preliminary testing found that current VLMs are not yet accurate enough to serve as success detectors, we use code generated by LLMs for this purpose instead in our experiments. However, we anticipate that VLM capabilities will rapidly improve, eventually reaching the accuracy needed for effective use. Our paper only suggests that one could use VLMs as success detectors, and this will be especially helpful as they improve in the future.
> >
> > ---
> >
> > > I want to emphasize that I think that the contribution of the paper is valuable to the field and that I'm willing to update my score if the mentioned issues are addressed.
> >
> > Thank you for recognizing the contribution of our work. Have we addressed all of your concerns? If not, please let us know and we will endeavor to do so. If so, we greatly appreciate you saying you are willing to update your score.

---

> > > ### Comment · Reviewer_PisJ · 2024-11-24
> > >
> > > Thanks for your thoughtful answers and for modifying the paper accordingly. I'm pleased to update my score in light of these modifications, which improve writing, reduce the paper's claims, and include more empirical evaluation in the additional material.
> > >
> > > Thanks for your response and the work you did to improve the paper.

---

> > > > ### Author Response · Authors · 2024-11-25
> > > >
> > > > Thank you for your kind words and for taking the time to review our work thoroughly. We greatly appreciate your feedback.

---

### Official Review · Reviewer_Cvga · 2024-11-02

**Soundness:** 2
**Presentation:** 3
**Contribution:** 2
**Rating:** 8
**Confidence:** 4

**Summary:**

This paper uses an LLM to generate environment code for an RL agent. This is combined with another foundation model that serves as a proxy for human models of interestingness. Together, the first LLM generates code whereas the second guides this process towards novel and interesting tasks. Theoretically, treating the environment as code is a much less constrained way to generate environments compared to much of the prior work. The paper also provides two experiments, one with simulated learning to focus on the environment generation, and another with actual RL agent training to identify how these two components interact.

**Strengths:**

- I agree, using code as an environment representation is indeed much more expressive than changing hand-designed features within a very constrained environment, as prior work does.
- It is great that this paper does provide a full pipeline that theoretically can be run for a long time to continuously generate novel tasks.

I am not giving it a lower score because I think the idea of using code as an environment representation is a good idea that future work can build on. Furthermore, demonstrating a full end-to-end pipeline is useful.
I am not giving it a higher score because I think the results are not that compelling. That is not to say that this approach does not hold promise---and I hope future work/the authors build on this---but the outcomes in this work are not that open ended.

**Weaknesses:**

While code as a representation is expressive in theory, all of the different environments seem quite similar. This is partly due to the agent in all of these tasks being the same.
	- However, if you remove the restriction of using this particular robot in a particular simulator, then the task may become too difficult for current LLMs, or you could end up generating only a certain type of task.
- It feels like the claims made are that we can express any possible task, and that, when run long enough, we will eventually obtain a very large number of semantically diverse environments; however, the results do not seem to substantiate this claim.
- Figure 2 is very hard to understand and parse, is there a better way to show the same information?
- This sentence is a bit hard to parse/can be rewritten
> Because the model has distilled a sense of interesting from reading the internet, it has a model ofi nterestingness (MoI) that emulates the human capacity for nuanced judgments of interestingness in open-ended learning (Zhang et al., 2023), which it brings to bear when generating tasks.

**Questions:**

- Is the code open source?
- To what extent will this be limited by the dataset of the coding LLM? Will there ever be truly novel environments generated?
- It is a bit unclear to me, but when starting the process, do you first sample a random task from the archive, and then use this as the reference task to choose the N most similar ones.
- Consider the following case:
	- You have 10 tasks in the archive, say 5 of type A and type B. Type A tasks are all somewhat similar to each other, but different to type B. As an example, type A tasks involve kicking a ball and type B involves crossing a bridge. Suppose the number of tasks given to the LLM if 5, say.
	- Would the following cyclic behaviour be possible? Please comment/explain how or how not, and what possible solutions there are.
		- The task generator receives 5 tasks of type A, and is asked to generate a novel and new task. Then it does this by generating a type B task. However, the post generation MOI rejects this because the archive already has many of these, so the task is not added. Whenever the randomly-chosen task is type A, this happens, and the opposite for type B.
		- If this happens, it seems like the process will just remain stuck and never generate anything new.
- Would there be axes of variation that are not explored at all by an LLM, potentially due to the bias in its data, or by virtue of how it predicts the next token? Concretely, suppose you apply this to a more general environment representation (where the LLM can change everything), and it happens to generate a physics-based R2D2 environment. Then, when generating new tasks, will it ever do something very different, like generating a game like Pong (different controls, doesn't use a robot anymore, etc.) or will it continuously generate variations of R2D2 physics tasks? Please elaborate on this

---

> ### Author Response · Authors · 2024-11-23
>
> Thank you for your insightful feedback. We now address each of your concerns and questions.
>
> ---
>
> > While code as a representation is expressive in theory, all of the different environments seem quite similar. This is partly due to the agent in all of these tasks being the same.
>
> OMNI-EPIC is not limited by the robot type. To demonstrate this, we conducted additional experiments using an Ant robot with a continuous action space (new Appendix O). These results show that OMNI-EPIC can generate diverse and complex environments even with non-discrete, larger action spaces for a different robot.
>
> We also note that the generated challenges are quite diverse (e.g., obstacle courses, kicking balls through goalposts, dodgeball, cleaning up construction sites, exploring buildings, and clearing dishes from cluttered restaurant tables). While they may superficially appear similar due to the use of the same robot in the same simulator, OMNI-EPIC represents a step toward the vision of generating any environment that can be produced with code. In this initial work, we focus on a single simulator, but future work could explore generating challenges in different simulators (or non-embodied domains such as math, writing, or protein-folding challenges) or for different robots.
>
> > It feels like the claims made are that we can express any possible task, and that, when run long enough, we will eventually obtain a very large number of semantically diverse environments; however, the results do not seem to substantiate this claim.
>
> Please see the previous answer. Also, we show that running OMNI-EPIC for long iterations generates semantically diverse tasks (Section 4), further supported by the cell coverage and archive diversity plots (Section 6).
>
> However, we acknowledge that our claims about the potential for expressing any possible task refer to what could be achieved with significantly more resources and time. Our current results provide only an initial glimpse of the system’s capabilities, and we recognize that realizing the full scope of this potential remains an open research question.
>
> > Figure 2 is very hard to understand and parse, is there a better way to show the same information?
>
> We have tried our best to present the generated tasks in the most readable way. The anonymous project website also contains an interactive figure, allowing readers to explore each environment more easily. We welcome any further suggestions for improving the readability of Figure 2.
>
> > This sentence is a bit hard to parse/can be rewritten
> Because the model has distilled a sense of interesting from reading the internet, it has a model ofi nterestingness (MoI) that emulates the human capacity for nuanced judgments of interestingness in open-ended learning (Zhang et al., 2023), which it brings to bear when generating tasks.
>
> We have rephrased the sentence.
>
> > Is the code open source?
>
> Yes! We clarified this in the revised manuscript (new Appendix A).
>
> > To what extent will this be limited by the dataset of the coding LLM? Will there ever be truly novel environments generated?
>
> Thank you for raising this important question. We implemented custom PyBullet environments specifically tailored to our unique settings, which differ from standard configurations. To the best of our knowledge, these particular implementations have not been coded or published online before, suggesting that OMNI-EPIC is generating entirely new environments.
>
> That said, whether we can generate truly novel environments (unlike anything humans have discussed or shared online) remains a fascinating open research problem that this work and other research on open-endedness bring into discussion. Verifying that environments are meaningfully new, and studying whether such novel environments can be produced indefinitely, are fascinating new research questions opened up by OMNI-EPIC. We have updated the manuscript to mention this.
>
> > It is a bit unclear to me, but when starting the process, do you first sample a random task from the archive, and then use this as the reference task to choose the N most similar ones.
>
> You are correct. When generating the next task, we first sample a random learned task from the archive and use it as the reference task to retrieve the N most similar tasks, including both successfully completed and those attempted but failed. These similar tasks are then provided as examples to the task generator to generate the next learnable and interesting task. We have updated the paper to make this clearer.

---

> > ### Author Response · Authors · 2024-11-23
> >
> > > Consider the following case: You have 10 tasks in the archive, say 5 of type A and type B… If this happens, it seems like the process will just remain stuck and never generate anything new.
> >
> > You are correct that, in theory, the process could become stuck in a scenario where only a limited set of task types are repeatedly generated. However, we believe this is highly unlikely in practice. Several factors would need to align perfectly, including the LLM behaving in an overly deterministic manner and the distribution of environments being very limited, for this situation to occur. Moreover, this potential issue is largely a consequence of current context length limitations. As context length increases, the likelihood of this problem occurring diminishes, though it can never be completely ruled out unless the entire archive can be used as context.
> >
> > > Would there be axes of variation that are not explored at all by an LLM, potentially due to the bias in its data, or by virtue of how it predicts the next token? … Then, when generating new tasks, will it ever do something very different, like generating a game like Pong … or will it continuously generate variations of R2D2 physics tasks?
> >
> > We believe that the system will be capable of generating significantly different environments, such as Pong, because they are interestingly different. Even with our current setup, we observe a wide variety of generated games (e.g., bowling, color sorting, conveyor belt challenges). If we were to loosen constraints on the agent type and simulator, we expect that the system would continue to produce even more diverse and varied environments. An open, fascinating research area motivated by work like OMNI-EPIC is whether the process of producing interestingly new environments could go on forever: we think it is possible the model of interestingness and the environment generator, when conditioned on the archive of already invented things, could keep innovating indefinitely.
> >
> > ---
> >
> > Have we answered all your questions and concerns? Please do not hesitate to bring up any additional questions. If we have, we hope you will consider increasing your score.

---

> > > ### Comment · Reviewer_Cvga · 2024-11-24
> > >
> > > Thank you for your response and updated version.
> > >
> > >
> > > > OMNI-EPIC is not limited by the robot type. To demonstrate this, we conducted additional experiments using an Ant robot with a continuous action space (new Appendix O). These results show that OMNI-EPIC can generate diverse and complex environments even with non-discrete, larger action spaces for a different robot.
> > >
> > > While I appreciate the additional experiment, I think it has the same weakness as the original, i.e., that it is just one robot morphology with some different terrain/obstacles. To really address this concern, I think the LLM would have to be able to choose the robot as well (as a proof of concept maybe selecting from a fixed set, but ultimately designing the morphology itself). An even more impressive showcase would be if the environment space consisted of all of the locomotion tasks you currently have, but also other tasks expressible in pybullet (e.g. games like pong/flappy bird, etc.).
> > >
> > >
> > > For figure 2, one option could be to make all the task circles transparent, except for the ones you highlight with images. Possibly making the lines between circles also thinner/more transparent could help visually. These are some suggestions so they may not necessarily help.
> > >
> > >
> > > > You are correct that, in theory, the process could become stuck in a scenario where only a limited set of task types are repeatedly generated. However, we believe this is highly unlikely in practice. Several factors would need to align perfectly, including the LLM behaving in an overly deterministic manner and the distribution of environments being very limited, for this situation to occur. Moreover, this potential issue is largely a consequence of current context length limitations. As context length increases, the likelihood of this problem occurring diminishes, though it can never be completely ruled out unless the entire archive can be used as context.
> > >
> > >
> > > I think it would be good to mention this as a possible limitation. While this particular version may be unlikely, other problems of a similar nature could potentially crop up.
> > >
> > >
> > >
> > > Then, finally, I just want to mention something reviewer xWEx brought up:
> > >
> > > > Thank you for raising this point. We would like to clarify that changes in surface forms, such as color, are not merely superficial in the context of RL. In RL, variations in visual attributes like color serve as a form of data augmentation, which can significantly impact the agent’s learning process and generalization capabilities. These variations can introduce new challenges and improve the robustness of the agent by requiring it to adapt to different observation inputs.
> > >
> > >
> > > While sure, the background is important for robustness purposes, I would argue that this is not the type of novelty we desire. This links to one of my previous points where one possible pathology of this approach is that the system will end up focusing very much on a limited number of niches (in this case, e.g., different backgrounds for virtually the same task) instead of exploring semantically diverse environments.

---

> > > > ### Author Response · Authors · 2024-11-26
> > > >
> > > > We thank the reviewer for following up on the rebuttal discussion and aim to address the remaining concerns below.
> > > >
> > > > > While I appreciate the additional experiment, I think it has the same weakness as the original, i.e., that it is just one robot morphology with some different terrain/obstacles. To really address this concern, I think the LLM would have to be able to choose the robot as well (as a proof of concept maybe selecting from a fixed set, but ultimately designing the morphology itself). An even more impressive showcase would be if the environment space consisted of all of the locomotion tasks you currently have, but also other tasks expressible in pybullet (e.g. games like pong/flappy bird, etc.).
> > > >
> > > > While we could conduct an experiment where the LLM selects the robot type, we believe this would not provide meaningful insights unless it were paired with vast (industrial-level) computational resources, which our academic lab does not have. Implementing robot selection would require unifying action spaces across different robot types – a challenge that has been addressed in previous works [1, 2], but demands extensive computational capacity to see the benefits of. Given our current compute limitations, the key question is: what specific insights would we gain from this experiment? We already know that OMNI-EPIC is capable of controlling a different robot with a different action space, as evidenced by the recent Ant robot experiments. At best, training one policy to control different morphologies would demonstrate that it is technically possible for one policy to handle multiple robot morphologies. The real scientific benefit would be to show positive transfer between the robots (i.e., that training one policy on all bodies is better than training separate, robot-specific policies), but those benefits would only show up with tremendous compute resources. We appreciate this suggestion as a potential avenue for future exploration and will add it to future work. We feel similarly about transfer across radically different domains or simulators. We know it is possible in theory, so just showing that it is technically possible would be uninformative; the real interesting question is if it shows positive transfer between domains. But training such generalist policies to see that beneficial positive transfer emerge requires vast, industrial compute resources [1, 2, 3].
> > > >
> > > > [1] Brohan, Anthony, et al. "Rt-1: Robotics transformer for real-world control at scale." arXiv preprint arXiv:2212.06817(2022).
> > > > [2] O'Neill, Abby, Abdul Rehman, Abhinav Gupta, Abhiram Maddukuri, Abhishek Gupta, Abhishek Padalkar, Abraham Lee et al. "Open x-embodiment: Robotic learning datasets and rt-x models." arXiv preprint arXiv:2310.08864 (2023).
> > > > [3] Team, Adaptive Agent, Jakob Bauer, Kate Baumli, Satinder Baveja, Feryal Behbahani, Avishkar Bhoopchand, Nathalie Bradley-Schmieg et al. "Human-timescale adaptation in an open-ended task space." arXiv preprint arXiv:2301.07608 (2023).
> > > >
> > > > > For figure 2, one option could be to make all the task circles transparent, except for the ones you highlight with images. Possibly making the lines between circles also thinner/more transparent could help visually. These are some suggestions so they may not necessarily help.
> > > >
> > > > Following your suggestion, we have plotted two additional versions of Figure 2 (Appendix D): one with all task circles gray except for the ones highlighted in the images, and another with thinner edges. While we believe the current Figure 2 (or the one with thinner edges) remains the most informative in conveying that OMNI-EPIC is capable of generating diverse tasks over iterations, we are happy to include the additional figures in the appendix for comparison and you can let us know which you think is best for inclusion in the final camera-ready copy of the paper (if it is published, which is still in question given the reviewer scores).
> > > >
> > > > > I think it would be good to mention this as a possible limitation. While this particular version may be unlikely, other problems of a similar nature could potentially crop up.
> > > >
> > > > We have incorporated this discussion into Appendix J for further elaboration.

---

> > > > > ### Author Response · Authors · 2024-11-26
> > > > >
> > > > > > While sure, the background is important for robustness purposes, I would argue that this is not the type of novelty we desire. This links to one of my previous points where one possible pathology of this approach is that the system will end up focusing very much on a limited number of niches (in this case, e.g., different backgrounds for virtually the same task) instead of exploring semantically diverse environments.
> > > > >
> > > > > Thank you for your comment. While it is true that some generated tasks may appear similar in nature (e.g., obstacle courses with varying backgrounds or colors), we demonstrate that, over the long run with simulated learning (Section 4), OMNI-EPIC is capable of producing a wide variety of semantically diverse tasks. OMNI-EPIC does generate tasks that differ not only in appearance but also in structure and objectives (e.g., conveyor belt item delivery vs. dodgeball arena). This highlights its ability to explore beyond limited niches. Additionally, since it looks for tasks that have high learning progress, it can discover whether the agent will continue to learn from various forms of data augmentation vs. entirely new types of tasks.
> > > > >
> > > > > ---
> > > > >
> > > > > We hope we have answered all your questions and concerns. If we have, we hope you will consider increasing your score. If not, please let us know what issues remain and we would be happy to address them.

---

> > > > > > ### Comment · Reviewer_Cvga · 2024-11-30
> > > > > >
> > > > > > Thank you for your response.
> > > > > >
> > > > > > I think Figure 7 is a reasonable way of showing the information, but I agree that none of the options are amazing. I will leave the final choice to you. The interactive figure on the website is great.
> > > > > >
> > > > > >
> > > > > > Ultimately, I think the idea behind this work is great. I tend to agree that illustrating a higher degree of open-endedness would likely be computationally infeasible for an academic lab (at least without using hardware-accelerated RL training). I will raise my score to 8, and hope that future work does push the boundary further, and building on this idea.

---

> > > > > > > ### Author Response · Authors · 2024-12-01
> > > > > > >
> > > > > > > Thank you for your kind words and for taking the time to review our work thoroughly. We greatly appreciate your feedback.

---

### Official Review · Reviewer_kcfh · 2024-11-03

**Soundness:** 1
**Presentation:** 1
**Contribution:** 2
**Rating:** 3
**Confidence:** 4

**Summary:**

The paper presents OMNI-EPIC, a system to design a large library of environments for reinforcement learning agents. OMNI-EPIC relies on Foundation Models, both large language models and vision language models. The system uses a pipeline where problems are specified in natural language, and then it uses an LLM as a synthesizer to generate programs encoding the environments. The environments are also evaluated with another LLM with respect to their interestingness; environments that aren't "interesting" are discarded. Finally, the system also has a module that checks whether a task was solved or not.

A few results of the system show that it is able to generate a diverse set of environments.

**Strengths:**

The paper attempts to tackle the ambitious problem of continually generating a diverse set of problems, from which an RL agent could learn. The idea of leveraging foundation models for this task is a promising idea. These models contain what one might call "general knowledge" that can be helpful in tackling this problem.

**Weaknesses:**

There are two key weaknesses with the paper that justifies my scores and recommendation.

**Scholarship**

First, the paper lacks scholarship, in the sense that the writing is sloppy (you read the sentences and say 'yes, I see what you mean, but what you wrote isn't technically correct.') and it lacks comparison with a very large body of works devoted to generating content in simulated environments: procedural content generation. I will give concrete examples and suggestions next.

The introduction of the paper talks about the grandiose plan of open-endedness, where systems would be able to come up with any computable environment. However, the paper is about generating environments in a restricted environment, with one specific simulator and small and discrete action spaces. It is not clear how the constrained experiments performed in this paper contribute to the grand goals of open-ended learning.

The environments that OMNI-EPIC generates could just be environments that humans implemented, made available online in GitHub, and an LLM was trained on. In this case, OMNI-EPIC is working as a retrieval system that can create different environments within a single codebase. While I see value in such a retrieval system, the paper is not about it. The paper doesn't even consider the possibility that the LLMs are simply retrieving knowledge and replicating it. From what I understand, this doesn't align with the open-ended goal of being... open-ended. A retrieval system will be bounded by whatever we created and made available online. The paper would have to go through a deep re-write to adjust these claims or to perform the proper evaluations to address these concerns.

Here are three examples of sentences that are sloppy.

1. *Because the model has distilled a sense of interesting from reading the internet.* The model does not read; we should use the correct technical terms. The model is trained on Internet data.
2. *Since code is Turing complete.* This is not true. You could have a language that is not Turing complete. The authors might be referring to commonly used general-purpose languages that are Turing complete.
3. *While our academic lab lacks the resources to train generalist agents on a large set of OMNI-EPIC-generated tasks, our findings suggest that this would be straightforward.* While I appreciate the use of the word "suggest," I don't see how we can conclude that learning a generalist agent is straightforward. Training RL agents is tricky (catastrophic forgetting and saturated neural networks are examples of issues one might come across). Evidence suggests that doing this in dozens of environments cannot be straightforward.

Overall, the paper currently reads like an informal report, which makes it difficult to pinpoint the problem that is being solved and the claims that are being made.

Finally, OMNI-EPIC is closely related to procedural content generation, an active research area in the computer games community. See the book *Procedural Content Generation in Games* by Noor Shaker, Julian Togelius, and Mark J. Nelson for an outdated overview.

**Evaluation**

The experiments are unfinished, as the environments generated with OMNI-EPIC are not used to actually evaluate RL algorithms and how such environments can potentialize them. The evaluation focuses on metrics of diversity of the environments used, instead of focusing on the actual goal of improving learning. If the goal is to learn a generalist agent (is that true?), then why not evaluate it that way? The current evaluation serves as a proxy for the actual evaluation, and it is not clear how good of a proxy this is.

The paper justifies why the evaluation is restricted by saying they do not have enough computational resources. While I sympathize with the issue, I cannot recommend acceptance of a paper that doesn't evaluate on the problem it is trying to solve.

**Questions:**

1. Is there a way you could have OMNI-EPIC design simpler problems for which you could thoroughly evaluate RL agents? Even with the amount of computational resources you have available in your research lab?
2. How can you verify whether OMNI-EPIC is solving a retrieval problem or an environment-generation problem?
3. Why is Darwin-completeness defined as the potential to generate any learning environment? How is Darwin's theory of evolution related to the ability to generate learning environments?

---

> ### Author Response · Authors · 2024-11-23
>
> Thank you for your constructive feedback and the opportunity to address your concerns.
>
> ---
>
> > First, the paper lacks scholarship, in the sense that the writing is sloppy…
>
> We are sorry you thought so, as we care deeply about excellent writing. We have fixed every issue you raised, specifically we:
> - Replaced “reading the internet” with “training on internet data”
> - Correcting our statement about Turing completeness to specify that not all but “the programming language used here (Python) is Turing Complete”
> - Removing the unsupported claim about “straightforward” generalist agent training and acknowledging that training generalist agents will require an extensive distribution of environments, considerable effort, and substantial computational resources.
>
> > the paper is about generating environments in a restricted environment, with one specific simulator and small and discrete action spaces. It is not clear how the constrained experiments performed in this paper contribute to the grand goals of open-ended learning.
>
> These limitations exist in the specific demo tasks for the paper, in part because we are a small academic lab, but they are not limitations of the method itself. For example, OMNI-EPIC is not limited by the simplified action space. To demonstrate this, we conducted new experiments using an Ant robot with a continuous action space (new Appendix O). These results show that OMNI-EPIC can generate diverse and complex environments even with non-discrete, larger action spaces. In the limit, any action space could be used, even extremely general ones like a keyboard + mouse movements, controls for any robot, etc.
>
> > The environments that OMNI-EPIC generates could just be environments that humans implemented, made available online in GitHub, and an LLM was trained on. In this case, OMNI-EPIC is working as a retrieval system that can create different environments within a single codebase.
> > How can you verify whether OMNI-EPIC is solving a retrieval problem or an environment-generation problem?
>
> We acknowledge the possibility that OMNI-EPIC may exhibit characteristics resembling a retrieval system. However, we do not think it is doing so. First, we are not able to find any similar environments online after doing extensive searching. Second, the generated environments are customized to work with our custom PyBullet settings, which differ from standard PyBullet configurations (no default plane loaded for more freedom of what objects and surfaces are created, different code structure for easier creation of new environments). To our best knowledge, these particular implementations of PyBullet environments have never been coded or published online before. This suggests that OMNI-EPIC is likely doing more than replicating existing code, although we cannot rule out that it is porting pre-existing abstract ideas for environments in this new setting.
>
> However, as you noted, even if OMNI-EPIC is just a retrieval system, being able to create an appropriate curriculum from human-generated environments would still be incredibly valuable to the field of open-endedness. By effectively sequencing diverse training environments, it offers a useful resource for agent learning. We have added this benefit to the paper.
>
> Thus, we think it is inventing new environments, but it is hard to know for sure. We have acknowledged that in the paper.
>
> Relatedly, we consider it an interesting research question motivated by OMNI-EPIC (but part of the larger discussion of whether AI can generate new knowledge) whether OMNI-EPIC can keep generating new, interesting challenges forever. This is a fascinating research area, and as yet we do not yet know the answer. It could be the case that a model of interestingness generalizes well outside the distribution of human data, and that the environment generator does as well, since both can look at the growing archive of “discoveries” (environments that have been created that have high learning progress and the model thought was interesting). Investigating that question is well beyond the scope (and budget!) of this paper, but it is very interesting future work. We have added this to the paper discussion.
>
> While OMNI-EPIC may leverage existing knowledge, we believe it still represents meaningful progress. In our paper, we will acknowledge the open questions regarding whether it might be doing retrieval only vs. inventing new things, and how investigating that is an interesting area of future research.

---

> > ### Author Response · Authors · 2024-11-23
> >
> > > it lacks comparison with a very large body of works devoted to generating content in simulated environments: procedural content generation
> > > OMNI-EPIC is closely related to procedural content generation, an active research area in the computer games community. See the book Procedural Content Generation in Games by Noor Shaker, Julian Togelius, and Mark J. Nelson for an outdated overview.
> >
> > We appreciate the reviewer's comment. While we have already discussed relevant works by Sudhakaran et al. (2023) and Todd et al. (2023), which we believe were the most pertinent Procedural Content Generation references for this paper, we have now included additional references, including the book Procedural Content Generation in Games by Shaker et al. (2016), as suggested.
> >
> > > the environments generated with OMNI-EPIC are not used to actually evaluate RL algorithms
> >
> > To clarify, we did demonstrate training RL agents on the environments generated by OMNI-EPIC in one of our experiments, and the results (including repeated runs) are presented in Section 5: Short Run with Learning. It is true that we also conducted other experiments without RL training, but that was to simulate what OMNI-EPIC could do in terms of producing diversity were it given a much larger compute budget.
> >
> > > The evaluation focuses on metrics of diversity of the environments used, instead of focusing on the actual goal of improving learning
> >
> > In Section 6: Quantitative Results, we use ANNECS and ANNECS-OMNI metrics to measure the system's progress in learning the generated environments. Additionally, we have included additional ablation studies (new Appendix G) to demonstrate how various components of OMNI-EPIC contribute to training RL agents.
> >
> > > If the goal is to learn a generalist agent (is that true?), then why not evaluate it that way?
> > >  I cannot recommend acceptance of a paper that doesn't evaluate on the problem it is trying to solve
> >
> > Thank you for the comment. In our paper, we aim to generate a diverse task distribution and create an effective curriculum within that distribution for either specialist or generalist agents. As mentioned in our results section, OMNI-EPIC successfully accomplished those goals, which are critical prerequisites for training specialist or generalist agents in an open-ended manner. Training generalist agents requires much more computational resources, so we chose to demonstrate OMNI-EPIC’s capabilities by training specialist agents. OMNI-EPIC generates a progressively complex and diverse set of tasks (Section 4, Section 5, new Appendix I), which, in turn, train a growing population of specialist agents. Furthermore, based on prior work by AdA Team et al. (2023), we believe our approach would also be highly beneficial for training generalist agents, albeit acknowledging the significant resources required for such an extensive evaluation. But the value of OMNI-EPIC is not just for training generalist agents, and doing so is not the problem we are trying to solve, and we thus do not believe it should be a prerequisite for publication.
> >
> > > Is there a way you could have OMNI-EPIC design simpler problems for which you could thoroughly evaluate RL agents?
> >
> > As noted above, the original paper did conduct experiments where agents learn to solve the tasks generated by OMNI-EPIC. However, due to compute constraints, we were unable to extend these experiments to include a larger number of tasks (beyond ~20) or train a generalist across all generated environments. While exploring these possibilities in a computationally cheaper domain is an interesting direction, we believe they are not strictly necessary to convey the vision and current contributions of OMNI-EPIC. Our existing experiments demonstrate that OMNI-EPIC generates a diverse and progressively challenging set of tasks, and prior work by AdA Team et al. (2023) has shown that generalist agents emerge when sufficiently diverse environments are trained on. While we intend to explore these additional experiments in future work, we hope that the current results sufficiently showcase OMNI-EPIC’s potential and contributions to open-ended learning, and hope you agree.
> >
> > > Why is Darwin-completeness defined as the potential to generate any learning environment? How is Darwin's theory of evolution related to the ability to generate learning environments?
> >
> > The term Darwin Completeness and the concept of generating any possible learning environment were introduced in prior work:  the AI-GA paper (Clune, 2020). That paper explains the connection to Darwin, but in short it is a nod to the vast, complex number of ecological niches (environments) on Earth, which Darwinian evolution innovates within. Our use of this language aligns with their framework, which emphasizes the necessity of a vast and diverse task space for open-ended learning. We hope this clarifies the relevance of the term in this context.

---

> > > ### Author Response · Authors · 2024-11-25
> > >
> > > Dear Reviewer kcfh,
> > >
> > > As we approach the end of the rebuttal period, we notice that you have not yet responded to our detailed answers to your concerns. Your score differs substantially from other reviewers who rated the paper more favorably. Since ICLR typically requires consensus among reviewers for acceptance, your current score could prevent the paper from being published despite the positive feedback from other reviewers.
> > >
> > > We have worked diligently to address each of your concerns by:
> > > - Fixing all writing issues you identified
> > > - Adding new experiments with continuous action spaces (Ant robot)
> > > - Clarifying our contribution relative to retrieval systems
> > > - Adding procedural content generation references
> > > - Explaining our evaluation choices and computational constraints
> > >
> > > Given these improvements and our detailed responses, would you be willing to review our rebuttal and reconsider your rating? If there are remaining concerns that would need to be addressed to merit a higher score, we would greatly appreciate your feedback before the rebuttal period ends.

---

> > > > ### Comment · Reviewer_kcfh · 2024-11-27
> > > >
> > > > Thank you for the detailed response. I apologize, but I didn't have the chance to read them carefully and I will need a few more days. I would like to read the paper again before commenting on your response. This message is just to say that I haven't forgotten about it.

---

> > > > > ### Author Response · Authors · 2024-11-27
> > > > >
> > > > > Thank you for your update. We understand how busy you are and deeply appreciate your taking the time to review our response and updated manuscript. We look forward to hearing back from you.

---

> > > ### Comment · Reviewer_kcfh · 2024-12-02
> > >
> > > Thank you for the detailed response to my review. Given the discrepancy between my score and the scores of the other reviewers, I decided to read the paper again, and that is what I did.
> > >
> > > I recognize that OMNI-EPIC represents a significant amount of work and an ambitious attempt to tackle a challenging problem. However, in its current form, I believe the paper would benefit from a substantial revision to clarify its contributions and address key areas for improvement. I will keep my score, as I still have concerns about both the write-up and the experiments. Naturally, the final decision is up to the Area Chairs. My goal is to provide them with a complete picture so they can make a more informed decision.
> > >
> > > I appreciate that the paper provides experiments with RL agents. However, I feel that these experiments do not fully address the key objective stated in Section 3.5: "a key objective of open-ended learning is to enable agents to master an ever-expanding set of tasks." This important goal is not evaluated in the paper. The focus is primarily on the diversity of the problems generated rather than on evaluating a generalist agent. When I asked about smaller-scale experiments, I was suggesting the inclusion of evaluations involving a generalist agent compared with baseline agents of similar abilities. Section 5 offers some anecdotal evidence of a generalist agent, but I believe it would strengthen the paper to include learning curves with statistically significant results and comparisons against baseline systems. For example, baselines using curriculum learning or machine teaching could serve as meaningful benchmarks. These experiments do not need to involve complex tasks—simple and diverse tasks would suffice to make the case.
> > >
> > > Regarding the write-up, I felt the paper might overstate its claims in some places, which could potentially misalign expectations with its contributions. For instance, the introduction discusses general intelligence and Darwin completeness, but much of the paper's work focuses on a procedural content generation (PCG) system based on large language models. Reframing the paper as primarily a contribution to PCG research might make it easier to situate the work within the literature and to decide on appropriate baselines and experiments. For example, in Appendix A, the statement "OMNI-EPIC opens a new era for games, where procedural content is automatically generated and tailored to the player's abilities, ensuring a consistently engaging experience," could be better supported by comparisons with established PCG baselines and evaluations.
> > >
> > > I also wanted to provide specific feedback on the use of the term "Darwin completeness." While I understand it is drawn from prior work, I think its usage here could be refined. For instance, the term is used inconsistently across the introduction and conclusion, where it sometimes appears to describe a property of the agent and other times a property of the environment. Clarifying its definition and ensuring consistency would help the reader understand its role in the paper. Additionally, the term "Darwin completeness" might not fully capture the intended concept, as Darwinian evolution typically involves adaptation to specific environments rather than the ability to adapt to *all possible environments*. A clearer explanation or alternative framing could avoid potential misunderstandings.
> > >
> > > Overall, I see potential in the ideas presented in OMNI-EPIC, and with further clarification of its framing, inclusion of stronger experimental evidence, and refinement of the terminology, I believe the paper could make a significant contribution to the field.

---

> > > > ### Author Response · Authors · 2024-12-03
> > > >
> > > > Thank you for your additional thoughtful feedback. We would like to address several key points you raised.
> > > >
> > > > ---
> > > >
> > > > > I recognize that OMNI-EPIC represents a significant amount of work and an ambitious attempt to tackle a challenging problem.
> > > >
> > > > Thank you for recognizing the amount of work we have put into OMNI-EPIC.
> > > >
> > > > > ... baselines using curriculum learning or machine teaching could serve as meaningful benchmarks.
> > > >
> > > > We want to clarify why comparing to curriculum learning methods would not be appropriate in this context: OMNI-EPIC creates problems on-the-fly, while curriculum learning methods require pre-defined target tasks (e.g., Mario levels, Minecraft objectives) and typically operate in a single domain. The fundamental contribution of OMNI-EPIC is not about having the best curriculum management system, but rather demonstrating the capability to generate and learn on a wide range of tasks. Traditional curriculum approaches cannot serve as meaningful baselines since they are designed for a fundamentally different purpose. In fact, the fact that there is no obvious algorithm to compare OMNI-EPIC to is one of the reasons we think this work is valuable: it opens up a new research frontier, rather than being a new/better way to do something others have tried before.
> > > >
> > > > > Reframing the paper as primarily a contribution to PCG research might make it easier to situate the work within the literature and to decide on appropriate baselines and experiments.
> > > >
> > > > We respectfully disagree with reframing OMNI-EPIC as primarily a PCG contribution. Traditional PCG systems are designed to generate content within a single game or domain (e.g., generating levels for Super Mario). In contrast, OMNI-EPIC can generate entirely new environments and games across any possible domain, including elements such as reward functions, domain randomization, termination conditions, and other features specific to RL that are not usually considered in PCG systems. PCG systems are also often motivated by and developed for different reasons, and are rarely motivated by creating open-ended learning algorithms. These fundamental differences make direct comparisons to PCG systems inappropriate, as they operate in fundamentally different scopes and have different things they are trying to accomplish (and thus should be evaluated on). For example, it would be inappropriate to critique PCG systems for not accomplishing open-ended goals, since that is not what they are designed for.
> > > >
> > > > > The focus is primarily on the diversity of the problems generated rather than on evaluating a generalist agent.
> > > >
> > > > While we are excited about the possibility to train generalist agents with OMNI-EPIC in the future, we believe your focus on generalist agents overlooks a key aspect of our work - the ability to train any type of agent, including (as we do in this paper) specialist agents. OMNI-EPIC's ability to generate appropriate training curricula for both generalist and specialist agents is an important contribution. The paper demonstrates how specialist agents can master specific skills through targeted environment generation, which is a valuable capability alongside generalist training. Given that we are simply introducing OMNI-EPIC as an interesting direction, we do not believe it should be a requirement for publication that we demonstrate both specialist and generalist agent training runs, especially given the vast financial resources that are required for the latter. As an academic lab, we try to make contributions with what we have, and we think our current experiments demonstrate the promise of OMNI-EPIC.
> > > >
> > > > > However, I feel that these experiments do not fully address the key objective stated in Section 3.5: "a key objective of open-ended learning is to enable agents to master an ever-expanding set of tasks."
> > > >
> > > > We would like to clarify that in Section 3.5, when we discuss mastering an ever-expanding set of tasks, we present this as one of the key objectives of open-ended learning as a field, rather than as the primary objective of our paper. Our work focuses on a crucial prerequisite: generating the diverse environments that would enable such learning. The ability to generate appropriate training environments is a necessary foundation for any system that aims to master an expanding set of tasks. Perhaps more importantly in terms of your evaluation and score, **the quoted sentence is entirely compatible with those agents being specialists**. Perennially producing ever more skilled specialists is also in line with that sentence, and we show that potential in our paper.

---

> > > > > ### Author Response · Authors · 2024-12-03
> > > > >
> > > > > > I also wanted to provide specific feedback on the use of the term "Darwin completeness." While I understand it is drawn from prior work, I think its usage here could be refined.
> > > > >
> > > > > Thank you for pointing out the need for clarity regarding Darwin completeness. We agree that we should be more precise with this terminology. In our paper, Darwin completeness is specifically a property of the environment generation system - it describes the capability to generate any environment that could foster open-ended evolution/learning. We will make this clearer in the manuscript. But as you note, the term already exists in the field and we are simply referring to it.
> > > > >
> > > > > ---
> > > > >
> > > > > Thank you again for your review. We hope you are open to reconsidering your score one last time with an open mind (and with an eye to avoiding the well known “anchoring bias”, which can occur to your original score). We really have made substantial improvements to the paper since you selected that score. If you believe the paper has improved, and/or have updated your views based on our comments and the other reviewers’ comments and scores, we would deeply appreciate you reassessing whether such a low score is warranted.
> > > > >
> > > > > Thank you,
> > > > > The OMNI-EPIC Authors

---

### Author Response · Authors · 2024-11-23

Thank you all for reviewing our work. We appreciate the comment that “leveraging foundation models for this task is a promising idea” (kcfh) and that OMNI-EPIC tackles “the ambitious problem of continually generating a diverse set of problems” (kcfh). We’re grateful for the observation that OMNI-EPIC contributes to the “open-endedness community” (PisJ) and serves as “an excellent example of how foundation models can be successfully used for this purpose” (PisJ). The use of “code as an environment representation” was noted as “much more expressive” than prior approaches (Cvga), and the system was recognized for presenting “a full pipeline that theoretically can be run for a long time to continuously generate novel tasks” (Cvga). Finally, OMNI-EPIC was described as “original and forward-looking” (xWEx) and as taking “first steps toward autotelic and open-ended learning” (xWEx).

We have addressed all concerns, which has significantly improved the manuscript. **Most importantly, we conducted additional experiments demonstrating that OMNI-EPIC is not limited to discrete or simplified action spaces and can generate diverse, complex environments, including those involving continuous action spaces, such as an Ant robot. We also performed human evaluation and additional ablation studies to demonstrate the necessity and effects of the different components of OMNI-EPIC. These new results confirm all previous findings and substantiate the claims regarding OMNI-EPIC’s ability to generate progressively challenging and diverse tasks.**

# The prominent additions are:

1. Clarifications and corrections to improve technical rigor, including rephrasing claims about Turing completeness, removing unsupported claims, and incorporating additional references to procedural content generation and intrinsic motivation research.
2. New experiments demonstrating OMNI-EPIC’s ability to generate diverse and complex environments for continuous action spaces, such as those involving an Ant robot, validating that it is not limited to discrete or simplified action spaces.
3. Ablation studies showing the necessity and effects of different components of OMNI-EPIC, including:
Using similar tasks as example inputs to the task generator as compared to uniformly sampling tasks from the archive.
The inclusion of both success and failure feedback in the task generator, which significantly improves performance.
The importance of learning progress in the task generator prompt.
4. Additional quantitative analyses including:
A comparison of task success rates between OMNI-EPIC (with transfer learning) and training from scratch, highlighting the benefits of transfer learning.
Diversity and curriculum progression metrics demonstrating how OMNI-EPIC generates increasingly challenging tasks over time.
5. Metrics on environment generation including:
The percentage of times the environment generator produced executable code, the percentage of times the task generator produced interesting tasks, and the percentage of times the generated tasks were solved.
Human evaluation of the percentage of generated environments that are solvable.
Human evaluation of how aligned the automatic success detector is with human evaluations.

These additions address the main concerns of reviewers. We feel this paper makes an important, helpful contribution that the ML community will benefit from if published. We hope the reviewers agree and will consider increasing their scores. See below and reviewer-specific responses for line-by-line replies to all reviewer comments.

---

### Meta-Review · Area_Chair_FdwX · 2024-12-19

**Metareview:**

This is an ambitious paper that provides a compelling demonstration that an LLM-based code synthesis pipeline, combined with inner-loop RL, can achieve continual discovery of novel environments. Albeit their experiments focus on a limited domain, the ideas and experimental decisions here are useful for the community to build on. Most reviewers agree that the work provides interesting ideas and insights, and that the experiments, as presented, are compelling. Echoing reviewer kcfh's concerns, I encourage the authors to clarify their discussion to emphasize that their work does not directly investigate whether the agents trained via OMNI-EPIC in fact do benefit in general capabilities from the evolutionary process they introduce. Showing the OMNI-EPIC agents show significant (and continued) improvements on held-out environment designs would be useful to support this latter hypothesis. In its current form, this work, however, does show OMNI-EPIC's potential as an engine for discovering novel environments with LLM-based code synthesis, which is the main contribution. It is this experimental result, alongside the proposed approach, that serves as the key, valuable contribution.

**Additional Comments On Reviewer Discussion:**

Most reviewers agreed that this paper presents an impressive and original contribution. One reviewer (kcfh) pointed out that the paper can improve its framing of its key contribution. Specifically, this work does not provide empirical evidence that the OMNI-EPIC agent is a general problem-solver. Rather the experiments show the proposed program synthesis pipeline is able to generate a wide diversity of environments in their restricted problem domain. The authors are expected to make this point clear in their final manuscript.

---

### Decision · Program_Chairs · 2025-01-22

Accept (Poster)